# DeepAFL: Deep Analytic Federated Learning

**Jianheng Tang[1,7], Yajiang Huang[† 2], Kejia Fan[2], Feijiang Han[3], Jiaxu Li[2], Jinfeng Xu[4], Run He[5], Anfeng Liu[2], Houbing Herbert Song[6], Huiping Zhuang[† 5], Yunhuai Liu[† 1,7]**

[1]Peking University, PRC; [2]Central South University, PRC; [3]University of Pennsylvania, USA;
[4]The University of Hong Kong, PRC; [5]South China University of Technology, PRC;
[6]University of Maryland, Baltimore County (UMBC), USA;
[7]Key Lab of High Confidence Software Technologies (PKU), Ministry of Education, PRC

## Abstract

Federated Learning (FL) is a popular distributed learning paradigm to break down data silo. Traditional FL approaches largely rely on gradient-based updates, facing significant issues about heterogeneity, scalability, convergence, and overhead, etc. Recently, some analytic-learning-based work has attempted to handle these issues by eliminating gradient-based updates via analytical (i.e., closed-form) solutions. Despite achieving superior invariance to data heterogeneity, these approaches are fundamentally limited by their single-layer linear model with a frozen pre-trained backbone. As a result, they can only achieve suboptimal performance due to their lack of representation learning capabilities. In this paper, to enable representable analytic models while preserving the ideal invariance to data heterogeneity for FL, we propose our *Deep Analytic Federated Learning* approach, named *DeepAFL*. Drawing inspiration from the great success of ResNet in gradient-based learning, we design gradient-free residual blocks in our DeepAFL with analytical solutions. We introduce an efficient layer-wise protocol for training our deep analytic models layer by layer in FL through least squares. Both theoretical analyses and empirical evaluations validate our DeepAFL's superior performance with its dual advantages in heterogeneity invariance and representation learning, outperforming state-of-the-art baselines by up to 5.68%–8.42% across three benchmark datasets. Related code is available at `https://github.com/tangent-heng/DeepAFL`.

## 1 Introduction

Federated Learning (FL) has emerged as a prominent paradigm that enables distributed learning to break down data silos (Fan et al., 2025c; Yang et al., 2023; Guo et al., 2025; Li et al., 2025c). The objective of FL is to allow a group of clients to collaboratively train a powerful and robust global model, while preserving their data privacy (Ren et al., 2025; Liu et al., 2024b). The field of FL has seen substantial growth across a wide range of applications (Zhou et al., 2024; Wu et al., 2025).

Traditional FL methods rely on a gradient-based optimization paradigm, exemplified by the classic FedAvg (McMahan et al., 2017) and its following variants (Li et al., 2021b; Yang et al., 2024). These methods typically necessitate iterative optimization processes to achieve convergence (Wang et al., 2024; Tan et al., 2022a). Yet, these gradient-based techniques are widely acknowledged to suffer from several major challenges (Ye et al., 2023a; Chai et al., 2024; He et al., 2025), as follows.

(1) **Heterogeneity Issues:** The data across clients are often Not Independently and Identically Distributed (Non-IID), which can severely impact model performance and convergence.
(2) **Scalability Issues:** As the number of clients increases, especially to a large scale (e.g., thousands of clients), the FL systems can experience substantial performance degradation.
(3) **Convergence Issues:** The FL methods may struggle to converge within limited aggregation rounds, particularly in challenging scenarios of non-IID data or large-scale clients.
(4) **Overhead Issues:** The overall FL process incurs significant overhead from multi-epoch training on each client and multi-round model aggregation across clients for convergence.

Many researchers have come to realize that the aforementioned challenges in FL are fundamentally rooted in the long-standing reliance on gradient-based updates, which are inherently sensitive and

---

[†]Corresponding Authors.

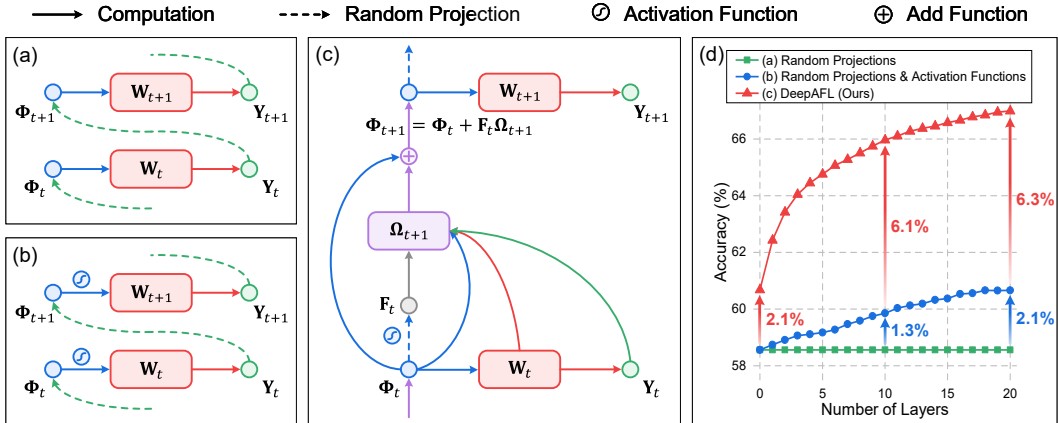

Figure 1: Comparing our proposed DeepAFL with the naive approaches for representation learning. (a) Illustration of multi-analytic layers with random projections. (b) Illustration of multi-analytic layers with random projections & activation functions. (c) Illustration of our proposed DeepAFL. (d) Among these, our DeepAFL exhibits the best performance improvements with increasing layers. For all these approaches, the activation function is GELU, the random projection dimension is 1024, the pre-trained model is the ResNet-18 used in AFL, and the evaluation dataset is the CIFAR-100.

costly in the distributed FL scenario (Ye et al., 2023a; Fanì et al., 2024; He et al., 2025). From this perspective, existing gradient-based methods can only superficially alleviate these issues rather than fundamentally address them. Therefore, a natural and promising avenue to fundamentally address these gradient-related issues is to eliminate gradient-based updates entirely (He et al., 2025).

As a prominent gradient-free technique, analytic learning has exhibited great promise by achieving analytical (closed-form) solutions through the least squares (Lai et al., 2025; Fan et al., 2025a). To introduce this technique into the FL, Analytic Federated Learning (AFL) has been proposed with the help of pre-trained models (He et al., 2025). The core idea of AFL is to leverage frozen pre-trained models (e.g., foundation models) for feature extraction on input data. Based on the extracted features, AFL can then build a single-layer linear model, which has an analytical solution. Through its specific protocols on local training and global aggregation, AFL achieves ideal invariance to data heterogeneity, as its final aggregated global model is equivalent to the centralized analytical solution.

Despite achieving State-Of-The-Art (SOTA) performance with its unique invariance to data heterogeneity, AFL is significantly limited by its single-layer linear model. On the one hand, this analytic model is foundational to AFL, as its simplicity with a convex optimization objective is a prerequisite for deriving an analytical solution. On the other hand, it also severely limits AFL's application, as it can only learn a **linear mapping** from the frozen backbone's features to the final output, thus **fundamentally failing to perform representation learning within the FL systems**. Consequently, due to the limited learning capacity of the linear analytic model, AFL is prone to underfitting, especially when the backbone itself is lightweight. Additionally, even if the backbone itself possesses sufficient feature extraction capabilities, the linear separability of its output features may still be insufficient.

Thus, an interesting and challenging problem arises: *can we deepen AFL's analytic model to enable its representation learning capabilities while simultaneously preserving its analytical solutions for invariance to data heterogeneity?* A quite naive approach is using a random projection after each analytic model to construct the input features for the next layer, as displayed in Figure 1(a). Yet, this approach brings almost no performance improvement, as all its layers are simply linear mappings. Then, we attempt to introduce an activation function after each feature to provide non-linearity, as illustrated in Figure 1(b). Based on the evaluation results presented in Figure 1(d), the deep activated random projections can, to some extent, enrich the feature representations. However, as the number of layers increases, the deepening approach (b) struggles to improve the performance through representation learning. Therefore, these naive approaches fall far short of our requirements.

Drawing inspiration from the great success of ResNet in gradient-based learning (He et al., 2016), in this paper, we adopt the similar skip connections to boost the representation of the analytic layers. Specifically, we model this representation learning as $\boldsymbol{\Phi}_t = \boldsymbol{\Phi}_{t-1} + g_t(\boldsymbol{\Phi}_{t-1})$, where $g_t(\cdot)$ represents a nonlinear feature transformation. In gradient-based learning, the residual blocks $g_t(\boldsymbol{\Phi}_{t-1})$ can

be easily learned, as Stochastic Gradient Descent (SGD) with backpropagation can automatically adjust the network weights to learn appropriate representations. Given that analytic models preclude gradient-based updates via backpropagation, a key technical challenge lies in how to effectively learn the residual blocks $g_t(\mathbf{\Phi}_{t-1})$ for meaningful boosting within the framework of analytic learning.

To enable representable analytic models while preserving their ideal invariance to data heterogeneity for FL, in this paper, we propose our ***Deep A**nalytic **F**ederated **L**earning* approach, named ***DeepAFL***. Inherited from AFL, our DeepAFL also employs a pre-trained backbone for initial feature extraction. Then, we propose to randomly project and activate these features to form the zero-layer features $\mathbf{\Phi}_0$, which can yield an immediate performance gain of about 2.1%, as shown in Figure 1(d). After this setup, our DeepAFL continuously refines the features $\mathbf{\Phi}_t$ layer by layer. Specifically, to obtain the residual block $g_t(\mathbf{\Phi}_{t-1})$ for layer $t$, we first build a nonlinear representation $\mathbf{F}_{t-1}$ from $\mathbf{\Phi}_{t-1}$ using a random projection layer with an activation function, as illustrated in Figure 1(c). Subsequently, we can obtain $g_t(\mathbf{\Phi}_{t-1}) = \mathbf{F}_{t-1}\mathbf{\Omega}_t$ by introducing a learnable transformation $\mathbf{\Omega}_t$ to adjust and scale the representation $\mathbf{F}_{t-1}$. We derive the optimal analytical solution for $\mathbf{\Omega}_t$ via *sandwiched least squares*. As shown in Figure 1(d), our DeepAFL exhibits a desired capability of deep representation learning, as its performance continuously and markedly improves with an increasing number of layers.

Our primary contributions are summarized as follows:

- Conceptually, we propose our DeepAFL, a novel approach that can achieve gradient-free representation learning while preserving ideal invariance to data heterogeneity in FL.

- Technically, we develop an efficient layer-wise protocol for learning deep analytic models via least squares. In our DeepAFL, the clients only need to conduct lightweight forward-propagation computations, so that the server can aggregate the global models layer by layer.

- Theoretically, we demonstrate two ideal properties of our DeepAFL: its invariance to data heterogeneity and its capability of representation learning. To the best of our knowledge, our DeepAFL represents the first to achieve both of these ideal properties simultaneously.

- Experimentally, we provide extensive evaluations on three benchmark datasets to show the superiority of our DeepAFL, which outperforms SOTA baselines by up to 5.68%–8.42%, thanks to its dual advantages in heterogeneity invariance and representation learning.

## 2    RELATED WORK

### 2.1    FEDERATED LEARNING

As a prominent distributed learning paradigm, FL allows multiple clients to collaboratively train a global model to break down data silos (Li et al., 2025b; Yang et al., 2023). Existing FL techniques are largely derived from FedAvg (McMahan et al., 2017) and rely on gradient-based optimization. Despite advancements in FL, their reliance on gradients also causes inherent issues about overhead, convergence, heterogeneity, and scalability (Ye et al., 2023a; Chai et al., 2024; He et al., 2025). Recently, AFL has introduced a new and promising wave that fundamentally handles these issues by avoiding gradient-based updates via analytic learning (He et al., 2025). Based on it, some subsequent research has extended this concept to personalized FL and federated continual learning (Fan et al., 2025b; Tang et al., 2025), achieving excellent performance. Nevertheless, a key limitation of existing analytic-learning-based FL approaches is their lack of representation learning capability.

### 2.2    ANALYTIC LEARNING

Analytic learning (Zhuang et al., 2022; 2023; 2024), also known as pseudoinverse learning (Cline, 1964; Guo et al., 2001), has emerged as a popular gradient-free technique to address gradient-related issues (Toh, 2018; Lanthaler & Nelsen, 2023; Prabhu et al., 2024; Zozoulenko et al., 2025; Fan et al., 2025a; Bolager et al., 2023). Its core idea is to directly derive analytical (i.e., closed-form) solutions using least squares, to eliminate gradient-based updates (Li et al., 2025a; Peng et al., 2025). Despite the extensive superiority shown by analytic learning, current approaches are largely limited by their single-layer linear models, which sacrifice the capabilities of deep representation learning. On the other hand, achieving representable deep analytic models while preserving their closed-form solutions stands as a great challenge, especially in the distributed scenario of FL. Drawing inspiration from the success of ResNet (He et al., 2016) in gradient-based learning, we fundamentally solve this problem by carefully designing gradient-free residual blocks with closed-form solutions.

## 3  OUR PROPOSED DEEPAFL

In this section, we elaborate in detail on our proposed DeepAFL. For clarity, we consider the typical FL setting involving one server and $K$ clients. Moreover, to align with existing research, we use the most prevalent task in FL, image recognition, as the prime example to describe the workflow of our DeepAFL (Yu et al., 2024; 2025; Miao et al., 2025; He et al., 2025). Each client's local dataset is denoted as $\mathcal{D}^k = \{\mathbf{X}^k, \mathbf{Y}^k\}$, where $\mathbf{X}^k \in \mathbb{R}^{N_k \times d_\mathrm{H} \times d_\mathrm{W} \times d_c}$ and $\mathbf{Y}^k \in \mathbb{R}^{N_k \times \mathrm{C}}$ represent the $N_k$ local samples and their corresponding labels, respectively. Here, $d_\mathrm{H} \times d_\mathrm{W} \times d_\mathrm{C}$ denotes the 3 dimensions (height, width, and channels) of the input images, and C denotes the number of output classes.

The objective of our proposed DeepAFL is to construct a deep residual analytic network comprising $T$ layers for deriving the features $\mathbf{\Phi}_T$ and the corresponding global classifier $\mathbf{W}_T$, in a distributed and gradient-free manner. In Section 3.1, we detail the motivation and insight of the proposed deep analytic learning in our DeepAFL, which is described from a centralized perspective for clarity. In Section 3.2, we elaborate on the specific workflow and implementation of our DeepAFL within the data-distributed FL scenario. In Section 3.3, we provide theoretical analyses of our DeepAFL for its validity, privacy, and efficiency, particularly showing its dual ideal properties.

### 3.1  DEEP RESIDUAL ANALYTIC LEARNING OF OUR DEEPAFL

Here, we aim to introduce the key motivation and insights behind the proposed deep residual analytic model in our DeepAFL. For clarity and brevity, we first adopt a centralized perspective and consider a full dataset $\mathcal{D} = \{\mathbf{X}, \mathbf{Y}\}$ to construct the $T$-layer deep residual analytic network. Specifically, we progressively learn better representations and refine the features $\{\mathbf{\Phi}_t\}_{t=0}^{T}$ layer by layer. Based on the features $\mathbf{\Phi}_t$ for each layer $t$, we construct a corresponding analytic classifier $\mathbf{W}_t$.

First of all, inherited from AFL (He et al., 2025), we employ a pre-trained backbone for our initial feature extraction from $\mathbf{X}$ to obtain $\widetilde{\mathbf{X}} \in \mathbb{R}^{N \times d_\mathrm{X}}$ via (1). This has been widely adopted as a common practice in many recent studies of FL (Nguyen et al., 2023; Piao et al., 2024; Yu et al., 2024; 2025).

$$\widetilde{\mathbf{X}} = \mathrm{Backbone}(\mathbf{X}, \mathbf{\Theta}), \tag{1}$$

where $\mathrm{Backbone}(\cdot)$ represents the pre-trained backbone with frozen parameters $\mathbf{\Theta}$. Based on this initial feature extraction of AFL, we further incorporate the activated random projection to boost the features' representation, thereby forming the zero-layer features $\mathbf{\Phi}_0 \in \mathbb{R}^{N \times d_\Phi}$, as follows.

$$\mathbf{\Phi}_0 = \sigma(\widetilde{\mathbf{X}}\mathbf{A}), \tag{2}$$

where $\sigma(\cdot)$ represents the activation function, and $\mathbf{A} \in \mathbb{R}^{d_\mathrm{X} \times d_\Phi}$ is the random projection matrix. This activated random projection can increase the feature dimension to a suitable size, boosting its linear separability, as widely validated in existing studies (Cover, 2006; Zhuang et al., 2022; 2024; Zhang et al., 2025). To progressively learn richer representations, we attempt to deepen the network. Yet, naive approaches to deepening the network fail to achieve it effectively, as shown in Figure 1.

Drawing inspiration from the great success of ResNet in gradient-based learning (He et al., 2016), we adopt the similar skip connections to boost the representations of the analytic layers. Specifically, we model this representation learning process as the feature updating formula in (3):

$$\mathbf{\Phi}_t = \mathbf{\Phi}_{t-1} + g_t(\mathbf{\Phi}_{t-1}), \forall t \in [1, T]. \tag{3}$$

Here, $g_t(\cdot)$ represents the residual block as a nonlinear feature transformation. We will specify the detailed gradient-free design of the residual block within our DeepAFL in (6) later. Built upon the obtained feature matrix $\mathbf{\Phi}_t$ for each layer $t$, we can construct a corresponding analytic classifier $\mathbf{W}_t$. Specifically, consistent with other existing analytic-learning-based approaches (He et al., 2025), the optimization objective for the analytic classifier $\mathbf{W}_t$ can be formulated as follows.

$$\mathbf{W}_t = \arg\min_{\mathbf{W}} \|\mathbf{Y} - \mathbf{\Phi}_t\mathbf{W}\|_\mathrm{F}^2 + \lambda\|\mathbf{W}\|_\mathrm{F}^2, \forall t \in [0, T], \tag{4}$$

where $\lambda$ denotes the regularization parameter and $\|\cdot\|_\mathrm{F}^2$ represents the Frobenius norm. Since cross-entropy loss does not admit a closed-form solution, we use the Mean Squared Error (MSE) loss here. In fact, the MSE loss is widely adopted in analytic learning (Zhuang et al., 2022; 2023; 2024) and can achieve performance comparable to that of the cross-entropy loss (Hui & Belkin, 2021). Using the least squares method, we can derive the optimal analytical solution to the objective (4), as given by (5). The detailed proof of the analytical solution (5) is provided in **Lemma 1** of Section 3.3.

$$\mathbf{W}_t = (\mathbf{\Phi}_t^\top\mathbf{\Phi}_t + \lambda\mathbf{I})^{-1}\mathbf{\Phi}_t^\top\mathbf{Y}, \forall t \in [0, T]. \tag{5}$$

After obtaining the analytic classifier $\mathbf{W}_t$, we then proceed with deep residual representation learning to update the $(t+1)$-th-layer features. Based on (3), the key to this problem is the gradient-free design of the residual blocks, which possess analytical solutions with the properties of **stochasticity**, **nonlinearity**, and **learnability**. Thus, in our DeepAFL, the residual block is instantiated as follows:

$$g_{t+1}(\mathbf{\Phi}_t) = \sigma(\mathbf{\Phi}_t\mathbf{B}_t)\mathbf{\Omega}_{t+1} = \mathbf{F}_t\mathbf{\Omega}_{t+1}, \forall t \in [0, T). \tag{6}$$

Here, $\mathbf{B}_t \in \mathbb{R}^{d_\Phi \times d_F}$ represents the random projection matrix for **stochasticity**, akin to that provided by SGD in gradient-based learning. Meanwhile, $\sigma(\cdot)$ means the activation function for **nonlinearity**, a component that is widely demonstrated to be essential in deep representation learning. Moreover, $\mathbf{\Omega}_{t+1} \in \mathbb{R}^{d_F \times d_\Phi}$ represents the trainable transformation matrix for providing **learnability**. In (6), we further define $\mathbf{F}_t = \sigma(\mathbf{\Phi}_t\mathbf{B}_t)$ as the hidden random feature, thereby isolating the trainable $\mathbf{\Omega}_{t+1}$. In Section 4, we will provide extensive ablation studies to show the effectiveness of these components.

Then, we focus on learning the optimal $\mathbf{\Omega}_{t+1}$ for effective representation boosting within the residual block. Here, our objective can be written as, given a fixed classifier $\mathbf{W}_t$ from the previous layer, to minimize the empirical risk by optimizing the new features $\mathbf{\Phi}_{t+1} = \mathbf{\Phi}_t + \mathbf{F}_t\mathbf{\Omega}_{t+1}$, as follows:

$$\begin{aligned}
\mathbf{\Omega}_{t+1} &= \arg\min_{\mathbf{\Omega}} \|\mathbf{Y} - (\mathbf{\Phi}_t + \mathbf{F}_t\mathbf{\Omega})\mathbf{W}_t\|_F^2 + \gamma\|\mathbf{\Omega}\|_F^2 \\
&= \arg\min_{\mathbf{\Omega}} \|\mathbf{R}_t - \mathbf{F}_t\mathbf{\Omega}\mathbf{W}_t\|_F^2 + \gamma\|\mathbf{\Omega}\|_F^2, \forall t \in [0, T).
\end{aligned} \tag{7}$$

Here, we define the residual of the current layer's classification as $\mathbf{R}_t = \mathbf{Y} - (\mathbf{\Phi}_t\mathbf{W}_t)$ for notational convenience. The optimization objective (7) can be seen as a special case of generalized Sylvester matrix equations (Wu et al., 2008; Ding et al., 2008; Duan, 2015; Zozoulenko et al., 2025), with the unknown variable $\mathbf{\Omega}$ being *sandwiched* between two known ones, $\mathbf{F}_t$ and $\mathbf{W}_t$. This structure enables the derivation of an analytical solution in (8). We term this as the *sandwiched least squares problem*, and provide a detailed proof of the analytical solution (8) in **Lemma 2** of Section 3.3.

$$\mathbf{\Omega}_{t+1} = \mathbf{V}_t[(\mathbf{V}_t^\top \mathbf{F}_t^\top \mathbf{R}_t \mathbf{W}_t^\top \mathbf{U}_t) \oslash (\gamma\mathbf{1} + \text{diag}(\mathbf{\Lambda}_t^F) \otimes \text{diag}(\mathbf{\Lambda}_t^W))]\mathbf{U}_t^\top, \forall t \in [0, T), \tag{8}$$

where $\mathbf{F}_t^\top \mathbf{F}_t = \mathbf{V}_t\mathbf{\Lambda}_t^F\mathbf{V}_t^\top$ and $\mathbf{W}_t\mathbf{W}_t^\top = \mathbf{U}_t\mathbf{\Lambda}_t^W\mathbf{U}_t^\top$ are spectral decompositions, while $\oslash$ and $\otimes$ represent element-wise division and outer product, respectively. Once we obtain the transformation matrix $\mathbf{\Omega}_{t+1}$, we can compute the next-layer features via the following recursive formula.

$$\mathbf{\Phi}_{t+1} = \mathbf{\Phi}_t + \mathbf{F}_t\mathbf{\Omega}_{t+1}, \forall t \in [0, T). \tag{9}$$

In summary, the construction of our deep residual analytic network proceeds in a layer-wise manner, involving the alternating derivation of $\mathbf{W}_t$ and $\mathbf{\Omega}_{t+1}$ via analytical solutions (5) and (8), respectively. Specifically, the solution procedure follows the sequence $\mathbf{W}_0 \mapsto \mathbf{\Omega}_1 \mapsto \mathbf{W}_1 \mapsto \cdots \mapsto \mathbf{\Omega}_T \mapsto \mathbf{W}_T$, beginning with the zero-layer classifier $\mathbf{W}_0$ and concluding with the final-layer classifier $\mathbf{W}_T$. For further clarity, we provide detailed illustrations of our DeepAFL's formulations in Figures 14–16 of Appendix F. Based on the above centralized process, we will then elaborate on the specific workflow and implementation of our DeepAFL for the data-distributed FL scenario in Section 3.2.

## 3.2 Federated Implementation of our DeepAFL

In this subsection, we display the federated implementation of our DeepAFL for the data-distributed FL scenario, where each client owns its local dataset $\mathcal{D}^k = \{\mathbf{X}^k, \mathbf{Y}^k\}$. The overall implementation workflow of our DeepAFL still follows the layer-wise procedure that is described in Section 3.1. Yet, in the FL setting, the analytical computation of the global weights $\mathbf{W}_t$ and $\mathbf{\Omega}_{t+1}$ requires additional aggregation of all clients' local knowledge to update the global knowledge.

First of all, each client $k$ performs feature extraction on its local data $\mathbf{X}^k$ to obtain its local zero-layer feature matrix $\mathbf{\Phi}_0^k \in \mathbb{R}^{N_k \times d_\Phi}$ similar to (1) and (2), as follows.

$$\mathbf{\Phi}_0^k = \sigma(\text{Backbone}(\mathbf{X}^k, \mathbf{\Theta})\mathbf{A}). \tag{10}$$

Then, for each layer $t \in [0, T]$, each client utilizes its features $\mathbf{\Phi}_t^k$ and labels $\mathbf{Y}^k$ to compute its local *Feature Auto-Correlation Matrix* $\mathbf{G}_t^k$ and *Label Cross-Correlation Matrix* $\mathbf{H}_t^k$, as follows.

$$\mathbf{G}_t^k = (\mathbf{\Phi}_t^k)^\top \mathbf{\Phi}_t^k, \quad \mathbf{H}_t^k = (\mathbf{\Phi}_t^k)^\top \mathbf{Y}^k. \tag{11}$$

Then, the server needs to aggregate $\mathbf{G}_t^{1:K}$ and $\mathbf{H}_t^{1:K}$ from the clients through (12). This process can be implemented via existing Secure Aggregation Protocols (Bonawitz et al., 2016; 2017; So et al., 2023). We provide related privacy analyses in Section 3.3.

$$\mathbf{G}_t^{1:K} = \sum_{k=1}^K \mathbf{G}_t^k, \quad \mathbf{H}_t^{1:K} = \sum_{k=1}^K \mathbf{H}_t^k. \tag{12}$$

Once obtaining $\mathbf{G}_t^{1:K}$ and $\mathbf{H}_t^{1:K}$, the server can derive the global classifier $\mathbf{W}_t$ via (13) and distribute it to all clients then. As shown in Section 3.3, regardless of how the data are distributed across clients, the global analytic classifier $\mathbf{W}_t$ obtained through (13) is exactly identical to the optimal solution of centralized analytic learning on the full dataset, thus achieving invariance to data heterogeneity.

$$\begin{aligned}
\mathbf{W}_t &= \left[(\mathbf{\Phi}_t^{1:K})^\top \mathbf{\Phi}_t^{1:K} + \lambda\mathbf{I}\right]^{-1} (\mathbf{\Phi}_t^{1:K})^\top \mathbf{Y}^{1:K} \\
&= (\mathbf{G}_t^{1:K} + \lambda\mathbf{I})^{-1}\mathbf{H}_t^{1:K}.
\end{aligned} \tag{13}$$

Then, each client is required to compute its hidden random features $\mathbf{F}_t^k$ and local residual matrix $\mathbf{R}_t^k$ based on the previously obtained features $\mathbf{\Phi}_t^k$ and classifiers $\mathbf{W}_t$, as follows.

$$\mathbf{F}_t^k = \sigma(\mathbf{\Phi}_t^k \mathbf{B}_t), \quad \mathbf{R}_t^k = \mathbf{Y}^k - (\mathbf{\Phi}_t^k \mathbf{W}_t). \tag{14}$$

Subsequently, each client utilizes the hidden random features $\mathbf{F}_t^k$ and local residual matrix $\mathbf{R}_t^k$ to get its local *Hidden Auto-Correlation Matrix* $\mathbf{\Pi}_t^k$ and *Residual Cross-Correlation Matrix* $\mathbf{\Upsilon}_t^k$ via (15).

$$\mathbf{\Pi}_t^k = (\mathbf{F}_t^k)^\top \mathbf{F}_t^k, \quad \mathbf{\Upsilon}_t^k = (\mathbf{F}_t^k)^\top \mathbf{R}_t^k. \tag{15}$$

Then, akin to $\mathbf{G}_t^k$ and $\mathbf{H}_t^k$, the server aggregates $\mathbf{\Pi}_t^{1:K}$ and $\mathbf{\Upsilon}_t^{1:K}$ from the clients via (16).

$$\mathbf{\Pi}_t^{1:K} = \sum\nolimits_{k=1}^{K} \mathbf{\Pi}_t^k, \quad \mathbf{\Upsilon}_t^{1:K} = \sum\nolimits_{k=1}^{K} \mathbf{\Upsilon}_t^k. \tag{16}$$

Once obtaining $\mathbf{\Pi}_t^{1:K}$ and $\mathbf{\Upsilon}_t^{1:K}$, the server can derive $\mathbf{\Omega}_{t+1}$ via (17) and distribute it to all clients.

$$\begin{aligned}
\mathbf{\Omega}_{t+1} &= \mathbf{V}_t[(\mathbf{V}_t^\top (\mathbf{F}_t^{1:K})^\top \mathbf{R}_t^{1:K} \mathbf{W}_t^\top \mathbf{U}_t) \oslash (\gamma\mathbf{1} + \mathrm{diag}(\mathbf{\Lambda}_t^{\mathrm{F}}) \otimes \mathrm{diag}(\mathbf{\Lambda}_t^{\mathrm{W}}))]\mathbf{U}_t^\top \\
&= \mathbf{V}_t[(\mathbf{V}_t^\top \mathbf{\Upsilon}_t^{1:K} \mathbf{W}_t^\top \mathbf{U}_t) \oslash (\gamma\mathbf{1} + \mathrm{diag}(\mathbf{\Lambda}_t^{\mathrm{F}}) \otimes \mathrm{diag}(\mathbf{\Lambda}_t^{\mathrm{W}}))]\mathbf{U}_t^\top,
\end{aligned} \tag{17}$$

where $\mathbf{V}_t$, $\mathbf{\Lambda}_t^{\mathrm{F}}$, $\mathbf{U}_t$, and $\mathbf{\Lambda}_t^{\mathrm{W}}$ are obtained by spectral decompositions, as follows.

$$\mathbf{\Pi}_t^{1:K} = (\mathbf{F}_t^{1:K})^\top \mathbf{F}_t^{1:K} = \mathbf{V}_t \mathbf{\Lambda}_t^{\mathrm{F}} \mathbf{V}_t^\top, \quad \mathbf{W}_t \mathbf{W}_t^\top = \mathbf{U}_t \mathbf{\Lambda}_t^{\mathrm{W}} \mathbf{U}_t^\top. \tag{18}$$

After receiving $\mathbf{\Omega}_{t+1}$ from the server, each client can thus update the next-layer features $\mathbf{\Phi}_{t+1}^k$ as:

$$\mathbf{\Phi}_{t+1}^k = \mathbf{\Phi}_t^k + \mathbf{F}_t^k \mathbf{\Omega}_{t+1}. \tag{19}$$

The preceding process continues layer-by-layer until the final-layer classifier $\mathbf{W}_T$ is completed. For clarity, we summarize the detailed training and inference procedures of our DeepAFL in Algorithm 1 and Algorithm 2, respectively, within Appendix A. In the next subsection, we will comprehensively provide theoretical analyses for our DeepAFL, including its validity, privacy, and efficiency.

## 3.3 THEORETICAL ANALYSES

**Validity Analyses**: Here, we provide detailed analyses of the validity of our DeepAFL. Specifically, we first derive the analytical solutions of our DeepAFL in Lemmas 1–2, and then demonstrate our DeepAFL's dual properties of heterogeneity invariance and representation learning in Theorems 1–2.

**Lemma 1**: For any least squares problem with the following form:

$$\mathbf{W}^* = \arg\min_{\mathbf{W}} \|\mathbf{Y} - \mathbf{\Phi}\mathbf{W}\|_{\mathrm{F}}^2 + \lambda\|\mathbf{W}\|_{\mathrm{F}}^2, \tag{20}$$

it yields a distinct analytical (i.e., closed-form) solution, which can be formulated as:

$$\mathbf{W}^* = (\mathbf{\Phi}^\top \mathbf{\Phi} + \lambda\mathbf{I})^{-1}\mathbf{\Phi}^\top \mathbf{Y}. \tag{21}$$

*Proof.* See Appendix B.1 for details.

**Lemma 2**: For any sandwiched least squares problem with the following form:

$$\mathbf{\Omega}^* = \arg\min_{\mathbf{\Omega}} \|\mathbf{R} - \mathbf{F}\mathbf{\Omega}\mathbf{W}\|_{\mathrm{F}}^2 + \gamma\|\mathbf{\Omega}\|_{\mathrm{F}}^2, \tag{22}$$

it yields a distinct analytical (i.e., closed-form) solution, which can be formulated as:

$$\mathbf{\Omega}^* = \mathbf{V}[(\mathbf{V}^\top \mathbf{F}^\top \mathbf{R}\mathbf{W}^\top \mathbf{U}) \oslash (\gamma\mathbf{1} + \mathrm{diag}(\mathbf{\Lambda}^{\mathrm{F}}) \otimes \mathrm{diag}(\mathbf{\Lambda}^{\mathrm{W}}))]\mathbf{U}^\top, \tag{23}$$

where $\mathbf{F}^\top \mathbf{F} = \mathbf{V}\mathbf{\Lambda}^{\mathrm{F}}\mathbf{V}^\top$ and $\mathbf{W}\mathbf{W}^\top = \mathbf{U}\mathbf{\Lambda}^{\mathrm{W}}\mathbf{U}^\top$ are spectral decompositions, while $\oslash$ and $\otimes$ represent element-wise division and outer product.

*Proof.* See Appendix B.1 for details.

**Theorem 1 (Invariance to Data Heterogeneity)**: In the FL scenario, let $\mathcal{D}$ be the full dataset, and $\mathcal{P} = \{\mathcal{D}^1, \mathcal{D}^2, \ldots, \mathcal{D}^K\}$ be any heterogeneous partition of $\mathcal{D}$ among $K$ clients, where $\mathcal{D}_i \cap \mathcal{D}_j = \emptyset$ for $i \neq j$ and $\bigcup_{i=1}^{K} \mathcal{D}_i = \mathcal{D}$. Given fixed seeds for the random projection matrices $\mathbf{A}$ and $\{\mathbf{B}_t\}_{t=0}^{T-1}$, the global model weights $\{\mathbf{W}_t\}_{t=0}^{T}$ and $\{\mathbf{\Omega}_t\}_{t=1}^{T}$ derived from DeepAFL are invariant to any data partition $\mathcal{P}$ with different heterogeneity, being identical to the centralized analytical solutions on $\mathcal{D}$.

*Proof.* See Appendix B.2 for details.

**Theorem 2 (Capability of Representation Learning)**: Let the empirical risk $\mathcal{H}(\mathbf{\Phi}, \mathbf{W})$, i.e., the loss function, for a given feature representation $\mathbf{\Phi}$ and a classifier $\mathbf{W}$ be defined as:

$$\mathcal{H}(\mathbf{\Phi}, \mathbf{W}) = \|\mathbf{Y} - \mathbf{\Phi}\mathbf{W}\|_{\mathrm{F}}^2, \tag{24}$$

where $\mathbf{Y}$ denotes the ground truth labels. Our DeepAFL yields a sequence of feature-classifier pairs $(\mathbf{\Phi}_t, \mathbf{W}_t)$ at each layer $t \in [0, T]$. When setting regularization parameters $\gamma$ and $\lambda$ to 0, the sequence of the empirical risks $\{\mathcal{H}(\mathbf{\Phi}_t, \mathbf{W}_t)\}_{t=0}^{T}$ in our DeepAFL keeps monotonically non-increasing, i.e.,

$$\mathcal{H}(\mathbf{\Phi}_t, \mathbf{W}_t) \geq \mathcal{H}(\mathbf{\Phi}_{t+1}, \mathbf{W}_{t+1}), \forall t \in [0, T). \tag{25}$$

Furthermore, as the number of layers $T$ within our DeepAFL increases (i.e., $T \to \infty$), the sequence of empirical risks is guaranteed to converge to a limit $\mathcal{H}^* \leq \mathcal{H}(\mathbf{\Phi}_0, \mathbf{W}_0)$.

*Proof.* See Appendix B.3 for details.

Notably, Theorem 2 demonstrates our DeepAFL's representation learning capability under the basic condition of no regularization ($\gamma = \lambda = 0$). In practice, we often employ regularization to enhance generalization performance and numerical stability. Thus, we introduce Theorem 3 in Appendix B.3 to extend our analysis to cover the regularized settings. In addition, we will further provide extensive empirical evidence in Section 4 to support these dual advantageous properties of our DeepAFL.

**Privacy Analyses**: For the privacy of our DeepAFL, several previous studies have provided similar analyses for *Auto-Correlation* and *Cross-Correlation Matrices* (Tan et al., 2022b; He et al., 2025; Fan et al., 2025b). Particularly, inferring each client's raw data is challenging as the size of its local dataset $N_k$ remains private. Furthermore, this aggregation process is a form of *Secure Multi-Party Computation*, and many existing protocols can be readily integrated into our DeepAFL to further enhance privacy (Bonawitz et al., 2016; 2017; So et al., 2023). See Appendix C for more details.

**Efficiency Analyses**: In our DeepAFL, the overall complexities of computation and communication for each client are $\mathcal{O}(TN_k(d_\Phi^2 + d_\Phi d_{\mathrm{F}} + d_{\mathrm{F}}^2))$ and $\mathcal{O}(T(d_\Phi^2 + d_{\mathrm{F}}^2))$, while those for the central server are $\mathcal{O}(T(d_\Phi^3 + d_{\mathrm{F}}^3 + d_\Phi^2 d_{\mathrm{F}} + d_\Phi d_{\mathrm{F}}^2))$ and $\mathcal{O}(TKd_\Phi d_{\mathrm{F}})$, respectively. See Appendix D for the detailed derivations and analyses. Notably, in Section 4, we will show the superior efficiency of our DeepAFL in comparison to existing gradient-based baselines. In our later experiments, we default to selecting projection dimensions $d_\Phi = d_{\mathrm{F}} = 1024$ to balance the effectiveness and efficiency.

## 4 EXPERIMENTAL EVALUATIONS

### 4.1 EXPERIMENTAL SETUP

**Datasets & Settings.** We conduct our experiments on three prominent benchmark datasets in FL: the CIFAR-10 (Krizhevsky & Hinton, 2009), CIFAR-100 (Krizhevsky & Hinton, 2009), and Tiny-ImageNet (Le & Yang, 2015). To simulate diverse data heterogeneity scenarios in FL, we use two common non-IID partitioning settings: Latent Dirichlet Allocation (Lin et al., 2020) (as Non-IID-1) and Sharding (Lin et al., 2020) (as Non-IID-2). We use the parameters $\alpha$ and $s$ to control the level of heterogeneity in these two non-IID settings, respectively. In both cases, smaller parameter values indicate more heterogeneous data distributions. Specifically, we set $\alpha \in \{0.1, 0.05\}$, $s \in \{2, 4\}$ for CIFAR-10, while setting $\alpha \in \{0.1, 0.01\}$, $s \in \{5, 10\}$ for CIFAR-100 and Tiny-ImageNet.

**Baselines & Metrics.** We compare our DeepAFL against 7 traditional gradient-based baselines, including FedAvg (McMahan et al., 2017), FedProx (Li et al., 2020) and MOON (Li et al., 2021a), FedGen (Zhu et al., 2021), FedDyn (Acar et al., 2021), FedNTD (Lee et al., 2022), and FedDisco (Ye et al., 2023b). Moreover, we also include the analytic learning-based method AFL (He et al., 2025) as a baseline to further highlight our advantages. For fair comparisons, we align the experimental benchmark with those of AFL (He et al., 2025) and use the same pre-trained ResNet-18 backbone for all methods. We employ the accuracy (%), computational cost (s), and communication cost (MB) as the main metrics. See Appendix E.2 for more details of our experimental implementation.

Table 1: Performance comparisons of the top-1 accuracy (%) among our DeepAFL and the baselines, on the CIFAR-100 and Tiny-ImageNet. The best result is highlighted in **bold**, and the second-best result is underlined. All the experiments were conducted three times, and the results are shown as $\mathrm{Mean}_{\pm\mathrm{Standard\ Error}}$. All the improvements of our DeepAFL were validated by Chi-squared tests. The results of all baselines are directly obtained from the given benchmark in AFL (He et al., 2025).

| Baseline | CIFAR-100 | | | | Tiny-ImageNet | | | |
|---|---|---|---|---|---|---|---|---|
| | Non-IID-1 | | Non-IID-2 | | Non-IID-1 | | Non-IID-2 | |
| | $\alpha = 0.1$ | $\alpha = 0.01$ | $s = 10$ | $s = 5$ | $\alpha = 0.1$ | $\alpha = 0.01$ | $s = 10$ | $s = 5$ |
| FedAvg (2017) | $56.62_{\pm0.12}$ | $32.99_{\pm0.20}$ | $55.76_{\pm0.13}$ | $48.33_{\pm0.15}$ | $46.04_{\pm0.27}$ | $32.63_{\pm0.19}$ | $39.06_{\pm0.26}$ | $29.66_{\pm0.19}$ |
| FedProx (2020) | $56.45_{\pm0.22}$ | $33.37_{\pm0.09}$ | $55.80_{\pm0.16}$ | $48.29_{\pm0.14}$ | $46.47_{\pm0.23}$ | $32.26_{\pm0.14}$ | $38.97_{\pm0.23}$ | $29.17_{\pm0.16}$ |
| MOON (2021a) | $56.58_{\pm0.02}$ | $33.34_{\pm0.11}$ | $55.70_{\pm0.25}$ | $48.34_{\pm0.19}$ | $46.21_{\pm0.14}$ | $32.38_{\pm0.20}$ | $38.79_{\pm0.14}$ | $29.24_{\pm0.30}$ |
| FedGen (2021) | $56.48_{\pm0.17}$ | $33.09_{\pm0.09}$ | $60.93_{\pm0.17}$ | $48.12_{\pm0.06}$ | $46.27_{\pm0.14}$ | $32.33_{\pm0.14}$ | $38.82_{\pm0.16}$ | $29.37_{\pm0.25}$ |
| FedDyn (2021) | $57.55_{\pm0.08}$ | $36.12_{\pm0.08}$ | $\underline{61.09}_{\pm0.09}$ | $\underline{59.34}_{\pm0.11}$ | $47.72_{\pm0.22}$ | $35.19_{\pm0.06}$ | $41.36_{\pm0.06}$ | $35.18_{\pm0.18}$ |
| FedNTD (2022) | $56.60_{\pm0.14}$ | $32.59_{\pm0.21}$ | $54.69_{\pm0.15}$ | $47.00_{\pm0.19}$ | $46.17_{\pm0.16}$ | $31.86_{\pm0.44}$ | $37.55_{\pm0.09}$ | $29.01_{\pm0.14}$ |
| FedDisco (2023b) | $55.79_{\pm0.04}$ | $25.72_{\pm0.08}$ | $54.65_{\pm0.09}$ | $45.86_{\pm0.18}$ | $47.48_{\pm0.06}$ | $27.15_{\pm0.10}$ | $38.86_{\pm0.12}$ | $27.72_{\pm0.18}$ |
| AFL (2025) | $\underline{58.56}_{\pm0.00}$ | $\underline{58.56}_{\pm0.00}$ | $58.56_{\pm0.00}$ | $58.56_{\pm0.00}$ | $\underline{54.67}_{\pm0.00}$ | $\underline{54.67}_{\pm0.00}$ | $\underline{54.67}_{\pm0.00}$ | $\underline{54.67}_{\pm0.00}$ |
| DeepAFL ($T = 5$) | $64.72_{\pm0.07}$ | $64.72_{\pm0.07}$ | $64.72_{\pm0.07}$ | $64.72_{\pm0.07}$ | $60.31_{\pm0.04}$ | $60.31_{\pm0.04}$ | $60.31_{\pm0.04}$ | $60.31_{\pm0.04}$ |
| DeepAFL ($T = 10$) | $65.96_{\pm0.05}$ | $65.96_{\pm0.05}$ | $65.96_{\pm0.05}$ | $65.96_{\pm0.05}$ | $61.37_{\pm0.07}$ | $61.37_{\pm0.07}$ | $61.37_{\pm0.07}$ | $61.37_{\pm0.07}$ |
| DeepAFL ($T = 20$) | $\mathbf{66.98}_{\pm0.04}$ | $\mathbf{66.98}_{\pm0.04}$ | $\mathbf{66.98}_{\pm0.04}$ | $\mathbf{66.98}_{\pm0.04}$ | $\mathbf{62.35}_{\pm0.01}$ | $\mathbf{62.35}_{\pm0.01}$ | $\mathbf{62.35}_{\pm0.01}$ | $\mathbf{62.35}_{\pm0.01}$ |
| Improvement ↑ | $\mathbf{8.42}_{p<0.05}$ | $\mathbf{8.42}_{p<0.05}$ | $\mathbf{5.89}_{p<0.05}$ | $\mathbf{7.64}_{p<0.05}$ | $\mathbf{7.68}_{p<0.05}$ | $\mathbf{7.68}_{p<0.05}$ | $\mathbf{7.68}_{p<0.05}$ | $\mathbf{7.68}_{p<0.05}$ |

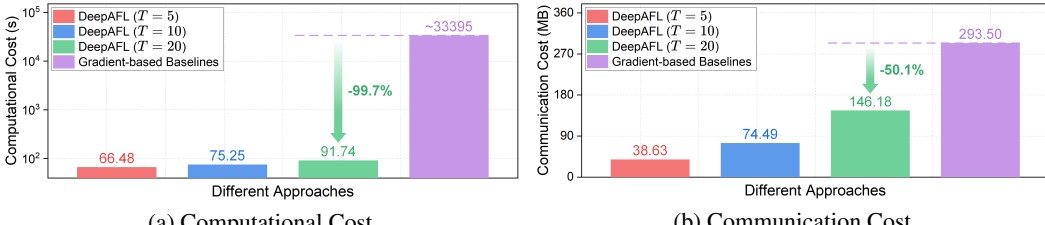

(a) Computational Cost  (b) Communication Cost

Figure 2: Efficiency evaluations on the CIFAR-100 dataset.

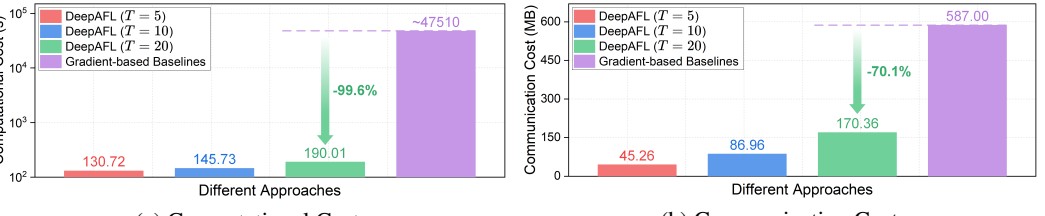

(a) Computational Cost  (b) Communication Cost

Figure 3: Efficiency evaluations on the Tiny-ImageNet dataset.

## 4.2 EXPERIMENTAL RESULTS

**Main Comparisons.** Here, we compare our DeepAFL extensively against baselines across various datasets and settings, as shown in Tables 1 and 2. For a fair comparison, we included the benchmark of the AFL (He et al., 2025) and show our DeepAFL's performance under $T \in \{5, 10, 20\}$. Notably, even with a small value of $T = 5$, our DeepAFL already surpasses all SOTA baselines, thanks to its dual advantages in heterogeneity invariance and representation learning. As $T$ increases ($5 \to 20$), the testing accuracy of our DeepAFL consistently enhances and eventually reaches $86.43\%$, $66.98\%$, and $62.35\%$ for the CIFAR-10, CIFAR-100, and Tiny-ImageNet, respectively, achieving impressive improvements of up to $5.68\%$–$8.42\%$ over the SOTA baselines. See Appendix E.3.1 for more details.

**Invariance Analyses.** In Tables 1 and 2, our DeepAFL shows invariant performance across various Non-IID settings on the same dataset, akin to the AFL. In contrast, other gradient-based baselines commonly suffer performance degradation as the degree of heterogeneity increases. This invariance property of our DeepAFL can be further extended to the number of clients. As shown in Figure 7, the superiority of our DeepAFL over the gradient-based baseline progressively widens with an increased number of clients ($100 \to 1000$) for the large-scale scenarios. These results empirically validate the ideal invariance of our DeepAFL as shown in Theorem 1. See Appendix E.3.2 for more details.

**Representation Analyses.** Another key advantage of our DeepAFL is its capability of deep representation learning. As shown in Tables 9–11, our DeepAFL exhibits a stable improvement in both training and testing accuracy as the number of layers $T$ increases. This consistent gain shows its effective ability to learn deep representations, handling the common issue of underfitting in traditional analytic learning. Moreover, while our main results only report our DeepAFL's performance at $T \in \{5, 10, 20\}$, we observe its performance continues to improve as $T$ further increases. Specifically, on the complex CIFAR-100 and Tiny-ImageNet, our DeepAFL's testing accuracy at $T = 50$ improves by over $1.5\%$ compared to that at $T = 20$. See Appendix E.3.3 for more details.

**Efficiency Evaluations.** We further give efficiency evaluations to show DeepAFL's superior balance between accuracy and efficiency. As shown in Figures 2–3, we show that, in addition to its SOTA performance, our DeepAFL also achieves superior efficiency compared to existing gradient-based baselines. In Tables 9– 11 and Figures 4–6, the total time cost of DeepAFL for 100 clients shows a minimal marginal increase with each additional layer, adding 1–2s per layer for the CIFAR-10 and CIFAR-100, and 3s per layer for the Tiny-ImageNet. Thus, when set to $T = 50$, our DeepAFL can achieve more than $9\%$ performance improvements compared to AFL on the complex CIFAR-100 and Tiny-ImageNet, with a time cost that is less than twice that of AFL, as the primary time-consuming backbone's forward pass only needs to be performed once. See Appendix E.3.4 for more details.

**Parameter Analyses.** Then, we provide comprehensive parameter sensitivity analyses of our Deep-AFL, including the regularization parameters $\lambda$, $\gamma$, the activation function $\sigma(\cdot)$, and the projection dimensions $d_\Phi$, $d_\mathrm{F}$. Specifically, as shown in Tables 3–4, for the regularization parameters, our Deep-AFL is insensitive to $\lambda$ but more sensitive to $\gamma$. For the simple CIFAR-10, we observe satisfactory performance with $\gamma \in [0.1, 0.5]$. For the more complex datasets, CIFAR-100 and Tiny-ImageNet, the optimal value for $\gamma$ is a smaller $0.01$. In Table 5, the activation function exhibits a notable impact on performance, with a difference of about $2\%$. Here, GELU is the optimal activation function, while Softshrink is the poorest yet still beats the model with no activation. As shown in Tables 6-8, larger dimensions for both $d_\Phi$ and $d_\mathrm{F}$ lead to better performance. Nonetheless, overly large dimensions can not only incur significant overhead due to their quadratic complexity, but also impact the numerical stability, causing crashing when $d_\Phi = d_\mathrm{F} = 2^{13}$. See Appendix E.3.5 for more details.

**Ablation Studies.** Finally, we conduct extensive ablation studies for the individual contributions of our DeepAFL's key components. Specifically, we compared our DeepAFL and four distinct ablation models across different layers on various datasets, as shown in Tables 12–17 and Figures 8–10. First, we observe that the residual skip connection in our DeepAFL has a dual role: it not only accelerates performance improvement as the number of layers $T$ increases, but also enhances the final converged performance. Second, ablating the random projection $\mathbf{B}_t$ of each layer with an identity matrix shows that the *stochasticity* provided by $\mathbf{B}_t$ is vital for DeepAFL to escape local optima and saddle points. Third, ablating the activation function $\sigma(\cdot)$ proves the necessity of its *nonlinearity* in our DeepAFL. Fourth, ablating the trainable transformation $\mathbf{\Omega}_{t+1}$ renders the ablation model ineffective, indicating that its provided *learnability* is also indispensable. See Appendix E.3.6 for more details. Meanwhile, we also provide comparisons of our DeepAFL against other deepening strategies in Appendix E.3.7.

## 5 CONCLUSION AND DISCUSSION

In this paper, we propose DeepAFL, as the first attempt in FL to achieve gradient-free representation learning while preserving strong invariance to data heterogeneity. By identifying and addressing a fundamental limitation in existing analytic-learning-based approaches, our DeepAFL exhibits dual advantages in both heterogeneity invariance and representation learning. By virtue of its gradient-free nature, our DeepAFL also achieves high efficiency by eliminating costly iterative computations and communications on gradients. Given its theoretical and empirical superiority, we believe our DeepAFL means a great advancement for the SOTA in the fields of both analytic learning and FL.

For future work, a natural and promising direction is to extend our deep residual analytic models in our DeepAFL to other well-suited fields for analytic learning. Specifically, we will go on to aim at continual learning, where all data arrives online and historical data is not accessible. While analytic learning has been proven to be effective in solving the catastrophic forgetting problem of continual learning, applying our DeepAFL to this field remains non-trivial. This is because the features learned from a previous phase may become invalid in a subsequent one if updating the weights layer by layer. As a result, we would need a new approach to incorporate the phase-wise recursions as well. Some further discussion is provided in Appendix N, and our usage of LLMs is declared in Appendix O.

## ACKNOWLEDGMENTS

This work is supported partly by the National Natural Science Foundation of China under grants 62576013 and 62306117, the National Key Research and Development of China under grant 2024YFC2607404, the Jiangsu Provincial Key Research and Development Program under Grants BE2022065-1, BE2022065-3, the Ningxia Domain-Specific Large Model Health Industry R&D 2024JBGS001, and the GJYC program of Guangzhou under grant 2024D03J0005.

## ETHICS STATEMENT

The authors have read and adhere to the ICLR Code of Ethics. We have carefully considered the potential ethical implications of our work and believe that our proposed approach, DeepAFL, aligns with the principles outlined in the provided ICLR Code of Ethics.

Our research is situated within the FL paradigm, which is specifically designed to address privacy and security concerns by enabling collaborative model training without requiring the sharing of raw, sensitive user data. Our proposed DeepAFL operates exclusively in a decentralized setting, with all local client data remaining on the respective devices. We provide detailed privacy analyses in our paper and discuss how our DeepAFL can be enhanced with existing proven privacy-preserving protocols. The empirical evaluations presented in this paper are based on well-established, publicly available benchmark datasets (i.e., CIFAR-10, CIFAR-100, and Tiny-ImageNet). This work did not involve any human subjects, the collection of new private data, or the use of sensitive information.

Furthermore, a core contribution of our proposed DeepAFL is its superior performance in FL, with its dual advantages in heterogeneity invariance and representation learning. To the best of our knowledge, the process of our DeepAFL does not introduce additional fairness and bias issues. On the contrary, its ability to achieve superior and invariant performance across various heterogeneous FL environments can help foster more equitable and generalizable models in real-world applications.

Lastly, in line with the principles of research reproducibility, we are committed to making our code open-source upon the paper's acceptance. We believe this will allow the broader research community to verify our findings, build upon our work, and ensure full transparency of our methodology.

## REPRODUCIBILITY STATEMENT

For reproducibility, we have made extensive efforts to provide the necessary details. We give detailed procedures of our approach in our paper, including comprehensive descriptions of the algorithmic steps and design choices. For our theoretical claims, the complete proofs and analyses are included in the Appendix. Codes are available at `https://github.com/tangent-heng/DeepAFL`.

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

## A    APPENDIX FOR PROCEDURES OF OUR DEEPAFL

---

**Algorithm 1** The Training Procedure of our Proposed DeepAFL

---

**Input:** The clients' local datasets $\{\mathcal{D}^1, \mathcal{D}^2, \cdots, \mathcal{D}^K\}$.
**Output:** The transformation matrices $\{\mathbf{\Omega}_t\}_{t=1}^T$ and the analytic classifiers $\{\mathbf{W}_t\}_{t=0}^T$.

1: **for** each layer $t \in \{0, 1, 2, \cdots, T\}$ **do**
2:    **// (a) Local Computation for Analytic Classifier Construction (Client side)**
3:    **for** each client $k \in \{1, 2, \cdots, K\}$ **do**
4:       **if** constructing the first layer, i.e., $t = 0$ **then**
5:          Extract and construct its local zero-layer feature matrix $\mathbf{\Phi}_0^k$ via (10)
6:       Compute its local *Feature Auto-Correlation Matrix* $\mathbf{G}_t^k$ using $\mathbf{\Phi}_t^k$ via (11);
7:       Compute its local *Label Cross-Correlation Matrix* $\mathbf{H}_t^k$ using $\mathbf{\Phi}_t^k$ and $\mathbf{Y}^k$ via (11);
8:       Transmit $\{\mathbf{G}_t^k, \mathbf{H}_t^k\}$ to the server;
9:    **// (b) Information Aggregation for Analytic Classifier Construction (Server side)**
10:    Aggregate $\{\mathbf{G}_t^k\}_{k=1}^K$ and $\{\mathbf{H}_t^k\}_{k=1}^K$ to obtain $\mathbf{G}_t^{1:K}$ and $\mathbf{H}_t^{1:K}$ via (12);
11:    Derive the global classifier $\mathbf{W}_t$ using the obtained $\mathbf{G}_t^{1:K}$ and $\mathbf{H}_t^{1:K}$ via (13);
12:    Transmit the global classifier $\mathbf{W}_t$ to all clients;
13:    **// (c) Local Computation for Residual Block Construction (Client side)**
14:    **for** each client $k \in \{1, 2, \cdots, K\}$ **do**
15:       **if** layer requirement not satisfied, i.e., $t < T$ **then**
16:          Compute its local hidden random feature $\mathbf{F}_t^k$ using $\mathbf{\Phi}_t^k$ via (14);
17:          Compute its local residual matrix $\mathbf{R}_t^k$ using $\mathbf{\Phi}_t^k$ and $\mathbf{W}_t$ via (14);
18:          Compute its local *Hidden Auto-Correlation Matrix* $\mathbf{\Pi}_t^k$ using $\mathbf{F}_t^k$ via (15);
19:          Compute its local *Residual Cross-Correlation Matrix* $\mathbf{\Upsilon}_t^k$ using $\mathbf{F}_t^k$ and $\mathbf{R}_t^k$ via (15);
20:          Transmit $\{\mathbf{\Pi}_t^k, \mathbf{\Upsilon}_t^k\}$ to the server;
21:    **// (d) Information Aggregation for Residual Block Construction (Server side)**
22:    **if** layer requirement not satisfied, i.e., $t < T$ **then**
23:       Aggregate $\{\mathbf{\Pi}_t^k\}_{k=1}^K$ and $\{\mathbf{\Upsilon}_t^k\}_{k=1}^K$ to obtain $\mathbf{\Pi}_t^{1:K}$ and $\mathbf{\Upsilon}_t^{1:K}$ via (16);
24:       Derive the transformation matrix $\mathbf{\Omega}_{t+1}$ using $\mathbf{\Pi}_t^{1:K}$ and $\mathbf{\Upsilon}_t^{1:K}$ via (17) and (18);
25:       Transmit the transformation matrix $\mathbf{\Omega}_{t+1}$ to all clients;
26:    **// (e) Feature Updating for Next Layer Construction (Client side)**
27:    **if** layer requirement not satisfied, i.e., $t < T$ **then**
28:       **for** each client $k \in \{1, 2, \cdots, K\}$ **do**
29:          Update its local feature matrix $\mathbf{\Phi}_t^k$ to obtain $\mathbf{\Phi}_{t+1}^k$ via (19);
30: **Return:** The transformation matrices $\{\mathbf{\Omega}_t\}_{t=1}^T$ and the analytic classifiers $\{\mathbf{W}_t\}_{t=0}^T$.

---

**Algorithm 2** The Inference Procedure of our Proposed DeepAFL

---

**Input:** Local sample $\mathbf{x}^i$, transformation matrices $\{\mathbf{\Omega}_t\}_{t=1}^T$, and analytic classifiers $\{\mathbf{W}_t\}_{t=0}^T$.
**Output:** Predicted label $\hat{y}^i$.

1: Extract and construct its local zero-layer feature $\mathbf{\Phi}_0^i$ using $\mathbf{x}^i$ via (10)
2: **for** each layer $t \in \{1, 2, \cdots, T\}$ **do**
3:    Compute its local hidden random feature $\mathbf{F}_t^i$ via (14);
4:    Update its local feature $\mathbf{\Phi}_{t-1}^i$ to obtain $\mathbf{\Phi}_t^i$ using $\mathbf{\Omega}_t$ and $\mathbf{F}_t^i$ via (19);
5: Calculate the predicted score vector $\hat{\mathbf{y}}^i = \mathbf{\Phi}_T^i \mathbf{W}_T$;
6: Calculate the predicted label $\hat{y}^i = \arg\max(\hat{\mathbf{y}}^i)$;
7: **Return:** Predicted label $\hat{y}^i$.

---

## B  APPENDIX FOR VALIDITY ANALYSES

In this section, we thoroughly analyze the validity of our DeepAFL, theoretically establishing several key propositions. First of all, in Appendix B.1, we derived the analytical solutions for the least squares problems addressed in this paper, which also constitute the centralized analytical solution. Next, in Appendix B.2, we demonstrate that the deep residual analytic networks derived from our DeepAFL is equivalent to the centralized closed-form solutions and remain invariant to data heterogeneity. Finally, in Appendix B.3, we validate DeepAFL's representation learning capacity, proving that the empirical risk of our DeepAFL is monotonically non-increasing with increasing layer depth.

### B.1  DERIVATION OF ANALYTICAL LEAST SQUARES SOLUTIONS

Here, we theoretically derive the analytical solutions to the least squares problems (4) and (7) addressed in this paper. In fact, these solutions constitute the centralized analytical solutions for both the global classifier and the transformation matrix. The detailed proofs are as follows.

**Lemma 1**: For any least squares problem with the following form:

$$\mathbf{W}^* = \arg\min_{\mathbf{W}} \|\mathbf{Y} - \mathbf{\Phi}\mathbf{W}\|_{\mathrm{F}}^2 + \lambda\|\mathbf{W}\|_{\mathrm{F}}^2, \tag{26}$$

it yields a distinct analytical (i.e., closed-form) solution, which can be formulated as:

$$\mathbf{W}^* = (\mathbf{\Phi}^\top\mathbf{\Phi} + \lambda\mathbf{I})^{-1}\mathbf{\Phi}^\top\mathbf{Y}. \tag{27}$$

*Proof.* Our proof strategy involves computing the derivative of the loss function in (26) with respect to $\mathbf{W}$ and subsequently setting it to zero to derive the optimal $\mathbf{W}^*$. Specifically, we first denote the loss function of the aforementioned least squares problem (26) as follows:

$$\mathcal{J}(\mathbf{W}) = \|\mathbf{Y} - \mathbf{\Phi}\mathbf{W}\|_{\mathrm{F}}^2 + \lambda\|\mathbf{W}\|_{\mathrm{F}}^2. \tag{28}$$

Subsequently, to facilitate subsequent differentiation, we expand the loss function (28) as follows:

$$\begin{aligned}
\mathcal{J}(\mathbf{W}) &= \|\mathbf{Y} - \mathbf{\Phi}\mathbf{W}\|_{\mathrm{F}}^2 + \lambda\|\mathbf{W}\|_{\mathrm{F}}^2 \\
&= \mathrm{Tr}((\mathbf{Y} - \mathbf{\Phi}\mathbf{W})^\top(\mathbf{Y} - \mathbf{\Phi}\mathbf{W})) + \lambda\mathrm{Tr}(\mathbf{W}^\top\mathbf{W}) \\
&= \mathrm{Tr}(\mathbf{Y}^\top\mathbf{Y}) + 2\mathrm{Tr}(\mathbf{Y}^\top\mathbf{\Phi}\mathbf{W}) + \mathrm{Tr}(\mathbf{W}^\top\mathbf{\Phi}^\top\mathbf{\Phi}\mathbf{W}) + \lambda\mathrm{Tr}(\mathbf{W}^\top\mathbf{W}),
\end{aligned} \tag{29}$$

where $\mathrm{Tr}(\cdot)$ represents the trace of a matrix. To minimize the objective function, we further compute the derivative of $\mathcal{J}(\mathbf{W})$ with respect to $\mathbf{W}$ as follows:

$$\frac{\partial\mathcal{J}(\mathbf{W})}{\partial\mathbf{W}} = -2\mathbf{\Phi}^\top\mathbf{Y} + 2\mathbf{\Phi}^\top\mathbf{\Phi}\mathbf{W} + 2\lambda\mathbf{W}. \tag{30}$$

By setting the derivative to zero, we can obtain:

$$(\mathbf{\Phi}^\top\mathbf{\Phi} + \lambda\mathbf{I})\mathbf{W}^* = \mathbf{\Phi}^\top\mathbf{Y}. \tag{31}$$

Since $(\mathbf{\Phi}^\top\mathbf{\Phi} + \lambda\mathbf{I})$ is full rank and hence invertible, the closed-form solution for the optimal $\mathbf{W}^*$ can be obtained as follows:

$$\mathbf{W}^* = (\mathbf{\Phi}^\top\mathbf{\Phi} + \lambda\mathbf{I})^{-1}\mathbf{\Phi}^\top\mathbf{Y}. \tag{32}$$

∎

**Lemma 2**: For any least squares problem with the following form:

$$\mathbf{\Omega}^* = \arg\min_{\mathbf{\Omega}} \|\mathbf{R} - \mathbf{F}\mathbf{\Omega}\mathbf{W}\|_{\mathrm{F}}^2 + \gamma\|\mathbf{\Omega}\|_{\mathrm{F}}^2, \tag{33}$$

it yields a distinct analytical (i.e., closed-form) solution, which can be formulated as:

$$\mathbf{\Omega}^* = \mathbf{V}[(\mathbf{V}^\top\mathbf{F}^\top\mathbf{R}\mathbf{W}^\top\mathbf{U}) \oslash (\gamma\mathbf{1} + \mathrm{diag}(\mathbf{\Lambda}^{\mathrm{F}}) \otimes \mathrm{diag}(\mathbf{\Lambda}^{\mathrm{W}}))]\mathbf{U}^\top, \tag{34}$$

where $\oslash$ and $\otimes$ represent element-wise division and outer product, respectively, while $\mathbf{F}^\top\mathbf{F} = \mathbf{V}\mathbf{\Lambda}^{\mathrm{F}}\mathbf{V}^\top$ and $\mathbf{W}\mathbf{W}^\top = \mathbf{U}\mathbf{\Lambda}^{\mathrm{W}}\mathbf{U}^\top$ are spectral decompositions.

*Proof.* Similar to the proof of **Lemma 1**, we begin by computing the derivative of the loss function in (33) with respect to $\boldsymbol{\Omega}$ and then set this derivative to zero to derive the optimal $\boldsymbol{\Omega}^*$. Specifically, we denote the loss function of the aforementioned least squares problem (33) as follows:

$$\mathcal{L}(\boldsymbol{\Omega}) = \|\mathbf{R} - \mathbf{F}\boldsymbol{\Omega}\mathbf{W}\|_{\mathrm{F}}^2 + \gamma\|\boldsymbol{\Omega}\|_{\mathrm{F}}^2, \tag{35}$$

which can be further expanded into the following form:

$$\begin{aligned}
\mathcal{L}(\boldsymbol{\Omega}) &= \|\mathbf{R} - \mathbf{F}\boldsymbol{\Omega}\mathbf{W}\|_{\mathrm{F}}^2 + \gamma\|\boldsymbol{\Omega}\|_{\mathrm{F}}^2 \\
&= \mathrm{Tr}((\mathbf{R} - \mathbf{F}\boldsymbol{\Omega}\mathbf{W})^{\top}(\mathbf{R} - \mathbf{F}\boldsymbol{\Omega}\mathbf{W})) + \gamma\mathrm{Tr}(\boldsymbol{\Omega}^{\top}\boldsymbol{\Omega}) \\
&= \mathrm{Tr}(\mathbf{R}^{\top}\mathbf{R}) - 2\mathrm{Tr}(\mathbf{R}^{\top}\mathbf{F}\boldsymbol{\Omega}\mathbf{W}) + \mathrm{Tr}(\mathbf{W}^{\top}\boldsymbol{\Omega}^{\top}\mathbf{F}^{\top}\mathbf{F}\boldsymbol{\Omega}\mathbf{W}) + \gamma\mathrm{Tr}(\boldsymbol{\Omega}^{\top}\boldsymbol{\Omega}).
\end{aligned} \tag{36}$$

Subsequently, by further differentiating (36), we can obtain:

$$\frac{\partial\mathcal{L}(\boldsymbol{\Omega})}{\partial\boldsymbol{\Omega}} = -2\mathbf{F}^{\top}\mathbf{R}\mathbf{W}^{\top} + 2\mathbf{F}^{\top}\mathbf{F}\boldsymbol{\Omega}\mathbf{W}\mathbf{W}^{\top} + 2\gamma\boldsymbol{\Omega}. \tag{37}$$

By setting the derivative to zero, we can obtain:

$$\mathbf{F}^{\top}\mathbf{F}\boldsymbol{\Omega}^*\mathbf{W}\mathbf{W}^{\top} + \gamma\boldsymbol{\Omega}^* = \mathbf{F}^{\top}\mathbf{R}\mathbf{W}^{\top}. \tag{38}$$

Since $\mathbf{F}^{\top}\mathbf{F}$ and $\mathbf{W}\mathbf{W}^{\top}$ are real symmetric matrices, we can spectrally decompose them to obtain:

$$\mathbf{F}^{\top}\mathbf{F} = \mathbf{V}\boldsymbol{\Lambda}^{\mathrm{F}}\mathbf{V}^{\top}, \quad \mathbf{W}\mathbf{W}^{\top} = \mathbf{U}\boldsymbol{\Lambda}^{\mathrm{W}}\mathbf{U}^{\top}, \tag{39}$$

where $\mathbf{V}^{\top}\mathbf{V} = \mathbf{I}$ and $\mathbf{U}^{\top}\mathbf{U} = \mathbf{I}$, and $\mathbf{I}$ denotes the identity matrix. By denoting $\mathbf{S} = \mathbf{V}^{\top}\boldsymbol{\Omega}^*\mathbf{U}$ and substituting (39) into (38), we obtain:

$$\begin{aligned}
&\mathbf{F}^{\top}\mathbf{F}\boldsymbol{\Omega}^*\mathbf{W}\mathbf{W}^{\top} + \gamma\boldsymbol{\Omega}^* = \mathbf{F}^{\top}\mathbf{R}\mathbf{W}^{\top} \\
&\implies \mathbf{V}\boldsymbol{\Lambda}^{\mathrm{F}}\mathbf{V}^{\top}\boldsymbol{\Omega}^*\mathbf{U}\boldsymbol{\Lambda}^{\mathrm{W}}\mathbf{U}^{\top} + \gamma\boldsymbol{\Omega}^* = \mathbf{F}^{\top}\mathbf{R}\mathbf{W}^{\top} \\
&\implies \mathbf{V}^{\top}\mathbf{V}\boldsymbol{\Lambda}^{\mathrm{F}}\mathbf{S}\boldsymbol{\Lambda}^{\mathrm{W}}\mathbf{U}^{\top}\mathbf{U} + \gamma\mathbf{V}^{\top}\boldsymbol{\Omega}^*\mathbf{U} = \mathbf{V}^{\top}\mathbf{F}^{\top}\mathbf{R}\mathbf{W}^{\top}\mathbf{U} \\
&\implies \boldsymbol{\Lambda}^{\mathrm{F}}\mathbf{S}\boldsymbol{\Lambda}^{\mathrm{W}} + \gamma\mathbf{S} = \mathbf{V}^{\top}\mathbf{F}^{\top}\mathbf{R}\mathbf{W}^{\top}\mathbf{U}.
\end{aligned} \tag{40}$$

Since $\boldsymbol{\Lambda}^{\mathrm{F}}$ and $\boldsymbol{\Lambda}^{\mathrm{W}}$ are both diagonal matrices, we proceed to expand them element-wise. Specifically, let $\lambda_i^{\mathrm{F}}$ and $\lambda_j^{\mathrm{W}}$ denote the $i$-th and $j$-th diagonal elements of $\boldsymbol{\Lambda}^{\mathrm{F}}$ and $\boldsymbol{\Lambda}^{\mathrm{W}}$, respectively. Concurrently, we denote the $(i, j)$-th entry of $\mathbf{S}$ and $\mathbf{V}^{\top}\mathbf{F}^{\top}\mathbf{R}\mathbf{W}^{\top}\mathbf{U}$ by $S_{i,j}$ and $(\mathbf{V}^{\top}\mathbf{F}^{\top}\mathbf{R}\mathbf{W}^{\top}\mathbf{U})_{i,j}$, respectively. According to (40), we obtain:

$$\lambda_i^{\mathrm{F}}S_{i,j}\lambda_j^{\mathrm{W}} + \gamma S_{i,j} = (\mathbf{V}^{\top}\mathbf{F}^{\top}\mathbf{R}\mathbf{W}^{\top}\mathbf{U})_{i,j}. \tag{41}$$

Meanwhile, we can further obtain:

$$S_{i,j} = \frac{(\mathbf{V}^{\top}\mathbf{F}^{\top}\mathbf{R}\mathbf{W}^{\top}\mathbf{U})_{i,j}}{\lambda_i^{\mathrm{F}}\lambda_j^{\mathrm{W}} + \gamma}. \tag{42}$$

Based on (42), we can extend $S_{i,j}$ to the entire matrix $\mathbf{S}$, yielding:

$$\mathbf{S} = (\mathbf{V}^{\top}\mathbf{F}^{\top}\mathbf{R}\mathbf{W}^{\top}\mathbf{U}) \oslash (\gamma\mathbf{1} + \mathrm{diag}(\boldsymbol{\Lambda}^{\mathrm{F}}) \otimes \mathrm{diag}(\boldsymbol{\Lambda}^{\mathrm{W}})), \tag{43}$$

where $\mathbf{1}$ is the all-ones matrix, while $\oslash$ and $\otimes$ represent element-wise division and outer product. Finally, leveraging the orthogonality of $\mathbf{V}$ and $\mathbf{U}$, we can obtain $\boldsymbol{\Omega}^* = \mathbf{V}\mathbf{S}\mathbf{U}^{\top}$. Substituting (43) into this expression, we can obtain:

$$\boldsymbol{\Omega}^* = \mathbf{V}(\mathbf{V}^{\top}\mathbf{F}^{\top}\mathbf{R}\mathbf{W}^{\top}\mathbf{U}) \oslash (\gamma\mathbf{1} + \mathrm{diag}(\boldsymbol{\Lambda}^{\mathrm{F}}) \otimes \mathrm{diag}(\boldsymbol{\Lambda}^{\mathrm{W}}))\mathbf{U}^{\top}. \tag{44}$$

■

### B.2 OUR DEEPAFL'S INVARIANCE TO DATA HETEROGENEITY

Here, we demonstrate that the deep residual analytic network derived from our DeepAFL exhibits ideal invariance to data heterogeneity, remaining invariant to any data partition across clients with different heterogeneity. Specifically, as established in Lemma 1 and Lemma 2, we have derived the centralized analytical solutions for both the global classifier and the transformation matrix. Here, we additionally prove that the solution obtained from our DeepAFL is identical to these centralized analytical solutions. The detailed proofs are as follows.

**Theorem 1 (Invariance to Data Heterogeneity):** In the FL scenario, let $\mathcal{D}$ be the full dataset, and $\mathcal{P} = \{\mathcal{D}^1, \mathcal{D}^2, \ldots, \mathcal{D}^K\}$ be any heterogeneous partition of $\mathcal{D}$ among $K$ clients, where $\mathcal{D}_i \cap \mathcal{D}_j = \emptyset$ for $i \neq j$ and $\bigcup_{i=1}^K \mathcal{D}_i = \mathcal{D}$. Given fixed seeds for the random projection matrices $\mathbf{A}$ and $\{\mathbf{B}_t\}_{t=0}^{T-1}$, the global model weights $\{\mathbf{W}_t\}_{t=0}^T$ and $\{\mathbf{\Omega}_t\}_{t=1}^T$ derived from DeepAFL are invariant to any data partition $\mathcal{P}$ with different heterogeneity, being identical to the centralized analytical solutions on $\mathcal{D}$.

*Proof.* As established in Lemma 1 and Lemma 2, for the complete dataset $\mathcal{D} = \{\mathbf{X}^{1:K}, \mathbf{Y}^{1:K}\}$, the centralized analytical solutions for $\{\mathbf{W}_t\}_{t=0}^T$ and $\{\mathbf{\Omega}_t\}_{t=1}^T$ can be expressed as:

$$\mathbf{W}_t = [(\mathbf{\Phi}_t^{1:K})^\top \mathbf{\Phi}_t^{1:K} + \lambda \mathbf{I}]^{-1}(\mathbf{\Phi}_t^{1:K})^\top \mathbf{Y}, \tag{45}$$

$$\mathbf{\Omega}_{t+1} = \mathbf{V}_t[(\mathbf{V}_t^\top(\mathbf{F}_t^{1:K})^\top \mathbf{R}_t^{1:K}\mathbf{W}_t^\top \mathbf{U}_t) \oslash (\gamma\mathbf{1} + \mathrm{diag}(\mathbf{\Lambda}_t^{\mathrm{F}}) \otimes \mathrm{diag}(\mathbf{\Lambda}_t^{\mathrm{W}}))]\mathbf{U}_t^\top, \tag{46}$$

where $(\mathbf{F}_t^{1:K})^\top \mathbf{F}_t^{1:K} = \mathbf{V}_t \mathbf{\Lambda}_t^{\mathrm{F}} \mathbf{V}_t^\top$ and $\mathbf{W}_t \mathbf{W}_t^\top = \mathbf{U}_t \mathbf{\Lambda}_t^{\mathrm{W}} \mathbf{U}_t^\top$ are spectral decompositions, while $\oslash$ and $\otimes$ represent element-wise division and outer product, respectively.

In fact, any heterogeneous partition $\mathcal{P}$ can be viewed as a permutation of the sample ordering within the complete dataset $\mathcal{D}$. Accordingly, for any heterogeneous partition $\mathcal{P} = \{\mathcal{D}^1, \mathcal{D}^2, \ldots, \mathcal{D}^K\}$, we define the corresponding permuted complete dataset as $\tilde{\mathcal{D}} = \{\tilde{\mathbf{X}}^{1:K}, \tilde{\mathbf{Y}}^{1:K}\}$, which satisfies:

$$\tilde{\mathbf{X}}^{1:K} = \pi \mathbf{X}^{1:K} \quad \tilde{\mathbf{Y}}^{1:K} = \pi \mathbf{Y}^{1:K}, \tag{47}$$

where $\pi$ is the corresponding permutation matrix and satisfies $\pi^\top \pi = \mathbf{I}$. Next, we employ mathematical induction to rigorously demonstrate that the global model weights $\{\tilde{\mathbf{W}}_t\}_{t=0}^T$ and $\{\tilde{\mathbf{\Omega}}_t\}_{t=1}^T$ obtained by our DeepAFL under any heterogeneous partition $\mathcal{P}$ are identical to their centralized analytical solutions, as presented in (45) and (46).

For the base case, we demonstrate that the initial $\tilde{\mathbf{W}}_0$ and $\tilde{\mathbf{\Omega}}_1$ obtained via our DeepAFL precisely satisfy (45) and (46). It can be readily deduced that the complete zero-layer feature matrix similarly satisfies $\tilde{\mathbf{\Phi}}_0^{1:K} = \pi \mathbf{\Phi}_0^{1:K}$. According to (13), the expression for $\tilde{\mathbf{W}}_0$ can be given by:

$$\begin{aligned}
\tilde{\mathbf{W}}_0 &= (\tilde{\mathbf{G}}_0^{1:K} + \lambda \mathbf{I})^{-1} \tilde{\mathbf{H}}_0^{1:K} \\
&= [(\tilde{\mathbf{\Phi}}_0^{1:K})^\top \tilde{\mathbf{\Phi}}_0^{1:K} + \lambda \mathbf{I}]^{-1}(\tilde{\mathbf{\Phi}}_0^{1:K})^\top \tilde{\mathbf{Y}}^{1:K} \\
&= [(\mathbf{\Phi}_0^{1:K})^\top \pi^\top \pi \mathbf{\Phi}_0^{1:K} + \lambda \mathbf{I}]^{-1}(\tilde{\mathbf{\Phi}}_0^{1:K})^\top \tilde{\mathbf{Y}}^{1:K} \\
&= [(\mathbf{\Phi}_0^{1:K})^\top \mathbf{\Phi}_0^{1:K} + \lambda \mathbf{I}]^{-1}(\tilde{\mathbf{\Phi}}_0^{1:K})^\top \tilde{\mathbf{Y}}^{1:K}.
\end{aligned} \tag{48}$$

It is evident that $\tilde{\mathbf{W}}_0$ coincides precisely with the corresponding centralized analytical solution (45). Consequently, the residual corresponding to each sample matches that obtained through centralized training, differing only in sample order. Thus, the complete residual matrix satisfies $\tilde{\mathbf{R}}_0^{1:K} = \pi \mathbf{R}_0^{1:K}$. Similarly, the complete hidden random features also satisfy $\tilde{\mathbf{F}}_0^{1:K} = \pi \mathbf{F}_0^{1:K}$. Drawing from (17) and (18), the $\tilde{\mathbf{\Omega}}_1$ is given by:

$$\begin{aligned}
\tilde{\mathbf{\Omega}}_1 &= \mathbf{V}_0[(\mathbf{V}_0^\top \tilde{\mathbf{\Upsilon}}_0^{1:K}\mathbf{W}_0^\top \mathbf{U}_0) \oslash (\gamma\mathbf{1} + \mathrm{diag}(\mathbf{\Lambda}_0^{\mathrm{F}}) \otimes \mathrm{diag}(\mathbf{\Lambda}_0^{\mathrm{W}}))]\mathbf{U}_0^\top \\
&= \mathbf{V}_0[(\mathbf{V}_0^\top(\tilde{\mathbf{F}}_0^{1:K})^\top \tilde{\mathbf{R}}_0^{1:K}\mathbf{W}_0^\top \mathbf{U}_0) \oslash (\gamma\mathbf{1} + \mathrm{diag}(\mathbf{\Lambda}_0^{\mathrm{F}}) \otimes \mathrm{diag}(\mathbf{\Lambda}_0^{\mathrm{W}}))]\mathbf{U}_0^\top \\
&= \mathbf{V}_0[(\mathbf{V}_0^\top(\mathbf{F}_0^{1:K})^\top \pi^\top \pi \mathbf{R}_0^{1:K}\mathbf{W}_0^\top \mathbf{U}_0) \oslash (\gamma\mathbf{1} + \mathrm{diag}(\mathbf{\Lambda}_0^{\mathrm{F}}) \otimes \mathrm{diag}(\mathbf{\Lambda}_0^{\mathrm{W}}))]\mathbf{U}_0^\top \\
&= \mathbf{V}_0[(\mathbf{V}_0^\top(\mathbf{F}_0^{1:K})^\top \mathbf{R}_0^{1:K}\mathbf{W}_0^\top \mathbf{U}_0) \oslash (\gamma\mathbf{1} + \mathrm{diag}(\mathbf{\Lambda}_0^{\mathrm{F}}) \otimes \mathrm{diag}(\mathbf{\Lambda}_0^{\mathrm{W}}))]\mathbf{U}_0^\top,
\end{aligned} \tag{49}$$

where the spectral decomposition results are also identical to those in the preceding formula (46): $(\tilde{\mathbf{F}}_0^{1:K})^\top \tilde{\mathbf{F}}_0^{1:K} = (\mathbf{F}_0^{1:K})^\top \pi^\top \pi \mathbf{F}_0^{1:K} = (\mathbf{F}_0^{1:K})^\top \mathbf{F}_0^{1:K} = \mathbf{V}_0 \mathbf{\Lambda}_0^{\mathrm{F}} \mathbf{V}_0^\top$ and $\tilde{\mathbf{W}}_0 \tilde{\mathbf{W}}_0^\top = \mathbf{W}_0 \mathbf{W}_0^\top = \mathbf{U}_0 \mathbf{\Lambda}_0^{\mathrm{W}} \mathbf{U}_0^\top$. It can be observed that the initial $\tilde{\mathbf{\Omega}}_1$ derived from our DeepAFL also precisely matches the centralized analytical solution. Thus, the base case holds for the theorem.

For the inductive step, assume the preceding $(t-1)$ layers $\{\tilde{\mathbf{W}}_i\}_{i=0}^{t-1}$ and $\{\tilde{\boldsymbol{\Omega}}_i\}_{i=1}^{t}$ satisfy (45) and (46). We further demonstrate that, based on these assumptions, the subsequent layer $\tilde{\mathbf{W}}_t$ and $\tilde{\boldsymbol{\Omega}}_{t+1}$ obtained via our DeepAFL also satisfy (45) and (46). Since the previous $(t-1)$ layers' global model weights exactly match the centralized analytical solution, the complete feature matrix $\tilde{\boldsymbol{\Phi}}_t^{1:K}$ and the complete hidden random features $\tilde{\mathbf{F}}_t^{1:K}$ satisfy $\tilde{\boldsymbol{\Phi}}_t^{1:K} = \pi \boldsymbol{\Phi}_t^{1:K}$ and $\tilde{\mathbf{F}}_t^{1:K} = \pi \mathbf{F}_t^{1:K}$. According to (13), the expression for $\tilde{\mathbf{W}}_t$ is thus obtained as follows:

$$
\begin{aligned}
\tilde{\mathbf{W}}_t &= (\tilde{\mathbf{G}}_t^{1:K} + \lambda \mathbf{I})^{-1} \tilde{\mathbf{H}}_t^{1:K} \\
&= [(\tilde{\boldsymbol{\Phi}}_t^{1:K})^\top \tilde{\boldsymbol{\Phi}}_t^{1:K} + \lambda \mathbf{I}]^{-1} (\tilde{\boldsymbol{\Phi}}_t^{1:K})^\top \tilde{\mathbf{Y}}^{1:K} \\
&= [(\boldsymbol{\Phi}_t^{1:K})^\top \pi^\top \pi \boldsymbol{\Phi}_t^{1:K} + \lambda \mathbf{I}]^{-1} (\tilde{\boldsymbol{\Phi}}_t^{1:K})^\top \tilde{\mathbf{Y}}^{1:K} \\
&= [(\boldsymbol{\Phi}_t^{1:K})^\top \boldsymbol{\Phi}_t^{1:K} + \lambda \mathbf{I}]^{-1} (\tilde{\boldsymbol{\Phi}}_t^{1:K})^\top \tilde{\mathbf{Y}}^{1:K}.
\end{aligned}
\tag{50}
$$

It is evident that the global classifier $\tilde{\mathbf{W}}_t$ satisfy (45), and thus the complete residual matrix satisfies $\tilde{\mathbf{R}}_t^{1:K} = \pi \mathbf{R}_t^{1:K}$. Subsequently, analogously to (49), we derive the analytical expression for $\tilde{\boldsymbol{\Omega}}_{t+1}$ yielded by our DeepAFL as follows:

$$
\begin{aligned}
\tilde{\boldsymbol{\Omega}}_{t+1} &= \mathbf{V}_t[(\mathbf{V}_t^\top \tilde{\boldsymbol{\Upsilon}}_t^{1:K} \mathbf{W}_t^\top \mathbf{U}_t) \oslash (\gamma \mathbf{1} + \mathrm{diag}(\boldsymbol{\Lambda}_t^\mathrm{F}) \otimes \mathrm{diag}(\boldsymbol{\Lambda}_t^\mathrm{W}))] \mathbf{U}_t^\top \\
&= \mathbf{V}_t[(\mathbf{V}_t^\top (\tilde{\mathbf{F}}_t^{1:K})^\top \tilde{\mathbf{R}}_t^{1:K} \mathbf{W}_t^\top \mathbf{U}_t) \oslash (\gamma \mathbf{1} + \mathrm{diag}(\boldsymbol{\Lambda}_t^\mathrm{F}) \otimes \mathrm{diag}(\boldsymbol{\Lambda}_t^\mathrm{W}))] \mathbf{U}_t^\top \\
&= \mathbf{V}_t[(\mathbf{V}_t^\top (\mathbf{F}_t^{1:K})^\top \pi^\top \pi \mathbf{R}_t^{1:K} \mathbf{W}_t^\top \mathbf{U}_t) \oslash (\gamma \mathbf{1} + \mathrm{diag}(\boldsymbol{\Lambda}_t^\mathrm{F}) \otimes \mathrm{diag}(\boldsymbol{\Lambda}_t^\mathrm{W}))] \mathbf{U}_t^\top \\
&= \mathbf{V}_t[(\mathbf{V}_t^\top (\mathbf{F}_t^{1:K})^\top \mathbf{R}_t^{1:K} \mathbf{W}_t^\top \mathbf{U}_t) \oslash (\gamma \mathbf{1} + \mathrm{diag}(\boldsymbol{\Lambda}_t^\mathrm{F}) \otimes \mathrm{diag}(\boldsymbol{\Lambda}_t^\mathrm{W}))] \mathbf{U}_t^\top,
\end{aligned}
\tag{51}
$$

where the spectral decomposition results are also identical to those in the preceding formula (46): $(\tilde{\mathbf{F}}_t^{1:K})^\top \tilde{\mathbf{F}}_t^{1:K} = (\mathbf{F}_t^{1:K})^\top \pi^\top \pi \mathbf{F}_t^{1:K} = (\mathbf{F}_t^{1:K})^\top \mathbf{F}_t^{1:K} = \mathbf{V}_t \boldsymbol{\Lambda}_t^\mathrm{F} \mathbf{V}_t^\top$ and $\tilde{\mathbf{W}}_t \tilde{\mathbf{W}}_t^\top = \mathbf{W}_t \mathbf{W}_t^\top = \mathbf{U}_t \boldsymbol{\Lambda}_t^\mathrm{W} \mathbf{U}_t^\top$. As a result, the transformation matrix $\tilde{\boldsymbol{\Omega}}_{t+1}$ derived from our DeepAFL also precisely matches the centralized analytical solution. Thus, the inductive step also holds for the theorem.

In summary, based on the aforementioned base case and inductive step, we establish via mathematical induction that the global model weights $\{\mathbf{W}_t\}_{t=0}^T$ and $\{\boldsymbol{\Omega}_t\}_{t=1}^T$ derived from our DeepAFL are invariant to any data partition $\mathcal{P}$ with different heterogeneity. In fact, the results obtained from our DeepAFL are identical to the centralized analytical solutions (45) and (46).

$\blacksquare$

### B.3 Our DeepAFL's Capability of Representation Learning

Here, we further demonstrate that the empirical risk monotonically non-increases with increasing layer depth and is guaranteed to converge to a limit, thus validating DeepAFL's representation learning capability. To begin, we analyze the general scenario without considering the regularization term in Theorem 2, proving that under this scenario, the empirical risk derived from our DeepAFL monotonically non-increases with increasing layer depth and converges to a limit.

**Theorem 2 (Capability of Representation Learning)**: Let the general empirical risk $\mathcal{H}(\boldsymbol{\Phi}, \mathbf{W})$ (i.e., the loss function), for a given feature representation $\boldsymbol{\Phi}$ and a classifier $\mathbf{W}$ be defined as:

$$
\mathcal{H}(\boldsymbol{\Phi}, \mathbf{W}) = \|\mathbf{Y} - \boldsymbol{\Phi}\mathbf{W}\|_\mathrm{F}^2,
\tag{52}
$$

where $\mathbf{Y}$ denotes the ground truth labels. Our DeepAFL yields a sequence of feature-classifier pairs $(\boldsymbol{\Phi}_t, \mathbf{W}_t)$ at each layer $t \in [0, T]$. When setting regularization parameters $\gamma$ and $\lambda$ to 0, the sequence of the empirical risks $\{\mathcal{H}(\boldsymbol{\Phi}_t, \mathbf{W}_t)\}_{t=0}^T$ in our DeepAFL keeps monotonically non-increasing, i.e.,

$$
\mathcal{H}(\boldsymbol{\Phi}_t, \mathbf{W}_t) \geq \mathcal{H}(\boldsymbol{\Phi}_{t+1}, \mathbf{W}_{t+1}), \forall t \in [0, T].
\tag{53}
$$

Furthermore, as the number of layers $T$ within our DeepAFL increases (i.e., $T \to \infty$), the sequence of empirical risks is guaranteed to converge to a limit $\mathcal{H}^* \leq \mathcal{H}(\boldsymbol{\Phi}_0, \mathbf{W}_0)$.

*Proof.* According to Algorithm 1, the construction of each feature-classifier pair $(\boldsymbol{\Phi}_{t+1}, \mathbf{W}_{t+1})$ proceeds in two alternating steps: (1) fixing the classifier $\mathbf{W}_t$ to construct the feature representation $\boldsymbol{\Phi}_{t+1}$; (2) fixing the updated feature representation $\boldsymbol{\Phi}_{t+1}$ to optimize the classifier $\mathbf{W}_{t+1}$. We will prove that the empirical risks in each of these substeps are non-increasing, thereby demonstrating that the sequence of empirical risks $\{\mathcal{H}(\boldsymbol{\Phi}_t, \mathbf{W}_t)\}_{t=0}^T$ is monotonically non-increasing.

First, let's focus on the substep of constructing the feature representation $\mathbf{\Phi}_{t+1}$ while keeping the global classifier $\mathbf{W}_t$ fixed. At this stage, the intermediate empirical risk for the feature-classifier pair $(\mathbf{\Phi}_{t+1}, \mathbf{W}_t)$ can be expressed as:

$$\mathcal{H}(\mathbf{\Phi}_{t+1}, \mathbf{W}_t) = \|\mathbf{Y} - \mathbf{\Phi}_{t+1}\mathbf{W}_t\|_{\mathrm{F}}^2, \tag{54}$$

where $\mathbf{\Phi}_{t+1}$ obtained through our DeepAFL satisfies $\mathbf{\Phi}_{t+1} = \mathbf{\Phi}_t + \mathbf{F}_t\mathbf{\Omega}_{t+1}$. Therefore, the aforementioned empirical risk in our DeepAFL can be further expressed as:

$$\mathcal{H}(\mathbf{\Phi}_{t+1}, \mathbf{W}_t) = \|\mathbf{Y} - (\mathbf{\Phi}_t + \mathbf{F}_t\mathbf{\Omega}_{t+1})\mathbf{W}_t\|_{\mathrm{F}}^2. \tag{55}$$

According to Lemma 2, the $\mathbf{\Omega}_{t+1}$ derived through our DeepAFL represents the optimal solution that minimizes the aforementioned empirical risk, thereby satisfying:

$$\mathcal{H}(\mathbf{\Phi}_{t+1}, \mathbf{W}_t) = \|\mathbf{Y} - (\mathbf{\Phi}_t + \mathbf{F}_t\mathbf{\Omega}_{t+1})\mathbf{W}_t\|_{\mathrm{F}}^2 \leq \|\mathbf{Y} - (\mathbf{\Phi}_t + \mathbf{F}_t\mathbf{\Omega})\mathbf{W}_t\|_{\mathrm{F}}^2, \forall \mathbf{\Omega}, \tag{56}$$

where, as a special case when $\mathbf{\Omega} = \mathbf{0}$, it holds that:

$$\|\mathbf{Y} - (\mathbf{\Phi}_t + \mathbf{F}_t\mathbf{0})\mathbf{W}_t\|_{\mathrm{F}}^2 = \|\mathbf{Y} - \mathbf{\Phi}_t\mathbf{W}_t\|_{\mathrm{F}}^2 = \mathcal{H}(\mathbf{\Phi}_t, \mathbf{W}_t). \tag{57}$$

Thus, it follows that:

$$\mathcal{H}(\mathbf{\Phi}_{t+1}, \mathbf{W}_t) \leq \mathcal{H}(\mathbf{\Phi}_t, \mathbf{W}_t). \tag{58}$$

Second, we further analyze the substep of constructing the global classifier $\mathbf{W}_{t+1}$ while keeping the feature representation $\mathbf{\Phi}_{t+1}$ fixed. When the regularization parameter $\lambda$ is set to zero, the $\mathbf{W}_{t+1}$ derived through our DeepAFL is obtained by minimizing the following empirical risk:

$$\mathbf{W}_{t+1} = \arg\min_{\mathbf{W}} \|\mathbf{Y} - \mathbf{\Phi}_{t+1}\mathbf{W}\|_{\mathrm{F}}^2 = \arg\min_{\mathbf{W}} \mathcal{H}(\mathbf{\Phi}_{t+1}, \mathbf{W}). \tag{59}$$

According to Lemma 1, the $\mathbf{W}_{t+1}$ derived through our DeepAFL represents the optimal solution that minimizes the aforementioned empirical risk. We can obtain:

$$\mathcal{H}(\mathbf{\Phi}_{t+1}, \mathbf{W}_{t+1}) = \|\mathbf{Y} - \mathbf{\Phi}_{t+1}\mathbf{W}_{t+1}\|_{\mathrm{F}}^2 \leq \|\mathbf{Y} - \mathbf{\Phi}_{t+1}\mathbf{W}\|_{\mathrm{F}}^2, \forall \mathbf{W}, \tag{60}$$

where, as a special case when $\mathbf{W} = \mathbf{W}_t$, it holds that:

$$\|\mathbf{Y} - \mathbf{\Phi}_{t+1}\mathbf{W}_t\|_{\mathrm{F}}^2 = \mathcal{H}(\mathbf{\Phi}_{t+1}, \mathbf{W}_t). \tag{61}$$

Thus, it follows that:

$$\mathcal{H}(\mathbf{\Phi}_{t+1}, \mathbf{W}_{t+1}) \leq \mathcal{H}(\mathbf{\Phi}_{t+1}, \mathbf{W}_t). \tag{62}$$

In summary, based on (58) and (62), it holds for each feature-classifier pair $(\mathbf{\Phi}_{t+1}, \mathbf{W}_{t+1})$ that:

$$\mathcal{H}(\mathbf{\Phi}_t, \mathbf{W}_t) \geq \mathcal{H}(\mathbf{\Phi}_{t+1}, \mathbf{W}_t) \geq \mathcal{H}(\mathbf{\Phi}_{t+1}, \mathbf{W}_{t+1}), \forall t \in [0, T). \tag{63}$$

Therefore, the sequence of empirical risks $\{\mathcal{H}(\mathbf{\Phi}_t, \mathbf{W}_t)\}_{t=0}^T$ is monotonically non-increasing. Given that $\mathcal{H}(\mathbf{\Phi}_t, \mathbf{W}_t) \geq 0$ and the sequence is monotonically non-increasing, by the Monotone Convergence Theorem, it necessarily converges to a limit $\mathcal{H}^* \geq 0$. Moreover, since the sequence begins at $\mathcal{H}(\mathbf{\Phi}_0, \mathbf{W}_0)$ and decreases at each layer, it holds that $\mathcal{H}^* \leq \mathcal{H}(\mathbf{\Phi}_0, \mathbf{W}_0)$. ∎

Considering our employment of regularization to enhance generalization performance and numerical stability, we further extend Theorem 2 to incorporate the regularized setting. To this end, we begin by analyzing the regularization terms in the regularized empirical risk, thereby defining it for the subsequent theorem, as detailed below:

(1) From the perspective of empirical risk, the empirical risk is a function of the global classifier $\mathbf{W}_t$ and the feature representation $\mathbf{\Phi}_t$. Accordingly, both should be subject to regularization. The feature representation $\mathbf{\Phi}_t$ is obtained through layer-by-layer updates and can essentially be expressed as:

$$\mathbf{\Phi}_t = \mathbf{\Phi}_{t-1} + (\mathbf{F}_{t-1}\mathbf{\Omega}_t) = \mathbf{\Phi}_{t-2} + (\mathbf{F}_{t-1}\mathbf{\Omega}_t + \mathbf{F}_{t-2}\mathbf{\Omega}_{t-1}) = \cdots = \mathbf{\Phi}_0 + \sum_{i=1}^t \mathbf{F}_{i-1}\mathbf{\Omega}_i, \tag{64}$$

where $\mathbf{F}_{i-1}$ is not trainable, providing stochasticity and nonlinearity to the features via fixed activation functions and random projections, whereas $\mathbf{\Omega}_i$ constitutes the trainable component for representation learning. Thus, regularizing all $\{\mathbf{\Omega}_i\}_{i=1}^t$ can be seen as a form of regularizing $\mathbf{\Phi}_t$. In addition, the global classifier $\mathbf{W}_t$ is itself trainable, and should be regularized directly.

(2) From the perspective of the model architecture, the global model derived from our DeepAFL after constructing the first $t$ layers encompasses the final global classifier $\mathbf{W}_t$ and all transformation matrices $\{\mathbf{\Omega}_i\}_{i=1}^t$. Therefore, regularization terms need to be introduced for both $\mathbf{W}_t$ and $\{\mathbf{\Omega}_i\}_{i=1}^t$.

Building upon the foregoing analyses of the regularized empirical risk, we further provide its explicit definition in (65). Moreover, in Theorem 3, we theoretically prove that the empirical risk of the resulting model is monotonically non-increasing as the layer depth increases.

**Theorem 3 (Regularized Capability of Representation Learning)**: In our DeepAFL, we define the regularized empirical risk $\mathcal{G}(t)$ (i.e., the regularized loss function) for each layer $t \in [0, T]$ as:

$$\mathcal{G}(t) = \underbrace{\|\mathbf{Y} - \mathbf{\Phi}_t \mathbf{W}_t\|_{\mathrm{F}}^2}_{(1)} + \lambda \underbrace{\|\mathbf{W}_t\|_{\mathrm{F}}^2}_{(2)} + \sum\nolimits_{i=1}^{t} \gamma \underbrace{\|\mathbf{\Omega}_i\|_{\mathrm{F}}^2}_{(3)}, \tag{65}$$

where $\gamma > 0$ and $\lambda > 0$ are non-negative regularization parameters. Moreover, $\mathbf{\Phi}_t$, $\mathbf{W}_t$, and $\mathbf{\Omega}_t$ are the feature representation, the analytic classifier, and the learned transformation in our DeepAFL at the layer $t$. Here, the term (1) is the original empirical risk $\mathcal{H}(\mathbf{\Phi}_t, \mathbf{W}_t)$, while the terms (2) and (3) are the regularization in our DeepAFL. The sequence of the regularized empirical risks $\{\mathcal{G}(t)\}_{t=0}^T$ in our DeepAFL remains monotonically non-increasing, i.e.,

$$\mathcal{G}(t) \geq \mathcal{G}(t+1), \forall t \in [0, T]. \tag{66}$$

Furthermore, as the number of layers $T$ within our DeepAFL increases (i.e., $T \to \infty$), the sequence of the regularized empirical risks is guaranteed to converge to a limit $\mathcal{G}^* \leq \mathcal{G}(0)$.

*Proof.* Analogous to Theorem 2, we sequentially analyze the construction of the feature representation $\mathbf{\Phi}_{t+1}$ (i.e., deriving the transformation matrix $\mathbf{\Omega}_{t+1}$) and the global classifier $\mathbf{W}_{t+1}$ at each layer, thereby demonstrating that the regularized empirical risks $\{\mathcal{G}(t)\}_{t=0}^T$ in our DeepAFL remain monotonically non-increasing.

First, let's focus on the substep of constructing the feature representation $\mathbf{\Phi}_{t+1}$ and the learned transformation $\mathbf{\Omega}_{t+1}$ while fixing the global classifier $\mathbf{W}_t$ and other transformations $\{\mathbf{\Omega}_i\}_{i=1}^t$. For notational convenience, we denote the intermediate regularized empirical risk between layers $t$ and $t+1$ (i.e., the risk associated with $\mathbf{\Phi}_{t+1}$, $\mathbf{W}_t$, and $\{\mathbf{\Omega}_i\}_{i=1}^{t+1}$) as $\hat{\mathcal{G}}(t)$, which can be expressed as:

$$\hat{\mathcal{G}}(t) = \|\mathbf{Y} - \mathbf{\Phi}_{t+1} \mathbf{W}_t\|_{\mathrm{F}}^2 + \lambda \|\mathbf{W}_t\|_{\mathrm{F}}^2 + \sum\nolimits_{i=1}^{t+1} \gamma \|\mathbf{\Omega}_i\|_{\mathrm{F}}^2, \tag{67}$$

where $\mathbf{\Phi}_{t+1}$ obtained through our DeepAFL satisfies $\mathbf{\Phi}_{t+1} = \mathbf{\Phi}_t + \mathbf{F}_t \mathbf{\Omega}_{t+1}$. Therefore, the aforementioned regularized empirical risk can be further expressed as:

$$\hat{\mathcal{G}}(t) = \|\mathbf{Y} - (\mathbf{\Phi}_t + \mathbf{F}_t \mathbf{\Omega}_{t+1}) \mathbf{W}_t\|_{\mathrm{F}}^2 + \lambda \|\mathbf{W}_t\|_{\mathrm{F}}^2 + \sum\nolimits_{i=1}^{t+1} \gamma \|\mathbf{\Omega}_i\|_{\mathrm{F}}^2. \tag{68}$$

According to Lemma 2, the $\mathbf{\Omega}_{t+1}$ derived through our DeepAFL represents the optimal solution that minimizes the regularized empirical risk:

$$\mathbf{\Omega}_{t+1} = \arg\min_{\mathbf{\Omega}} \|\mathbf{Y} - (\mathbf{\Phi}_t + \mathbf{F}_t \mathbf{\Omega}) \mathbf{W}_t\|_{\mathrm{F}}^2 + \gamma \|\mathbf{\Omega}\|_{\mathrm{F}}^2. \tag{69}$$

Therefore, we can obtain:

$$\|\mathbf{Y} - (\mathbf{\Phi}_t + \mathbf{F}_t \mathbf{\Omega}_{t+1}) \mathbf{W}_t\|_{\mathrm{F}}^2 + \gamma \|\mathbf{\Omega}_{t+1}\|_{\mathrm{F}}^2 \leq \|\mathbf{Y} - (\mathbf{\Phi}_t + \mathbf{F}_t \mathbf{\Omega}) \mathbf{W}_t\|_{\mathrm{F}}^2 + \gamma \|\mathbf{\Omega}\|_{\mathrm{F}}^2, \ \forall \mathbf{\Omega}, \tag{70}$$

where $\mathbf{\Omega} = \mathbf{0}$ serves as a special case, from which we further derive:

$$\|\mathbf{Y} - (\mathbf{\Phi}_t + \mathbf{F}_t \mathbf{\Omega}_{t+1}) \mathbf{W}_t\|_{\mathrm{F}}^2 + \gamma \|\mathbf{\Omega}_{t+1}\|_{\mathrm{F}}^2 \leq \|\mathbf{Y} - \mathbf{\Phi}_t \mathbf{W}_t\|_{\mathrm{F}}^2. \tag{71}$$

By adding $\lambda \|\mathbf{W}_t\|_{\mathrm{F}}^2$ and $\sum_{i=1}^t \gamma \|\mathbf{\Omega}_i\|_{\mathrm{F}}^2$ to both sides of inequality (71), we obtain:

$$\begin{aligned} \hat{\mathcal{G}}(t) &= \|\mathbf{Y} - (\mathbf{\Phi}_t + \mathbf{F}_t \mathbf{\Omega}_{t+1}) \mathbf{W}_t\|_{\mathrm{F}}^2 + \lambda \|\mathbf{W}_t\|_{\mathrm{F}}^2 + \sum\nolimits_{i=1}^{t+1} \gamma \|\mathbf{\Omega}_i\|_{\mathrm{F}}^2 \\ &\leq \|\mathbf{Y} - \mathbf{\Phi}_t \mathbf{W}_t\|_{\mathrm{F}}^2 + \lambda \|\mathbf{W}_t\|_{\mathrm{F}}^2 + \sum\nolimits_{i=1}^{t} \gamma \|\mathbf{\Omega}_i\|_{\mathrm{F}}^2 = \mathcal{G}(t). \end{aligned} \tag{72}$$

Second, we further analyze the substep of constructing the global classifier $\mathbf{W}_{t+1}$ while keeping the feature representation $\mathbf{\Phi}_{t+1}$ and transformation matrices $\{\mathbf{\Omega}_i\}_{i=1}^{t+1}$ fixed. According to Lemma 1, the $\mathbf{W}_{t+1}$ represents the optimal solution that minimizes the regularized empirical risk:

$$\mathbf{W}_{t+1} = \arg\min_{\mathbf{W}} \|\mathbf{Y} - \mathbf{\Phi}_{t+1} \mathbf{W}\|_{\mathrm{F}}^2 + \lambda \|\mathbf{W}\|_{\mathrm{F}}^2. \tag{73}$$

Therefore, we can obtain:

$$\|\mathbf{Y} - \mathbf{\Phi}_{t+1}\mathbf{W}_{t+1}\|_{\mathrm{F}}^2 + \lambda\|\mathbf{W}_{t+1}\|_{\mathrm{F}}^2 \leq \|\mathbf{Y} - \mathbf{\Phi}_{t+1}\mathbf{W}\|_{\mathrm{F}}^2 + \lambda\|\mathbf{W}\|_{\mathrm{F}}^2, \ \forall \mathbf{W}, \tag{74}$$

where $\mathbf{W} = \mathbf{W}_t$ serves as a special case, from which we further derive:

$$\|\mathbf{Y} - \mathbf{\Phi}_{t+1}\mathbf{W}_{t+1}\|_{\mathrm{F}}^2 + \lambda\|\mathbf{W}_{t+1}\|_{\mathrm{F}}^2 \leq \|\mathbf{Y} - \mathbf{\Phi}_{t+1}\mathbf{W}_t\|_{\mathrm{F}}^2 + \lambda\|\mathbf{W}_t\|_{\mathrm{F}}^2. \tag{75}$$

By adding $\sum_{i=1}^{t+1} \gamma\|\mathbf{\Omega}_i\|_{\mathrm{F}}^2$ to both sides of inequality (75), we obtain:

$$\begin{aligned}
\mathcal{G}(t+1) &= \|\mathbf{Y} - \mathbf{\Phi}_{t+1}\mathbf{W}_{t+1}\|_{\mathrm{F}}^2 + \lambda\|\mathbf{W}_{t+1}\|_{\mathrm{F}}^2 + \sum_{i=1}^{t+1} \gamma\|\mathbf{\Omega}_i\|_{\mathrm{F}}^2 \\
&\leq \|\mathbf{Y} - \mathbf{\Phi}_{t+1}\mathbf{W}_t\|_{\mathrm{F}}^2 + \lambda\|\mathbf{W}_t\|_{\mathrm{F}}^2 + \sum_{i=1}^{t+1} \gamma\|\mathbf{\Omega}_i\|_{\mathrm{F}}^2 = \hat{\mathcal{G}}(t).
\end{aligned} \tag{76}$$

In summary, based on (72) and (76), it holds for each layer that:

$$\mathcal{G}(t) \geq \hat{\mathcal{G}}(t) \geq \mathcal{G}(t+1), \ \forall t \in [0, T). \tag{77}$$

Therefore, the sequence of the regularized empirical risks $\{\mathcal{G}(t)\}_{t=0}^T$ in our DeepAFL remains monotonically non-increasing. Given that $\mathcal{G}(t) \geq 0$, by the well-known Monotone Convergence Theorem, it necessarily converges to a limit $\mathcal{G}^* \geq 0$. Moreover, since the sequence begins at $\mathcal{G}(0)$ and decreases at each layer, it holds that $\mathcal{G}^* \leq \mathcal{G}(0)$.

∎

## C  APPENDIX FOR PRIVACY ANALYSES

In this section, we provide detailed privacy analyses of our DeepAFL, providing multi-layered analyses to substantiate our DeepAFL's robust privacy preservation, as follows:

First, in our DeepAFL, all clients need only submit their locally computed *Auto-Correlation* and *Cross-Correlation Matrices*, and numerous studies (Tan et al., 2022b; He et al., 2025; Fan et al., 2025b) have shown the privacy advantages of uploading such information. Specifically, according to (15), the uploaded *Cross-Correlation Matrices* $\mathbf{\Pi}_t^k$ and $\mathbf{\Upsilon}_t^k$ are essentially prototype matrices, where each column corresponds to the unaveraged class and residual prototypes for the corresponding class. Consequently, the information uploaded within DeepAFL is akin to that of prototype-based FL methods (Tan et al., 2022b), which have provided a detailed introduction to the inherent privacy benefits of transmitting prototypes. Additionally, existing analytical learning-based FL methods (He et al., 2025; Fan et al., 2025b) also require the uploading of *Auto-Correlation* and *Cross-Correlation Matrices* similarly. Thus, our DeepAFL shares the same privacy advantages as these existing methods.

Second, since clients are not required to upload their local dataset sizes $N_k$, it is pretty hard to infer the clients' private data (i.e., $\mathbf{X}^k$, $\mathbf{Y}^k$, $\mathbf{\Phi}_t^k$, $\mathbf{F}_t^k$, and $\mathbf{R}_t^k$) from the information they upload. Specifically, in our DeepAFL, each client uploads only its local matrices $\mathbf{G}_t^k \in \mathbb{R}^{d_\Phi \times d_\Phi}$, $\mathbf{H}_t^k \in \mathbb{R}^{d_\Phi \times \mathrm{C}}$, $\mathbf{\Pi}_t^k \in \mathbb{R}^{d_\mathrm{F} \times d_\mathrm{F}}$, and $\mathbf{\Upsilon}_t^k \in \mathbb{R}^{d_\mathrm{F} \times \mathrm{C}}$. Evidently, clients need not upload their local dataset sizes $N_k$, which also cannot be deduced from the dimensions of the aforementioned matrices. Moreover, the dimensions of all private data are directly tied to $N_k$, i.e., $\mathbf{X}^k \in \mathbb{R}^{N_k \times d_\mathrm{H} \times d_\mathrm{W} \times d_\mathrm{C}}$, $\mathbf{Y}^k \in \mathbb{R}^{N_k \times \mathrm{C}}$, $\mathbf{\Phi}_t^k \in \mathbb{R}^{N_k \times d_\Phi}$, $\mathbf{F}_t^k \in \mathbb{R}^{N_k \times d_\mathrm{F}}$, and $\mathbf{R}_t^k \in \mathbb{R}^{N_k \times \mathrm{C}}$. Therefore, without knowledge of $N_k$, the dimensions of the aforementioned private data cannot be determined. This results in infinitely many possible instantiations, rendering it fundamentally impossible to infer the clients' private data.

Third, many existing privacy-preserving techniques can be directly integrated into our DeepAFL, enabling the server to operate without requiring access to each client's specific matrices. Specifically, in our DeepAFL, the server only needs to utilize the aggregated values of the clients' uploaded matrices for constructing the global classifier and residual block, without needing the individual client upload results. Therefore, this can be regarded as a form of *Secure Multi-Party Computation*. Existing *Secure Aggregation Protocols* (Bonawitz et al., 2016; 2017; So et al., 2023) can be employed to aggregate clients' uploaded matrices without accessing their specific matrices. Additionally, techniques such as *Homomorphic Encryption* (Hu & Li, 2025) and *Differential Privacy* (Hu et al., 2024) can also be integrated into our DeepAFL to further preserve client privacy.

## D    APPENDIX FOR EFFICIENCY ANALYSES

In this section, we provide detailed efficiency analyses of our DeepAFL, focusing on the computation and communication complexities on both the client and server sides. Overall, for constructing all $T$ layers, the total computation and communication complexities of each client are $\mathcal{O}(TN_k(d_\Phi^2 + d_\Phi d_F + d_F^2))$ and $\mathcal{O}(T(d_\Phi^2 + d_F^2))$, while those for the server are $\mathcal{O}(T(d_\Phi^3 + d_F^3 + d_\Phi^2 d_F + d_\Phi d_F^2))$ and $\mathcal{O}(TKd_\Phi d_F)$ respectively. Notably, although the complexities of our DeepAFL are approximately $T$ times greater than those of AFL to construct a deep network for superior performance, they remain far lower than those of gradient-based methods. The detailed analyses are as follows:

**Analysis of Computation Complexity**: Here, we thoroughly analyze the computation complexity within our DeepAFL. Notably, each layer's construction within our DeepAFL merely requires single-round lightweight analytic computation by the clients and server, obviating the need for multi-round gradient-based iterative updates.

• *The Client Side*: First, for the construction of each layer $t$, each client $k$ computes its local *Feature Auto-Correlation Matrix* $\mathbf{G}_t^k$ and *Label Cross-Correlation Matrix* $\mathbf{H}_t^k$ via (11). Given that $\mathbf{\Phi}_t^k \in \mathbb{R}^{N_k \times d_\Phi}$ and $\mathbf{Y}_t^k \in \mathbb{R}^{N_k \times C}$, the computation complexity of the above calculations is $\mathcal{O}(N_k d_\Phi^2 + CN_k d_\Phi)$. Second, each client further constructs its local hidden random features $\mathbf{F}_t^k$ and local residual matrix $\mathbf{R}_t^k$ via (14), with a complexity of $\mathcal{O}(N_k d_\Phi d_F + CN_k d_\Phi)$. Third, each client computes its local *Hidden Auto-Correlation Matrix* $\mathbf{\Pi}_t^k$ and *Residual Cross-Correlation Matrix* $\mathbf{\Upsilon}_t^k$ via (15). Given that $\mathbf{F}_t^k \in \mathbb{R}^{N_k \times d_F}$ and $\mathbf{R}_t^k \in \mathbb{R}^{N_k \times C}$, the computation complexity of executing (15) can be calculated $\mathcal{O}(N_k d_F^2 + CN_k d_F)$. Fourth, the client updates the next-layer feature $\mathbf{\Phi}_{t+1}^k$ with a complexity of $\mathcal{O}(N_k d_\Phi d_F)$. In summary, the complexity for each client to construct each layer is $\mathcal{O}(CN_k(d_F + d_\Phi) + N_k(d_\Phi^2 + d_\Phi d_F + d_F^2))$, and the overall complexity for constructing all $T$ layers is $\mathcal{O}(TCN_k(d_F + d_\Phi) + TN_k(d_\Phi^2 + d_\Phi d_F + d_F^2))$.

• *The Server Side*: First, for the construction of each layer $t$, the server aggregates all received $\{\mathbf{G}_t^k\}_{k=1}^K$ and $\{\mathbf{H}_t^k\}_{k=1}^K$ and subsequently derives the global classifier $\mathbf{W}_t$ via (12) and (13). Given that $\mathbf{G}_t^k \in \mathbb{R}^{d_\Phi \times d_\Phi}$ and $\mathbf{H}_t^k \in \mathbb{R}^{d_\Phi \times C}$, the computation complexity of these calculations is $\mathcal{O}(d_\Phi^3 + Kd_\Phi^2 + Cd_\Phi^2 + KCd_\Phi)$. Second, the server aggregates all received $\{\mathbf{\Pi}_t^k\}_{k=1}^K$ and $\{\mathbf{\Upsilon}_t^k\}_{k=1}^K$ via (16) with a complexity of $\mathcal{O}(Kd_F^2 + KCd_F)$. Third, the server performs spectral decompositions and subsequently derives the transformation matrix $\mathbf{\Omega}_{t+1}$ via (17) and (18) with a computation complexity of $\mathcal{O}(d_F^3 + d_\Phi^3 + C(d_F^2 + d_\Phi^2) + d_F d_\Phi(d_F + d_\Phi + C))$. In summary, the complexity for the server to construct each layer is $\mathcal{O}(d_F^3 + d_\Phi^3 + (C+K)(d_F^2 + d_\Phi^2) + d_F d_\Phi(d_F + d_\Phi + C) + KC(d_F + d_\Phi))$, and the overall complexity for constructing all $T$ layers is $\mathcal{O}(T(d_F^3 + d_\Phi^3) + T(C+K)(d_F^2 + d_\Phi^2) + Td_F d_\Phi(d_F + d_\Phi + C) + TKC(d_F + d_\Phi))$.

**Analysis of Communication Complexity**: Here, we further thoroughly analyze the communication complexity within our DeepAFL. Notably, DeepAFL requires only a single communication exchange between the client and server for constructing each layer, in contrast to gradient-based methods that depend on multiple rounds of communication.

• *The Client Side*: According to Algorithm 1, during the construction of each layer $t$, each client $k$ is only required to upload its local *Feature Auto-Correlation Matrix* $\mathbf{G}_t^k$, *Label Cross-Correlation Matrix* $\mathbf{H}_t^k$, *Hidden Auto-Correlation Matrix* $\mathbf{\Pi}_t^k$, and *Residual Cross-Correlation Matrix* $\mathbf{\Upsilon}_t^k$. Given that $\mathbf{G}_t^k \in \mathbb{R}^{d_\Phi \times d_\Phi}$, $\mathbf{H}_t^k \in \mathbb{R}^{d_\Phi \times C}$, $\mathbf{\Pi}_t^k \in \mathbb{R}^{d_F \times d_F}$, and $\mathbf{\Upsilon}_t^k \in \mathbb{R}^{d_F \times C}$, the client's total communication complexity for constructing all $T$ layers can be derived as $\mathcal{O}(T(d_\Phi^2 + d_F^2) + TC(d_\Phi + d_F))$.

• *The Server Side*: For each layer $t$, the server similarly needs only to transmit once to all clients the global classifier $\mathbf{W}_t \in \mathbb{R}^{d_\Phi \times C}$ and the transformation matrix $\mathbf{\Omega}_t \in \mathbb{R}^{d_F \times d_\Phi}$, with a communication complexity of $\mathcal{O}(Kd_\Phi(C + d_F))$. Consequently, the server's total communication complexity for constructing all $T$ layers can be expressed as $\mathcal{O}(TKd_\Phi(C + d_F))$.

**Simplification of Complexity Results**: Here, we further simplify the complexity results based on the relative magnitudes of their constituent terms. Specifically, in practical scenarios, the total number of layers $T$, the number of clients $K$, and the number of classes $C$ are substantially smaller than the sample size $N_k$ in each client's local dataset, as well as the feature dimensions $d_\Phi$ and $d_F$. Consequently, the computation complexity for each client and the server can be simplified to $\mathcal{O}(TN_k(d_\Phi^2 + d_\Phi d_F + d_F^2))$ and $\mathcal{O}(T(d_\Phi^3 + d_F^3 + d_\Phi^2 d_F + d_\Phi d_F^2))$. Similarly, the communication complexity for each client and the server can be simplified to $\mathcal{O}(T(d_\Phi^2 + d_F^2))$ and $\mathcal{O}(TKd_\Phi d_F)$.

# E APPENDIX FOR EXPERIMENTAL EVALUATIONS

## E.1 ADDITIONAL RESULTS FOR EXPERIMENTAL EVALUATIONS

Table 2: Performance comparisons of the top-1 accuracy (%) among our DeepAFL and the baselines, on the CIFAR-10. The best result is highlighted in **bold**, and the second-best result is underlined. All the experiments were conducted three times, and the results are shown as $\text{Mean}_{\pm\text{Standard Error}}$. All the improvements of our DeepAFL were validated by Chi-squared tests at the level of $p < 0.05$. The results of all baselines are directly obtained from the given benchmark in AFL (He et al., 2025).

| | CIFAR-10 | | | |
|---|---|---|---|---|
| | Non-IID-1 | | Non-IID-2 | |
| Baseline | $\alpha = 0.1$ | $\alpha = 0.05$ | $s = 4$ | $s = 2$ |
| FedAvg (2017) | $64.02_{\pm0.18}$ | $60.52_{\pm0.39}$ | $68.47_{\pm0.13}$ | $57.81_{\pm0.03}$ |
| FedProx (2020) | $64.07_{\pm0.08}$ | $60.39_{\pm0.09}$ | $68.46_{\pm0.08}$ | $57.61_{\pm0.12}$ |
| MOON (2021a) | $63.84_{\pm0.03}$ | $60.28_{\pm0.17}$ | $68.47_{\pm0.15}$ | $57.72_{\pm0.15}$ |
| FedGen (2021) | $64.14_{\pm0.24}$ | $60.65_{\pm0.19}$ | $68.24_{\pm0.28}$ | $57.02_{\pm0.18}$ |
| FedDyn (2021) | $64.77_{\pm0.11}$ | $60.35_{\pm0.54}$ | $73.50_{\pm0.11}$ | $64.07_{\pm0.09}$ |
| FedNTD (2022) | $64.64_{\pm0.02}$ | $61.16_{\pm0.33}$ | $70.24_{\pm0.11}$ | $58.77_{\pm0.18}$ |
| FedDisco (2023b) | $63.83_{\pm0.08}$ | $59.90_{\pm0.05}$ | $65.04_{\pm0.11}$ | $58.78_{\pm0.02}$ |
| AFL (2025) | $\underline{80.75}_{\pm0.00}$ | $\underline{80.75}_{\pm0.00}$ | $\underline{80.75}_{\pm0.00}$ | $\underline{80.75}_{\pm0.00}$ |
| DeepAFL ($T = 5$) | $85.20_{\pm0.05}$ | $85.20_{\pm0.05}$ | $85.20_{\pm0.05}$ | $85.20_{\pm0.05}$ |
| DeepAFL ($T = 10$) | $85.93_{\pm0.09}$ | $85.93_{\pm0.09}$ | $85.93_{\pm0.09}$ | $85.93_{\pm0.09}$ |
| DeepAFL ($T = 20$) | $\mathbf{86.43}_{\pm0.07}$ | $\mathbf{86.43}_{\pm0.07}$ | $\mathbf{86.43}_{\pm0.07}$ | $\mathbf{86.43}_{\pm0.07}$ |
| Improvement ↑ | $\mathbf{5.68}_{\,p<0.05}$ | $\mathbf{5.68}_{\,p<0.05}$ | $\mathbf{5.68}_{\,p<0.05}$ | $\mathbf{5.68}_{\,p<0.05}$ |

Table 3: Accuracy of our DeepAFL under varying regularization parameters $\lambda$.

| Datasets | Layers | $\lambda = 0.01$ | $\lambda = 0.05$ | $\lambda = 0.1$ | $\lambda = 0.5$ | $\lambda = 1$ | $\lambda = 5$ | $\lambda = 10$ |
|---|---|---|---|---|---|---|---|---|
| CIFAR-10 | $T = 5$ | $85.14_{\pm0.05}$ | $85.15_{\pm0.04}$ | $85.14_{\pm0.03}$ | $85.11_{\pm0.05}$ | $85.13_{\pm0.04}$ | $85.17_{\pm0.05}$ | $\mathbf{85.20}_{\pm0.05}$ |
| | $T = 10$ | $85.87_{\pm0.08}$ | $85.90_{\pm0.09}$ | $85.86_{\pm0.08}$ | $85.92_{\pm0.08}$ | $85.91_{\pm0.08}$ | $85.92_{\pm0.10}$ | $\mathbf{85.93}_{\pm0.09}$ |
| | $T = 20$ | $86.34_{\pm0.07}$ | $86.31_{\pm0.08}$ | $86.37_{\pm0.07}$ | $86.27_{\pm0.06}$ | $86.33_{\pm0.08}$ | $86.39_{\pm0.08}$ | $\mathbf{86.43}_{\pm0.07}$ |
| CIFAR-100 | $T = 5$ | $64.61_{\pm0.01}$ | $64.62_{\pm0.02}$ | $64.59_{\pm0.02}$ | $64.61_{\pm0.02}$ | $64.62_{\pm0.02}$ | $64.64_{\pm0.01}$ | $\mathbf{64.66}_{\pm0.02}$ |
| | $T = 10$ | $65.95_{\pm0.07}$ | $\mathbf{65.96}_{\pm0.06}$ | $65.95_{\pm0.08}$ | $65.94_{\pm0.06}$ | $65.94_{\pm0.06}$ | $65.91_{\pm0.04}$ | $65.91_{\pm0.05}$ |
| | $T = 20$ | $66.91_{\pm0.06}$ | $66.91_{\pm0.06}$ | $66.92_{\pm0.06}$ | $\mathbf{66.96}_{\pm0.05}$ | $66.94_{\pm0.03}$ | $66.87_{\pm0.03}$ | $66.95_{\pm0.03}$ |
| Tiny-ImageNet | $T = 5$ | $60.23_{\pm0.02}$ | $60.24_{\pm0.02}$ | $60.25_{\pm0.01}$ | $60.24_{\pm0.02}$ | $60.22_{\pm0.02}$ | $\mathbf{60.26}_{\pm0.02}$ | $60.23_{\pm0.02}$ |
| | $T = 10$ | $61.34_{\pm0.08}$ | $61.33_{\pm0.08}$ | $61.35_{\pm0.07}$ | $\mathbf{61.36}_{\pm0.07}$ | $61.34_{\pm0.08}$ | $61.32_{\pm0.08}$ | $61.32_{\pm0.07}$ |
| | $T = 20$ | $62.34_{\pm0.02}$ | $62.33_{\pm0.04}$ | $62.34_{\pm0.02}$ | $62.32_{\pm0.02}$ | $\mathbf{62.35}_{\pm0.03}$ | $62.33_{\pm0.02}$ | $62.34_{\pm0.01}$ |

Table 4: Accuracy of our DeepAFL under varying regularization parameters $\gamma$.

| Datasets | Layers | $\gamma = 0.01$ | $\gamma = 0.05$ | $\gamma = 0.1$ | $\gamma = 0.5$ | $\gamma = 1$ | $\gamma = 5$ | $\gamma = 10$ |
|---|---|---|---|---|---|---|---|---|
| CIFAR-10 | $T = 5$ | $85.06_{\pm0.05}$ | $85.14_{\pm0.06}$ | $\mathbf{85.15}_{\pm0.04}$ | $85.10_{\pm0.02}$ | $85.10_{\pm0.04}$ | $85.04_{\pm0.04}$ | $84.99_{\pm0.04}$ |
| | $T = 10$ | $85.87_{\pm0.06}$ | $85.90_{\pm0.07}$ | $\mathbf{85.91}_{\pm0.26}$ | $85.86_{\pm0.10}$ | $85.84_{\pm0.08}$ | $85.78_{\pm0.08}$ | $85.65_{\pm0.07}$ |
| | $T = 20$ | $86.21_{\pm0.09}$ | $86.21_{\pm0.08}$ | $86.32_{\pm0.08}$ | $\mathbf{86.37}_{\pm0.06}$ | $86.33_{\pm0.08}$ | $86.26_{\pm0.07}$ | $86.23_{\pm0.07}$ |
| CIFAR-100 | $T = 5$ | $\mathbf{64.72}_{\pm0.07}$ | $64.66_{\pm0.02}$ | $64.63_{\pm0.01}$ | $64.58_{\pm0.04}$ | $64.49_{\pm0.00}$ | $63.77_{\pm0.07}$ | $63.21_{\pm0.10}$ |
| | $T = 10$ | $\mathbf{65.96}_{\pm0.05}$ | $65.94_{\pm0.06}$ | $65.95_{\pm0.08}$ | $65.81_{\pm0.08}$ | $65.69_{\pm0.06}$ | $64.82_{\pm0.15}$ | $64.26_{\pm0.15}$ |
| | $T = 20$ | $\mathbf{66.98}_{\pm0.04}$ | $66.94_{\pm0.04}$ | $66.94_{\pm0.03}$ | $66.82_{\pm0.07}$ | $66.74_{\pm0.07}$ | $66.20_{\pm0.09}$ | $65.61_{\pm0.16}$ |
| Tiny-ImageNet | $T = 5$ | $\mathbf{60.31}_{\pm0.04}$ | $60.28_{\pm0.02}$ | $60.24_{\pm0.02}$ | $60.13_{\pm0.04}$ | $59.89_{\pm0.01}$ | $59.04_{\pm0.04}$ | $58.41_{\pm0.05}$ |
| | $T = 10$ | $\mathbf{61.37}_{\pm0.07}$ | $61.36_{\pm0.07}$ | $61.35_{\pm0.08}$ | $61.28_{\pm0.12}$ | $61.19_{\pm0.11}$ | $60.44_{\pm0.04}$ | $59.65_{\pm0.07}$ |
| | $T = 20$ | $\mathbf{62.35}_{\pm0.01}$ | $62.33_{\pm0.09}$ | $62.26_{\pm0.02}$ | $62.24_{\pm0.05}$ | $62.14_{\pm0.02}$ | $61.53_{\pm0.03}$ | $60.98_{\pm0.02}$ |

Table 5: Accuracy of our DeepAFL under varying activation functions $\sigma(\cdot)$.

| Datasets | Layers | None | GELU | ReLU | LeakyReLU | Tanh | Hardwish | Softshrink |
|---|---|---|---|---|---|---|---|---|
| CIFAR-10 | $T=5$ | $83.23_{\pm0.10}$ | $\mathbf{85.13}_{\pm0.04}$ | $84.37_{\pm0.09}$ | $84.49_{\pm0.12}$ | $84.42_{\pm0.07}$ | $84.78_{\pm0.03}$ | $83.77_{\pm0.12}$ |
| | $T=10$ | $83.08_{\pm0.17}$ | $\mathbf{85.89}_{\pm0.10}$ | $85.04_{\pm0.12}$ | $84.54_{\pm0.49}$ | $84.92_{\pm0.12}$ | $85.41_{\pm0.04}$ | $84.13_{\pm0.05}$ |
| | $T=20$ | $82.99_{\pm0.06}$ | $\mathbf{86.34}_{\pm0.05}$ | $85.47_{\pm0.17}$ | $83.85_{\pm1.48}$ | $85.62_{\pm0.05}$ | $86.05_{\pm0.06}$ | $84.25_{\pm0.07}$ |
| CIFAR-100 | $T=5$ | $60.82_{\pm0.09}$ | $\mathbf{64.62}_{\pm0.01}$ | $64.26_{\pm0.02}$ | $64.29_{\pm0.01}$ | $63.28_{\pm0.12}$ | $64.02_{\pm0.02}$ | $62.89_{\pm0.13}$ |
| | $T=10$ | $60.82_{\pm0.09}$ | $\mathbf{65.98}_{\pm0.08}$ | $65.39_{\pm0.09}$ | $65.48_{\pm0.10}$ | $64.27_{\pm0.17}$ | $65.07_{\pm0.20}$ | $63.91_{\pm0.14}$ |
| | $T=20$ | $60.83_{\pm0.09}$ | $\mathbf{66.95}_{\pm0.02}$ | $66.68_{\pm0.08}$ | $66.72_{\pm0.10}$ | $64.02_{\pm1.02}$ | $66.04_{\pm0.06}$ | $64.78_{\pm0.08}$ |
| Tiny-ImageNet | $T=5$ | $56.71_{\pm0.07}$ | $\mathbf{60.23}_{\pm0.01}$ | $60.17_{\pm0.04}$ | $60.16_{\pm0.02}$ | $58.77_{\pm0.04}$ | $59.40_{\pm0.04}$ | $58.80_{\pm0.07}$ |
| | $T=10$ | $56.70_{\pm0.08}$ | $61.33_{\pm0.07}$ | $61.35_{\pm0.05}$ | $\mathbf{61.36}_{\pm0.08}$ | $59.68_{\pm0.07}$ | $60.48_{\pm0.09}$ | $59.83_{\pm0.04}$ |
| | $T=20$ | $56.70_{\pm0.08}$ | $\mathbf{62.33}_{\pm0.02}$ | $62.31_{\pm0.06}$ | $62.31_{\pm0.07}$ | $60.74_{\pm0.04}$ | $61.47_{\pm0.01}$ | $60.70_{\pm0.09}$ |

Table 6: Accuracy of our DeepAFL under varying dimensions $d_\Phi$. For a comprehensive and consistent analysis, all the experimental results were obtained by fixing the other dimension $d_\mathrm{F}$ at 1024.

| Datasets | Layers | $d_\Phi = 2^7$ | $d_\Phi = 2^8$ | $d_\Phi = 2^9$ | $d_\Phi = 2^{10}$ | $d_\Phi = 2^{11}$ | $d_\Phi = 2^{12}$ | $d_\Phi = 2^{13}$ |
|---|---|---|---|---|---|---|---|---|
| CIFAR-10 | $T=5$ | $81.48_{\pm0.30}$ | $83.31_{\pm0.12}$ | $84.58_{\pm0.08}$ | $85.07_{\pm0.02}$ | $85.48_{\pm0.03}$ | $\mathbf{85.76}_{\pm0.7}$ | $85.57_{\pm0.33}$ |
| | $T=10$ | $82.28_{\pm0.30}$ | $84.41_{\pm0.01}$ | $85.55_{\pm0.11}$ | $\mathbf{85.96}_{\pm0.10}$ | $85.94_{\pm0.01}$ | $85.94_{\pm0.12}$ | $85.77_{\pm0.28}$ |
| | $T=20$ | $82.68_{\pm0.27}$ | $84.95_{\pm0.13}$ | $86.01_{\pm0.01}$ | $86.28_{\pm0.15}$ | $86.41_{\pm0.01}$ | $\mathbf{86.48}_{\pm0.11}$ | $83.12_{\pm0.39}$ |
| CIFAR-100 | $T=5$ | $57.61_{\pm0.23}$ | $62.21_{\pm0.14}$ | $64.06_{\pm0.07}$ | $64.63_{\pm0.02}$ | $65.39_{\pm0.05}$ | $66.46_{\pm0.12}$ | $\mathbf{67.46}_{\pm0.01}$ |
| | $T=10$ | $58.20_{\pm0.19}$ | $63.45_{\pm0.19}$ | $65.29_{\pm0.07}$ | $65.93_{\pm0.10}$ | $66.36_{\pm0.04}$ | $\mathbf{67.20}_{\pm0.06}$ | $67.19_{\pm0.44}$ |
| | $T=20$ | $58.51_{\pm0.25}$ | $64.29_{\pm0.36}$ | $66.64_{\pm0.10}$ | $66.93_{\pm0.04}$ | $67.39_{\pm0.06}$ | $67.85_{\pm0.05}$ | $\mathbf{67.94}_{\pm0.02}$ |
| Tiny-ImageNet | $T=5$ | $44.94_{\pm0.24}$ | $56.91_{\pm0.14}$ | $59.29_{\pm0.26}$ | $60.21_{\pm0.04}$ | $60.77_{\pm0.23}$ | $61.84_{\pm0.19}$ | $\mathbf{62.75}_{\pm0.22}$ |
| | $T=10$ | $44.11_{\pm0.11}$ | $57.51_{\pm0.18}$ | $60.68_{\pm0.24}$ | $61.43_{\pm0.04}$ | $61.90_{\pm0.15}$ | $62.42_{\pm0.24}$ | $\mathbf{63.13}_{\pm0.41}$ |
| | $T=20$ | $43.10_{\pm0.19}$ | $58.09_{\pm0.15}$ | $61.70_{\pm0.19}$ | $62.33_{\pm0.03}$ | $62.88_{\pm0.18}$ | $63.11_{\pm0.29}$ | $\mathbf{63.70}_{\pm0.19}$ |

Table 7: Accuracy of our DeepAFL under varying dimensions $d_\mathrm{F}$. For a comprehensive and consistent analysis, all the experimental results were obtained by fixing the other dimension $d_\Phi$ at 1024.

| Datasets | Layers | $d_\mathrm{F} = 2^7$ | $d_\mathrm{F} = 2^8$ | $d_\mathrm{F} = 2^9$ | $d_\mathrm{F} = 2^{10}$ | $d_\mathrm{F} = 2^{11}$ | $d_\mathrm{F} = 2^{12}$ | $d_\mathrm{F} = 2^{13}$ |
|---|---|---|---|---|---|---|---|---|
| CIFAR-10 | $T=5$ | $83.48_{\pm0.10}$ | $83.76_{\pm0.09}$ | $84.24_{\pm0.14}$ | $85.14_{\pm0.04}$ | $86.16_{\pm0.06}$ | $\mathbf{86.51}_{\pm0.04}$ | $85.55_{\pm0.11}$ |
| | $T=10$ | $83.58_{\pm0.11}$ | $84.03_{\pm0.10}$ | $84.84_{\pm0.08}$ | $85.89_{\pm0.09}$ | $\mathbf{86.50}_{\pm0.13}$ | $86.42_{\pm0.04}$ | $84.90_{\pm0.09}$ |
| | $T=20$ | $83.83_{\pm0.11}$ | $84.49_{\pm0.09}$ | $85.34_{\pm0.04}$ | $86.31_{\pm0.08}$ | $86.53_{\pm0.05}$ | $\mathbf{85.59}_{\pm0.10}$ | $83.40_{\pm0.10}$ |
| CIFAR-100 | $T=5$ | $61.22_{\pm0.06}$ | $61.82_{\pm0.11}$ | $63.11_{\pm0.11}$ | $64.63_{\pm0.01}$ | $66.49_{\pm0.12}$ | $\mathbf{67.81}_{\pm0.01}$ | $67.38_{\pm0.09}$ |
| | $T=10$ | $61.54_{\pm0.10}$ | $62.57_{\pm0.10}$ | $64.08_{\pm0.04}$ | $65.96_{\pm0.08}$ | $67.59_{\pm0.06}$ | $\mathbf{67.92}_{\pm0.08}$ | $66.12_{\pm0.04}$ |
| | $T=20$ | $61.92_{\pm0.14}$ | $63.41_{\pm0.12}$ | $65.23_{\pm0.18}$ | $66.94_{\pm0.03}$ | $\mathbf{68.20}_{\pm0.03}$ | $67.14_{\pm0.07}$ | $64.10_{\pm0.10}$ |
| Tiny-ImageNet | $T=5$ | $56.92_{\pm0.02}$ | $57.55_{\pm0.11}$ | $58.69_{\pm0.05}$ | $60.24_{\pm0.02}$ | $61.82_{\pm0.07}$ | $63.36_{\pm0.08}$ | $\mathbf{63.82}_{\pm0.04}$ |
| | $T=10$ | $57.23_{\pm0.07}$ | $58.30_{\pm0.09}$ | $59.58_{\pm0.08}$ | $61.34_{\pm0.07}$ | $62.89_{\pm0.06}$ | $\mathbf{63.84}_{\pm0.10}$ | $62.89_{\pm0.04}$ |
| | $T=20$ | $57.79_{\pm0.05}$ | $59.18_{\pm0.08}$ | $60.86_{\pm0.03}$ | $62.34_{\pm0.03}$ | $\mathbf{63.85}_{\pm0.11}$ | $63.48_{\pm0.03}$ | $60.81_{\pm0.08}$ |

Table 8: Accuracy of our DeepAFL under varying dimensions $d_\Phi$ and $d_\mathrm{F}$. For a comprehensive and consistent analysis, all the experimental results were obtained by ensuring $d_\Phi = d_\mathrm{F} = d$.

| Datasets | Layers | $d = 2^7$ | $d = 2^8$ | $d = 2^9$ | $d = 2^{10}$ | $d = 2^{11}$ | $d = 2^{12}$ | $d = 2^{13}$ |
|---|---|---|---|---|---|---|---|---|
| CIFAR-10 | $T=5$ | $77.01_{\pm0.31}$ | $80.62_{\pm0.17}$ | $83.54_{\pm0.09}$ | $85.04_{\pm0.12}$ | $86.19_{\pm0.03}$ | $\mathbf{86.54}_{\pm0.04}$ | / |
| | $T=10$ | $77.96_{\pm0.21}$ | $81.62_{\pm0.19}$ | $84.24_{\pm0.08}$ | $85.77_{\pm0.04}$ | $\mathbf{86.58}_{\pm0.04}$ | $86.47_{\pm0.12}$ | / |
| | $T=20$ | $78.69_{\pm0.11}$ | $82.38_{\pm0.08}$ | $85.07_{\pm0.07}$ | $86.22_{\pm0.02}$ | $86.71_{\pm0.13}$ | $\mathbf{84.76}_{\pm0.34}$ | / |
| CIFAR-100 | $T=5$ | $49.88_{\pm0.27}$ | $57.15_{\pm0.10}$ | $61.74_{\pm0.04}$ | $64.63_{\pm0.00}$ | $66.86_{\pm0.13}$ | $\mathbf{68.59}_{\pm0.08}$ | / |
| | $T=10$ | $51.36_{\pm0.13}$ | $58.76_{\pm0.12}$ | $63.22_{\pm0.02}$ | $65.96_{\pm0.06}$ | $68.20_{\pm0.06}$ | $\mathbf{69.10}_{\pm0.03}$ | / |
| | $T=20$ | $53.21_{\pm0.10}$ | $60.26_{\pm0.07}$ | $64.72_{\pm0.11}$ | $66.93_{\pm0.03}$ | $68.75_{\pm0.06}$ | $\mathbf{69.12}_{\pm0.08}$ | / |
| Tiny-ImageNet | $T=5$ | $44.94_{\pm0.23}$ | $53.11_{\pm0.13}$ | $57.44_{\pm0.05}$ | $60.22_{\pm0.03}$ | $62.29_{\pm0.05}$ | $\mathbf{63.95}_{\pm0.07}$ | / |
| | $T=10$ | $44.20_{\pm0.09}$ | $54.47_{\pm0.21}$ | $58.72_{\pm0.07}$ | $61.17_{\pm0.01}$ | $63.13_{\pm0.06}$ | $\mathbf{64.63}_{\pm0.10}$ | / |
| | $T=20$ | $43.83_{\pm0.12}$ | $56.18_{\pm0.11}$ | $59.92_{\pm0.02}$ | $62.34_{\pm0.02}$ | $64.08_{\pm0.06}$ | $\mathbf{64.68}_{\pm0.04}$ | / |

Table 9: Analyses of our DeepAFL's representation learning capability on the CIFAR-10 with increasing network depth. "Training Acc" and "Testing Acc" show the accuracy (%) on the training and test sets. "Time Cost" denotes the required training time (s). The symbol $\Delta$ represents the difference between two consecutive cases, which reflects the marginal effect of deepening the network.

| Layers | Training Acc | $\Delta_{\text{Training Acc}}$ | Testing Acc | $\Delta_{\text{Testing Acc}}$ | Time Cost | $\Delta_{\text{Time Cost}}$ |
|---|---|---|---|---|---|---|
| AFL | $83.81_{\pm 0.00}$ | / | $80.75_{\pm 0.00}$ | / | $52.36_{\pm 0.24}$ | / |
| $T = 0$ | $85.35_{\pm 0.02}$ | $1.54_{\pm 0.02}$ | $83.29_{\pm 0.20}$ | $2.54_{\pm 0.20}$ | $58.02_{\pm 0.17}$ | $5.66_{\pm 0.13}$ |
| $T = 5$ | $89.03_{\pm 0.03}$ | $3.68_{\pm 0.02}$ | $85.13_{\pm 0.08}$ | $1.84_{\pm 0.08}$ | $65.08_{\pm 0.39}$ | $7.06_{\pm 0.22}$ |
| $T = 10$ | $90.61_{\pm 0.05}$ | $1.58_{\pm 0.03}$ | $85.89_{\pm 0.09}$ | $0.75_{\pm 0.11}$ | $71.91_{\pm 0.28}$ | $6.83_{\pm 0.29}$ |
| $T = 15$ | $91.80_{\pm 0.04}$ | $1.19_{\pm 0.04}$ | $86.18_{\pm 0.08}$ | $0.29_{\pm 0.04}$ | $78.86_{\pm 0.24}$ | $6.95_{\pm 0.15}$ |
| $T = 20$ | $92.74_{\pm 0.03}$ | $0.94_{\pm 0.01}$ | $86.34_{\pm 0.06}$ | $0.16_{\pm 0.06}$ | $85.71_{\pm 0.22}$ | $6.85_{\pm 0.17}$ |
| $T = 25$ | $93.49_{\pm 0.02}$ | $0.75_{\pm 0.01}$ | $86.40_{\pm 0.06}$ | $0.06_{\pm 0.06}$ | $92.57_{\pm 0.19}$ | $6.86_{\pm 0.11}$ |
| $T = 30$ | $94.10_{\pm 0.03}$ | $0.61_{\pm 0.01}$ | $86.46_{\pm 0.06}$ | $0.06_{\pm 0.05}$ | $99.29_{\pm 0.33}$ | $6.72_{\pm 0.19}$ |
| $T = 35$ | $94.63_{\pm 0.03}$ | $0.53_{\pm 0.03}$ | $86.72_{\pm 0.06}$ | $0.26_{\pm 0.01}$ | $105.79_{\pm 0.31}$ | $6.50_{\pm 0.21}$ |
| $T = 40$ | $95.08_{\pm 0.03}$ | $0.44_{\pm 0.02}$ | $86.76_{\pm 0.06}$ | $0.04_{\pm 0.02}$ | $112.40_{\pm 0.21}$ | $6.61_{\pm 0.06}$ |
| $T = 45$ | $95.43_{\pm 0.03}$ | $0.35_{\pm 0.01}$ | $\mathbf{86.77}_{\pm 0.06}$ | $0.01_{\pm 0.00}$ | $119.00_{\pm 0.17}$ | $6.60_{\pm 0.13}$ |
| $T = 50$ | $\mathbf{95.79}_{\pm 0.03}$ | $0.36_{\pm 0.00}$ | $86.72_{\pm 0.06}$ | $-0.05_{\pm 0.11}$ | $\mathbf{125.72}_{\pm 0.31}$ | $6.72_{\pm 0.08}$ |

Table 10: Analyses of our DeepAFL's representation learning capability on the CIFAR-100 with increasing network depth. "Training Acc" and "Testing Acc" show the accuracy (%) on the training and test sets. "Time Cost" denotes the required training time (s). The symbol $\Delta$ represents the difference between two consecutive cases, which reflects the marginal effect of deepening the network.

| Layers | Training Acc | $\Delta_{\text{Training Acc}}$ | Testing Acc | $\Delta_{\text{Testing Acc}}$ | Time Cost | $\Delta_{\text{Time Cost}}$ |
|---|---|---|---|---|---|---|
| AFL | $61.55_{\pm 0.00}$ | / | $58.56_{\pm 0.00}$ | / | $50.05_{\pm 0.29}$ | / |
| $T = 0$ | $65.66_{\pm 0.04}$ | $4.11_{\pm 0.04}$ | $60.81_{\pm 0.15}$ | $2.25_{\pm 0.15}$ | $56.90_{\pm 0.17}$ | $6.85_{\pm 0.19}$ |
| $T = 5$ | $74.93_{\pm 0.04}$ | $9.27_{\pm 0.04}$ | $64.62_{\pm 0.02}$ | $3.81_{\pm 0.08}$ | $66.48_{\pm 0.21}$ | $9.58_{\pm 0.23}$ |
| $T = 10$ | $79.41_{\pm 0.08}$ | $4.48_{\pm 0.04}$ | $65.98_{\pm 0.13}$ | $1.36_{\pm 0.09}$ | $75.25_{\pm 0.33}$ | $8.77_{\pm 0.03}$ |
| $T = 15$ | $82.69_{\pm 0.09}$ | $3.28_{\pm 0.04}$ | $66.59_{\pm 0.03}$ | $0.61_{\pm 0.09}$ | $83.50_{\pm 0.12}$ | $8.25_{\pm 0.09}$ |
| $T = 20$ | $85.15_{\pm 0.03}$ | $2.45_{\pm 0.06}$ | $66.95_{\pm 0.04}$ | $0.36_{\pm 0.02}$ | $91.74_{\pm 0.29}$ | $8.24_{\pm 0.22}$ |
| $T = 25$ | $87.21_{\pm 0.02}$ | $2.06_{\pm 0.03}$ | $67.40_{\pm 0.10}$ | $0.45_{\pm 0.05}$ | $100.36_{\pm 0.37}$ | $8.62_{\pm 0.18}$ |
| $T = 30$ | $88.84_{\pm 0.06}$ | $1.63_{\pm 0.03}$ | $67.71_{\pm 0.06}$ | $0.31_{\pm 0.03}$ | $108.37_{\pm 0.09}$ | $8.01_{\pm 0.03}$ |
| $T = 35$ | $90.14_{\pm 0.06}$ | $1.30_{\pm 0.00}$ | $67.97_{\pm 0.12}$ | $0.26_{\pm 0.07}$ | $116.50_{\pm 0.19}$ | $8.13_{\pm 0.11}$ |
| $T = 40$ | $91.29_{\pm 0.09}$ | $1.15_{\pm 0.02}$ | $68.19_{\pm 0.07}$ | $0.22_{\pm 0.03}$ | $124.74_{\pm 0.17}$ | $8.24_{\pm 0.03}$ |
| $T = 45$ | $92.15_{\pm 0.11}$ | $0.86_{\pm 0.01}$ | $68.32_{\pm 0.11}$ | $0.13_{\pm 0.05}$ | $133.06_{\pm 0.09}$ | $8.32_{\pm 0.02}$ |
| $T = 50$ | $\mathbf{92.93}_{\pm 0.06}$ | $0.78_{\pm 0.03}$ | $\mathbf{68.51}_{\pm 0.09}$ | $0.19_{\pm 0.02}$ | $\mathbf{141.38}_{\pm 0.08}$ | $8.32_{\pm 0.03}$ |

Table 11: Analyses of our DeepAFL's representation learning capability on the Tiny-ImageNet with increasing network depth. "Training Acc" and "Testing Acc" show the accuracy (%) on the training and test sets. "Time Cost" denotes the required training time (s). The symbol $\Delta$ represents the difference between two consecutive cases, which reflects the marginal effect of deepening the network.

| Layers | Training Acc | $\Delta_{\text{Training Acc}}$ | Testing Acc | $\Delta_{\text{Testing Acc}}$ | Time Cost | $\Delta_{\text{Time Cost}}$ |
|---|---|---|---|---|---|---|
| AFL | $57.39_{\pm 0.00}$ | / | $54.67_{\pm 0.00}$ | / | $125.31_{\pm 0.38}$ | / |
| $T = 0$ | $60.30_{\pm 0.02}$ | $2.91_{\pm 0.02}$ | $56.72_{\pm 0.14}$ | $2.05_{\pm 0.14}$ | $130.72_{\pm 0.22}$ | $5.41_{\pm 0.08}$ |
| $T = 5$ | $66.89_{\pm 0.06}$ | $6.59_{\pm 0.02}$ | $60.24_{\pm 0.02}$ | $3.52_{\pm 0.07}$ | $145.73_{\pm 0.13}$ | $15.01_{\pm 0.11}$ |
| $T = 10$ | $70.14_{\pm 0.03}$ | $3.24_{\pm 0.02}$ | $61.34_{\pm 0.13}$ | $1.10_{\pm 0.06}$ | $160.75_{\pm 0.23}$ | $15.02_{\pm 0.03}$ |
| $T = 15$ | $72.64_{\pm 0.05}$ | $2.50_{\pm 0.02}$ | $61.94_{\pm 0.08}$ | $0.60_{\pm 0.11}$ | $175.76_{\pm 0.34}$ | $15.01_{\pm 0.02}$ |
| $T = 20$ | $74.75_{\pm 0.02}$ | $2.11_{\pm 0.03}$ | $62.34_{\pm 0.02}$ | $0.40_{\pm 0.03}$ | $190.01_{\pm 0.09}$ | $14.25_{\pm 0.11}$ |
| $T = 25$ | $76.58_{\pm 0.03}$ | $1.83_{\pm 0.01}$ | $62.66_{\pm 0.05}$ | $0.32_{\pm 0.02}$ | $204.64_{\pm 0.12}$ | $14.63_{\pm 0.13}$ |
| $T = 30$ | $78.31_{\pm 0.02}$ | $1.73_{\pm 0.02}$ | $62.94_{\pm 0.03}$ | $0.29_{\pm 0.02}$ | $219.56_{\pm 0.11}$ | $14.92_{\pm 0.03}$ |
| $T = 35$ | $79.87_{\pm 0.03}$ | $1.56_{\pm 0.01}$ | $63.24_{\pm 0.16}$ | $0.30_{\pm 0.09}$ | $234.99_{\pm 0.23}$ | $15.53_{\pm 0.07}$ |
| $T = 40$ | $81.32_{\pm 0.03}$ | $1.45_{\pm 0.02}$ | $63.48_{\pm 0.18}$ | $0.24_{\pm 0.05}$ | $249.95_{\pm 0.13}$ | $14.94_{\pm 0.04}$ |
| $T = 45$ | $82.62_{\pm 0.04}$ | $1.30_{\pm 0.03}$ | $63.65_{\pm 0.18}$ | $0.16_{\pm 0.01}$ | $264.31_{\pm 0.05}$ | $14.36_{\pm 0.11}$ |
| $T = 50$ | $\mathbf{83.82}_{\pm 0.09}$ | $1.20_{\pm 0.03}$ | $\mathbf{63.74}_{\pm 0.04}$ | $0.10_{\pm 0.08}$ | $\mathbf{279.68}_{\pm 0.14}$ | $15.37_{\pm 0.13}$ |

Table 12: Ablation study of our DeepAFL's training accuracy on the CIFAR-10. We evaluate four distinct ablation models to explore the contributions of our key components in the representations: (1) Ablating the residual skip connection, (2) Ablating the random projection $\mathbf{B}_t$, (3) Ablating the activation function $\sigma(\cdot)$, and (4) Ablating the trainable transformation $\mathbf{\Omega}_{t+1}$. The value in parentheses ($\downarrow$) indicates the performance drop compared to our full DeepAFL under identical conditions.

| Layers | DeepAFL | Ablation (1) | Ablation (2) | Ablation (3) | Ablation (4) |
|---|---|---|---|---|---|
| $T=0$ | $85.35_{\pm0.02}$ | $85.35_{\pm0.02}$ ($\downarrow 0.00$) | $85.35_{\pm0.02}$ ($\downarrow 0.00$) | $85.35_{\pm0.02}$ ($\downarrow 0.00$) | $85.35_{\pm0.02}$ ($\downarrow 0.00$) |
| $T=5$ | $89.03_{\pm0.03}$ | $85.23_{\pm0.05}$ ($\downarrow 3.80$) | $86.86_{\pm0.03}$ ($\downarrow 2.17$) | $85.35_{\pm0.02}$ ($\downarrow 3.68$) | $86.13_{\pm0.03}$ ($\downarrow 2.90$) |
| $T=10$ | $90.61_{\pm0.05}$ | $85.28_{\pm0.05}$ ($\downarrow 5.33$) | $86.97_{\pm0.03}$ ($\downarrow 3.64$) | $85.35_{\pm0.02}$ ($\downarrow 5.26$) | $86.30_{\pm0.04}$ ($\downarrow 4.31$) |
| $T=15$ | $91.80_{\pm0.04}$ | $85.32_{\pm0.07}$ ($\downarrow 6.48$) | $87.03_{\pm0.04}$ ($\downarrow 4.77$) | $85.35_{\pm0.02}$ ($\downarrow 6.45$) | $86.38_{\pm0.04}$ ($\downarrow 5.42$) |
| $T=20$ | $92.74_{\pm0.03}$ | $85.33_{\pm0.07}$ ($\downarrow 7.41$) | $87.06_{\pm0.03}$ ($\downarrow 5.68$) | $85.35_{\pm0.02}$ ($\downarrow 7.39$) | $86.43_{\pm0.04}$ ($\downarrow 6.31$) |
| $T=25$ | $93.49_{\pm0.02}$ | $85.34_{\pm0.06}$ ($\downarrow 8.15$) | $87.07_{\pm0.03}$ ($\downarrow 6.42$) | $85.35_{\pm0.02}$ ($\downarrow 8.14$) | $86.45_{\pm0.03}$ ($\downarrow 7.04$) |
| $T=30$ | $94.10_{\pm0.03}$ | $85.37_{\pm0.06}$ ($\downarrow 8.73$) | $87.09_{\pm0.03}$ ($\downarrow 7.01$) | $85.35_{\pm0.02}$ ($\downarrow 8.75$) | $86.48_{\pm0.02}$ ($\downarrow 7.62$) |
| $T=35$ | $94.63_{\pm0.03}$ | $85.39_{\pm0.06}$ ($\downarrow 9.24$) | $87.10_{\pm0.03}$ ($\downarrow 7.53$) | $85.36_{\pm0.02}$ ($\downarrow 9.27$) | $86.45_{\pm0.04}$ ($\downarrow 8.18$) |
| $T=40$ | $95.08_{\pm0.03}$ | $85.41_{\pm0.05}$ ($\downarrow 9.67$) | $87.10_{\pm0.03}$ ($\downarrow 7.98$) | $85.35_{\pm0.02}$ ($\downarrow 9.73$) | $86.47_{\pm0.01}$ ($\downarrow 8.61$) |
| $T=45$ | $95.43_{\pm0.03}$ | $85.40_{\pm0.05}$ ($\downarrow 10.0$) | $87.11_{\pm0.03}$ ($\downarrow 8.32$) | $85.35_{\pm0.02}$ ($\downarrow 10.1$) | $86.45_{\pm0.02}$ ($\downarrow 8.98$) |
| $T=50$ | $95.79_{\pm0.03}$ | $85.42_{\pm0.04}$ ($\downarrow 10.4$) | $87.11_{\pm0.02}$ ($\downarrow 8.68$) | $85.35_{\pm0.02}$ ($\downarrow 10.4$) | $86.47_{\pm0.04}$ ($\downarrow 9.32$) |

Table 13: Ablation study of our DeepAFL's testing accuracy on the CIFAR-10. We evaluate four distinct ablation models to explore the contributions of our key components in the representations: (1) Ablating the residual skip connection, (2) Ablating the random projection $\mathbf{B}_t$, (3) Ablating the activation function $\sigma(\cdot)$, and (4) Ablating the trainable transformation $\mathbf{\Omega}_{t+1}$. The value in parentheses ($\downarrow$) indicates the performance drop compared to our full DeepAFL under identical conditions.

| Layers | DeepAFL | Ablation (1) | Ablation (2) | Ablation (3) | Ablation (4) |
|---|---|---|---|---|---|
| $T=0$ | $83.29_{\pm0.20}$ | $83.29_{\pm0.20}$ ($\downarrow 0.00$) | $83.29_{\pm0.20}$ ($\downarrow 0.00$) | $83.29_{\pm0.20}$ ($\downarrow 0.00$) | $83.29_{\pm0.20}$ ($\downarrow 0.00$) |
| $T=5$ | $85.13_{\pm0.08}$ | $83.03_{\pm0.08}$ ($\downarrow 2.10$) | $83.92_{\pm0.21}$ ($\downarrow 1.21$) | $83.33_{\pm0.23}$ ($\downarrow 1.80$) | $83.87_{\pm0.16}$ ($\downarrow 1.26$) |
| $T=10$ | $85.89_{\pm0.09}$ | $83.06_{\pm0.07}$ ($\downarrow 2.83$) | $83.99_{\pm0.21}$ ($\downarrow 1.90$) | $83.34_{\pm0.23}$ ($\downarrow 2.55$) | $83.95_{\pm0.14}$ ($\downarrow 1.94$) |
| $T=15$ | $86.18_{\pm0.08}$ | $83.05_{\pm0.08}$ ($\downarrow 3.13$) | $84.00_{\pm0.24}$ ($\downarrow 2.18$) | $83.33_{\pm0.23}$ ($\downarrow 2.85$) | $84.08_{\pm0.28}$ ($\downarrow 2.10$) |
| $T=20$ | $86.34_{\pm0.06}$ | $83.03_{\pm0.08}$ ($\downarrow 3.31$) | $84.01_{\pm0.25}$ ($\downarrow 2.33$) | $83.32_{\pm0.23}$ ($\downarrow 3.02$) | $84.02_{\pm0.22}$ ($\downarrow 2.32$) |
| $T=25$ | $86.40_{\pm0.06}$ | $82.99_{\pm0.11}$ ($\downarrow 3.41$) | $84.00_{\pm0.29}$ ($\downarrow 2.40$) | $83.33_{\pm0.23}$ ($\downarrow 3.07$) | $84.08_{\pm0.27}$ ($\downarrow 2.32$) |
| $T=30$ | $86.46_{\pm0.06}$ | $83.02_{\pm0.11}$ ($\downarrow 3.44$) | $84.01_{\pm0.27}$ ($\downarrow 2.45$) | $83.33_{\pm0.23}$ ($\downarrow 3.13$) | $84.06_{\pm0.34}$ ($\downarrow 2.40$) |
| $T=35$ | $86.72_{\pm0.06}$ | $82.94_{\pm0.12}$ ($\downarrow 3.78$) | $83.99_{\pm0.27}$ ($\downarrow 2.73$) | $83.33_{\pm0.24}$ ($\downarrow 3.39$) | $84.03_{\pm0.33}$ ($\downarrow 2.69$) |
| $T=40$ | $86.76_{\pm0.06}$ | $82.93_{\pm0.16}$ ($\downarrow 3.83$) | $84.02_{\pm0.25}$ ($\downarrow 2.74$) | $83.33_{\pm0.24}$ ($\downarrow 3.43$) | $84.00_{\pm0.28}$ ($\downarrow 2.76$) |
| $T=45$ | $86.77_{\pm0.06}$ | $82.96_{\pm0.15}$ ($\downarrow 3.81$) | $84.01_{\pm0.28}$ ($\downarrow 2.76$) | $83.34_{\pm0.23}$ ($\downarrow 3.43$) | $83.91_{\pm0.31}$ ($\downarrow 2.86$) |
| $T=50$ | $86.72_{\pm0.06}$ | $82.97_{\pm0.16}$ ($\downarrow 3.75$) | $84.02_{\pm0.28}$ ($\downarrow 2.70$) | $83.34_{\pm0.24}$ ($\downarrow 3.38$) | $83.97_{\pm0.25}$ ($\downarrow 2.75$) |

Table 14: Ablation study of our DeepAFL's training accuracy on the CIFAR-100. We evaluate four distinct ablation models to explore the contributions of our key components in the representations: (1) Ablating the residual skip connection, (2) Ablating the random projection $\mathbf{B}_t$, (3) Ablating the activation function $\sigma(\cdot)$, and (4) Ablating the trainable transformation $\mathbf{\Omega}_{t+1}$. The value in parentheses ($\downarrow$) indicates the performance drop compared to our full DeepAFL under identical conditions.

| Layers | DeepAFL | Ablation (1) | Ablation (2) | Ablation (3) | Ablation (4) |
|---|---|---|---|---|---|
| $T=0$ | $65.66_{\pm0.04}$ | $65.66_{\pm0.04}$ ($\downarrow 0.00$) | $65.66_{\pm0.04}$ ($\downarrow 0.00$) | $65.66_{\pm0.04}$ ($\downarrow 0.00$) | $65.66_{\pm0.04}$ ($\downarrow 0.00$) |
| $T=5$ | $74.93_{\pm0.04}$ | $65.81_{\pm0.04}$ ($\downarrow 9.12$) | $69.80_{\pm0.11}$ ($\downarrow 5.13$) | $65.66_{\pm0.04}$ ($\downarrow 9.27$) | $65.67_{\pm0.07}$ ($\downarrow 9.26$) |
| $T=10$ | $79.41_{\pm0.08}$ | $66.32_{\pm0.03}$ ($\downarrow 13.1$) | $70.16_{\pm0.10}$ ($\downarrow 9.25$) | $65.66_{\pm0.04}$ ($\downarrow 13.8$) | $65.70_{\pm0.06}$ ($\downarrow 13.7$) |
| $T=15$ | $82.69_{\pm0.09}$ | $66.60_{\pm0.11}$ ($\downarrow 16.1$) | $70.38_{\pm0.09}$ ($\downarrow 12.3$) | $65.66_{\pm0.03}$ ($\downarrow 17.0$) | $65.72_{\pm0.08}$ ($\downarrow 17.0$) |
| $T=20$ | $85.15_{\pm0.03}$ | $66.86_{\pm0.08}$ ($\downarrow 18.3$) | $70.46_{\pm0.08}$ ($\downarrow 14.7$) | $65.66_{\pm0.03}$ ($\downarrow 19.5$) | $65.72_{\pm0.09}$ ($\downarrow 19.4$) |
| $T=25$ | $87.21_{\pm0.02}$ | $67.10_{\pm0.12}$ ($\downarrow 20.1$) | $70.57_{\pm0.08}$ ($\downarrow 16.6$) | $65.67_{\pm0.04}$ ($\downarrow 21.5$) | $65.74_{\pm0.10}$ ($\downarrow 21.5$) |
| $T=30$ | $88.84_{\pm0.06}$ | $67.25_{\pm0.09}$ ($\downarrow 21.6$) | $70.63_{\pm0.10}$ ($\downarrow 18.2$) | $65.67_{\pm0.03}$ ($\downarrow 23.2$) | $65.77_{\pm0.10}$ ($\downarrow 23.1$) |
| $T=35$ | $90.14_{\pm0.06}$ | $67.40_{\pm0.11}$ ($\downarrow 22.7$) | $70.65_{\pm0.10}$ ($\downarrow 19.5$) | $65.67_{\pm0.04}$ ($\downarrow 24.5$) | $65.76_{\pm0.08}$ ($\downarrow 24.4$) |
| $T=40$ | $91.29_{\pm0.09}$ | $67.56_{\pm0.12}$ ($\downarrow 23.7$) | $70.69_{\pm0.09}$ ($\downarrow 20.6$) | $65.68_{\pm0.04}$ ($\downarrow 25.6$) | $65.78_{\pm0.09}$ ($\downarrow 25.5$) |
| $T=45$ | $92.15_{\pm0.11}$ | $67.70_{\pm0.08}$ ($\downarrow 24.5$) | $70.73_{\pm0.10}$ ($\downarrow 21.4$) | $65.68_{\pm0.04}$ ($\downarrow 26.5$) | $65.77_{\pm0.08}$ ($\downarrow 26.4$) |
| $T=50$ | $92.93_{\pm0.06}$ | $67.80_{\pm0.11}$ ($\downarrow 25.1$) | $70.78_{\pm0.09}$ ($\downarrow 22.2$) | $65.67_{\pm0.04}$ ($\downarrow 27.3$) | $65.76_{\pm0.06}$ ($\downarrow 27.2$) |

Table 15: Ablation study of our DeepAFL's testing accuracy on the CIFAR-100. We evaluate four distinct ablation models to explore the contributions of our key components in the representations: (1) Ablating the residual skip connection, (2) Ablating the random projection $\mathbf{B}_t$, (3) Ablating the activation function $\sigma(\cdot)$, and (4) Ablating the trainable transformation $\mathbf{\Omega}_{t+1}$. The value in parentheses ($\downarrow$) indicates the performance drop compared to our full DeepAFL under identical conditions.

| Layers | DeepAFL | Ablation (1) | Ablation (2) | Ablation (3) | Ablation (4) |
|---|---|---|---|---|---|
| $T = 0$ | $\mathbf{60.81}_{\pm 0.15}$ | $60.81_{\pm 0.15}\ (\downarrow 0.00)$ | $60.81_{\pm 0.15}\ (\downarrow 0.00)$ | $60.81_{\pm 0.15}\ (\downarrow 0.00)$ | $60.81_{\pm 0.15}\ (\downarrow 0.00)$ |
| $T = 5$ | $\mathbf{64.62}_{\pm 0.02}$ | $60.91_{\pm 0.15}\ (\downarrow 3.71)$ | $62.43_{\pm 0.09}\ (\downarrow 2.19)$ | $60.80_{\pm 0.16}\ (\downarrow 3.82)$ | $60.80_{\pm 0.10}\ (\downarrow 3.82)$ |
| $T = 10$ | $\mathbf{65.98}_{\pm 0.13}$ | $61.17_{\pm 0.10}\ (\downarrow 4.81)$ | $62.55_{\pm 0.06}\ (\downarrow 3.43)$ | $60.81_{\pm 0.16}\ (\downarrow 5.17)$ | $60.82_{\pm 0.05}\ (\downarrow 5.16)$ |
| $T = 15$ | $\mathbf{66.59}_{\pm 0.03}$ | $61.48_{\pm 0.06}\ (\downarrow 5.11)$ | $62.55_{\pm 0.08}\ (\downarrow 4.04)$ | $60.82_{\pm 0.16}\ (\downarrow 5.77)$ | $60.85_{\pm 0.05}\ (\downarrow 5.74)$ |
| $T = 20$ | $\mathbf{66.95}_{\pm 0.04}$ | $61.82_{\pm 0.10}\ (\downarrow 5.13)$ | $62.55_{\pm 0.09}\ (\downarrow 4.40)$ | $60.82_{\pm 0.16}\ (\downarrow 6.13)$ | $60.84_{\pm 0.12}\ (\downarrow 6.11)$ |
| $T = 25$ | $\mathbf{67.40}_{\pm 0.10}$ | $61.95_{\pm 0.11}\ (\downarrow 5.45)$ | $62.61_{\pm 0.05}\ (\downarrow 4.79)$ | $60.82_{\pm 0.16}\ (\downarrow 6.58)$ | $60.83_{\pm 0.07}\ (\downarrow 6.57)$ |
| $T = 30$ | $\mathbf{67.71}_{\pm 0.06}$ | $62.12_{\pm 0.12}\ (\downarrow 5.59)$ | $62.60_{\pm 0.09}\ (\downarrow 5.11)$ | $60.82_{\pm 0.16}\ (\downarrow 6.89)$ | $60.77_{\pm 0.04}\ (\downarrow 6.94)$ |
| $T = 35$ | $\mathbf{67.97}_{\pm 0.12}$ | $62.26_{\pm 0.13}\ (\downarrow 5.71)$ | $62.63_{\pm 0.11}\ (\downarrow 5.34)$ | $60.83_{\pm 0.16}\ (\downarrow 7.14)$ | $60.81_{\pm 0.06}\ (\downarrow 7.16)$ |
| $T = 40$ | $\mathbf{68.19}_{\pm 0.07}$ | $62.28_{\pm 0.06}\ (\downarrow 5.91)$ | $62.64_{\pm 0.08}\ (\downarrow 5.55)$ | $60.82_{\pm 0.15}\ (\downarrow 7.37)$ | $60.88_{\pm 0.05}\ (\downarrow 7.31)$ |
| $T = 45$ | $\mathbf{68.32}_{\pm 0.11}$ | $62.37_{\pm 0.12}\ (\downarrow 5.95)$ | $62.63_{\pm 0.12}\ (\downarrow 5.69)$ | $60.83_{\pm 0.16}\ (\downarrow 7.49)$ | $60.89_{\pm 0.11}\ (\downarrow 7.43)$ |
| $T = 50$ | $\mathbf{68.51}_{\pm 0.09}$ | $62.39_{\pm 0.13}\ (\downarrow 6.12)$ | $62.64_{\pm 0.11}\ (\downarrow 5.87)$ | $60.84_{\pm 0.16}\ (\downarrow 7.67)$ | $60.89_{\pm 0.06}\ (\downarrow 7.62)$ |

Table 16: Ablation study of our DeepAFL's training accuracy on the Tiny-ImageNet. We evaluate four distinct ablation models to explore the contributions of our key components in the representations: (1) Ablating the residual skip connection, (2) Ablating the random projection $\mathbf{B}_t$, (3) Ablating the activation function $\sigma(\cdot)$, and (4) Ablating the trainable transformation $\mathbf{\Omega}_{t+1}$. The value in parentheses ($\downarrow$) indicates the performance drop compared to our full DeepAFL under identical conditions.

| Layers | DeepAFL | Ablation (1) | Ablation (2) | Ablation (3) | Ablation (4) |
|---|---|---|---|---|---|
| $T = 0$ | $\mathbf{60.30}_{\pm 0.02}$ | $60.30_{\pm 0.02}\ (\downarrow 0.00)$ | $60.30_{\pm 0.02}\ (\downarrow 0.00)$ | $60.30_{\pm 0.02}\ (\downarrow 0.00)$ | $60.30_{\pm 0.02}\ (\downarrow 0.00)$ |
| $T = 5$ | $\mathbf{66.89}_{\pm 0.06}$ | $60.84_{\pm 0.03}\ (\downarrow 6.05)$ | $63.21_{\pm 0.03}\ (\downarrow 3.68)$ | $60.30_{\pm 0.01}\ (\downarrow 6.59)$ | $60.31_{\pm 0.02}\ (\downarrow 6.58)$ |
| $T = 10$ | $\mathbf{70.14}_{\pm 0.03}$ | $61.87_{\pm 0.05}\ (\downarrow 8.27)$ | $63.46_{\pm 0.03}\ (\downarrow 6.68)$ | $60.30_{\pm 0.01}\ (\downarrow 9.84)$ | $60.31_{\pm 0.01}\ (\downarrow 9.83)$ |
| $T = 15$ | $\mathbf{72.64}_{\pm 0.05}$ | $62.53_{\pm 0.04}\ (\downarrow 10.1)$ | $63.59_{\pm 0.03}\ (\downarrow 9.05)$ | $60.31_{\pm 0.01}\ (\downarrow 12.3)$ | $60.32_{\pm 0.02}\ (\downarrow 12.3)$ |
| $T = 20$ | $\mathbf{74.75}_{\pm 0.02}$ | $62.97_{\pm 0.03}\ (\downarrow 11.8)$ | $63.68_{\pm 0.03}\ (\downarrow 11.1)$ | $60.31_{\pm 0.01}\ (\downarrow 14.4)$ | $60.33_{\pm 0.02}\ (\downarrow 14.4)$ |
| $T = 25$ | $\mathbf{76.58}_{\pm 0.03}$ | $63.30_{\pm 0.02}\ (\downarrow 13.3)$ | $63.74_{\pm 0.03}\ (\downarrow 12.8)$ | $60.31_{\pm 0.01}\ (\downarrow 16.3)$ | $60.33_{\pm 0.03}\ (\downarrow 16.3)$ |
| $T = 30$ | $\mathbf{78.31}_{\pm 0.02}$ | $63.58_{\pm 0.04}\ (\downarrow 14.7)$ | $63.77_{\pm 0.03}\ (\downarrow 14.5)$ | $60.31_{\pm 0.01}\ (\downarrow 18.0)$ | $60.35_{\pm 0.02}\ (\downarrow 18.0)$ |
| $T = 35$ | $\mathbf{79.87}_{\pm 0.03}$ | $63.76_{\pm 0.02}\ (\downarrow 16.1)$ | $63.80_{\pm 0.03}\ (\downarrow 16.1)$ | $60.31_{\pm 0.01}\ (\downarrow 19.6)$ | $60.36_{\pm 0.02}\ (\downarrow 19.5)$ |
| $T = 40$ | $\mathbf{81.32}_{\pm 0.11}$ | $63.85_{\pm 0.02}\ (\downarrow 17.5)$ | $63.82_{\pm 0.02}\ (\downarrow 17.5)$ | $60.31_{\pm 0.01}\ (\downarrow 21.0)$ | $60.35_{\pm 0.02}\ (\downarrow 21.0)$ |
| $T = 45$ | $\mathbf{82.62}_{\pm 0.04}$ | $63.98_{\pm 0.03}\ (\downarrow 18.6)$ | $63.84_{\pm 0.02}\ (\downarrow 18.8)$ | $60.31_{\pm 0.01}\ (\downarrow 22.3)$ | $60.35_{\pm 0.02}\ (\downarrow 22.3)$ |
| $T = 50$ | $\mathbf{83.82}_{\pm 0.09}$ | $64.07_{\pm 0.01}\ (\downarrow 19.8)$ | $63.86_{\pm 0.01}\ (\downarrow 20.0)$ | $60.31_{\pm 0.01}\ (\downarrow 23.5)$ | $60.36_{\pm 0.01}\ (\downarrow 23.5)$ |

Table 17: Ablation study of our DeepAFL's testing accuracy on the Tiny-ImageNet. We evaluate four distinct ablation models to explore the contributions of our key components in the representations: (1) Ablating the residual skip connection, (2) Ablating the random projection $\mathbf{B}_t$, (3) Ablating the activation function $\sigma(\cdot)$, and (4) Ablating the trainable transformation $\mathbf{\Omega}_{t+1}$. The value in parentheses ($\downarrow$) indicates the performance drop compared to our full DeepAFL under identical conditions.

| Layers | DeepAFL | Ablation (1) | Ablation (2) | Ablation (3) | Ablation (4) |
|---|---|---|---|---|---|
| $T = 0$ | $\mathbf{56.72}_{\pm 0.14}$ | $56.72_{\pm 0.14}\ (\downarrow 0.00)$ | $56.72_{\pm 0.14}\ (\downarrow 0.00)$ | $56.72_{\pm 0.14}\ (\downarrow 0.00)$ | $56.72_{\pm 0.14}\ (\downarrow 0.00)$ |
| $T = 5$ | $\mathbf{60.24}_{\pm 0.02}$ | $57.39_{\pm 0.11}\ (\downarrow 2.85)$ | $58.16_{\pm 0.24}\ (\downarrow 2.08)$ | $56.69_{\pm 0.13}\ (\downarrow 3.55)$ | $56.69_{\pm 0.11}\ (\downarrow 3.55)$ |
| $T = 10$ | $\mathbf{61.34}_{\pm 0.13}$ | $58.42_{\pm 0.16}\ (\downarrow 2.92)$ | $58.22_{\pm 0.17}\ (\downarrow 3.12)$ | $56.70_{\pm 0.13}\ (\downarrow 4.64)$ | $56.69_{\pm 0.14}\ (\downarrow 4.65)$ |
| $T = 15$ | $\mathbf{61.94}_{\pm 0.08}$ | $59.01_{\pm 0.10}\ (\downarrow 2.93)$ | $58.20_{\pm 0.14}\ (\downarrow 3.74)$ | $56.70_{\pm 0.13}\ (\downarrow 5.24)$ | $56.69_{\pm 0.12}\ (\downarrow 5.25)$ |
| $T = 20$ | $\mathbf{62.34}_{\pm 0.02}$ | $59.21_{\pm 0.08}\ (\downarrow 3.13)$ | $58.25_{\pm 0.19}\ (\downarrow 4.09)$ | $56.70_{\pm 0.13}\ (\downarrow 5.64)$ | $56.68_{\pm 0.12}\ (\downarrow 5.66)$ |
| $T = 25$ | $\mathbf{62.66}_{\pm 0.05}$ | $59.37_{\pm 0.14}\ (\downarrow 3.29)$ | $58.25_{\pm 0.16}\ (\downarrow 4.41)$ | $56.70_{\pm 0.14}\ (\downarrow 5.96)$ | $56.70_{\pm 0.10}\ (\downarrow 5.96)$ |
| $T = 30$ | $\mathbf{62.94}_{\pm 0.03}$ | $59.49_{\pm 0.10}\ (\downarrow 3.45)$ | $58.24_{\pm 0.17}\ (\downarrow 4.70)$ | $56.70_{\pm 0.14}\ (\downarrow 6.24)$ | $56.70_{\pm 0.12}\ (\downarrow 6.24)$ |
| $T = 35$ | $\mathbf{63.24}_{\pm 0.16}$ | $59.54_{\pm 0.07}\ (\downarrow 3.70)$ | $58.27_{\pm 0.18}\ (\downarrow 4.97)$ | $56.70_{\pm 0.14}\ (\downarrow 6.54)$ | $56.72_{\pm 0.12}\ (\downarrow 6.52)$ |
| $T = 40$ | $\mathbf{63.48}_{\pm 0.18}$ | $59.66_{\pm 0.08}\ (\downarrow 3.82)$ | $58.34_{\pm 0.16}\ (\downarrow 5.14)$ | $56.70_{\pm 0.14}\ (\downarrow 6.78)$ | $56.73_{\pm 0.12}\ (\downarrow 6.75)$ |
| $T = 45$ | $\mathbf{63.65}_{\pm 0.18}$ | $59.72_{\pm 0.10}\ (\downarrow 3.93)$ | $58.35_{\pm 0.18}\ (\downarrow 5.30)$ | $56.70_{\pm 0.14}\ (\downarrow 6.95)$ | $56.72_{\pm 0.10}\ (\downarrow 6.93)$ |
| $T = 50$ | $\mathbf{63.74}_{\pm 0.04}$ | $59.81_{\pm 0.11}\ (\downarrow 3.93)$ | $58.39_{\pm 0.15}\ (\downarrow 5.35)$ | $56.70_{\pm 0.14}\ (\downarrow 7.04)$ | $56.72_{\pm 0.08}\ (\downarrow 7.02)$ |

Table 18: Training accuracy comparison of our DeepAFL against the other four deepening strategies for the analytic networks on the CIFAR-10, including (1) cascades w/ random feature, (2) cascades w/ random feature & label encoding, (3) cascades w/ activated random feature, (4) cascades w/ activated random feature & label encoding. All the strategies share the same zero-layer features to ensure fairness. The value in parentheses ($\downarrow$) indicates the performance lags behind our DeepAFL.

| Layers | DeepAFL | Strategy (1) | Strategy (2) | Strategy (3) | Strategy (4) |
|---|---|---|---|---|---|
| $T=0$ | $85.35_{\pm 0.02}$ | $85.35_{\pm 0.02} (\downarrow 0.00)$ | $85.35_{\pm 0.02} (\downarrow 0.00)$ | $85.35_{\pm 0.02} (\downarrow 0.00)$ | $85.35_{\pm 0.02} (\downarrow 0.00)$ |
| $T=5$ | $89.03_{\pm 0.03}$ | $85.35_{\pm 0.02} (\downarrow 3.68)$ | $85.36_{\pm 0.03} (\downarrow 3.67)$ | $85.68_{\pm 0.01} (\downarrow 3.35)$ | $85.68_{\pm 0.02} (\downarrow 3.35)$ |
| $T=10$ | $90.61_{\pm 0.05}$ | $85.35_{\pm 0.03} (\downarrow 5.26)$ | $85.36_{\pm 0.03} (\downarrow 5.25)$ | $85.68_{\pm 0.01} (\downarrow 4.93)$ | $85.67_{\pm 0.02} (\downarrow 4.94)$ |
| $T=15$ | $91.80_{\pm 0.04}$ | $85.35_{\pm 0.03} (\downarrow 6.45)$ | $85.35_{\pm 0.03} (\downarrow 6.45)$ | $85.70_{\pm 0.00} (\downarrow 6.10)$ | $85.68_{\pm 0.03} (\downarrow 6.12)$ |
| $T=20$ | $92.74_{\pm 0.03}$ | $85.35_{\pm 0.03} (\downarrow 7.39)$ | $85.35_{\pm 0.03} (\downarrow 7.39)$ | $85.70_{\pm 0.01} (\downarrow 7.04)$ | $85.67_{\pm 0.01} (\downarrow 7.07)$ |
| $T=25$ | $93.49_{\pm 0.02}$ | $85.35_{\pm 0.03} (\downarrow 8.14)$ | $85.35_{\pm 0.03} (\downarrow 8.14)$ | $85.71_{\pm 0.01} (\downarrow 7.78)$ | $85.68_{\pm 0.02} (\downarrow 7.81)$ |
| $T=30$ | $94.10_{\pm 0.03}$ | $85.36_{\pm 0.03} (\downarrow 8.74)$ | $85.35_{\pm 0.03} (\downarrow 8.75)$ | $85.71_{\pm 0.01} (\downarrow 8.39)$ | $85.67_{\pm 0.03} (\downarrow 8.43)$ |
| $T=35$ | $94.63_{\pm 0.03}$ | $85.36_{\pm 0.03} (\downarrow 9.27)$ | $85.35_{\pm 0.03} (\downarrow 9.28)$ | $85.71_{\pm 0.01} (\downarrow 8.92)$ | $85.68_{\pm 0.03} (\downarrow 8.95)$ |
| $T=40$ | $95.08_{\pm 0.03}$ | $85.35_{\pm 0.02} (\downarrow 9.73)$ | $85.35_{\pm 0.03} (\downarrow 9.73)$ | $85.71_{\pm 0.01} (\downarrow 9.37)$ | $85.68_{\pm 0.03} (\downarrow 9.40)$ |
| $T=45$ | $95.43_{\pm 0.03}$ | $85.35_{\pm 0.03} (\downarrow 10.1)$ | $85.35_{\pm 0.03} (\downarrow 10.1)$ | $85.71_{\pm 0.00} (\downarrow 9.72)$ | $85.69_{\pm 0.03} (\downarrow 9.74)$ |
| $T=50$ | $95.79_{\pm 0.03}$ | $85.36_{\pm 0.03} (\downarrow 10.4)$ | $85.35_{\pm 0.03} (\downarrow 10.4)$ | $85.70_{\pm 0.01} (\downarrow 10.1)$ | $85.68_{\pm 0.03} (\downarrow 10.1)$ |

Table 19: Testing accuracy comparison of our DeepAFL against the other four deepening strategies for the analytic networks on the CIFAR-10, including (1) cascades w/ random feature, (2) cascades w/ random feature & label encoding, (3) cascades w/ activated random feature, (4) cascades w/ activated random feature & label encoding. All the strategies share the same zero-layer features to ensure fairness. The value in parentheses ($\downarrow$) indicates the performance lags behind our DeepAFL.

| Layers | DeepAFL | Strategy (1) | Strategy (2) | Strategy (3) | Strategy (4) |
|---|---|---|---|---|---|
| $T=0$ | $83.29_{\pm 0.20}$ | $83.30_{\pm 0.20} (\downarrow 0.00)$ | $83.30_{\pm 0.20} (\downarrow 0.00)$ | $83.30_{\pm 0.20} (\downarrow 0.00)$ | $83.30_{\pm 0.20} (\downarrow 0.00)$ |
| $T=5$ | $85.13_{\pm 0.08}$ | $83.30_{\pm 0.22} (\downarrow 1.83)$ | $83.31_{\pm 0.24} (\downarrow 1.82)$ | $83.62_{\pm 0.20} (\downarrow 1.51)$ | $83.61_{\pm 0.20} (\downarrow 1.52)$ |
| $T=10$ | $85.89_{\pm 0.09}$ | $83.29_{\pm 0.23} (\downarrow 2.60)$ | $83.31_{\pm 0.23} (\downarrow 2.58)$ | $83.63_{\pm 0.23} (\downarrow 2.26)$ | $83.60_{\pm 0.20} (\downarrow 2.29)$ |
| $T=15$ | $86.18_{\pm 0.08}$ | $83.32_{\pm 0.23} (\downarrow 2.86)$ | $83.31_{\pm 0.23} (\downarrow 2.87)$ | $83.62_{\pm 0.20} (\downarrow 2.56)$ | $83.60_{\pm 0.19} (\downarrow 2.58)$ |
| $T=20$ | $86.34_{\pm 0.06}$ | $83.29_{\pm 0.24} (\downarrow 3.05)$ | $83.31_{\pm 0.24} (\downarrow 3.03)$ | $83.62_{\pm 0.22} (\downarrow 2.72)$ | $83.62_{\pm 0.23} (\downarrow 2.72)$ |
| $T=25$ | $86.40_{\pm 0.06}$ | $83.29_{\pm 0.24} (\downarrow 3.11)$ | $83.31_{\pm 0.23} (\downarrow 3.09)$ | $83.63_{\pm 0.22} (\downarrow 2.77)$ | $83.63_{\pm 0.21} (\downarrow 2.77)$ |
| $T=30$ | $86.46_{\pm 0.06}$ | $83.31_{\pm 0.24} (\downarrow 3.15)$ | $83.29_{\pm 0.23} (\downarrow 3.17)$ | $83.64_{\pm 0.22} (\downarrow 2.82)$ | $83.60_{\pm 0.22} (\downarrow 2.86)$ |
| $T=35$ | $86.72_{\pm 0.06}$ | $83.30_{\pm 0.23} (\downarrow 3.42)$ | $83.31_{\pm 0.24} (\downarrow 3.41)$ | $83.64_{\pm 0.22} (\downarrow 3.08)$ | $83.59_{\pm 0.20} (\downarrow 3.13)$ |
| $T=40$ | $86.76_{\pm 0.06}$ | $83.31_{\pm 0.25} (\downarrow 3.45)$ | $83.32_{\pm 0.24} (\downarrow 3.44)$ | $83.66_{\pm 0.20} (\downarrow 3.10)$ | $83.62_{\pm 0.20} (\downarrow 3.14)$ |
| $T=45$ | $86.77_{\pm 0.06}$ | $83.31_{\pm 0.24} (\downarrow 3.46)$ | $83.30_{\pm 0.24} (\downarrow 3.47)$ | $83.65_{\pm 0.23} (\downarrow 3.12)$ | $83.60_{\pm 0.22} (\downarrow 3.17)$ |
| $T=50$ | $86.72_{\pm 0.06}$ | $83.31_{\pm 0.23} (\downarrow 3.41)$ | $83.30_{\pm 0.24} (\downarrow 3.42)$ | $83.64_{\pm 0.22} (\downarrow 3.08)$ | $83.60_{\pm 0.22} (\downarrow 3.12)$ |

Table 20: Training accuracy comparison of our DeepAFL against the other four deepening strategies for the analytic networks on the CIFAR-100, including (1) cascades w/ random feature, (2) cascades w/ random feature & label encoding, (3) cascades w/ activated random feature, (4) cascades w/ activated random feature & label encoding. All the strategies share the same zero-layer features to ensure fairness. The value in parentheses ($\downarrow$) indicates the performance lags behind our DeepAFL.

| Layers | DeepAFL | Strategy (1) | Strategy (2) | Strategy (3) | Strategy (4) |
|---|---|---|---|---|---|
| $T=0$ | $65.66_{\pm 0.04}$ | $65.66_{\pm 0.04} (\downarrow 0.00)$ | $65.66_{\pm 0.04} (\downarrow 0.00)$ | $65.66_{\pm 0.04} (\downarrow 0.00)$ | $65.66_{\pm 0.04} (\downarrow 0.00)$ |
| $T=5$ | $74.93_{\pm 0.04}$ | $65.64_{\pm 0.06} (\downarrow 9.29)$ | $65.58_{\pm 0.07} (\downarrow 9.35)$ | $66.18_{\pm 0.08} (\downarrow 8.75)$ | $65.30_{\pm 0.05} (\downarrow 9.63)$ |
| $T=10$ | $79.41_{\pm 0.08}$ | $65.64_{\pm 0.05} (\downarrow 13.8)$ | $65.58_{\pm 0.06} (\downarrow 13.8)$ | $66.67_{\pm 0.08} (\downarrow 12.7)$ | $65.39_{\pm 0.06} (\downarrow 14.0)$ |
| $T=15$ | $82.69_{\pm 0.09}$ | $65.64_{\pm 0.05} (\downarrow 17.0)$ | $65.58_{\pm 0.06} (\downarrow 17.1)$ | $67.11_{\pm 0.11} (\downarrow 15.6)$ | $65.51_{\pm 0.04} (\downarrow 17.2)$ |
| $T=20$ | $85.15_{\pm 0.03}$ | $65.64_{\pm 0.06} (\downarrow 19.5)$ | $65.58_{\pm 0.07} (\downarrow 19.6)$ | $67.52_{\pm 0.11} (\downarrow 17.6)$ | $65.60_{\pm 0.07} (\downarrow 19.6)$ |
| $T=25$ | $87.21_{\pm 0.02}$ | $65.64_{\pm 0.05} (\downarrow 21.6)$ | $65.57_{\pm 0.07} (\downarrow 21.6)$ | $67.86_{\pm 0.13} (\downarrow 19.4)$ | $65.70_{\pm 0.12} (\downarrow 21.5)$ |
| $T=30$ | $88.84_{\pm 0.06}$ | $65.64_{\pm 0.05} (\downarrow 23.2)$ | $65.56_{\pm 0.07} (\downarrow 23.3)$ | $68.14_{\pm 0.11} (\downarrow 20.7)$ | $65.74_{\pm 0.12} (\downarrow 23.1)$ |
| $T=35$ | $90.14_{\pm 0.06}$ | $65.64_{\pm 0.05} (\downarrow 24.5)$ | $65.58_{\pm 0.06} (\downarrow 24.6)$ | $68.40_{\pm 0.12} (\downarrow 21.7)$ | $65.84_{\pm 0.09} (\downarrow 24.3)$ |
| $T=40$ | $91.29_{\pm 0.09}$ | $65.64_{\pm 0.05} (\downarrow 25.6)$ | $65.58_{\pm 0.05} (\downarrow 25.7)$ | $68.64_{\pm 0.14} (\downarrow 22.6)$ | $65.95_{\pm 0.15} (\downarrow 25.3)$ |
| $T=45$ | $92.15_{\pm 0.11}$ | $65.64_{\pm 0.05} (\downarrow 26.5)$ | $65.58_{\pm 0.07} (\downarrow 26.6)$ | $68.82_{\pm 0.16} (\downarrow 23.3)$ | $66.04_{\pm 0.17} (\downarrow 26.1)$ |
| $T=50$ | $92.93_{\pm 0.06}$ | $65.64_{\pm 0.05} (\downarrow 27.3)$ | $65.59_{\pm 0.07} (\downarrow 27.3)$ | $68.99_{\pm 0.19} (\downarrow 23.9)$ | $66.16_{\pm 0.13} (\downarrow 26.8)$ |

Table 21: Testing accuracy comparison of our DeepAFL against the other four deepening strategies for the analytic networks on the CIFAR-100, including (1) cascades w/ random feature, (2) cascades w/ random feature & label encoding, (3) cascades w/ activated random feature, (4) cascades w/ activated random feature & label encoding. All the strategies share the same zero-layer features to ensure fairness. The value in parentheses ($\downarrow$) indicates the performance lags behind our DeepAFL.

| Layers | DeepAFL | Strategy (1) | Strategy (2) | Strategy (3) | Strategy (4) |
|---|---|---|---|---|---|
| $T = 0$ | $\mathbf{60.81}_{\pm 0.15}$ | $60.81_{\pm 0.15}\,(\downarrow 0.00)$ | $60.81_{\pm 0.15}\,(\downarrow 0.00)$ | $60.81_{\pm 0.15}\,(\downarrow 0.00)$ | $60.81_{\pm 0.15}\,(\downarrow 0.00)$ |
| $T = 5$ | $\mathbf{64.62}_{\pm 0.02}$ | $60.80_{\pm 0.13}\,(\downarrow 3.82)$ | $60.75_{\pm 0.14}\,(\downarrow 3.87)$ | $61.23_{\pm 0.07}\,(\downarrow 3.39)$ | $60.57_{\pm 0.04}\,(\downarrow 4.05)$ |
| $T = 10$ | $\mathbf{65.98}_{\pm 0.13}$ | $60.78_{\pm 0.10}\,(\downarrow 5.20)$ | $60.74_{\pm 0.11}\,(\downarrow 5.24)$ | $61.67_{\pm 0.09}\,(\downarrow 4.31)$ | $60.69_{\pm 0.05}\,(\downarrow 5.29)$ |
| $T = 15$ | $\mathbf{66.59}_{\pm 0.03}$ | $60.78_{\pm 0.12}\,(\downarrow 5.81)$ | $60.74_{\pm 0.11}\,(\downarrow 5.85)$ | $61.98_{\pm 0.03}\,(\downarrow 4.61)$ | $60.85_{\pm 0.02}\,(\downarrow 5.74)$ |
| $T = 20$ | $\mathbf{66.95}_{\pm 0.04}$ | $60.79_{\pm 0.10}\,(\downarrow 6.16)$ | $60.73_{\pm 0.11}\,(\downarrow 6.22)$ | $62.21_{\pm 0.02}\,(\downarrow 4.74)$ | $60.95_{\pm 0.07}\,(\downarrow 6.00)$ |
| $T = 25$ | $\mathbf{67.40}_{\pm 0.10}$ | $60.78_{\pm 0.10}\,(\downarrow 6.62)$ | $60.72_{\pm 0.11}\,(\downarrow 6.68)$ | $62.31_{\pm 0.04}\,(\downarrow 5.09)$ | $60.99_{\pm 0.06}\,(\downarrow 6.41)$ |
| $T = 30$ | $\mathbf{67.71}_{\pm 0.06}$ | $60.78_{\pm 0.12}\,(\downarrow 6.93)$ | $60.73_{\pm 0.11}\,(\downarrow 6.98)$ | $62.53_{\pm 0.08}\,(\downarrow 5.18)$ | $61.12_{\pm 0.08}\,(\downarrow 6.59)$ |
| $T = 35$ | $\mathbf{67.97}_{\pm 0.12}$ | $60.78_{\pm 0.12}\,(\downarrow 7.19)$ | $60.72_{\pm 0.10}\,(\downarrow 7.25)$ | $62.73_{\pm 0.12}\,(\downarrow 5.24)$ | $61.16_{\pm 0.06}\,(\downarrow 6.81)$ |
| $T = 40$ | $\mathbf{68.19}_{\pm 0.07}$ | $60.78_{\pm 0.12}\,(\downarrow 7.41)$ | $60.74_{\pm 0.10}\,(\downarrow 7.45)$ | $62.82_{\pm 0.11}\,(\downarrow 5.37)$ | $61.20_{\pm 0.01}\,(\downarrow 6.99)$ |
| $T = 45$ | $\mathbf{68.32}_{\pm 0.11}$ | $60.78_{\pm 0.11}\,(\downarrow 7.54)$ | $60.73_{\pm 0.11}\,(\downarrow 7.59)$ | $62.96_{\pm 0.11}\,(\downarrow 5.36)$ | $61.27_{\pm 0.01}\,(\downarrow 7.05)$ |
| $T = 50$ | $\mathbf{68.51}_{\pm 0.09}$ | $60.77_{\pm 0.12}\,(\downarrow 7.74)$ | $60.71_{\pm 0.10}\,(\downarrow 7.80)$ | $63.05_{\pm 0.11}\,(\downarrow 5.46)$ | $61.37_{\pm 0.06}\,(\downarrow 7.14)$ |

Table 22: Training accuracy comparison of our DeepAFL against the other four deepening strategies for the analytic networks on the Tiny-ImageNet, including (1) cascades w/ random feature, (2) cascades w/ random feature & label encoding, (3) cascades w/ activated random feature, (4) cascades w/ activated random feature & label encoding. All the strategies share the same zero-layer features to ensure fairness. The value in parentheses ($\downarrow$) indicates the performance lags behind our DeepAFL.

| Layers | DeepAFL | Strategy (1) | Strategy (2) | Strategy (3) | Strategy (4) |
|---|---|---|---|---|---|
| $T = 0$ | $\mathbf{60.30}_{\pm 0.02}$ | $60.30_{\pm 0.02}\,(\downarrow 0.00)$ | $60.30_{\pm 0.02}\,(\downarrow 0.00)$ | $60.30_{\pm 0.02}\,(\downarrow 0.00)$ | $60.30_{\pm 0.02}\,(\downarrow 0.00)$ |
| $T = 5$ | $\mathbf{66.89}_{\pm 0.06}$ | $60.20_{\pm 0.02}\,(\downarrow 6.69)$ | $59.84_{\pm 0.02}\,(\downarrow 7.05)$ | $59.92_{\pm 0.03}\,(\downarrow 6.97)$ | $56.14_{\pm 0.08}\,(\downarrow 10.8)$ |
| $T = 10$ | $\mathbf{70.14}_{\pm 0.03}$ | $60.20_{\pm 0.01}\,(\downarrow 9.94)$ | $59.83_{\pm 0.01}\,(\downarrow 10.3)$ | $60.00_{\pm 0.02}\,(\downarrow 10.1)$ | $54.78_{\pm 0.07}\,(\downarrow 15.4)$ |
| $T = 15$ | $\mathbf{72.64}_{\pm 0.05}$ | $60.21_{\pm 0.01}\,(\downarrow 12.4)$ | $59.82_{\pm 0.04}\,(\downarrow 12.8)$ | $60.08_{\pm 0.03}\,(\downarrow 12.6)$ | $54.40_{\pm 0.07}\,(\downarrow 18.2)$ |
| $T = 20$ | $\mathbf{74.75}_{\pm 0.02}$ | $60.20_{\pm 0.01}\,(\downarrow 14.5)$ | $59.81_{\pm 0.01}\,(\downarrow 14.9)$ | $60.17_{\pm 0.04}\,(\downarrow 14.6)$ | $54.23_{\pm 0.11}\,(\downarrow 20.5)$ |
| $T = 25$ | $\mathbf{76.58}_{\pm 0.03}$ | $60.19_{\pm 0.01}\,(\downarrow 16.4)$ | $59.80_{\pm 0.02}\,(\downarrow 16.8)$ | $60.25_{\pm 0.04}\,(\downarrow 16.3)$ | $54.12_{\pm 0.11}\,(\downarrow 22.5)$ |
| $T = 30$ | $\mathbf{78.31}_{\pm 0.02}$ | $60.21_{\pm 0.01}\,(\downarrow 18.1)$ | $59.81_{\pm 0.04}\,(\downarrow 18.5)$ | $60.33_{\pm 0.04}\,(\downarrow 18.0)$ | $54.03_{\pm 0.12}\,(\downarrow 24.3)$ |
| $T = 35$ | $\mathbf{79.87}_{\pm 0.03}$ | $60.21_{\pm 0.01}\,(\downarrow 19.7)$ | $59.81_{\pm 0.02}\,(\downarrow 20.1)$ | $60.44_{\pm 0.07}\,(\downarrow 19.4)$ | $54.06_{\pm 0.12}\,(\downarrow 25.8)$ |
| $T = 40$ | $\mathbf{81.32}_{\pm 0.11}$ | $60.20_{\pm 0.01}\,(\downarrow 21.1)$ | $59.80_{\pm 0.03}\,(\downarrow 21.5)$ | $60.52_{\pm 0.05}\,(\downarrow 20.8)$ | $53.99_{\pm 0.12}\,(\downarrow 27.3)$ |
| $T = 45$ | $\mathbf{82.62}_{\pm 0.04}$ | $60.20_{\pm 0.01}\,(\downarrow 22.4)$ | $59.80_{\pm 0.03}\,(\downarrow 22.8)$ | $60.62_{\pm 0.06}\,(\downarrow 22.0)$ | $54.07_{\pm 0.08}\,(\downarrow 28.6)$ |
| $T = 50$ | $\mathbf{83.82}_{\pm 0.09}$ | $60.20_{\pm 0.02}\,(\downarrow 23.6)$ | $59.78_{\pm 0.03}\,(\downarrow 24.0)$ | $60.71_{\pm 0.05}\,(\downarrow 23.1)$ | $54.05_{\pm 0.11}\,(\downarrow 29.8)$ |

Table 23: Testing accuracy comparison of our DeepAFL against the other four deepening strategies for the analytic networks on the Tiny-ImageNet, including (1) cascades w/ random feature, (2) cascades w/ random feature & label encoding, (3) cascades w/ activated random feature, (4) cascades w/ activated random feature & label encoding. All the strategies share the same zero-layer features to ensure fairness. The value in parentheses ($\downarrow$) indicates the performance lags behind our DeepAFL.

| Layers | DeepAFL | Strategy (1) | Strategy (2) | Strategy (3) | Strategy (4) |
|---|---|---|---|---|---|
| $T = 0$ | $\mathbf{56.72}_{\pm 0.14}$ | $56.72_{\pm 0.14}\,(\downarrow 0.00)$ | $56.72_{\pm 0.14}\,(\downarrow 0.00)$ | $56.72_{\pm 0.14}\,(\downarrow 0.00)$ | $56.72_{\pm 0.14}\,(\downarrow 0.00)$ |
| $T = 5$ | $\mathbf{60.24}_{\pm 0.02}$ | $56.69_{\pm 0.12}\,(\downarrow 3.55)$ | $56.32_{\pm 0.10}\,(\downarrow 3.92)$ | $56.31_{\pm 0.10}\,(\downarrow 3.93)$ | $53.26_{\pm 0.12}\,(\downarrow 6.98)$ |
| $T = 10$ | $\mathbf{61.34}_{\pm 0.13}$ | $56.61_{\pm 0.13}\,(\downarrow 4.73)$ | $56.26_{\pm 0.11}\,(\downarrow 5.08)$ | $56.40_{\pm 0.06}\,(\downarrow 4.94)$ | $52.01_{\pm 0.06}\,(\downarrow 9.33)$ |
| $T = 15$ | $\mathbf{61.94}_{\pm 0.08}$ | $56.62_{\pm 0.14}\,(\downarrow 5.32)$ | $56.25_{\pm 0.13}\,(\downarrow 5.69)$ | $56.49_{\pm 0.10}\,(\downarrow 5.45)$ | $51.74_{\pm 0.12}\,(\downarrow 10.2)$ |
| $T = 20$ | $\mathbf{62.34}_{\pm 0.02}$ | $56.64_{\pm 0.14}\,(\downarrow 5.70)$ | $56.23_{\pm 0.08}\,(\downarrow 6.11)$ | $56.61_{\pm 0.10}\,(\downarrow 5.73)$ | $51.63_{\pm 0.09}\,(\downarrow 10.7)$ |
| $T = 25$ | $\mathbf{62.66}_{\pm 0.05}$ | $56.62_{\pm 0.14}\,(\downarrow 6.04)$ | $56.27_{\pm 0.09}\,(\downarrow 6.39)$ | $56.66_{\pm 0.12}\,(\downarrow 6.00)$ | $51.66_{\pm 0.16}\,(\downarrow 11.0)$ |
| $T = 30$ | $\mathbf{62.94}_{\pm 0.03}$ | $56.63_{\pm 0.09}\,(\downarrow 6.31)$ | $56.23_{\pm 0.09}\,(\downarrow 6.71)$ | $56.75_{\pm 0.21}\,(\downarrow 6.19)$ | $51.47_{\pm 0.16}\,(\downarrow 11.5)$ |
| $T = 35$ | $\mathbf{63.24}_{\pm 0.16}$ | $56.61_{\pm 0.14}\,(\downarrow 6.63)$ | $56.22_{\pm 0.04}\,(\downarrow 7.02)$ | $56.79_{\pm 0.15}\,(\downarrow 6.45)$ | $51.51_{\pm 0.21}\,(\downarrow 11.7)$ |
| $T = 40$ | $\mathbf{63.48}_{\pm 0.18}$ | $56.62_{\pm 0.12}\,(\downarrow 6.86)$ | $56.26_{\pm 0.02}\,(\downarrow 7.22)$ | $56.83_{\pm 0.24}\,(\downarrow 6.65)$ | $51.47_{\pm 0.15}\,(\downarrow 12.0)$ |
| $T = 45$ | $\mathbf{63.65}_{\pm 0.18}$ | $56.61_{\pm 0.13}\,(\downarrow 7.04)$ | $56.22_{\pm 0.09}\,(\downarrow 7.43)$ | $56.99_{\pm 0.20}\,(\downarrow 6.66)$ | $51.42_{\pm 0.09}\,(\downarrow 12.2)$ |
| $T = 50$ | $\mathbf{63.74}_{\pm 0.04}$ | $56.61_{\pm 0.14}\,(\downarrow 7.13)$ | $56.23_{\pm 0.13}\,(\downarrow 7.51)$ | $57.11_{\pm 0.18}\,(\downarrow 6.63)$ | $51.47_{\pm 0.12}\,(\downarrow 12.3)$ |

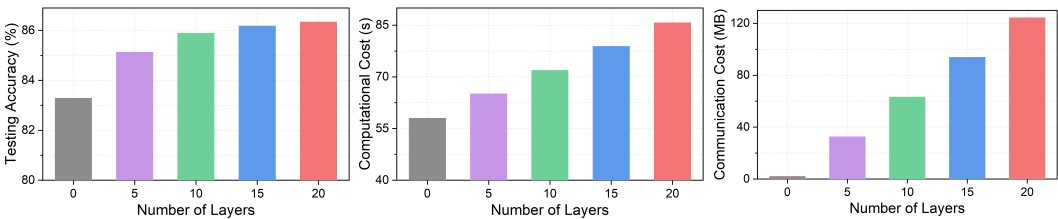

(a) Testing Accuracy v.s. Layers     (b) Computational Cost v.s. Layers     (c) Communication Cost v.s. Layers

Figure 4: Accuracy-Efficiency balance of our DeepAFL on the CIFAR-10.

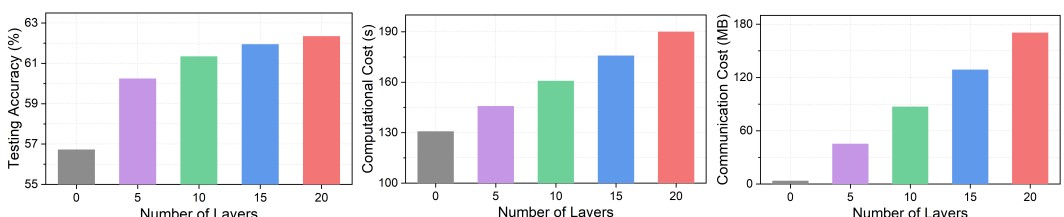

(a) Testing Accuracy v.s. Layers     (b) Computational Cost v.s. Layers     (c) Communication Cost v.s. Layers

Figure 5: Accuracy-Efficiency balance of our DeepAFL on the CIFAR-100.

(a) Testing Accuracy v.s. Layers     (b) Computational Cost v.s. Layers     (c) Communication Cost v.s. Layers

Figure 6: Accuracy-Efficiency balance of our DeepAFL on the Tiny-ImageNet.

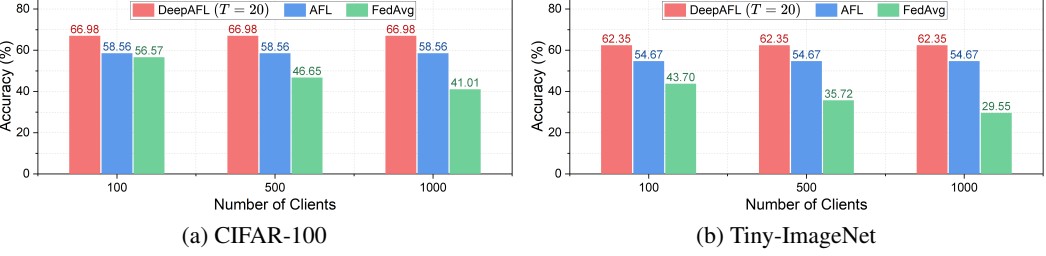

(a) CIFAR-100                  (b) Tiny-ImageNet

Figure 7: Scalability analyses of our DeepAFL against baselines with different numbers of clients.

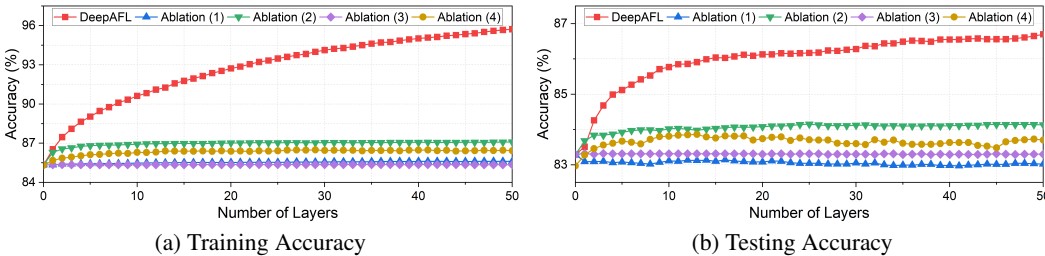

(a) Training Accuracy             (b) Testing Accuracy

Figure 8: Ablation study of our DeepAFL on the CIFAR-10.

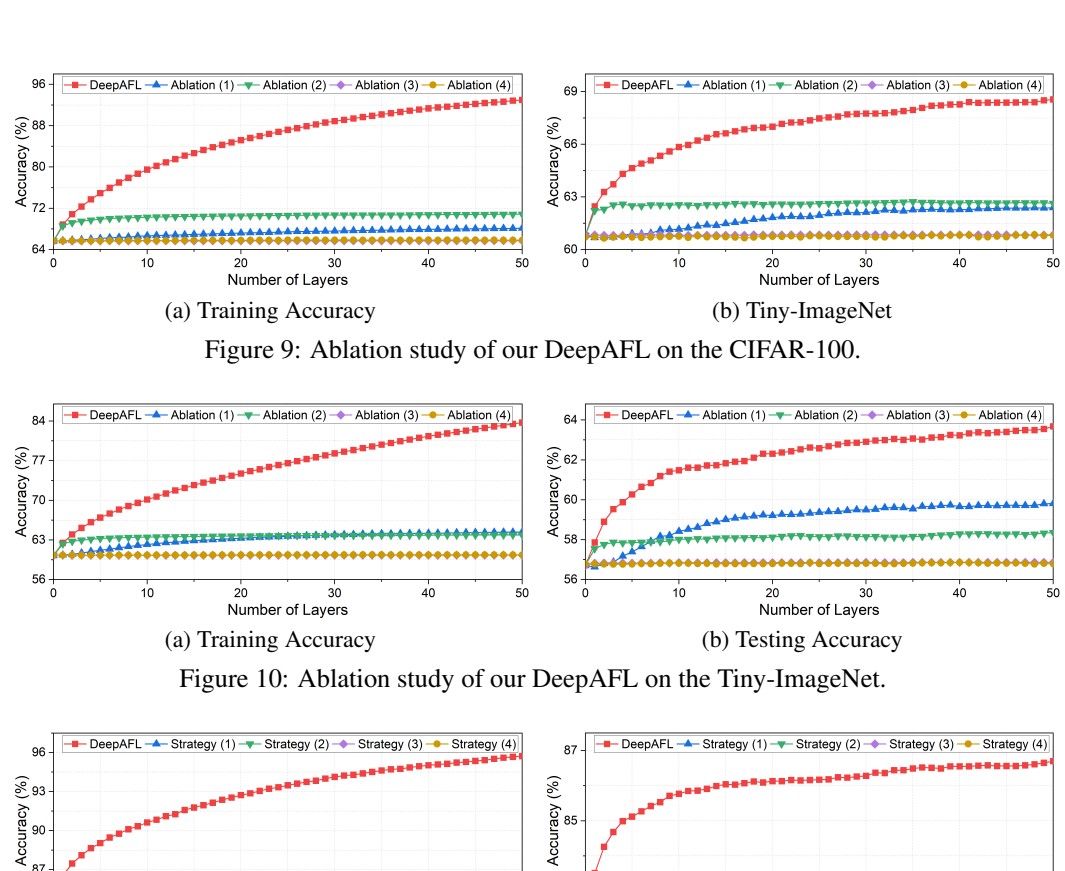

(a) Training Accuracy

(b) Tiny-ImageNet

Figure 9: Ablation study of our DeepAFL on the CIFAR-100.

(a) Training Accuracy

(b) Testing Accuracy

Figure 10: Ablation study of our DeepAFL on the Tiny-ImageNet.

(a) Training Accuracy

(b) Testing Accuracy

Figure 11: Comparison of our DeepAFL with other deepening strategies on the CIFAR-10.

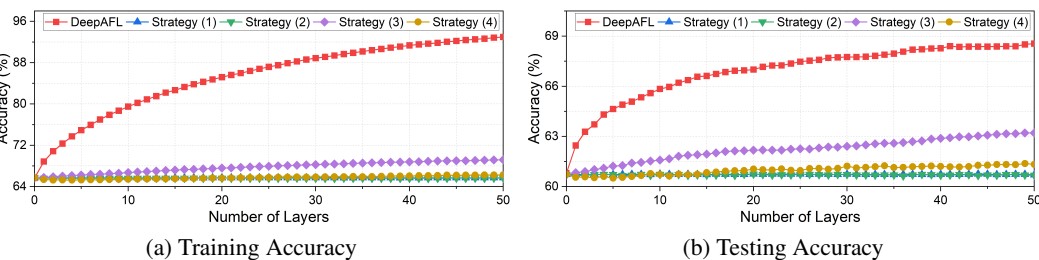

(a) Training Accuracy

(b) Testing Accuracy

Figure 12: Comparison of our DeepAFL with other deepening strategies on the CIFAR-100.

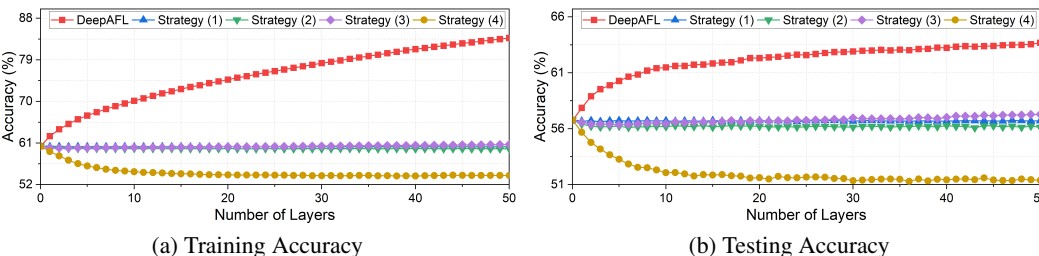

(a) Training Accuracy

(b) Testing Accuracy

Figure 13: Comparison of our DeepAFL with other deepening strategies on the Tiny-ImageNet.

### E.2 Implementation Details of the Experiments

Here, we detail the implementation details of our experiments. Specifically, the number of clients is set to 100 for all methods. To simulate diverse data heterogeneity scenarios, two common non-IID partitioning settings are employed: LDA (Lin et al., 2020) (as Non-IID-1) and Sharding (Lin et al., 2020) (as Non-IID-2). Under the Non-IID-1 setting, the dataset is allocated to all clients via the Dirichlet distribution, with the parameter $\alpha$ modulating the heterogeneity level. Under the Non-IID-2 setting, the dataset is sorted by label and divided into equal-sized shards for distribution among clients, where the number of shards per client $s$ controls the heterogeneity level. Smaller values of $\alpha$ and $s$ both indicate more heterogeneous data distributions. We set $\alpha \in \{0.1, 0.05\}$ and $s \in \{2, 4\}$ for CIFAR-10, while setting $\alpha \in \{0.1, 0.01\}$ and $s \in \{5, 10\}$ for CIFAR-100 and Tiny-ImageNet.

To ensure a fair comparison, in our main experiments, the results of all baselines are directly taken from the provided data in AFL (He et al., 2025). Furthermore, for all three datasets (i.e., CIFAR-10, CIFAR-100, and Tiny-ImageNet), we adopted exactly the same pre-processing procedure as AFL, resizing all input images to $224 \times 224$. Regarding the specific parameters of our DeepAFL, we adopt the following *default settings*. First, we employ GELU as the activation function and set the projection dimensions to $d_\Phi = d_F = 1024$ across all three datasets. Additionally, the regularization parameter $\lambda$ is set to 10 for CIFAR-10, and 1 for both CIFAR-100 and Tiny-ImageNet. Then, for the regularization parameter $\gamma$, we set it to 0.1 for CIFAR-10, and 0.01 for both CIFAR-100 and Tiny-ImageNet. To ensure experimental transparency, we provide the detailed parameter settings for each experiment in the extended analyses of our evaluation in Section E.3.

As for the metrics employed in our experiments, we adopt the top-1 accuracy on the testing sets as the primary metric for evaluating the performance of all approaches. In addition, since the top-1 accuracy on the training sets also serves as a strong indicator of the representation learning capability, we extensively employ this metric to analyze how the representations of different models evolve with increasing network depth. Moreover, to assess the efficiency of all approaches, we measure the required training time (s) and the volume of transmitted data (MB) as indicators of computational and communication cost, respectively. To analyze the marginal effect of increasing network depth, we use the symbol $\Delta$ to denote the difference between two consecutive cases of the the metric.

All the experiments in our paper are executed three times per setting, and we report the mean and standard error of the experimental results. Furthermore, all the experimental evaluations in this paper are conducted using PyTorch on NVIDIA RTX 4090 GPUs. Notably, for transparency, the related codes will be made publicly accessible as open-sourced upon the acceptance of this paper, allowing the broader research community to verify our findings and build upon our work.

### E.3 Detailed Analyses on Experimental Evaluations

#### E.3.1 Main Comparisons

In our main results reported in Tables 1 and 2, we employ GELU as the activation function and set the projection dimensions to $d_\Phi = d_F = 1024$ across all three datasets. The regularization parameter $\lambda$ is fixed at 10 for CIFAR-10 and at 1 for both CIFAR-100 and Tiny-ImageNet. Similarly, the regularization parameter $\gamma$ is set to 0.1 for CIFAR-10 and 0.01 for CIFAR-100 and Tiny-ImageNet. To ensure a fair comparison, the results of all baselines in Tables 1 and 2 are directly taken from the benchmark data provided in AFL (He et al., 2025). It is worth noting that, since our DeepAFL involves random projections, its performance may vary under different random seeds. To mitigate the influence of randomness, all main experiments in this paper were repeated three times, and the results are reported as $\mathrm{Mean} \pm \mathrm{Standard\ Error}$. The improvements of our DeepAFL were validated by Chi-squared tests, all of which were found to be statistically significant at the $p = 0.05$ level.

From the results, we observe that the performance of DeepAFL is consistent across different levels of heterogeneity, which empirically supports the heterogeneity invariance in Theorem 1. Moreover, as the network depth $T$ increases, DeepAFL consistently achieves significant performance gains, empirically supporting its representation learning capability in Theorems 2–3. It is also worth noting that although we report the performance of DeepAFL under $T \in \{5, 10, 20\}$ in the main results, its performance continues to improve as $T$ increases. This is further confirmed by our extended experiments with $T \in [0, 50]$. Specifically, on the more complex CIFAR-100 and Tiny-ImageNet datasets, the testing accuracy of DeepAFL at $T = 50$ surpasses that at $T = 20$ by more than $1.5\%$.

### E.3.2  INVARIANCE ANALYSES

Here, we conduct detailed analyses to demonstrate DeepAFL's ideal property of invariance to data heterogeneity. As shown in Tables 1 and 2, the performance of all gradient-based methods deteriorates markedly as data heterogeneity increases (i.e., as the partitioning parameters $\alpha$ or $s$ decrease). In contrast, due to its ideal property of invariance to data heterogeneity, our DeepAFL maintains stable performance. This property of DeepAFL can be further extended to invariance with respect to the number of clients. As shown in Figure 7, the performance of DeepAFL remains entirely consistent across all scenarios, while its advantage over gradient-based methods (e.g., FedAvg) becomes increasingly pronounced as the number of clients $K$ grows. Quantitatively, at $K = 100$, DeepAFL achieves performance gains of $10.41\%$ and $18.65\%$ over FedAvg on the CIFAR-100 and Tiny-ImageNet, respectively, which expand to $25.97\%$ and $32.8\%$ at $K = 1000$. Notably, the results of AFL and FedAvg in Figure 7 are also taken from the given data in AFL (He et al., 2025).

### E.3.3  REPRESENTATION ANALYSES

Here, we provide a comprehensive analysis of DeepAFL's capability for deep representation learning. To this end, we examine how its training accuracy and testing accuracy evolve as the number of layers $T$ increases, as shown in Tables 9–11. As observed, on the simple CIFAR-10 dataset, DeepAFL's performance improves steadily, reaching optimality at $T = 45$. On the complex CIFAR-100 and Tiny-ImageNet datasets, it scales without overfitting up to $T = 50$, yielding over $1.5\%$ improvement relative to $T = 20$. This indicates that for datasets with higher complexity, deeper models can be constructed to enhance representation capacity and achieve superior performance. Moreover, a particularly interesting and evident observation is that AFL exhibits consistently low training accuracy, which can likely be attributed to the underfitting limitations of its simple single-layer linear model. In contrast, DeepAFL is able to significantly improve training accuracy by deepening the network, showing that its representation learning capability can effectively overcome the underfitting issues in traditional analytic learning. In fact, our DeepAFL only requires minimal computational overhead when increasing depth, with more detailed analyses presented in Appendix E.3.4.

### E.3.4  EFFICIENCY EVALUATIONS

Here, we present the comprehensive efficiency evaluations of our DeepAFL. To highlight its superior balance between accuracy and efficiency, Figures 2–3 compare DeepAFL with baselines in terms of computational and communication cost. Specifically, to ensure a fair comparison, we selected the baseline with the lowest computational and communication cost (i.e., FedAvg) as the representative benchmark to highlight the superiority of our DeepAFL. Notably, the computational costs of all baselines are directly adopted from the existing results reported in AFL (He et al., 2025). Moreover, Tables 9–11 and Figures 4–6 provide further details on how our DeepAFL achieves a balance between accuracy and efficiency through the flexible adjustment of $T$.

Specifically, as shown in Figures 2–3, compared with gradient-based baselines, our DeepAFL (with $T = 20$) achieves at least a $99.7\%$ reduction in computational cost and a $50.2\%$ reduction in communication cost on the CIFAR-100. On the Tiny-ImageNet, the advantages of DeepAFL correspond to a $99.6\%$ reduction in computational cost and a $70.1\%$ reduction in communication cost. By flexibly adjusting the number of layers $T$, for example, setting $T = 5$, users can obtain greater efficiency advantages. In summary, although the cost of our DeepAFL is inevitably higher than that of AFL due to the additional deep layers required for enhanced representations, it remains substantially lower than that of gradient-based baselines. This advantage stems from our DeepAFL's gradient-free manner, which avoids the iterative costly computation and communication associated with gradients.

Subsequently, we also report the detailed accuracy-efficiency balance of our DeepAFL with varying numbers of layers, as shown in Tables 9–11 and Figures 4–6. Encouragingly, the construction of each additional layer in DeepAFL requires no more than 3s across all three datasets, providing an intuitive demonstration of its efficiency. Compared with AFL, even when building a 50-layer deep network, DeepAFL incurs less than a twofold increase in training time while delivering performance improvements of at most up to $31.38\%$ on the training set and $9.95\%$ on the testing set. It is worth noting that, there is considerable potential for our DeepAFL to further reduce communication cost in the future through compression techniques such as matrix factorization, as its primary communication information is the *Auto-Correlation* and *Cross-Correlation* Matrices.

### E.3.5 PARAMETER ANALYSES

Here, we provide comprehensive analyses for the parameter sensitivity of our DeepAFL, encompassing the regularization parameters $\lambda$, $\gamma$, the activation function $\sigma(\cdot)$, and the projection dimensions $d_\Phi$, $d_F$. Specifically, we keep all parameters' default settings, varying only the parameter under investigation to assess its sensitivity. The specific analyses for each parameter are detailed below:

First, we explore the regularization parameters $\lambda$ and $\gamma$, which regularize the global classifier and the transformation matrix, respectively. As illustrated in Tables 3–4, DeepAFL is insensitive to $\lambda$ but sensitive to $\gamma$. This evidence may stem from the fact that our deep residual network employs only the final global classifier during inference, yet relies on all transformation matrices. Specifically, on the simple CIFAR-10 dataset, the best performance is achieved when $\lambda = 10$ and $\gamma \in [0.1, 0.5]$. For the complex CIFAR-100 and Tiny-ImageNet datasets, the optimal values of $\lambda$ are found to be highly dispersed, while the optimal $\gamma$ consistently occurs at the smaller value of $\gamma = 0.01$.

Second, we further analyze the impact of different activation functions on DeepAFL's performance. The detailed experimental results are shown in Table 5. The selection of the activation function markedly influences DeepAFL's performance. More specifically, a performance variance of about 2% can be observed among different activation functions, with GELU emerging as the optimal choice and Softshrink as the least effective. Moreover, compared to omitting the activation function entirely, employing GELU yields up to 5% gains, underscoring its critical role. More comprehensive analysis of this significance is available in the corresponding ablation study in Appendix E.3.6.

Third, we study the effects of varying projection dimensions $d_\Phi$ and $d_F$ on our DeepAFL in Tables 6–8. As these projection dimensions increase, DeepAFL's performance rises initially before declining. This behavior can be attributed to the fact that, while larger dimensions can enhance the model's expressive power by capturing more information, they also concurrently increase the propensity for overfitting and compromise numerical stability. Notably, the model crashes when $d_\Phi = d_F = 2^{13}$. Furthermore, as detailed in Appendix D, the complexity of DeepAFL scales at least quadratically with these dimensions, and excessively large values incur substantial overhead, cautioning against the indiscriminate pursuit of larger dimensions. That's why we select $d_\Phi = d_F = 2^{10}$ as default.

### E.3.6 ABLATION STUDIES

Here, we present the detailed analyses of our ablation studies to dissect the individual contributions of residual skip connections, random projections, activation functions, and trainable transformations in our DeepAFL. Specifically, we construct four ablation models, with each omitting one of these key components, and compare them against our full DeepAFL across varying layer depths on diverse datasets, as shown in Tables 12–17 and Figures 8–10. Moreover, the ablation studies are conducted under identical conditions for fair comparisons, with all parameters aligned with those used in the main experiments. The detailed analyses are provided below:

First, we focus on analyzing the contribution of the residual skip connections in our DeepAFL, with the corresponding ablation model denoted as Ablation (1). For the simple CIFAR-10, ablating the residual connections prevents performance improvements as the network depth increases, as shown in Tables 12–13 and Figure 8, indicating that this dataset is highly sensitive to such skip connections. On the CIFAR-100 and Tiny-ImageNet datasets, performance continues to improve even without skip connections, exhibiting an initial rise followed by convergence, as shown in Tables 14–17 and Figures 9–10. Meanwhile, the ablation model attains respectable performance on Tiny-ImageNet, yet exhibits markedly limited gains on CIFAR-100. These disparities across datasets may stem from their differing complexities, with CIFAR-10, CIFAR-100, and Tiny-ImageNet containing 10, 100, and 200 classes, respectively. Notably, compared to DeepAFL, the ablation model yields substantially inferior performance across all cases. This degradation stems from the fact that, without skip connections, the model encounters an information bottleneck akin to that in gradient-based deep learning. Crucially, the importance of residual skip connections is theoretically grounded, as their ablation would invalidate Theorems 2 and 3. Specifically, the residual skip connections are crucial because they enable a special case: if the transformation matrix is a zero matrix, no update is applied to the representation features. This functionality guarantees that the representation features learned at each layer will be at least as good as those from the preceding layer. Thus, without the residual skip connections, each layer essentially constructs new representation features from the hidden random features, thereby forfeiting these valuable theoretical guarantees.

Second, we further analyze the contributions of the random projections, which give the ***stochasticity*** to our DeepAFL. Ablating the random projections $\{\mathbf{B}_i\}_{i=1}^{T}$ can be achieved by simply setting them to the identity matrices, and we denote this ablated model as Ablation (2). While this ablated model still yields performance improvements across all datasets, its performance growth rate is markedly slower compared to DeepAFL, and it reaches convergence earlier. Specifically, at $T = 10$, the ablation model reaches convergence across all three datasets, exhibiting performance gaps of $3.64\%$ to $9.25\%$ on the training set and $1.90\%$ to $3.43\%$ on the testing set compared to our DeepAFL. Subsequently, DeepAFL's performance continues to improve with increasing layers, further widening the performance gap. By $T = 50$, this gap further expands to $8.68\%$ to $22.2\%$ on the training set and $2.70\%$ to $5.87\%$ on the testing set. In Figures 8–10, we can observe that without random projections to introduce the essential ***stochasticity***, the model tends to converge to local optima or saddle points, even though the projection itself does not alter the feature dimensionality, as $d_\Phi = d_F = 2^{10}$.

Third, we focus on analyzing the contributions of the activation function in our DeepAFL, which introduces ***nonlinearity*** to our DeepAFL. The corresponding ablation model is named Ablation (3). It can be observed that the performance of this ablation model remains nearly invariant as the layer depth $T$ increases. Specifically, the performance fluctuation from $T = 0$ to $T = 50$ consistently falls below $0.05\%$ across all datasets, thereby underscoring the critical importance of the activation function in DeepAFL. This is because, without the activation function, each residual block merely performs a purely linear transformation on the feature representations, fundamentally limiting the model's representative power and preventing further performance improvement with added depth.

Fourth, we study the contributions of the trainable transformation $\mathbf{\Omega}_{t+1}$, which imparts ***learnability*** to our DeepAFL. The corresponding ablation model is denoted as ablation (4). As illustrated in Figures 8–10, this ablation model exhibits modest performance gains followed by rapid convergence on the simple CIFAR-10, while showing nearly no improvement on the complex CIFAR-100 and Tiny-ImageNet. This evidence arises because, without the trainable transformation for ***learnability***, each residual block essentially functions as a fixed random nonlinear feature updater. For the simple CIFAR-10, these updaters can fortuitously render random features progressively more discriminative to a limited extent. For complex CIFAR-100 and Tiny-ImageNet, relying solely on random feature updates fails to capture the high-level features, leading to negligible performance improvement.

### E.3.7 COMPARING DEEPAFL WITH OTHER DEEPENING STRATEGIES

Here, we further compare our DeepAFL with the other four alternative deepening strategies to highlight its superiority. Specifically, we design four distinct deepening strategies: (1) cascades with random features, (2) cascades with random features and label encoding, (3) cascades with activated random features, and (4) cascades with activated random features and label encoding. Notably, the strategies (1) and (3) here are similar to the naive approaches (a) and (b) in Figure 1. However, in our comparison, strategies (1) and (3) additionally incorporate random projection and activation after the backbone to form the zero-layer features $\mathbf{\Phi}_0$, thereby aligning more closely with our DeepAFL at the starting point to ensure fairness in comparison. These two naive strategies are consistent with most existing attempts in the literature to deepen analytic networks (Low et al., 2019). Meanwhile, strategies (2) and (4) are essentially extensions of (1) and (3) through the incorporation of label encoding, which is an established technique in analytic learning that introduces an additional linear mapping for the labels at each layer to facilitate the training of deep analytic networks (Zhuang et al., 2021; 2025). The detailed experimental results are presented in Tables 18–23 and Figures 11–13.

From the results, our DeepAFL consistently outperforms all other deepening strategies across all datasets, thereby underscoring its superiority. Across all datasets, strategies (1) and (2) achieve very similar performance, exhibiting nearly no improvement with an increasing number of layers. Next, we turn our attention to strategies (3) and (4), which differ in that strategy (4) incorporates additional label encoding. On the CIFAR-10, strategies (3) and (4) yield comparable performance, both showing some improvement followed by rapid convergence. On the CIFAR-100, strategy (4) lags substantially behind strategy (3), despite both showing gains with increasing layer depth. This disparity widens further on the Tiny-ImageNet, where strategy (4) experiences a pronounced performance decline from added depth. The varying effects of label encoding across datasets may stem from differences in the number of classes, as these datasets contain 10, 100, and 200 classes, respectively. Taken together, these results suggest that existing deepening strategies are of very limited effectiveness and fall far short of the performance achieved by our proposed DeepAFL.

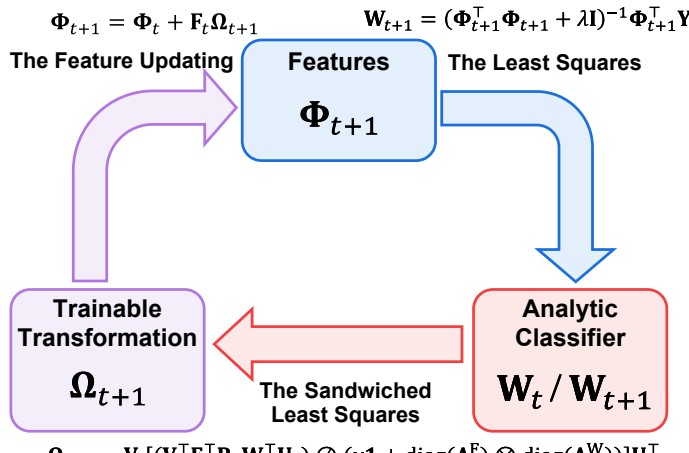

Figure 14: The training process for each layer in our DeepAFL.

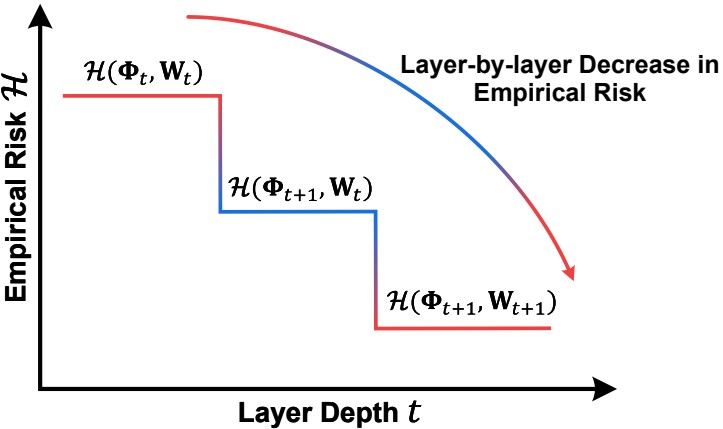

Figure 15: Layer-wise reduction of empirical risk in our DeepAFL.

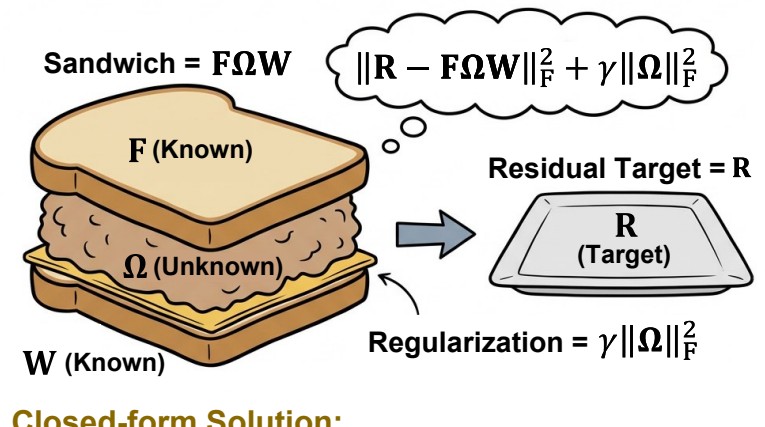

Figure 16: The sandwiched least-squares process in our DeepAFL.

Table 24: Performance comparisons of the top-1 accuracy (%) among our DeepAFL and two new baselines (i.e., FedAWA (Shi et al., 2025) and (Liu et al., 2024a)) on CIFAR-10, CIFAR-100, and Tiny-ImageNet. The best result is highlighted in **bold**, and the second-best result is underlined.

| Baseline | CIFAR-10 | | CIFAR-100 | | Tiny-ImageNet | |
|---|---|---|---|---|---|---|
| | $\alpha = 0.1$ | $\alpha = 0.05$ | $\alpha = 0.1$ | $\alpha = 0.1$ | $\alpha = 0.1$ | $\alpha = 0.1$ |
| FedAWA (2025) | 79.42% | 76.37% | 58.94% | 50.48% | 59.07% | 52.33% |
| FedLPA (2024a) | 49.57% | 42.56% | 40.04% | 22.55% | 39.48% | 26.67% |
| AFL (2025) | 80.75% | 80.75% | 58.56% | 58.56% | 54.67% | 54.67% |
| DeepAFL ($T = 5$) | 85.20% | 85.20% | 64.72% | 64.72% | 60.31% | 60.31% |
| DeepAFL ($T = 10$) | 85.93% | 85.93% | 65.96% | 65.96% | 61.37% | 61.37% |
| DeepAFL ($T = 20$) | **86.43%** | **86.43%** | **66.98%** | **66.98%** | **62.35%** | **62.35%** |
| Improvement ↑ | **5.68%** | **5.68%** | **8.04%** | **8.42%** | **3.28%** | **7.68%** |

## F  MORE EXPLANATIONS ON THE DETAILED PROCESS OF DEEPAFL

Here, we provide comprehensive descriptions of our DeepAFL process to enhance clarity. Specifically, as illustrated in Figures 14–16, we elaborate on the training process of each layer, demonstrate the layer-by-layer empirical risk reduction, and visually depict the sandwiched least-squares mechanism that underpins our DeepAFL. Further details are presented below.

First of all, let's focus on the training process for each layer in our DeepAFL. As illustrated in Figure 14, each layer entails the sequential update and computation of the trainable transformation matrix $\boldsymbol{\Omega}_{t+1}$, the feature matrix $\boldsymbol{\Phi}_{t+1}$, and the analytic classifier $\mathbf{W}_{t+1}$. Specifically, for each $(t + 1)$-th layer, the trainable transformation matrix $\boldsymbol{\Omega}_{t+1}$ is initially derived through the sandwiched least squares process, utilizing the preceding analytic classifier $\mathbf{W}_t$ alongside $\mathbf{F}_t$ and $\mathbf{R}_t$. Subsequently, based on the newly acquired $\boldsymbol{\Omega}_{t+1}$, the feature matrix is updated to yield $\boldsymbol{\Phi}_{t+1}$ according to Equation (9). Finally, employing the derived feature matrix $\boldsymbol{\Phi}_{t+1}$, the analytic classifier is further updated to obtain $\mathbf{W}_{t+1}$ using the least squares process. Notably, as a special case, the zero-layer feature matrix $\boldsymbol{\Phi}_0$ is obtained directly via feature extraction using Equations (1) and (2), thereby eliminating the need to compute a trainable transformation matrix $\boldsymbol{\Omega}_0$. The aforementioned process is executed layer-by-layer, adhering to the update sequence $\boldsymbol{\Phi}_0 \mapsto \mathbf{W}_0 \mapsto \cdots \mapsto \mathbf{W}_{T-1} \mapsto \boldsymbol{\Omega}_T \mapsto \boldsymbol{\Phi}_T \mapsto \mathbf{W}_T$ until the network construction is complete.

Subsequently, we present intuitive and visual explanations of the monotonic decrease in empirical risk as the depth of our DeepAFL increases. The empirical risk $\mathcal{H}(\boldsymbol{\Phi}_t, \mathbf{W}_t)$ is determined by the feature matrix $\boldsymbol{\Phi}_t$ and the analytic classifier $\mathbf{W}_t$. As previously analyzed, within each layer, Deep-AFL sequentially updates the feature matrix and the analytic classifier to yield improved $\boldsymbol{\Phi}_{t+1}$ and $\mathbf{W}_{t+1}$. Consequently, the empirical risk undergoes two reductions within each layer, as depicted in Figure 15. We now detail these two reduction steps within each layer. Specifically, the feature matrix $\boldsymbol{\Phi}_t$ is first updated to obtain $\boldsymbol{\Phi}_{t+1}$ via the trainable transformation matrix $\boldsymbol{\Omega}_{t+1}$, which constitutes the optimal solution to Equation (7) for minimizing the empirical risk with respect to feature optimization. Thus, the updated empirical risk $\mathcal{H}(\boldsymbol{\Phi}_{t+1}, \mathbf{W}_t)$ is lower than the initial risk $\mathcal{H}(\boldsymbol{\Phi}_t, \mathbf{W}_t)$. Subsequently, with $\boldsymbol{\Phi}_{t+1}$ held fixed, the new analytic classifier $\mathbf{W}_{t+1}$ is computed as the optimal solution to Equation (4) for minimizing the local empirical risk. Thus, the final empirical risk $\mathcal{H}(\boldsymbol{\Phi}_{t+1}, \mathbf{W}_{t+1})$ is also lower than the risk $\mathcal{H}(\boldsymbol{\Phi}_{t+1}, \mathbf{W}_t)$. By executing this procedure layer-by-layer, the empirical risk is progressively reduced, thereby leading to enhanced performance.

Finally, as we term this optimization process for $\boldsymbol{\Omega}$ as the *sandwiched least squares problem*, we would like to provide a detailed illustration of its process. Specifically, the optimization objective in Equation (7) can be viewed as a special case of generalized Sylvester matrix equations (Wu et al., 2008; Ding et al., 2008; Duan, 2015). This structure is characterized by the unknown variable $\boldsymbol{\Omega}$ being *sandwiched* between two known matrices, $\mathbf{F}$ and $\mathbf{W}$. In this context, the "*sandwich*" structure, therefore, refers to this three-matrix product form $\mathbf{F}\boldsymbol{\Omega}\mathbf{W}$. The empirical risk minimization aims to minimize the residual, meaning the sandwich term should be as close as possible to the residual target $\mathbf{R}$. Meanwhile, the regularization term $\gamma\|\boldsymbol{\Omega}\|_F^2$ is applied to constrain the magnitude of $\boldsymbol{\Omega}$ and prevent it from becoming excessively large. Consequently, this particular structure facilitates the derivation of a distinct analytical solution, as presented in Equation (8).

Table 25: Accuracy of gradient-based baselines with varying layers on CIFAR-100.

| Layers | $T=1$ | $T=2$ | $T=5$ | $T=10$ | $T=20$ |
|---|---|---|---|---|---|
| FedAvg | 56.62% | 55.83% | <5% | <3% | <1% |
| FedDyn | 57.55% | 57.22% | 56.26% | <3% | <1% |

# G MORE EXPERIMENTAL RESULTS ON RECENT BASELINES

Following a reviewer's suggestion, we implemented two new baselines (i.e., FedAWA (Shi et al., 2025) and FedLPA (Liu et al., 2024a)) across the three benchmark datasets utilized in our main experiment: CIFAR-10, CIFAR-100, and ImageNet-R. All experimental settings were maintained to be consistent with the other baselines established in our manuscript. Specifically, because FedLPA is classified as a one-shot communication FL method, and we empirically observed that its performance essentially converges within 50 local epochs, we set the local epochs to 50 for this baseline. This setting allows us to mitigate unnecessary computational overhead for FedLPA, thereby more fully demonstrating its potential for efficiency. We present the results of these two new baselines alongside AFL and our DeepAFL for additional comparison, as shown in Table 24.

In terms of accuracy, FedAWA exhibits very strong results, approaching (on CIFAR-10) or even surpassing AFL (on CIFAR-100 and Tiny-ImageNet) when $\alpha = 0.1$. Notably, even though FedAWA outperforms AFL, our DeepAFL still consistently achieves the best performance across all scenarios. Furthermore, as a gradient-based one-shot method, FedLPA's performance is compromised by data heterogeneity, as it lacks the inherent invariance demonstrated by our DeepAFL. Consequently, despite our best efforts to tune its hyperparameters, FedLPA still performs poorly. Furthermore, as the degree of Non-IID data increases, a pronounced performance degradation is observed for both of the newly introduced gradient-based baselines (i.e., FedAWA and FedLPA). This observation further highlights the advantage of the inherent invariance property of our DeepAFL.

In terms of efficiency, using the CIFAR-100 dataset as an example, FedAWA requires approximately 10 hours, while FedLPA requires approximately 2.5 hours. In sharp contrast, our DeepAFL completes the task in less than 100 seconds (specifically, 91.74 seconds), achieving a speedup exceeding $90\times$. Notably, even though FedLPA is also classified as a one-shot FL approach, and we have already minimized its number of local epochs to fully reflect its efficiency potential, its overall overhead remains significantly higher than that of DeepAFL. This phenomenon is primarily due to FedLPA still being gradient-based and facing time-consuming backpropagation. Conversely, our DeepAFL is in a forward-only manner, further highlighting its efficiency advantage beyond its one-shot nature.

# H MORE EXPLANATIONS ON OUR DEEPAFL'S EXPERIMENTAL SETUP

Here, we would like to clarify our experimental setup. To maintain fairness in comparisons, the results for all baselines in our paper are directly employed from the benchmark provided in the original AFL paper (He et al., 2025). Specifically, the gradient-based approaches in the original AFL paper adhered to a similar setup as AFL, where the backbone network is entirely frozen during the FL process, and only the one-layer classifier is trained.

Moreover, to address potential concerns that gradient-based baselines might benefit from multiple trainable layers, we conduct additional experiments on FedAvg and FedDyn with multiple trainable layers ($T \in \{2, 5, 10, 20\}$) on CIFAR-100 under the Dirichlet distribution ($\alpha = 0.1$). For these experiments, the feature dimensions of the additional layers are aligned with those of our DeepAFL (i.e., 1024) to ensure consistency. Detailed results are presented in Table 25.

Through these results, we can observe that the performance of the gradient-based methods gradually and constantly degrades as $T$ increases. Specifically, when $T$ increases beyond a certain threshold (e.g., $T \geq 10$), the excessive number of parameters, combined with the Non-IID data in the FL scenario, may cause the training process to collapse. These findings fully indicate that directly increasing the number of trainable layers after the backbone for these baselines, similar to our DeepAFL setup, has a detrimental effect on their performance. Consequently, far from benefiting from multiple trainable layers, these gradient-based baselines exhibit significantly inferior performance under such configurations, thereby validating the propriety of our experimental setup.

# I  MORE ANALYSES ON DEEPAFL'S SCALABILITY WITH VIT BACKBONES

Table 26: Analyses of DeepAFL's scalability with the ViT-B-16-I-1K backbone on CIFAR-100.

| Layers | Testing Acc | $\Delta_{\text{Testing Acc}}$ | $\Delta_{\text{AFL Acc}}$ | Time Cost | $\Delta_{\text{Time Cost}}$ | $\Delta_{\text{AFL Cost}}$ |
|---|---|---|---|---|---|---|
| AFL | 75.45% | / | / | 107.19s | / | / |
| DeepAFL ($T = 5$) | 78.05% | 2.60% | 2.60% | 118.97s | 11.78s | 11.78s |
| DeepAFL ($T = 10$) | 79.01% | 0.96% | 3.56% | 126.55s | 7.58s | 19.37s |
| DeepAFL ($T = 15$) | 79.73% | 0.72% | 4.28% | 133.79s | 7.24s | 26.60s |
| DeepAFL ($T = 20$) | 80.13% | 0.40% | 4.68% | 141.02s | 7.23s | 33.83s |
| DeepAFL ($T = 25$) | 80.33% | 0.20% | 4.88% | 148.56s | 7.54s | 41.37s |
| DeepAFL ($T = 30$) | 80.51% | 0.18% | 5.06% | 156.21s | 7.65s | 49.02s |

Table 27: Analyses of DeepAFL's scalability with the ViT-B-16-I-21K backbone on CIFAR-100.

| Layers | Testing Acc | $\Delta_{\text{Testing Acc}}$ | $\Delta_{\text{AFL Acc}}$ | Time Cost | $\Delta_{\text{Time Cost}}$ | $\Delta_{\text{AFL Time}}$ |
|---|---|---|---|---|---|---|
| AFL | 86.35% | / | / | 107.19s | / | / |
| DeepAFL ($T = 5$) | 87.71% | 1.36% | 1.36% | 118.97s | 11.78s | 11.78s |
| DeepAFL ($T = 10$) | 88.15% | 0.44% | 1.80% | 126.55s | 7.58s | 19.37s |
| DeepAFL ($T = 15$) | 88.49% | 0.34% | 2.14% | 133.79s | 7.24s | 26.60s |
| DeepAFL ($T = 20$) | 88.80% | 0.31% | 2.45% | 141.02s | 7.23s | 33.83s |
| DeepAFL ($T = 25$) | 88.95% | 0.15% | 2.60% | 148.56s | 7.54s | 41.37s |
| DeepAFL ($T = 30$) | 89.08% | 0.13% | 2.73% | 156.21s | 7.65s | 49.02s |

Here, we present detailed analyses of DeepAFL's practical scalability with ViT backbones. Given that our DeepAFL is built upon AFL (He et al., 2025), which uses ResNet-18 as its primary backbone, we also employ the aligned backbone in our main experiments. This alignment facilitates a clearer and more transparent comparison between the two methods. Here, we further extend our evaluation by adopting larger ViT models to investigate our DeepAFL's practical scalability.

Specifically, we select two versions of the ViT-B-16 backbone: **ViT-B-16-I-1K** (pre-trained on the ImageNet-1K with approximately 1.28 million image) and **ViT-B-16-I-21K** (pre-trained on the larger ImageNet-21K dataset with approximately 14 million images). Owing to the significantly broader pre-training scale, ViT-B-16-I-21K typically yields better performance than ViT-B-16-I-1K. Since the backbone sizes of both ViT models are identical (Base-16), the runtime for our DeepAFL on both models remains essentially the same. Furthermore, all experiments are conducted on the CIFAR-100 dataset with 100 clients using a single NVIDIA RTX 4090 GPU.

The detailed results are presented in Tables 26–27, where "Testing Acc" and "Time Cost" represent the test accuracy and total wall-clock runtime. The $\Delta$ columns denote the step-wise change relative to the preceding row, while the $\Delta_{\text{AFL}}$ columns indicate the cumulative difference from AFL.

From these results, it is evident that applying DeepAFL to both ViT backbones leads to significant performance improvements with minimal time cost, thereby demonstrating its robust scalability. Specifically, at $T = 30$, our DeepAFL achieves an accuracy gain of 5.06% over AFL with the ViT-B-16-I-1K backbone, and an improvement of 2.73% with the ViT-B-16-I-21K backbone. Remarkably, these substantial improvements are achieved with a runtime of merely 156.21 s, which is only 49.02 s higher than AFL. It is important to note that this reported runtime encompasses the cumulative time consumed by all 100 clients to complete the training process across all layers on a single 4090 GPU. Thus, the average cost per client is less than 1.56 s. In stark contrast, even the simplest and most efficient FL baseline (e.g., FedAvg) requires a Total Wall-Clock Runtime of over 33,000 s.

Furthermore, although we observe that the performance improvement of DeepAFL exhibits diminishing marginal returns, this phenomenon is entirely expected and normal. First, as the backbone capability strengthens, the feature representations are already highly optimized, making it increasingly challenging to extract further performance gains beyond such a high baseline. Second, while increasing the number of layers enhances the fitting capacity of DeepAFL, the fixed volume of training data inevitably leads to a saturation limit in model performance. Notably, such diminishing returns are also common in gradient-based methods. Therefore, the ability of DeepAFL to achieve stable and significant improvements across various backbones without increasing any training data volume is highly noteworthy. Especially when considering the negligible runtime cost, the performance gains delivered by DeepAFL can be regarded as incurring virtually no time overhead.

## J MORE ANALYSES ON PARTIAL CLIENT PARTICIPATION

Table 28: Performance evaluation of DeepAFL on CIFAR-100 under partial client participation.

| Layers | Consistent Participation | | | | | | Inconsistent Participation | | | | | |
|---|---|---|---|---|---|---|---|---|---|---|---|---|
| | $\eta = 100\%$ | $\eta = 90\%$ | $\eta = 80\%$ | $\eta = 70\%$ | $\eta = 60\%$ | $\eta = 50\%$ | $\eta = 100\%$ | $\eta = 90\%$ | $\eta = 80\%$ | $\eta = 70\%$ | $\eta = 60\%$ | $\eta = 50\%$ |
| $T = 5$ | 64.72% | 64.53% | 64.38% | 63.92% | 62.85% | 61.54% | 64.72% | 64.21% | 64.03% | 64.05% | 62.19% | 61.36% |
| $T = 10$ | 65.96% | 65.46% | 65.26% | 64.49% | 64.76% | 63.17% | 65.96% | 65.36% | 65.05% | 64.45% | 64.09% | 62.93% |
| $T = 15$ | 66.56% | 66.39% | 66.20% | 65.38% | 65.17% | 65.08% | 66.56% | 66.22% | 66.17% | 65.64% | 64.60% | 64.98% |
| $T = 20$ | 66.98% | 66.76% | 66.66% | 66.47% | 66.27% | 65.14% | 66.98% | 66.57% | 66.51% | 66.32% | 65.50% | 64.81% |

Here, we provide detailed analyses of our DeepAFL's performance in scenarios involving partial client participation, thereby further highlighting its robustness. The detailed analyses are as follows.

First, let's focus on the minor mechanism adjustments required to accommodate partial client participation. As detailed in Section 3.2, constructing each new layer in DeepAFL entails two rounds of communication, corresponding to the computation of the classifier weights $\mathbf{W}_t$ and the trainable transformation matrix $\mathbf{\Omega}_{t+1}$. For notational convenience, let $\mathcal{S}$ denote the complete set of clients, where the total number of clients is $|\mathcal{S}| = K$. Consequently, the aggregation processes for $\mathbf{W}_t$ (i.e., Equation (12)) and $\mathbf{\Omega}_{t+1}$ (i.e., Equation (16)) can be expressed as:

$$\mathbf{G}_t^{1:K} = \mathbf{G}_t^{\mathcal{S}} = \sum_{k \in \mathcal{S}} \mathbf{G}_t^k, \quad \mathbf{H}_t^{1:K} = \mathbf{H}_t^{\mathcal{S}} = \sum_{k \in \mathcal{S}} \mathbf{H}_t^k, \quad (78)$$

$$\mathbf{\Pi}_t^{1:K} = \mathbf{\Pi}_t^{\mathcal{S}} = \sum_{k \in \mathcal{S}} \mathbf{\Pi}_t^k, \quad \mathbf{\Upsilon}_t^{1:K} = \mathbf{\Upsilon}_t^{\mathcal{S}} = \sum_{k \in \mathcal{S}} \mathbf{\Upsilon}_t^k. \quad (79)$$

When only a partial set of clients participates in the aggregation for DeepAFL, we denote the subset of clients contributing to the classifier construction at layer $t$ as $\mathcal{S}_t^{\mathbf{W}}$, and the subset contributing to the trainable transformation matrix construction as $\mathcal{S}_t^{\mathbf{\Omega}}$. Consequently, the aggregated matrices will be constructed using contributions from only these participating subsets:

$$\mathbf{G}_t^{\mathcal{S}_t^{\mathbf{W}}} = \sum_{k \in \mathcal{S}_t^{\mathbf{W}}} \mathbf{G}_t^k, \quad \mathbf{H}_t^{\mathcal{S}_t^{\mathbf{W}}} = \sum_{k \in \mathcal{S}_t^{\mathbf{W}}} \mathbf{H}_t^k, \quad (80)$$

$$\mathbf{\Pi}_t^{\mathcal{S}_t^{\mathbf{\Omega}}} = \sum_{k \in \mathcal{S}_t^{\mathbf{\Omega}}} \mathbf{\Pi}_t^k, \quad \mathbf{\Upsilon}_t^{\mathcal{S}_t^{\mathbf{\Omega}}} = \sum_{k \in \mathcal{S}_t^{\mathbf{\Omega}}} \mathbf{\Upsilon}_t^k. \quad (81)$$

Subsequently, the server utilizes these aggregated matrices from the partial participants to compute $\mathbf{W}_t$ and $\mathbf{\Omega}_{t+1}$ as before. Notably, since only a subset of clients contributed data, the effective full dataset should be redefined as the union of data held by all participating clients. Thus, the invariance property of our DeepAFL needs to be redefined as being identical to the centralized analytical solution over the effective full dataset comprising the data of the participating clients. In summary, our DeepAFL can handle partial participation easily and effectively without substantial procedural adjustments, while maintaining its analytical advantage of being invariant to data heterogeneity.

Second, we conducted experiments to evaluate the performance of our DeepAFL under varying client participation rates $\eta$. For simplicity, we assume that the participation rate remains consistent across aggregation processes within the same layer, i.e., $\eta = |\mathcal{S}_t^{\mathbf{W}}|/|\mathcal{S}| = |\mathcal{S}_t^{\mathbf{\Omega}}|/|\mathcal{S}|$. Furthermore, the client subsets $\mathcal{S}_t^{\mathbf{W}}$ and $\mathcal{S}_t^{\mathbf{\Omega}}$ corresponding to $\eta$ are randomly sampled from $\mathcal{S}$. Specifically, to ensure a comprehensive evaluation, we consider two distinct cases: (1) **Consistent Participation** and (2) **Inconsistent Participation**. These cases dictate whether the client subsets for the two aggregation processes within the same layer are identical or distinct, i.e., $\forall t, \mathcal{S}_t^{\mathbf{W}} = \mathcal{S}_t^{\mathbf{\Omega}}$ or $\forall t, \mathcal{S}_t^{\mathbf{W}} \neq \mathcal{S}_t^{\mathbf{\Omega}}$. Intuitively, the case (2) presents a greater challenge for DeepAFL, as the inconsistency within a single layer may heighten the risk of model instability.

Building upon these two cases, and using full participation ($\eta = 100\%$) as the control group, we conducted a broad range of analyses with $\eta$ ranging from $90\%$ to $50\%$, to thoroughly assess the performance variation of our DeepAFL on CIFAR-100. As shown in Table 28, our DeepAFL exhibits substantial robustness to partial client participation in both cases. Specifically, for a high participation rate of $\eta \geq 70\%$, the maximum accuracy degradation observed at $T = 20$ remains below 0.7% across both cases. Furthermore, even under the extremely challenging scenario of $\eta = 50\%$, where up to half of the clients drop out, DeepAFL still maintains high accuracy at $T = 20$, achieving 65.14% and 64.81% for both cases. Even at a low layer count ($T = 5$) and the most severe dropout rate ($\eta = 50\%$), the performance of our DeepAFL (61.54% and 61.36%) still significantly outperforms the AFL's accuracy (58.56%) achieved under the condition of 100% client participation. This underscores the strong robustness of DeepAFL in handling partial client participation scenarios.

## K   MORE DISCUSSIONS ON DEEPAFL'S ROBUSTNESS TO NOISES

Table 29: Performance evaluation of DeepAFL on CIFAR-100 under noisy training environments.

| Layers | $\tau = 0\%$ | $\tau = 10\%$ | $\tau = 20\%$ | $\tau = 30\%$ | $\tau = 40\%$ | $\tau = 50\%$ |
|---|---|---|---|---|---|---|
| DeepAFL ($T = 5$) | 64.72% | 64.20% | 63.63% | 63.25% | 62.51% | 60.91% |
| DeepAFL ($T = 10$) | 65.96% | 65.43% | 65.09% | 63.96% | 62.98% | 60.99% |
| DeepAFL ($T = 15$) | 66.56% | 66.11% | 65.66% | 64.29% | 63.15% | 60.97% |
| DeepAFL ($T = 20$) | 66.98% | 66.38% | 66.11% | 64.69% | 63.35% | 60.45% |

Here, we further demonstrate that our DeepAFL, utilizing the MSE loss, is less sensitive to noise compared to gradient-based methods employing the CE loss. Specifically, we elaborate on this point from both logical and experimental perspectives, as detailed below.

From the logical perspective, a crucial factor that makes a method sensitive to noise is overfitting. Specifically, if a method tends to overfit, then when there is even slight noise in the input data, it will tend to fit this noise. In contrast, the analytic learning methods using MSE loss are inherently less susceptible to overfitting, typically exhibiting a tendency toward underfitting instead. Indeed, addressing this underfitting limitation by augmenting the model's fitting capacity through deep representation learning is the primary motivation for proposing our DeepAFL.

Furthermore, Tables 9–11 in Appendix E.1 list the training and testing accuracies for AFL and our DeepAFL across various datasets, corroborating that analytic learning methods are not prone to overfitting. Taking the CIFAR-100 dataset as an example, the training set accuracy for AFL is only 61.55%, which is quite low, especially compared to its testing set accuracy of 58.56%. This phenomenon suggests that AFL suffers from severe underfitting. Moreover, when we gradually increase the depth $T$ from 0 to 50 in our DeepAFL, both training and testing accuracies rise steadily. The fact that testing accuracy improves commensurately with training accuracy confirms that DeepAFL avoids overfitting, even at depths of up to 50 layers.

More specifically, DeepAFL's ability to maintain superior representation learning without overfitting stems from the fact that its analytic classifier with MSE loss is linear. Because of this, its model complexity is quite low, making it very difficult to overfit and thus robust to noise. In contrast, CE loss is designed to induce steeper gradients to facilitate backpropagation optimization. While this characteristic accelerates training, it simultaneously renders gradient-based methods more prone to overfitting and hypersensitive to data noise. Consequently, noise robustness constitutes the inherent advantage of our DeepAFL (with MSE loss) compared to gradient-based methods (with CE loss).

From the experimental perspective, we further verify the noise robustness of our DeepAFL by explicitly simulating noisy training environments. Specifically, we simulate the noise in the training data by randomly flipping the labels of a proportion $\tau$ of the training data. The selected samples have their labels randomly changed to another class label. Using $\tau = 0\%$ as the control group, we conduct extensive analyses with $\tau \in \{10\%, 20\%, 30\%, 40\%, 50\%\}$ to thoroughly assess performance variations of our DeepAFL on CIFAR-100. The detailed results are reported in Table 29.

These results demonstrate that DeepAFL exhibits substantial robustness to data noise. Specifically, under moderate noise levels ($\tau \leq 20\%$), the maximum accuracy drop for our DeepAFL (at $T = 20$) is less than 1%. Furthermore, even under the extremely challenging scenario of $\tau = 50\%$, where half of the training labels are corrupted, DeepAFL maintains high accuracy. Crucially, even at a low layer count ($T = 5$) and the most severe noise rate ($\tau = 50\%$), the performance of our DeepAFL (i.e., 60.91%) still significantly outperforms the AFL's accuracy (i.e., 58.56%) achieved under the condition of 100% accurate data labels with no noise. Since AFL represents the state-of-the-art baseline on Non-IID-1 for CIFAR-100, these results also imply that our DeepAFL, even with 50% label corruption, outperforms all gradient-based baselines under ideal label accuracy.

Additionally, as detailed in Appendix J, we also examine the robustness of DeepAFL under partial client participation. Similar experiments conducted with client dropout rates ranging from 10% to 50% show that DeepAFL maintains highly stable and superior performance. Notably, DeepAFL's performance under label flipping is slightly weaker than under client dropouts. This is expected, as client dropouts simply reduce training data volume, whereas label flipping introduces the additional negative effect of data poisoning. Moreover, the performance gap between these two scenarios remains small, which further substantiates DeepAFL's superior robustness.

## L  MORE DISCUSSIONS ON DEEPAFL'S USE CASES

Here, we provide detailed discussions on DeepAFL's use cases. Specifically, our DeepAFL requires a given backbone to serve as an effective feature extractor for training, but it does not mandate how this backbone is obtained. Indeed, leveraging pretrained models or foundation models as the backbone in FL has emerged as a very common and popular research practice in recent years (Nguyen et al., 2023; Piao et al., 2024; Yu et al., 2024; 2025). Accordingly, we elaborate below on several common ways for obtaining such pretrained backbones, corresponding to different use cases of our DeepAFL, as well as the pervasiveness of pre-trained backbones in FL and the necessity of FL even with such backbones. The detailed discussions are presented as follows:

First, we want to highlight that our DeepAFL requires a pretrained backbone for feature extraction, but it does not mandate how this backbone is obtained. For the sake of brevity, we do not discuss this in detail within the main text. Herein, we introduce three common ways for obtaining pretrained backbones, which collectively illustrate DeepAFL's various use cases:

- **Supervised Pre-training:** This is the most prevalent case and constitutes the primary setting for the experimental comparisons presented in our paper. In this scenario, our Deep-AFL can be viewed as a collaborative fine-tuning approach for FL, where clients adapt a shared public backbone to private data with different feature distributions.

- **Self-Supervised Pre-training:** With the emergence of large-scale pre-trained models, there is frequently insufficient labeled data for supervised pre-training. In such cases, self-supervised backbones have gained prominence (such as auto-encoders via reconstruction, and DINO series via contrastive learning). These self-supervised models, however, inherently lack an integrated classifier or regressor, possessing only robust generalizable feature extraction capabilities without direct predictive functionality. Therefore, deploying such backbones in FL necessitates training a matching classifier or regressor. In this context, our DeepAFL serves as a direct and highly efficient FL training approach to obtain this classifier or regressor, effectively leveraging the backbone's established capabilities.

- **Domain Adaptation:** If the pre-trained backbone presents a domain shift relative to the FL system's target data, the server can also perform initial feature domain adaptation by fine-tuning the backbone (using a set of domain-specific or compliant public data). Once this adaptation is complete, the backbone is then frozen, and our DeepAFL can be implemented for subsequent FL training. This case can be viewed as an engineering strategy to mitigate the issues arising from a cross-domain backbone. Given that domain adaptation can be performed centrally by the server and DeepAFL incurs only negligible time costs, the aggregate overhead across both stages is still considered to be very small.

In summary, DeepAFL can not only leverage backbones obtained through supervised pre-training, as demonstrated in our experiments, but also effectively utilize other types of backbones for different use cases. Specifically, when using backbones from self-supervised pre-training, our DeepAFL effectively solves the meaningful problem of constructing task-specific classifier in FL. Furthermore, when faced with the potential for domain gaps, this problem can be mitigated through domain adaptation as described above, which can be viewed as an engineering strategy.

Second, we further discuss the pervasiveness of pre-trained backbones in FL and the necessity of FL even with such backbones. Indeed, incorporating pre-trained backbones is a widely embraced practice in recent research to introduce prior knowledge and stabilize FL (Nguyen et al., 2023; Piao et al., 2024; Yu et al., 2024; 2025). Specifically, a key commonality between our work and these studies is the focus on using foundation models for conducting FL, rather than using FL for building foundation models. Notably, many existing pre-trained model-based FL works are mechanistically constrained to necessitate large foundation models. In contrast, DeepAFL accommodates backbones of varying sizes and capabilities, which renders it particularly suitable for resource-constrained edge environments compared to related works. Moreover, we demonstrate that a pre-trained backbone does not negate the need for FL. Leveraging results from the original AFL paper on CIFAR-100 ($\alpha = 0.1$ and $K = 100$) (He et al., 2025), we observe that purely local training without global aggregation yields maximum and average test accuracies of merely $16.36\%$ and $12.04\%$. Conversely, traditional FL method FedAvg and AFL achieve $56.62\%$ and $58.56\%$, while our DeepAFL ($T = 20$) further elevates performance to $66.98\%$. These results conclusively demonstrate that despite the availability of a pre-trained backbone, FL remains indispensable for achieving high performance.

# M   DEEPAFL'S REPRESENTATION LEARNING

In this section, we present detailed discussions on the representation learning capabilities of Deep-AFL. Specifically, we first outline how the capacity of DeepAFL aligns with fundamental characteristics of representation learning. We then analyze the intermediate features produced by DeepAFL as empirical evidence supporting these capabilities. The specific discussions are provided below.

## M.1   ANALYSES ON THE CONCEPT OF REPRESENTATION LEARNING

Here, we demonstrate that the capabilities of DeepAFL align with the fundamental characteristics of representation learning. To this end, we first outline these fundamental characteristics and then conduct a point-by-point analysis showing how our DeepAFL satisfies each one. Subsequently, we further substantiate our claim by analyzing the architectural similarities between DeepAFL and the Multi-Layer Perceptron (MLP). The detailed analyses are presented below.

First of all, drawing upon the seminal work of Bengio et al. (Bengio et al., 2013), we can conceptually distill two fundamental characteristics of representation learning: **(1) Automated Feature Extraction**: This entails learning feature transformations (which are often non-linear) directly from the data, thereby avoiding the complexity and reliance on expert knowledge associated with manual feature engineering. **(2) Utility for Downstream Tasks**: The extracted features are beneficial and useful for adapting to specific downstream tasks, meaning they provide utility when building predictors (e.g., classifiers), ultimately leading to enhanced model performance.

In fact, our DeepAFL directly satisfies these two foundational characteristics: (1) The feature transformations involve trainable parameters $\mathbf{\Omega}$, which are automatically learned from the data via the "sandwiched" least squares. Moreover, the use of the activation function $\sigma(\cdot)$ ensures necessary non-linearity in the layer-wise transformations. (2) The feature transformations show evident and significant benefits for our downstream task (i.e., classification). Specifically, when coupled with an analytic classifier, the accuracy of our DeepAFL consistently improves on the training and test sets as $T$ increases (as shown in Tables 9–11 of our paper). In summary, based on these conceptual analyses, our DeepAFL indeed aligns with the fundamental concept of representation learning.

Subsequently, to further clarify our DeepAFL's alignment with fundamental representation learning concepts, we further draw an analogy between our DeepAFL and the MLP. Given that the MLP is recognized as the most fundamental DNN, which indisputably achieves representation learning, this analogy serves to demonstrate that our DeepAFL similarly achieves these capabilities.

Specifically, since our DeepAFL incorporates a residual block structure, we also introduce skip connections into the MLP, which does not impair its representation learning capability (but rather makes it easier to train). At this point, what the residual block of the MLP learns is nothing more than subjecting the features from the previous layer to an affine transformation $\mathbf{W}_{t+1}$ followed by an activation function $\sigma(\cdot)$. Here, the affine transformation $\mathbf{W}_{t+1}$ constitutes the learnable parameters of the MLP, as follows:

$$g_{t+1}(\mathbf{\Phi}_t) = \sigma(\mathbf{\Phi}_t \mathbf{W}_{t+1}). \tag{82}$$

Consequently, when multiple layers are stacked, the MLP functions as a continuous alternation of affine transformations and non-linear activation functions. Crucially, a clear parallel emerges between our DeepAFL and the MLP, as DeepAFL similarly involves an alternating stack of affine transformations and non-linear activation functions:

$$g_{t+1}(\mathbf{\Phi}_t) = \sigma(\mathbf{\Phi}_t \mathbf{B}_t)\mathbf{\Omega}_{t+1}. \tag{83}$$

The primary distinction is that the learnable parameters $\mathbf{W}_{t+1}$ in the MLP are applied prior to the activation function, while the parameters $\mathbf{\Omega}_{t+1}$ in our DeepAFL are applied after the activation. This localized difference is effectively diminished when multiple layers are stacked. Furthermore, more complex DNNs also share this philosophy of feature re-weighting via affine transformations and non-linear activation functions. For instance, a CNN can be broadly viewed as an MLP with weight sharing and local connectivity. Therefore, the inherent structural similarity between DeepAFL and the MLP confirms that DeepAFL is aligned with the core concepts of representation learning.

Based on the analyses above, our DeepAFL not only satisfies the fundamental characteristics of representation learning but also exhibits a strong structural similarity to the MLP. This dual validation confirms that DeepAFL aligns with the core concepts of representation learning.

## M.2 ANALYSES ON INTERMEDIATE FEATURES

Table 30: Separability analyses of DeepAFL's intermediate features on CIFAR-100.

| Layer | Training CSR | Testing CSR | Training IFS | Testing IFS | Training DM | Testing DM |
|---|---|---|---|---|---|---|
| DeepAFL ($T = 0$) | 1.583 | 1.529 | 3.25 | 3.13 | 3.21 | 3.09 |
| DeepAFL ($T = 5$) | 1.565 | 1.520 | 3.20 | 3.11 | 3.16 | 3.07 |
| DeepAFL ($T = 10$) | 1.555 | 1.516 | 3.17 | 3.10 | 3.14 | 3.06 |
| DeepAFL ($T = 15$) | 1.546 | 1.513 | 3.15 | 3.09 | 3.12 | 3.05 |
| DeepAFL ($T = 20$) | 1.539 | 1.510 | 3.13 | 3.08 | 3.11 | 3.04 |

Here, we present experimental analyses of the intermediate features generated by DeepAFL to empirically validate its representation learning capabilities. Specifically, we first conduct additional experiments designed to assess the separability of the intermediate features produced by our DeepAFL, thereby substantiating the model's advanced learning capacity. Furthermore, we analyze DeepAFL's utility for the downstream task to further validate its representation learning capabilities.

First, let's focus on the additional experiments for investigating the separability of intermediate features produced by our DeepAFL, which are conducted using the following three metrics:

- **Compactness-Separation-Ratio (CSR)**: This metric evaluates the ratio of the average intra-class squared distance (i.e., compactness) to the average inter-class squared distance (i.e., separation). A lower CSR value indicates better separability.

- **Inverse Fisher Score (IFS)**: It is defined as the reciprocal of the Fisher Score, which is a widely adopted metric to distinguish different classes. The IFS is calculated by the ratio of the intra-class variance to inter-class variance. A lower FS value implies better separability.

- **Discriminative Measure (DM)**: It is a metric derived from the core principles of the Linear Discriminant Analysis (LDA), quantifying the ratio of within-class scatter to between-class scatter. A lower DM value signifies better separability.

In summary, all the additional metrics are unified such that a lower value indicates better representation separability. Using these metrics, we conducted additional experiments on the CIFAR-100 dataset to analyze the intermediate features generated by our DeepAFL. The detailed experimental results are presented in Table 30. From the results, it is evident that all these metrics show better separability as the layer count of our DeepAFL increases on both the training and test sets, which further substantiates our DeepAFL's representation learning capabilities. Furthermore, we observe that the metrics on the test set generally perform better (i.e., yield lower values) than those on the training set. This is likely because the training set contains a much more samples, leading to greater inherent noise and, consequently, slightly inflated metric values.

Meanwhile, we can also observe that DeepAFL exhibits a diminishing marginal returns phenomenon as the layer count increases, but this is entirely expected and normal since we did not increase any training data volume. This phenomenon is also common in gradient-based methods. For instance, simply increasing the depth of a DNN can not lead to continuous accuracy improvement up to 100% without encountering a bottleneck. Even the high performance achieved by LLMs is a result of scaling the model alongside vast increases in training data volume (i.e., the scaling law).

Second, as discussed in Appendix M.1, the utility for the downstream task (i.e., classification accuracy) is the most direct and fundamental metric to validate the representation learning capabilities of DeepAFL. This perspective also aligns with the classical work by Yoshua Bengio et al. (Alain & Bengio, 2016), which asserts that better linear separability is indicative of a more meaningful representation. In fact, in Tables 9–11, we have reported the training and testing accuracies achieved by our DeepAFL when utilizing intermediate features from various depths (i.e., DeepAFL with different numbers of layers). Let's take the results on the CIFAR-100 dataset as an example. The baseline AFL struggles with severe underfitting, achieving a training accuracy of only 61.55%. In stark contrast, our DeepAFL (at $T = 20$) propels the training accuracy to 85.15%, representing a massive absolute improvement of 23.6%. This improved representation translates directly to generalization, yielding a substantial 8.39% increase in test accuracy. Crucially, these gains are achieved with high efficiency. The total runtime for our DeepAFL is only 91.74 s, which is a marginal increase of less than 42 s compared to AFL (50.05 s). These results show that DeepAFL learns significantly more discriminative and useful representations than AFL.

## N ADDITIONAL DISCUSSION

Beyond the discussion in our main text, we provide additional discussion here. A key feature of our DeepAFL is its capability to perform gradient-free representation learning after the feature extraction backbone. This stands in contrast to traditional methods that typically apply representation learning directly to the backbone model. A natural question arises: what are the differences and advantages of our approach compared to direct representation learning on the backbone?

Direct representation learning on the backbone typically involves two main strategies.

- The first strategy is to fine-tune the backbone *during the FL training process*. This aims to adapt the model to the data distribution within the FL system. However, this approach requires enabling gradient-based training in the FL training, which can be severely impacted by data heterogeneity. Moreover, there is a high risk of damaging the knowledge acquired during pre-training, known as catastrophic forgetting. As many existing works have shown that fine-tuning the backbone with gradients during FL training may be counterproductive, it is precisely why we developed DeepAFL for gradient-free representation learning.
- The second strategy is to build a more complex backbone *during the pre-training process*, leveraging massive pre-training datasets to enhance its robustness. Our approach is orthogonal to this strategy, as our DeepAFL is designed to further improve the representations during the FL training process, given any fixed, pre-trained backbone. This training-process representation learning is both necessary and meaningful, as the pre-trained backbone will inevitably exhibit a domain shift when applied to a new FL system.

In addition, another noteworthy aspect of our work is the effectiveness of residual connections in our DeepAFL, especially since its gradient-free nature seems to be at odds with the original gradient-based motivation for these connections in ResNet. We believe that it is a profound theoretical question that warrants future research. Here, we would like to offer several intuitive explanations for it:

- First, our comprehensive ablation studies show that the model without residual connections struggles to improve its training accuracy as the number of layers increases, let alone testing accuracy. This situation mirrors the challenges faced by gradient-based deep networks before the introduction of ResNet. Deep networks without residual connections can similarly suffer from an information bottleneck as they continually update feature representations. By adding a skip connection, each layer's representation learning is built upon the previous layer's foundation, enabling continuous and cumulative improvements in representations.
- Second, as we prove in Theorems 2 and 3, our DeepAFL's representation learning capability is primarily reflected in the non-increasing nature of its empirical risk with added layers. This beneficial property is ensured by the use of skip connections. When network features converge, each residual block can be just set to zero, at a minimum, to achieve an unchanging empirical risk, thereby guaranteeing that the overall empirical risk of our DeepAFL will not increase. Without this design, the ablated model would no longer be able to guarantee these theorems, severely compromising its representation learning capability.
- Third, the philosophy of our DeepAFL intrinsically aligns with that of *Gradient Boosting*. Specifically, under the chosen MSE loss function, the negative gradient of the loss function is precisely the residual. Therefore, by continuously fitting the residuals from previous layers, our DeepAFL can be seen as learning *pseudo-gradients*. This perspective may further help unify the understanding of both gradient-based and gradient-free learning approaches.

In summary, we believe our DeepAFL represents a significant first step. There is substantial potential for future theoretical and practical exploration based on this foundation, which we hope will further advance various fields, including FL, analytic learning, and representation learning, etc.

## O STATEMENT ON THE USAGES OF LARGE LANGUAGE MODELS

In adherence to the ICLR 2026 policy, we report the use of Large Language Models (LLMs) during the preparation of this paper. Here, the only usages of LLMs were to aid and polish the writing and illustration. Specifically, the LLMs were utilized to improve sentence structure, enhance clarity, ensure grammatical accuracy, and optimize the illustration. Furthermore, all the core scientific contributions (including the hypothesis formulations, the theoretical analyses, the experimental designs, the data analyses, and the final conclusions, etc.) are the original work of the human authors.

