# OpenReview forum: "DeepAFL: Deep Analytic Federated Learning"
_ICLR.cc/2026/Conference — ICLR 2026 Poster_

### Official Review · Reviewer_ZKVc · 2025-10-27

**Soundness:** 3
**Presentation:** 3
**Contribution:** 3
**Rating:** 6
**Confidence:** 3

**Summary:**

This paper presents DeepAFL, a framework designed to achieve invariance to data heterogeneity in federated learning (FL) while reducing computational cost. Unlike existing analytical federated learning (AFL) methods that are limited to a single layer, DeepAFL extends the idea to multiple nonlinear layers, thereby enabling richer representation learning. Specifically, building upon pre-trained features, DeepAFL updates the model in a layer-wise manner, where it learns an analytical classifier for each layer and subsequently learns a transformation to adapt the layer’s features for improved alignment with the class labels. In the FL setting, local clients transmit information matrices only once to the server, which then aggregates the final model. Experimental results show that DeepAFL outperforms both gradient-based and analytical FL approaches, demonstrating strong invariance to data heterogeneity and improved computational efficiency.

**Strengths:**

- The paper tackles a well-motivated problem that constitutes a significant and timely advancement for analytical federated learning.

- The proposed DeepAFL demonstrates promising performance compared to existing gradient-based and analytical FL methods, while maintaining strong computational efficiency and invariance to data heterogeneity.

- The experiments are comprehensive and examine multiple aspects, including hyperparameter settings, activation functions, and ablation analyses.

**Weaknesses:**

- To obtain an analytical solution, DeepAFL relies on the mean squared error (MSE) loss; however, MSE can be sensitive to noise in the data when applied to classification tasks. In realistic federated learning (FL) scenarios, such as hospital applications, clients may contain data annotated by different labelers, leading to slight inconsistencies due to varying labeling standards. The experiments in the paper primarily use clean and well-organized datasets, such as CIFAR and TinyImageNet. Therefore, it remains unclear whether DeepAFL can achieve competitive performance under noisier, real-world conditions compared to gradient-based FL approaches that employ cross-entropy loss.

- Although the paper highlights the importance of representation learning for analytical FL, it lacks visualizations or analyses to demonstrate whether the learned features are semantically meaningful. Moreover, the generalization ability of these features remains unexplored, for example, their cross-domain generalization capability.


- The gradient-based FL baselines used for comparison are mostly from studies prior to 2023, lacking evaluation against more recent and advanced approaches such as FedAWA [1] and one-shot FL methods like FedLPA [2].


[1] 2025 CVPR FedAWA: Adaptive Optimization of Aggregation Weights in Federated Learning Using Client Vectors


[2] 2024 NeurIPS FedLPA: One-shot Federated Learning with Layer-Wise Posterior Aggregation

**Questions:**

Besides the weakness shown in the above section, please also see the following questions:

Q1: DeepAFL uses ResNet-18 as a pre-trained backbone and trains $T$ additional layers on top of it. Do the gradient-based approaches also follow a similar setup, or do they fine-tune the ResNet-18 backbone? If not, would gradient-based methods potentially perform better, or be more stable, if they likewise kept the pre-trained backbone fixed and only trained additional layers?

Q2: Since each intermediate layer in DeepAFL is trained with an auxiliary classifier that shares the same objective and class labels, all layers are encouraged to align closely with the same targets without awareness of how subsequent layers will utilize their features. Could this design lead to redundant representations across layers? For instance, if a given layer already achieves linear separability, the following layer may have little room to learn additional useful information. Would it be possible to incorporate a reconstruction-based objective, such as an autoencoder with MSE, to encourage more diverse and informative feature learning?

---

> ### Author Response · Authors · 2025-11-21
> **Author Response to Reviewer ZKVc (Part 1 of 9)**
>
> We sincerely thank you for your thorough feedback and positive recognition.
>
> Here, we would like to respectfully address your raised Weakness (W) and Question (Q) below.
>
> # W1: Noise Robustness for the MSE Loss
>
> > To obtain an analytical solution, DeepAFL relies on the mean squared error (MSE) loss; however, MSE can be sensitive to noise in the data when applied to classification tasks. In realistic federated learning (FL) scenarios, such as hospital applications, clients may contain data annotated by different labelers, leading to slight inconsistencies due to varying labeling standards. The experiments in the paper primarily use clean and well-organized datasets, such as CIFAR and TinyImageNet. Therefore, it remains unclear whether DeepAFL can achieve competitive performance under noisier, real-world conditions compared to gradient-based FL approaches that employ cross-entropy loss.
>
> Thank you very much for your insightful comment!
>
> We understand your concern is that the Mean Squared Error (MSE) loss appears different from the Cross-Entropy (CE) loss commonly used in classification tasks, and that our used MSE loss may be more sensitive to noise in the data.
>
> First and foremost, we wish to propose a potentially counter-intuitive argument that is **contrary to your concern**: our analytic classifier utilizing the MSE loss is actually **less sensitive to noise** compared to gradient-based methods utilizing the CE loss.
>
> ## W1-1: Logical Analyses
>
> Let's first analyze this logically. What is the key factor that makes a method sensitive to noise? We believe one crucial factor that everyone can agree upon is **overfitting**. If a method tends to overfit, then when there is even slight noise in the input data, it will tend to fit this noise. We describe this phenomenon as being sensitive to noise (or not robust).
>
> Here, to avoid ambiguity, we first refer to the Wikipedia definition of overfitting:
>
> > Overfitting means creating a model that matches (memorizes) the training set so closely that the model fails to make correct predictions on new data. An overfit model is analogous to an invention that performs well in the lab (i.e., training set) but is worthless in the real world (i.e., test set).
>
> In reality, the analytic-learning method with MSE loss does not tend to overfit. On the contrary, it typically suffers from **underfitting**. In fact, this is precisely the original motivation for proposing our DeepAFL—to enhance the model's fitting capacity through deep representation learning and thus improve performance.
>
> As shown in Tables 9-11 of our paper, we list the training and testing accuracies of AFL and our DeepAFL across different datasets. Taking the CIFAR-100 dataset as an example, the training set accuracy for AFL is only **61.55%**, which is quite low, especially compared to its test set accuracy of **58.56%**. This phenomenon suggests that AFL suffers from **severe underfitting**.
>
> Moreover, returning to our DeepAFL, as we gradually increase $T$ from 0 up to 50 on CIFAR-100, not only does its training accuracy steadily increase, but its test accuracy also steadily increases. Note that because the test accuracy increases stably along with the training accuracy, this indicates that our DeepAFL **does not exhibit overfitting** up to 50 layers. (As we mentioned earlier, the primary characteristic of overfitting is good performance on the training set but poor performance on the test set).
>
> In summary, both AFL and our DeepAFL, which are trained using the MSE loss, show virtually **no signs of overfitting**, and are thus **theoretically expected to be insensitive to noise**.
>
> To analyze this point in greater detail, the secret to our DeepAFL's ability to maintain superior representation learning without overfitting lies in the fact that **its analytic classifier with MSE loss is linear**. Because of this, its model complexity is quite low, making it very difficult to overfit and thus robust to noise.
>
> In contrast, when the CE loss was introduced for backpropagation optimization, one of its main advantages was its ability to make the loss curve steeper, making it easier to train. Therefore, gradient-based methods with CE loss are actually **more prone to overfitting** and consequently, **more sensitive to data noise**.
>
> Logically, therefore, robustness is not a weakness but actually a **unique advantage** of AFL/DeepAFL (with MSE loss) compared to gradient-based methods (with CE loss).
>
> Next, we would like to further provide additional **empirical evidence** to demonstrate our DeepAFL's superior robustness against noise in the data. Due to space constraints, we will elaborate on these results and analyses in the next comment block.

---

> ### Author Response · Authors · 2025-11-21
> **Author Response to Reviewer ZKVc (Part 2 of 9)**
>
> # W1: Noise Robustness for the MSE Loss (Continued)
>
> ## W1-2: Experimental Evaluations
>
> Here, we would like to provide additional **empirical evidence** to demonstrate our DeepAFL's superior robustness against noise in the data.
>
> Specifically, we simulate the noise in the training data by randomly flipping the labels of a proportion $\tau$ of the training data. The selected samples have their labels randomly changed to another class label.
>
> Thus, using $\tau=0\\%$ as the control group, we conduct a broad range of experimental analyses with $\tau$ values including $10\\%$, $20\\%$, $30\\%$, $40\\%$, and $50\\%$, to thoroughly assess the performance variation of our DeepAFL on CIFAR-100.
>
> The detailed results (Test Accuracy in % across layers $T$) are presented below:
>
> | $T$  | $\tau=0\\%$ | $\tau=10\\%$ | $\tau=20\\%$ | $\tau=30\\%$ | $\tau=40\\%$ | $\tau=50\\%$ |
> | ---- | ---------- | ----------- | ----------- | ----------- | ----------- | ----------- |
> | 5    | 64.72      | 64.20       | 63.63       | 63.25       | 62.51       | 60.91       |
> | 10   | 65.96      | 65.43       | 65.09       | 63.96       | 62.98       | 60.99       |
> | 15   | 66.56      | 66.11       | 65.66       | 64.29       | 63.15       | 60.97       |
> | 20   | 66.98      | 66.38       | 66.11       | 64.69       | 63.35       | 60.45       |
>
> From the results, we can observe that our DeepAFL exhibits **substantial robustness** under training data noise.
>
> Specifically, for a moderate noise rate of $\tau \leq 20\\%$, the maximum accuracy degradation observed for our DeepAFL (at $T=20$) is less than $1$ percentage point (e.g., $66.98\\% - 66.11\\% = 0.87\\%$).
>
> Furthermore, even under the **extremely challenging scenario** of $\tau=50\\%$ (where up to half of the training data labels are erroneous), our DeepAFL still maintains a high accuracy.
>
> Crucially, even at a low layer count ($T=5$) and the most severe noise rate ($\tau=50\\%$), the performance of our DeepAFL (i.e., $60.91\\%$) still **significantly outperforms the AFL's accuracy (i.e., $58.56\\%$) achieved under the condition of $100\\%$ accurate data labels with no noise**.
>
> Since AFL is already the state-of-the-art baseline in Non-IID-1 on CIFAR-100, the results above also imply that our DeepAFL, even when $50\\%$ of the data labels are randomly corrupted, can exceed the performance of all gradient-based baselines achieved under the condition of 100% accurate data labels.
>
> It is highly undoubted that for a gradient-based method, having $\tau=20\\%$ to $\tau=50\\%$ of the data labels being completely incorrect would be an unimaginably large disaster for gradient-based training with CE loss.
>
> We believe these impressive results are sufficiently compelling to show the **extreme robustness of our DeepAFL**, as we have already analyzed logically.
>
> By the way, another reviewer, **f9VW**, also raised a similar concern regarding the robustness of our DeepAFL. That question focused on whether DeepAFL would suffer a significant performance drop when faced with **partial client participation** (i.e., when a portion of clients drop out).
>
> In response to that question, we conducted similar additional experiments, showing that under client dropout rates ranging from $10\\%$ to $50\\%$ (i.e., only $60\\%$ to $90\\%$ of clients participate in training), our DeepAFL maintains **highly stable and superior performance**.
>
> Notably, if you carefully compare the experimental results provided in that response with the results presented here, you will observe that at the same rate ($\eta=1-\tau$), the performance of DeepAFL under label flipping attack is slightly weaker than under client dropouts (though both are at a very high level).
>
> This additional observation is reasonable, as client dropouts merely reduce the volume of training data, whereas label flipping introduces the additional negative effect of data poisoning. Consequently, model performance under data poisoning should logically be weaker than simply discarding that portion of the data. Moreover, the performance gap between these two scenarios for our DeepAFL is quite small, which **further validates its superior robustness**.
>
> ----
>
> Thank you again for your insightful comment! We will incorporate the above results and analyses into our final paper. And we firmly believe this addition will significantly enhance the clarity of our paper.
>
> We sincerely hope that our detailed explanations and clarifications can adequately address your concerns and alleviate any potential misunderstanding regarding our DeepAFL.
>
> We welcome any further questions you may have.

---

> ### Author Response · Authors · 2025-11-21
> **Author Response to Reviewer ZKVc (Part 3 of 9)**
>
> # W2: Additional Analyses for Representation Learning
>
> > Although the paper highlights the importance of representation learning for analytical FL, it lacks visualizations or analyses to demonstrate whether the learned features are semantically meaningful. Moreover, the generalization ability of these features remains unexplored, for example, their cross-domain generalization capability.
>
> Thank you very much for your constructive comment!
>
> We understand your main concerns and have summarized them into two distinct sub-issues:
>
> 1. Providing more evidence and analyses for the intermediate features.
> 2. Exploring the generalization ability of our DeepAFL beyond empirical risks.
>
> ---
>
> Regarding the first sub-issue, we respectfully highlight that the most direct and fundamental evidence that the learned features of our DeepAFL are meaningful is their **utility for our downstream task** (i.e., the classification accuracy).
>
> This is because we maintain the linear analytic classifier at each layer (i.e., the same least-squares optimization objective with consistent parameters). Thus, the classification accuracy at each layer directly reflects the **separability of the features** under the linear analytic classifier, as also established by **Yoshua Bengio et al.** [1].
>
> In fact, **we have already provided quantitative intermediate analyses of training and testing accuracies in Tables 9-10 of our paper.**
>
> To reinforce this point, we would like to reiterate the impressive performance gains achieved by DeepAFL on both training and testing sets. Let's take the results on the CIFAR-100 dataset as an example. The baseline AFL struggles with severe underfitting, achieving a training accuracy of only **61.55%**. In stark contrast, our DeepAFL (at $T=20$) propels the training accuracy to **85.15%**, representing a massive absolute improvement of **23.6%**. This improved representation translates directly to generalization, yielding a substantial **8.39%** increase in test accuracy.
>
> Crucially, these gains are achieved with high efficiency. The total runtime for DeepAFL is only **91.74 s**, which is a marginal increase of less than 42 s compared to AFL (50.05 s). These results compellingly show that DeepAFL learns significantly more discriminative and useful representations than AFL.
>
> Moreover, in direct response to your valuable request, we have introduced **three additional metrics** to analyze the separability of the intermediate features learned by our DeepAFL. The new results and analyses will be provided in the upcoming section **W2-1**.
>
> ---
>
> Regarding the second sub-issue, we would like to categorize the "generalization ability" you mentioned into the following two main meanings for discussion:
>
> - **Generalization from Training Data to Test Data:** Since our theory only provides the non-increase of empirical error, we surmise you are curious whether this theory can be extended to an analysis of the generalization error. In this case, we assume that the training and test data remain IID, which is the common practice and prerequisite in almost all supervised statistical learning theory.
> - **Generalization (or Transferability) from IID to Out-Of-Distribution (OOD):** Whether the features learned by our DeepAFL can be applied to other cross-domain applications with transferability.
>
> For **Training-to-Testing generalization**, we want to state that our proposed monotonic decrease of empirical risk within our DeepAFL (as proven in Theorems 2 and 3 of our paper) is a very strong theoretical result. We can readily extend it to the analyses of the generalization error bound, as the complexity of the linear analytic classifier remains constant at each layer. We will provide more detailed theoretical analyses in the subsequent section **W2-2**.
>
> As for **IID-to-OOD generalization (i.e., transferability)**, we want to clarify that "transferability" is not a necessary condition for successful representation learning but pertains to the distinct field of transfer learning. Since we did not claim the transferability ability of our DeepAFL, its lack of immediate cross-domain transfer capability does not constitute a severe weakness for our current work.
>
> In fact, **the vast majority of current gradient-based supervised methods also fall far short of robust transferability**, as they tend to focus narrowly on the current task and inherently suffer when faced with out-of-distribution data.
>
> Furthermore, as the development of analytic-learning-based methods is still in its nascent stages, our current core aim is to **achieve direct performance improvement with marginal additional cost**.
>
> While we do not deny that further exploring how to "improve the transferability of the underlying model" is a promising and interesting direction for future work, **it clearly exceeds the scope of our current work.**
>
> Thanks again for your constructive comment, and we will go on to provide more results and analyses in the upcoming W2-1 and W2-2.

---

> ### Author Response · Authors · 2025-11-21
> **Author Response to Reviewer ZKVc (Part 4 of 9)**
>
> # W2: Additional Analyses for Representation Learning (Continued)
>
> ## W2-1: Additional Empirical Evidence for Representation Learning
>
> In response to your valuable comment, we have introduced 3 more metrics to analyze the separability of the intermediate features learned by our DeepAFL, which further substantiates its superior representation learning capabilities.
>
> Specifically, we adopted the following 3 additional metrics for layer-wise feature analyses:
>
> * **Compactness-Separation- Ratio (CSR):** It evaluates the ratio of the average intra-class squared distance (compactness) to the average inter-class squared distance (separation). A **lower CSR value** indicates better separability.
> * **Inverse Fisher Score (IFS):** It is defined as the reciprocal of the Fisher Score, which is a widely adopted metric to distinguish different classes. The IFS is calculated by the ratio of the intra-class variance to inter-class variance. A **lower FS value** implies better separability.
> * **Discriminative Measure (DM):** It is a metric derived from the core principles of Linear Discriminant Analysis (LDA), quantifying the ratio of within-class scatter to between-class scatter. Specifically, a **lower DM value** signifies better separability.
>
> In summary, all the additional metrics are unified such that a **lower** value indicates **better** representation separability. Thus, we conducted additional experiments on the CIFAR-100 dataset, and the results are presented below:
>
> | Layer | Training CSR | Testing CSR | Training IFS | Testing IFS | Training DM | Testing DM |
> | ----- | ------------ | ----------- | ------------ | ----------- | ----------- | ---------- |
> | 0     | 1.583        | 1.529       | 3.25         | 3.13        | 3.21        | 3.09       |
> | 5     | 1.565        | 1.520       | 3.2          | 3.11        | 3.16        | 3.07       |
> | 10    | 1.555        | 1.516       | 3.17         | 3.10        | 3.14        | 3.06       |
> | 15    | 1.546        | 1.513       | 3.15         | 3.09        | 3.12        | 3.05       |
> | 20    | 1.539        | 1.510       | 3.13         | 3.08        | 3.11        | 3.04       |
>
> From the results, it is evident that additional metrics all show better separability as the layer count of our DeepAFL increases on both the training and test sets, which further substantiates our DeepAFL's representation learning capabilities.
>
> Furthermore, we observe that the metrics on the **test set generally perform better (i.e., yield lower values)** than those on the training set. This is likely because the training set contains a much more samples, leading to greater inherent noise and, consequently, slightly inflated metric values. By the way, this observation also indirectly suggests that the direct classification accuracy may be a more authoritative measure than these other layer-wise analytical metrics, as we previously noted.
>
> Moreover, we also acknowledge that our DeepAFL exhibits a diminishing marginal returns phenomenon as the layer count increases, but this is entirely expected and normal since we did not increase any training data volume.
>
> This phenomenon is also common in gradient-based methods (e.g., simply increasing the depth of a DNN can not lead to continuous accuracy improvement up to 100% without encountering a bottleneck). Even the high performance achieved by large models like LLMs is a result of scaling the model *alongside* vast increases in training data volume (i.e., the scaling law).
>
> Thus, without any increase in the training data volume, we should not overly focus on the absolute improvement of these metrics. What is more important for analyzing DeepAFL's representation learning is observing the **consistent improving trend** of these metrics as the number of layers increases.
>
> Thank you again for your valuable comment, and we will incorporate these results into our final paper.

---

> ### Author Response · Authors · 2025-11-21
> **Author Response to Reviewer ZKVc (Part 5 of 9)**
>
> # W2: Additional Analyses for Representation Learning (Continued)
>
> ## W2-2: Additional Theoretical Analyses for Representation Learning
>
> Then we want to further provide additional theoretical analyses for the Training-to-Testing generalization of our DeepAFL. Specifically, the monotonic decrease of empirical risk within our DeepAFL (as proven in Theorems 2 and 3 of our paper) is in fact a very **strong theoretical result**.
>
> Crucially, this conclusion can be straightforwardly **extended to imply a better Generalization Error Bound (GEB)** in statistical learning theory. Intuitively, this is because our DeepAFL does **not increase the complexity of the linear analytic classifier**. At each layer, we utilize the same analytically-optimized linear classifier based on the least-squares method, as defined in Equation (4). Specifically, during the layer-wise optimization process for representation learning in DeepAFL, the data sample size ($N$), the feature dimension ($d _ {\Phi}$), and the regularization parameter ($\lambda$ & $\gamma$) all remain constant.
>
> Below, we provide a brief theoretical explanation.
>
> In statistical learning theory, the Generalization Error Bound (GEB) typically follows the core form:
>
> $\mathcal{R}(f) \leq \hat{\mathcal{R}}(f) + \text{Complexity Term}$
>
> where:
>
> - $\mathcal{R}(f)$ is the **Generalization Risk** (or Generalization Error);
> - $\hat{\mathcal{R}}(f)$ is the **Empirical Risk** (or Empirical Error);
> - The **Complexity Term**, also known as the complexity gap, measures the model's complexity (usually related to the sample size, feature dimension, and the data's geometric structure).
>
> The size of the GEB, $\mathcal{B}(f)$, is what we are typically concerned with:
>
> $\mathcal{B}(f) = \hat{\mathcal{R}}(f) + \text{Complexity Term}$
>
> According to standard VC-dimension and Rademacher complexity generalization bounds [1], [2], for a analytic linear hypothesis class in $\mathbb{R}^d$, the generalization gap/complexity is bounded by $O\left(\frac{\sqrt{d_{\Phi} \log (N/\delta)}}{\sqrt{N}}\right)$.
>
> In our analysis:
>
> - $N$ (data size) remains **constant**;
> - $d _ {\Phi}=1024$ (feature dimension) remains **constant**;
> - $\delta \in (0,1)$ (confidence level) remains **constant**.
>
> Therefore, the linear analytic classifiers at each layer of our DeepAFL can be considered to **share the same Complexity Term**.
>
> Notably, the above bound implicitly assumes that the input features $\Phi$ are bounded. This premise is satisfied in our DeepAFL because we incorporate constant **L2-normalization as a constraint** throughout the training process, as shown in Equations (4) and (7) of our paper.
>
> Now, considering the analytic classifiers $f _ t$ and $f _ {t+1}$ trained on the representations of layer $t$ and $(t+1)$ respectively, we have:
>
> $\mathcal{B}({f} _ {t}) = \hat{\mathcal{R}}(f _ {t}) + \text{Complexity Term} _ {t}$
>
> $\mathcal{B}({f} _ {t+1}) = \hat{\mathcal{R}}(f _ {t+1}) + \text{Complexity Term} _ {t+1}$
>
> As we have established that the **Complexity Term is constant** across layers, and our paper proves that the empirical risk is non-increasing (i.e., $\hat{\mathcal{R}}(f _ {t})\geq \hat{\mathcal{R}}(f _ {t+1})$), it directly follows that:
>
> $\mathcal{B}({f} _ {t})\geq \mathcal{B}({f} _ {t+1})$
>
> In this way, we have successfully shown that the theory introduced in our paper can be easily extended to deduce that the **GEB** of the analytic classifier is also **non-increasing**. This result provides **stronger evidence** for the effectiveness and generalization of our DeepAFL's representation learning capabilities.
>
> Thank you very much for your valuable and constructive comment. It has reminded us that these important theoretical implications should be **more clearly elucidated** for readers to fully appreciate our DeepAFL's representation learning capabilities.
>
> We will incorporate these new analyses into our final paper. And we firmly believe this addition will significantly **enhance the quality, clarity, and theoretical soundness** of our manuscript.
>
> ---
>
> ## References
>
> [1] Alain, Guillaume, and Bengio, Yoshua. Understanding intermediate layers using linear classifier probes. *International Conference on Learning Representations (ICLR)*, 2016.
>
> [2] Vapnik, Vladimir N. An overview of statistical learning theory. *IEEE Transactions on Neural Networks*, 1999.
>
> [3] Mohri, Mehryar, Afshin Rostamizadeh, and Ameet Talwalkar. Foundations of machine learning. *MIT Press*, 2018.
>
> ---
>
> We sincerely hope that our detailed explanations and clarifications can adequately address your concerns. Once again, thank you very much for your valuable comments in terms of representation learning!

---

> ### Author Response · Authors · 2025-11-21
> **Author Response to Reviewer ZKVc (Part 6 of 9)**
>
> # W3: Additional Baselines for Experimental Comparisons
>
> > The gradient-based FL baselines used for comparison are mostly from studies prior to 2023, lacking evaluation against more recent and advanced approaches such as FedAWA [1] and one-shot FL methods like FedLPA [2].
>
> Thank you very much for your valuable feedback regarding more recent baselines for experimental comparisons. We are currently dedicating our maximum effort to reproduce the two approaches you mentioned, i.e., FedAWA and FedLPA.
>
> The experiments are **actively underway**, and we will share the results with you in the next few days.
>
> Thank you again for your valuable feedback and patience!
>
>
>
> # Q1: Experimental Setup
>
> > DeepAFL uses ResNet-18 as a pre-trained backbone and trains additional T layers on top of it. Do the gradient-based approaches also follow a similar setup, or do they fine-tune the ResNet-18 backbone? If not, would gradient-based methods potentially perform better, or be more stable, if they likewise kept the pre-trained backbone fixed and only trained additional layers?
>
> Thank you very much for your valuable question.
>
> First of all, we would like to clarify that, to maintain fairness in comparisons, the results for all baselines in our paper were directly employed from the benchmark provided in the original AFL paper.
>
> Specifically, within the original AFL paper, the gradient-based approaches do indeed follow a similar setup to AFL, freezing the ResNet-18 backbone and only training the one-layer classifier, as you noted.
>
> We understand the readers may raise potential concerns that the gradient-based baseline might require the inclusion of multiple trainable layers in the experimental setup (similar to our DeepAFL settings where $T=5, 10, 20$). Consequently, we have been conducting some relevant experiments over the past few days and have observed some interesting phenomena.
>
> Specifically, we took two baselines, FedAvg and FedDyn, as examples and conducted the aforementioned experiments on CIFAR-100 under the Dirichlet distribution with $\alpha=0.1$. We kept the gradient-based methods' pre-trained backbone fixed and only trained their additional layers. The feature dimension of the additional layers for the gradient-based methods was kept consistent with that of our DeepAFL (i.e., 1024).
>
> As we mainly showcase the performance of our DeepAFL at $T=20$ in our paper, we first conducted experiments for FedAvg and FedDyn at $T=20$. However, the results indicated that their test accuracy after 500 rounds of training remained extremely low ($<1\\%$), demonstrating a severe performance collapse.
>
> Subsequently, we further conducted experiments for $T=2, 5, \text{and } 10$. The results are presented in the table below:
>
> |        | FedAvg | FedDyn |
> | ------ | ------ | ------ |
> | $T=1$  | 56.62% | 57.55% |
> | $T=2$  | 55.83% | 57.22% |
> | $T=5$  | <5%    | 56.26% |
> | $T=10$ | <3%    | <3%    |
> | $T=20$ | <1%    | <1%    |
>
> Through these results, we can observe that the performance of the gradient-based methods gradually and constantly degrades as $T$ increases.
>
> When $T$ increases beyond a certain threshold (e.g., $T\geq10$), the excessive number of parameters, combined with the Non-IID data in the FL scenario, may cause the training process to collapse.
>
> These results fully show that **directly increasing the number of network layers after the backbone for the baselines, similar to our DeepAFL setup, would have a detrimental effect on their performance**.
>
>  We will incorporate the above results and analyses into our final paper. And we sincerely hope that our detailed explanations and clarifications can adequately address your concerns.
>
> Thank you again for your insightful question!

---

> ### Author Response · Authors · 2025-11-21
> **Author Response to Reviewer ZKVc (Part 7 of 9)**
>
> # Q2: Reducing Redundant Representations for Better Performance
>
> > Since each intermediate layer in DeepAFL is trained with an auxiliary classifier that shares the same objective and class labels, all layers are encouraged to align closely with the same targets without awareness of how subsequent layers will utilize their features. Could this design lead to redundant representations across layers? For instance, if a given layer already achieves linear separability, the following layer may have little room to learn additional useful information. Would it be possible to incorporate a reconstruction-based objective, such as an auto-encoder with MSE, to encourage more diverse and informative feature learning?
>
> Thank you very much for your **creative and insightful question.**
>
> **We are genuinely impressed by your professional observation!** The issue you raised is **absolutely correct** and, in fact, represents a very interesting phenomenon we observed early in the development of our DeepAFL.
>
> As designed in the current version of DeepAFL, each new layer's residual block, $g _ {t+1}(\mathbf{\Phi} _ {t})$, is built upon the feature information from the immediately preceding layer, $\mathbf{\Phi} _ {t}$, as follows:
>
> $\mathbf{\Phi} _ {t+1} =\mathbf{\Phi} _ {t}+g _ {t+1}(\mathbf{\Phi} _ {t})$
>
> $g _ {t+1}(\mathbf{\Phi} _ {t}) = \sigma(\mathbf{\Phi} _ {t}\mathbf{B} _ {t})\mathbf{\Omega} _ {t+1}$
>
> As you correctly pointed out, if the feature information $\mathbf{\Phi} _ {t}$ already achieves good linear separability or fitting, the following layer may indeed have little room to learn additional useful information, potentially leading to representational redundancy.
>
> Now, please allow us to discuss your great suggestion to incorporate a reconstruction-based objective below:
>
> > Would it be possible to incorporate a reconstruction-based objective, such as an auto-encoder with MSE, to encourage more diverse and informative feature learning?
>
> We understand your intention is to add a reconstruction loss during the feature optimization process in DeepAFL to promote feature diversity. Our original feature optimization follows the objective we term the *sandwiched* least squares problem:
>
> $\mathbf{\Omega} _ {t+1}= \arg\underset{\mathbf{\Omega}}{\min} \
>   \|\mathbf{Y}-(\mathbf{\Phi} _ {t} + \mathbf{F} _ t\mathbf{\Omega})\mathbf{W} _ t\|^2 _ \text{F}+{\mathbf{\gamma} \| \mathbf{\Omega} \|^{2} _ \text{F}}$
>
> This objective includes empirical risk minimization and L2 regularization, and we can derive its analytical (i.e., closed-form) solution based on Lemma 2 of our paper.
>
> We surmise you propose incorporating a reconstruction-based objective on top of this, forming a multi-objective optimization problem. For example:
>
> $\mathbf{\Omega} _ {t+1}= \arg\underset{\mathbf{\Omega}}{\min} \ \rho
>   \|\mathbf{Y}-(\mathbf{\Phi} _ {t} + \mathbf{F} _ t\mathbf{\Omega})\mathbf{W} _ t\|^2 _ \text{F}+ \ (1-\rho)
>   \|\mathbf{\Phi} _ {0}-(\mathbf{\Phi} _ {t} + \mathbf{F} _ t\mathbf{\Omega})\mathbf{Q} _ t\|^2 _ \text{F}+{\mathbf{\gamma} \| \mathbf{\Omega} \|^{2} _ \text{F}}$
>
> Here, the first term is the original empirical risk minimization, the second term is the reconstruction-based objective (similar to an auto-encoder with MSE) you suggested, and the third term is the regularization. $\rho$ acts as a weighting factor to balance the prediction MSE and the reconstruction MSE.
>
> In this formulation, the term $\mathbf{\Phi} _ {t+1}=(\mathbf{\Phi} _ {t} + \mathbf{F} _ t\mathbf{\Omega})$ can be considered the encoded feature, and the projection matrix $\mathbf{Q} _ t$ can be viewed as a decoder to map the feature back to the initial feature $\mathbf{\Phi} _ {0}$. Theoretically, this approach would ensure that the newly constructed feature $\mathbf{\Phi} _ {t+1}$ at each layer contains information necessary to reconstruct the initial feature $\mathbf{\Phi} _ {0}$.
>
> However, deriving the closed-form solution for this new and coupled formula becomes **much more complex** than our basic *sandwiched* least squares problem.
>
> If you are interested in the closed-form solution and the resulting performance of this new formulation, we would for sure be delighted to discuss it further with you then.
>
> But here, for the sake of simplicity and immediate verification, we opted to introduce a **simpler modification** to our DeepAFL to verify whether reducing representational redundancy indeed helps boost performance.
>
> Due to space constraints, we will provide the detailed design and experimental results in the next comment block.

---

> ### Author Response · Authors · 2025-11-21
> **Author Response to Reviewer ZKVc (Part 8 of 9)**
>
> # Q2: Reducing Redundant Representations for Better Performance (Continued)
>
> Here, we aim to introduce a simple modification to our DeepAFL to verify whether reducing representational redundancy indeed helps boost performance.
>
> Specifically, we can readily extend our DeepAFL by directly modifying the input of the residual block $g _ {t+1}(\cdot)$. For instance, the simplest modification is to let the residual block at each layer no longer depend on the **adjacent preceding feature** $\mathbf{\Phi} _ {t}$, but instead depend on the **initial feature $\mathbf{\Phi} _ {0}$**. Since the initial feature $\mathbf{\Phi} _ {0}$ has not yet been modified or adjusted by DeepAFL, it is expected to retain a richer set of information from the original input. The modified formula is described as follows:
>
> $\mathbf{\Phi} _ {t+1} =\mathbf{\Phi} _ {t}+g _ {t+1}(\mathbf{\Phi} _ {0})$
>
> $g _ {t+1}(\mathbf{\Phi} _ {0}) = \sigma(\mathbf{\Phi} _ {0}\mathbf{B} _ {t})\mathbf{\Omega} _ {t+1}$
>
> The only difference between the modified version and the original version of our DeepAFL lies in **the input of the residual block $g _ {t+1}(\cdot)$**.
>
> We have conducted experiments with this modified version of DeepAFL. The results demonstrate that such a simple adjustment provides a **stable performance uplift** for our DeepAFL, **without introducing any additional computational overhead**.
>
> We compare the training and testing accuracy of the original and modified DeepAFL using the CIFAR-100 dataset as an example:
>
> | Model | Training Acc. (Original, %) | Test Acc. (Original, %) | Training Acc. (Modified, %) | Test Acc. (Modified, %) | Training Acc. Improvement (%) | Test Acc. Improvement (%) |
> | - | :-: | :-: | :-: | :-: | :-: | :-: |
> | AFL | 61.55 | 58.56 | 61.55 | 58.56 | \ | \ |
> | DeepAFL $(T=0)$  |65.71|60.81|65.71|60.81|0.00|0.00|
> | DeepAFL $(T=5)$  |74.93|64.62|77.00|65.11|2.07|0.49|
> | DeepAFL $(T=10)$ |79.41|65.98|82.21|66.66|2.80|0.68|
> | DeepAFL $(T=15)$ |82.69|66.59|85.65|67.60|2.96|1.01|
> | DeepAFL $(T=20)$ |85.15|66.95|88.05|68.02|2.90|1.07|
> | DeepAFL $(T=25)$ |87.21|67.40|89.94|68.23|2.73|0.83|
> | DeepAFL $(T=30)$ |88.84|67.71|91.72|68.60|2.88|0.89|
>
> The results clearly show that by utilizing the initial features $\mathbf{\Phi} _ {0}$, the model consistently achieves higher training accuracy (by **over 2 percentage points**) and testing accuracy (by **about 0.5 to 1 percentage point**).
>
> Crucially, at $T=30$, the test set accuracy of the **modified version is already comparable to the performance of the original version at $T=50$**, effectively saving the computational cost of **20 layers**.
>
> Considering this modification is so elegant and introduces no additional overhead, we believe it already confirms your hypothesis: reducing redundancy indeed boosts performance of our DeepAFL. **We will explicitly discuss this 'Input-Skip' variant of our DeepAFL in the revised paper as a promising direction inspired by your review.**
>
> If delving deeper, we can find that the core principle of our proposed modification is very similar to that of your suggested reconstruction objective: both methods ensure that the knowledge source learned by each residual block is not solely confined to the potentially redundant information of the adjacent layer:
>
> * Your proposed method of incorporating reconstruction effectively adds a constraint aimed at maintaining the feature's reconstruction capability while enhancing its separability, thereby **indirectly reducing feature redundancy**.
>
> * Our modification directly utilizes the **initial feature $\mathbf{\Phi} _ {0}$ as the source**, providing completely unprocessed original information that has not been adjusted by DeepAFL. This serves to **directly circumvent feature redundancy** between layers.
>
> By the way, besides the basic modification we introduced, there is significant room for other design choices and manipulations, such as:
>
> - Combining the information from $\mathbf{\Phi} _ {0}$ and $\mathbf{\Phi} _ {t}$ (either by summation or concatenation) to serve as the input for the residual block $g _ {t+1}(\cdot)$.
> - Incorporating outputs from other positions within the backbone (e.g., the second-to-last layer) as multi-scale information to be used as the input for the residual block $g _ {t+1}(\cdot)$.
> - Employing different information source strategies for the residual block $g _ {t+1}(\cdot)$ in the odd and even layers of our DeepAFL to enhance representational diversity.
> - ......
>
> In fact, our early experiments have already indicated that these alternative versions could yield performance gains over our original DeepAFL. Yet, for this paper, we ultimately chose the current version of DeepAFL, despite the potential for redundant representations. This decision was primarily based on considerations of **clarity and accessibility** for the reader.
>
> Due to space constraints, we will elaborate on **our rationale and earlier considerations** in the next comment block.

---

> ### Author Response · Authors · 2025-11-21
> **Author Response to Reviewer ZKVc (Part 9 of 9)**
>
> # Q2: Reducing Redundant Representations for Better Performance (Continued)
>
> Our decision to maintain the current version of our DeepAFL was primarily based on the following reasons and considerations:
>
> * The main objective of our DeepAFL is to introduce **gradient-free deep representation learning** to achieve direct performance improvement based on AFL with marginal additional cost. In fact, we have successfully demonstrated this goal in this paper, and introducing further "tricks" might dilute our core insight.
>
> * The performance improvement of our DeepAFL over the AFL baseline is **already quite substantial** (e.g., up to an **8.39% increase** in test accuracy on CIFAR-100 at $T=20$). While further optimization is possible, the magnitude of these incremental gains seems less critical than the main proof of concept.
>
> * From a descriptive standpoint, the overall process of our current DeepAFL is highly consistent with that of a standard MLP: each new layer learns based on the information derived from the preceding layer. We thought maintaining this simplicity is conducive to readers drawing analogies and facilitating understanding.
>
> * The gradient-free representation learning methods are inherently rare, and the complex closed-form solutions (especially **our sandwiched least-squares problem**) involved can pose a challenge to readers. Thus, we opted not to introduce greater structural complexity that could increase the reading burden.
>
> In summary, we truly admire your professional insight! **Inspired by your review, we will incorporate the above discussion and analyses into our final paper**. We believe including these contents is highly helpful for **enhancing the depth and quality of our paper**, and it is particularly promising for driving a series of foreseeable future research based on our DeepAFL.
>
> P.S. Your suggestion to incorporate a reconstruction-based objective is very interesting. We think your idea is indeed feasible by extending the closed-form solution of our original sandwiched least-squares problem. Yet, we were concerned that deriving the exact closed-form solution might be **overly complex** in our first-period rebuttal. Given that we have introduced other evidence and analyses, we decided not to elaborate on this issue in detail here to avoid making our first-period rebuttal overly lengthy. If you are indeed interested in it, please let us know, and we would certainly be delighted to discuss it further with you then.
>
> ----
>
> This concludes our responses for your first-period comments.
>
> The process of responding to your questions **prompted significant reflection and yielded substantial new insights for us**, helping us to gain a deeper understanding of our own DeepAFL and seriously consider its potential future work directions.
>
> It is really a pleasure and an honor to engage in such a meaningful discussion with a professional of your caliber!
>
> We sincerely hope that our detailed explanations and clarifications can adequately address your concerns. Please feel free to raise any further questions you may have.
>
> **We would be extremely grateful if you would consider increasing your score to further support our work.**
>
> Once again, thank you very much for your valuable review. Looking forward to your feedback!

---

> ### Author Response · Authors · 2025-11-26
> **Additional Results for New Baselines (W3)**
>
> Dear Reviewer ZKVc:
>
> As promised in our initial response regarding your comment W3, we have now completed the additional comparative experiments with the two new baselines you recommended, **FedAWA and FedLPA**.
>
> We are pleased to present these additional results to fully address your concerns.
>
> Specifically, we implemented FedAWA and FedLPA across the three datasets utilized in our experiment: CIFAR-10, CIFAR-100, and ImageNet-R.
>
> All experimental settings were maintained to be consistent with the other baselines established in our paper. Specifically, because FedLPA is classified as a one-shot communication FL method, and we empirically observed that its performance essentially converges within 50 local epochs, we set the local epochs to 50 for this baseline.
>
> The specific results on CIFAR-10 are as follows:
>
> |                    | $\alpha=0.1$ | $\alpha=0.05$ |
> | :----------------: | :----------: | :-----------: |
> |       FedAWA       |    79.42%    |    76.37%     |
> |       FedLPA       |    49.57%    |    42.56%     |
> |        AFL         |    80.75%    |    80.75%     |
> | DeepAFL $(T = 5)$  |    85.20%    |    85.20%     |
> | DeepAFL $(T = 10)$ |    85.93%    |    85.93%     |
> | DeepAFL $(T = 20)$ |    86.43%    |    86.43%     |
>
> The specific results on CIFAR-100 are as follows:
>
> |                    | $\alpha=0.1$ | $\alpha=0.01$ |
> | :----------------: | :----------: | :-----------: |
> |       FedAWA       |    58.94%    |    50.48%     |
> |       FedLPA       |    40.04%    |    22.55%     |
> |        AFL         |    58.56%    |    58.56%     |
> | DeepAFL $(T = 5)$  |    64.72%    |    64.72%     |
> | DeepAFL $(T = 10)$ |    65.96%    |    65.96%     |
> | DeepAFL $(T = 20)$ |    66.98%    |    66.98%     |
>
> The specific results on Tiny-ImageNet are as follows:
>
> |                    | $\alpha=0.1$ | $\alpha=0.01$ |
> | :----------------: | :----------: | :-----------: |
> |       FedAWA       |    59.07%    |    52.33%     |
> |       FedLPA       |    39.48%    |    26.67%     |
> |        AFL         |    54.67%    |    54.67%     |
> | DeepAFL $(T = 5)$  |    60.31%    |    60.31%     |
> | DeepAFL $(T = 10)$ |    61.37%    |    61.37%     |
> | DeepAFL $(T = 20)$ |    62.35%    |    62.35%     |
>
> In terms of **accuracy**, FedAWA exhibits very strong results, approaching (on CIFAR-10) or even surpassing AFL (on CIFAR-100 and Tiny-ImageNet) when $\alpha=0.1$. Notably, even though FedAWA outperforms AFL, our DeepAFL still consistently achieves the best performance across all scenarios.
> Furthermore, as a gradient-based one-shot method, FedLPA's performance is compromised by data heterogeneity, as it lacks the inherent invariance demonstrated by our DeepAFL. Consequently, despite our best efforts to tune its hyperparameters, FedLPA still performs poorly. Furthermore, as the degree of Non-IID data increases, a pronounced performance degradation is observed for both of the newly introduced gradient-based baselines (i.e., FedAWA and FedLPA). This observation further highlights the advantage of the inherent invariance property of our DeepAFL.
>
> In terms of **efficiency**, using the CIFAR-100 dataset as an example, FedAWA requires approximately 10 hours, while FedLPA requires approximately 2.5 hours. In sharp contrast, our DeepAFL completes the task in less than 100 seconds (specifically, 91.74 seconds), achieving a speedup exceeding $90\times$. Notably, even though FedLPA is also classified as a one-shot FL approach, and we have already minimized its number of local epochs to fully reflect its efficiency potential, its overall overhead remains significantly higher than that of DeepAFL. This phenomenon is primarily due to FedLPA still being gradient-based and facing time-consuming backpropagation. Conversely, our DeepAFL is in a forward-only manner, further highlighting its efficiency advantage beyond its one-shot nature. In terms of **efficiency**, using the CIFAR-100 dataset as an example, FedAWA requires approximately 10 hours, while FedLPA requires approximately 2.5 hours. In contrast, our DeepAFL completes in less than 100 seconds, achieving a speedup exceeding $90\times$.
>
> In summary, compared to the new baselines you recommended, **our DeepAFL maintains substantial advantages in both accuracy and efficiency**, further highlighting its superiority.
>
> We believe these additional comparisons significantly strengthen our paper's evaluation, and we have already incorporated these results into our revised PDF (*Appendix I*, Page 41).
>
> With this, all of our results and analyses in response to your initial comments are now complete.
> Thank you again for your valuable guidance and patience!
>
> **We sincerely hope that our response has adequately addressed your concerns. And we would be extremely grateful if you would consider increasing your score to further support our work.**
>
> We are pleased to address any further concerns you may have, and we wish you all the best!

---

> ### Author Response · Authors · 2025-12-03
> **Additional Analyses and Results for Incorporating A Reconstruction-Based Objective (Q2, Part 1 of 2)**
>
> Dear Reviewer ZKVc:
>
> In our initial responses to your Q2, we briefly outlined the objective function that incorporates a reconstruction-based term, as you suggested. However, we did not elaborate on this issue in detail at that time to avoid making our first-period rebuttal overly lengthy.
>
> We previously stated:
>
> > "If you are indeed interested in it, please let us know, and we would certainly be delighted to discuss it further with you then."
>
> In light of the unexpected circumstances regarding the recent ICLR updates, we realize and understand that you may still be interested in this topic but unable to express your interest further. **Therefore, to fully address all your potential concerns in a responsible manner, we would like to provide detailed analyses and results for this approach.**
>
>
>
> ## Q2-new-1: Closed-Form Solution for the Multi-Objective Optimization Problem
>
> Specifically, let us revisit the multi-objective optimization problem we previously derived (in Part 8 of our initial responses to you):
>
> $\mathbf{\Omega}  _  {t+1}= \arg\underset{\mathbf{\Omega}}{\min} \ \rho \|\mathbf{Y}-(\mathbf{\Phi}  _  {t} + \mathbf{F}  _  t\mathbf{\Omega})\mathbf{W}  _  t\|^2  _  \text{F}+ \ (1-\rho) \|\mathbf{\Phi}  _  {0}-(\mathbf{\Phi}  _  {t} + \mathbf{F}  _  t\mathbf{\Omega})\mathbf{Q}  _  t\|^2  _  \text{F}+{\mathbf{\gamma} \| \mathbf{\Omega} \|^{2}  _  \text{F}}$
>
> Here, the **first term** is the original empirical risk minimization (prediction MSE), the **second term** is the reconstruction-based objective (similar to an autoencoder with MSE) you suggested, and the **third term** is the regularization. $\rho$ acts as a weighting factor to balance the prediction MSE and the reconstruction MSE.
>
> We define two residuals as $\mathbf{R} _ t^{W} = \mathbf{Y} - \mathbf{\Phi} _ {t}\mathbf{W} _ t$ and $\mathbf{R} _ t^{Q} = \mathbf{\Phi} _ {0} - \mathbf{\Phi} _ {t}\mathbf{Q} _ t$. The optimization objective can then be reformulated into a ***"sandwiched" form***:
>
> $\mathbf{\Omega}  _  {t+1}= \arg\underset{\mathbf{\Omega}}{\min} \ \rho \|\mathbf{R} _ t^{W} - \mathbf{F} _ t\mathbf{\Omega}\mathbf{W} _ t\|^2 _ \text{F}+ \ (1-\rho) \|\mathbf{R} _ t^{Q} - \mathbf{F} _ t\mathbf{\Omega}\mathbf{Q} _ t\|^2 _ \text{F}+{\mathbf{\gamma} \| \mathbf{\Omega} \|^{2}  _  \text{F}}$
>
> This can be seen as an extension of our original sandwiched least squares, as it involves **two sandwiches**, $\mathbf{F} _ t\mathbf{\Omega}\mathbf{W} _ t$ and $\mathbf{F} _ t\mathbf{\Omega}\mathbf{Q} _ t$, respectively.
>
> We derived the **closed-form solution** for the above optimization objective as follows:
>
> $\mathbf{\Omega} _ {t+1} = \mathbf{V} _ t\left[(\mathbf{V} _ t^\top \mathbf{O} _ {t} \mathbf{U} _ t) \oslash \left(\gamma\mathbf{1} + \text{diag}(\mathbf{\Lambda} _ t^\text{F}) \otimes \text{diag}(\mathbf{\Lambda} _ t^\text{WQ})\right)\right] \mathbf{U} _ t^\top$
>
> Here, we have:
>
> * $\mathbf{O} _ {t}= \rho(\mathbf{F} _ t^\top \mathbf{R} _ t^{W} \mathbf{W} _ t^\top)+ (1-\rho) (\mathbf{F} _ t^\top \mathbf{R} _ t^{Q} \mathbf{Q} _ t^\top)$
> * $\rho \mathbf{W} _ t \mathbf{W} _ t^\top +(1-\rho) \mathbf{Q} _ t \mathbf{Q} _ t^\top= \mathbf{U} _ t \mathbf{\Lambda} _ t^\text{WQ} \mathbf{U} _ t^\top$
> * $\mathbf{F} _ t^\top \mathbf{F} _ t = \mathbf{V} _ t \mathbf{\Lambda} _ t^\text{F} \mathbf{V} _ t^\top$
>
> While the formula might appear complex, it is actually quite **intuitive**. Specifically, it introduces weighted terms into our original sandwiched least squares closed-form solution. If we take the extreme case of $\rho=1$, the above formula directly degenerates to Equation (8) of our paper.
>
> Thus, we can adjust the procedure of our DeepAFL based on this new design. When solving for each analytic layer, we can calculate an additional $\mathbf{Q} _ t$ after computing $\mathbf{W} _ t$, maintaining a similar closed-form solution for both:
>
> $\mathbf{W} _ t = (\mathbf{\Phi} _ t^\top \mathbf{\Phi} _ t + \lambda\mathbf{I})^{-1} \mathbf{\Phi} _ t^\top \mathbf{Y}$
>
> $\mathbf{Q} _ t = (\mathbf{\Phi} _ t^\top \mathbf{\Phi} _ t + \lambda\mathbf{I})^{-1} \mathbf{\Phi} _ t^\top \mathbf{\Phi} _ {0}$
>
> Subsequently, $\mathbf{\Omega} _ {t+1}$ is derived using the solution we introduced above, and the feature $\mathbf{\Phi}  _  {t+1}$ is updated using the similar residual approach:
>
> $\mathbf{\Phi}  _  {t+1} =\mathbf{\Phi}  _  {t}+\sigma(\mathbf{\Phi}  _  {t}\mathbf{B}  _  {t})\mathbf{\Omega}  _  {t+1}$
>
> Following this line of thought, we have recently conducted additional experiments to investigate whether your proposed scheme for incorporating a reconstruction-based objective is suitable for our gradient-free learning.
>
> Regrettably, despite our best efforts in hyper-parameter tuning, the performance of the modified DeepAFL did not show significant improvement.
>
> Due to space constraints, we will provide the detailed experimental results in the next comment block.

---

> > ### Author Response · Authors · 2025-12-03
> > **Additional Analyses and Results for Incorporating A Reconstruction-Based Objective (Q2, Part 2 of 2)**
> >
> > ## Q2-new-2: Experimental Results for the Multi-Objective Optimization Problem
> >
> > Here, we provide the detailed results of the aforementioned modified approach.
> >
> > Specifically, we adjusted the weighting factor $\rho \in [0.0, 1.0]$ with an interval of $0.1$ to obtain 11 different sets of results, as shown in the table below:
> >
> > | Test  Acc. (%)   | $\rho=1.0$ | $\rho=0.9$ | $\rho=0.8$ | $\rho=0.7$ | $\rho=0.6$ | $\rho=0.5$ | $\rho=0.4$ | $\rho=0.3$ | $\rho=0.2$ | $\rho=0.1$ | $\rho=0.0$ |
> > | ---------------- | ---------- | ---------- | ---------- | ---------- | ---------- | ---------- | ---------- | ---------- | ---------- | ---------- | ---------- |
> > | DeepAFL $(T=0)$  | 60.81      | 60.81      | 60.81      | 60.81      | 60.81      | 60.81      | 60.81      | 60.81      | 60.81      | 60.81      | 60.81      |
> > | DeepAFL $(T=5)$  | 64.62      | 64.57      | 64.58      | 64.60      | 64.59      | 64.55      | 64.57      | 64.54      | 64.43      | 64.46      | 60.87      |
> > | DeepAFL $(T=10)$ | 65.98      | 65.82      | 65.87      | 65.78      | 65.73      | 65.89      | 65.97      | 65.85      | 65.94      | 65.79      | 60.89      |
> > | DeepAFL $(T=20)$ | 66.95      | 66.80      | 66.82      | 66.92      | 66.84      | 67.01      | 66.85      | 66.79      | 66.88      | 66.79      | *1.15      |
> >
> > From the results above, it is evident that the newly introduced reconstruction-based objective (similar to an autoencoder with MSE) does not improve the performance of DeepAFL.
> >
> > Specifically, the modified approach exhibits its best performance of **$67.01\%$ at $T=20$ when $\rho=0.5$**. However, this is only a marginal increase of $0.06\%$ compared to our original DeepAFL (i.e., $\rho=1.0$).
> >
> > We believe such a minor performance gain is highly likely to be attributed to random chance. Moreover, the performance of this modified approach is worse than our original DeepAFL ($\rho=1.0$) in the vast majority of other settings.
> >
> > At the other extreme, when $\rho=0.0$, the original empirical risk minimization term is totally disabled. This setting is equivalent to directly optimizing a deep auto-encoder for only reconstruction in a gradient-free manner. In this case, the performance improvement remains negligible even as the number of analytic layers $T$ increases. Furthermore, numerical issues occur when the number of analytic layers is large ($T=20$).
> >
> > We hypothesize that the underlying reason for these results is that simultaneously optimizing for both empirical risk reduction and reconstruction may **hinder DeepAFL's ability to effectively optimize feature representations**, especially given that the learning process itself is **gradient-free**.
> >
> > In essence, we consider that this reconstruction loss acts primarily as a **regularizer** within our DeepAFL, constraining the features from deviating excessively and requiring them to retain information from the initial feature $\mathbf{\Phi} _ {0}$.
> >
> > However, since our DeepAFL already includes an $L _ 2$ regularization term, this additional reconstruction loss is unlikely to provide effective gains as a regularizer. Instead, it may thus limit the model's performance due to over-constraining the model parameters.
> >
> > Let us now reflect on the alternative strategy proposed in our initial response (i.e., directly utilizing the initial feature $\mathbf{\Phi} _ {0}$ as the source) and its corresponding results (in Part 8 of our initial responses to you), where we did demonstrate that **reducing redundant representations can contribute to better performance** of DeepAFL.
> >
> > It is worth noting that the new finding here (that adding reconstruction loss does not yield better performance) is **not contradictory** to our previous conclusion. The main reason is that the potential direct negative effect of the added reconstruction loss may outweigh its indirect positive effect in reducing redundant representations.
> >
> > It suggests that future efforts aimed at optimizing DeepAFL must identify that different strategies for achieving the same objective (e.g., reducing redundant representations) can have vastly different impacts. Thus, careful design and comparison are essential to selecting truly effective strategies.
> >
> > We sincerely hope that our additional analyses can help you better understand our DeepAFL and its gradient-free learning manner.
> >
> > Thank you again for your support of our work, even though you are no longer able to discuss it further with us.

---

### Official Review · Reviewer_ofER · 2025-10-29

**Soundness:** 2
**Presentation:** 4
**Contribution:** 2
**Rating:** 6
**Confidence:** 4

**Summary:**

This paper proposes DeepAFL, an extension of Analytic Federated Learning (AFL) that introduces analytic residual blocks and deeper analytic layers to enable federated representation learning without gradient calculation and communication. The method alternates between solving for the classifier and transformation matrices​ in closed form, using aggregated feature–label correlation matrices across clients. This paper provides theoretical proofs for heterogeneity invariance and monotonic empirical risk reduction, and the analysing of privacy and efficiency. Experiments evaluate DeepAFL on CIFAR-10, CIFAR-100, and Tiny-ImageNet under Non-IID settings, showing 5–8% accuracy gains over gradient-based FL baselines (FedAvg, FedGen, FedDisco, etc) and the original AFL.

**Strengths:**

1. The paper is well-written and clearly structured, with equations and derivations easy to follow. The figures and notations are well explained.

2. Extending analytic learning to a deeper residual form and formulating closed-form residual updates is technically neat and easily understood.

3. This work provides sound theoretical analyses to support the heterogeneous invariance and representation learning capabilities.

4. Due to no backpropagation, this work achieves quite optimization cost reduction, which is demonstrated by experiments.

**Weaknesses:**

### **1. The experimental setup is methodologically unfair.**

DeepAFL aggregates global feature–label statistics and computes a closed-form model that the authors explicitly state is identical to centralized analytic learning, hence it is inherently immune to data heterogeneity. In contrast, gradient-based baselines (FedAvg, FedProx, MOON, etc.) naturally degrade under Non-IID settings. Evaluating all methods on strongly heterogeneous partitions, therefore, gives DeepAFL an artificial advantage: the method bypasses heterogeneity by design while the baselines must deal with it.

A fair comparison should instead be conducted under IID conditions or against a centralized analytic baseline, so that any remaining advantage would reflect DeepAFL’s genuine strengths in efficiency or analytic formulation rather than the absence of gradient-related heterogeneity effects.

### **2. The representation learning ability is not convincing.**

While the paper repeatedly emphasizes that *DeepAFL achieves deep analytic representation learning*, the method it introduces does not actually justify that claim. According to Section 3.1, each residual analytic block is defined as
$g_{t+1}(\Phi_t) = \sigma(\Phi_t B_t)\Omega_{t+1},$
where $B_t$ is a random projection matrix, $\sigma(\cdot)$ is a nonlinear activation, and $\Omega_{t+1}$ is computed analytically via least squares. This design simply reprojects and reweights the *fixed backbone features* rather than creating new or more abstract representations. All transformations occur within the same frozen feature space extracted by the pretrained backbone, without modifying or adapting the backbone itself. Consequently, DeepAFL indeed enhances the expressiveness of the analytic mapping in this fixed space, but it does not improve the representational capacity or transferability of the underlying model. In other words, the pretrained backbone remains the real bottleneck of representation learning, and DeepAFL does not attempt to address it.

Moreover, the theoretical and empirical evidence provided does not demonstrate genuine feature learning. Theorem 2 merely proves that the empirical risk decreases monotonically with layer depth, a sign of better fitting, not of learned representations. Similarly, the “representation analyses” in Section 4.2 only report accuracy gains with increasing layers, which reflect improved regression capability but not new feature semantics or cross-domain adaptability.

### **3.The overall problem setting is conceptually misguided.**
DeepAFL assumes that all clients share a frozen, pre-trained backbone such as an ImageNet-trained ResNet-18.
However, if this backbone has never been trained on data related to the clients’ tasks, its extracted features may already be poorly aligned with the actual data distributions.
This feature-domain mismatch is exactly the challenge that federated learning is meant to address—adapting a shared model to diverse client data without violating privacy.
The authors themselves note that “a domain shift between the backbone’s pre-training data and the FL training data can further impact AFL’s performance”, yet the proposed method and theory ignore this issue.
All experiments assume that the backbone provides perfectly relevant and transferable features, effectively sidestepping the main difficulty that would justify using FL.
To be meaningful, DeepAFL should be analyzed and validated in settings where the backbone features are imperfect or mismatched with client data, since that is where its claimed advantages would actually matter.

**Questions:**

1. Could the authors clarify the real motivation and use case of DeepAFL? If all clients already share a strong frozen backbone, what problem is this method actually addressing? Would it make more sense to frame DeepAFL as a collaborative fine-tuning approach, where clients adapt a shared public backbone to private data with different feature distributions?

2. The current experiments assume that the backbone provides perfect and well-aligned features. Could the authors evaluate DeepAFL under domain-gap conditions, where the backbone has never seen the clients’ data or task, to test whether the proposed analytic layers can compensate for feature misalignment?

3. Please include results under IID partitions or with a centralized analytic baseline to verify whether DeepAFL’s improvements come from genuine analytic efficiency rather than its immunity to Non-IID effects.

4. What exactly does “representation learning” mean in this work? If the backbone is frozen and the residual blocks only reweight existing features, are new representations actually being learned? Some analysis of intermediate features, such as layer-wise visualization, separability metrics, or similarity, would help substantiate this claim.

5. Theoretical analyses currently assume ideal, fixed backbone features. Do the invariance and convergence results still hold when the features are imperfect, noisy, or mismatched with the clients’ data? If not, discussing or extending the theory to such realistic cases would greatly improve the paper’s relevance.

---

> ### Author Response · Authors · 2025-11-22
> **Author Response to Reviewer ofER (Part 1 of 12)**
>
> We sincerely thank you for your thorough feedback and positive recognition.
>
> Here, we would like to respectfully address your raised Weakness (W) and Question (Q) below.
>
> # W1 & Q3: Additional Results of the IID Condition and Centralized Analytic Baseline
>
> > DeepAFL aggregates global feature–label statistics and computes a closed-form model that the authors explicitly state is identical to centralized analytic learning, hence it is inherently immune to data heterogeneity. In contrast, gradient-based baselines (FedAvg, FedProx, MOON, etc.) naturally degrade under Non-IID settings. Evaluating all methods on strongly heterogeneous partitions, therefore, gives DeepAFL an artificial advantage: the method bypasses heterogeneity by design while the baselines must deal with it.
> >
> > A fair comparison should instead be conducted under IID conditions or against a centralized analytic baseline, so that any remaining advantage would reflect DeepAFL’s genuine strengths in efficiency or analytic formulation rather than the absence of gradient-related heterogeneity effects.
>
> Thank you for your constructive comment.
>
> First of all, we would like to emphasize that **Non-IID data** is widely acknowledged as a **pervasive issue** in FL, and virtually most recent work on FL attempts to mitigate this issue. Consequently, we maintain that effectively addressing the Non-IID challenge is a **critical and practical concern** within the FL community, and is **by no means an artificial setting**.
>
> Thus, we argue that the perfect IID conditions you posit for fair comparisons are **nearly non-existent** in real-world FL systems and consequently lack practical significance. Furthermore, if IID conditions were considered the sole benchmark for comparisons, then the **utility of most recent advances in gradient-based FL methods would also appear insignificant**, as they **tend to perform similarly to, or even worse than**, the traditional FedAvg under perfect IID conditions. In summary, we think that using the perfect IID conditions as the **primary experimental setup** would appear even more unfair and impractical.
>
> Yet, we fully agree that providing **more results in addition to our main Non-IID setting** would be beneficial as supplementary material to strengthen the rigor of our paper. Therefore, we are very happy to accept your suggestion in Q3:
>
> > Please include results **under IID partitions** or **with a centralized analytic baseline** to verify whether DeepAFL’s improvements come from genuine analytic efficiency rather than its immunity to Non-IID effects.
>
> Specifically, we use FedAvg as an example to analyze its performance under Dirichlet distributions ($\alpha=0.005, 0.01, 0.1, 1$) and the completely IID condition. Moreover, as you suggested including a centralized analytic baseline for comparison, we further introduced the *Centralized Analytic Probe* to enhance the richness of this analysis. Theoretically, the results of the *Centralized Analytic Probe* should be identical to those of AFL, due to AFL's invariance to data heterogeneity.
>
> The specific comparison results are shown below:
>
> | Acc. (%) | $\alpha=0.005$ | $\alpha=0.01$ | $\alpha=0.1$ | $\alpha=1$ | IID   |
> | -- | -- | -- | -- | -- | -- |
> | FedAvg | 24.74 | 33.09 | 56.57 | 57.72 | 57.89 |
> | Centralized Analytic Probe | 58.56 | 58.56 | 58.56 | 58.56 | 58.56 |
> | AFL    | 58.56 | 58.56 | 58.56 | 58.56 | 58.56 |
> | DeepAFL (T=20) | 66.98 | 66.98 | 66.98 | 66.98 | 66.98 |
>
> **Notably, for fair comparison, the FedAvg results here, across these different settings, are directly obtained from the data provided in the original paper of AFL.** Since we have previously discussed that other gradient-based methods (which are typically improvements upon FedAvg) generally do not outperform FedAvg under completely IID conditions, we believe using FedAvg as the representative example for this IID analysis is sufficient.
>
> From the above results, it is evident that as the degree of Non-IID decreases, the performance of the gradient-based baseline (i.e., FedAvg) improves gradually as expected, reaching its best result under the idealized IID condition.
>
> However, even under the IID scenario, the performance of FedAvg remains weaker than that of the *Centralized Analytic Probe*, as its distributed aggregation introduces a performance loss, making it inferior to the centralized analytic result. In contrast, AFL, benefiting from its invariance, maintains performance identical to that of the *Centralized Analytic Probe*. Based on it, our DeepAFL, benefiting from its representation learning capability, further achieves a stable performance improvement over AFL while preserving its invariance to data heterogeneity.
>
> Thank you again for your valuable comment and suggestion. We will incorporate these results and analyses into our final paper. We sincerely hope that our detailed explanations and clarifications can adequately address your concerns.

---

> ### Author Response · Authors · 2025-11-22
> **Author Response to Reviewer ofER (Part 2 of 12)**
>
> # W2 & Q4: Clarification on Representation Learning
>
> > While the paper repeatedly emphasizes that *DeepAFL achieves deep analytic representation learning*, the method it introduces does not actually justify that claim. According to Section 3.1, each residual analytic block is defined as
> >
> > $g _ {t+1}(\mathbf{\Phi} _ {t}) = \sigma(\mathbf{\Phi} _ {t}\mathbf{B} _ {t})\mathbf{\Omega} _ {t+1}$
> >
> > where is a random projection matrix, is a nonlinear activation, and is computed analytically via least squares. This design simply reprojects and reweights the *fixed backbone features* rather than creating new or more abstract representations. All transformations occur within the same frozen feature space extracted by the pretrained backbone, without modifying or adapting the backbone itself. Consequently, DeepAFL indeed enhances the expressiveness of the analytic mapping in this fixed space, but it does not improve the representational capacity or transferability of the underlying model. In other words, the pretrained backbone remains the real bottleneck of representation learning, and DeepAFL does not attempt to address it.
> >
> > Moreover, the theoretical and empirical evidence provided does not demonstrate genuine feature learning. Theorem 2 merely proves that the empirical risk decreases monotonically with layer depth, a sign of better fitting, not of learned representations. Similarly, the “representation analyses” in Section 4.2 only report accuracy gains with increasing layers, which reflect improved regression capability but not new feature semantics or cross-domain adaptability.
>
> > What exactly does “representation learning” mean in this work? If the backbone is frozen and the residual blocks only reweight existing features, are new representations actually being learned? Some analyses of intermediate features, such as layer-wise visualization, separability metrics, or similarity, would help substantiate this claim.
>
> Thank you very much for your thoughtful comment and question regarding the representation learning capabilities of our deepAFL.
>
> We understand your main concerns and have summarized them as follows:
>
> 1. **Conceptualization:** What exactly does representation learning mean in this work?
> 2. **Justification:** If the backbone is frozen and the residual blocks only re-weight existing features, are new representations actually being learned?
> 3. **Empirical Evidence:** Is there sufficient evidence of intermediate features (e.g., separability) to substantiate the claim of representation learning?
> 4. **Theoretical Generalization:** Is proving the empirical risk decreases monotonically with layer depth sufficient to demonstrate better generalization of learned representations?
>
> We sincerely appreciate this opportunity to present our clarification, which we will address point-by-point in the following 4 comment blocks.

---

> ### Author Response · Authors · 2025-11-22
> **Author Response to Reviewer ofER (Part 3 of 12)**
>
> ## W2 & Q4 (1): The Capability of Our DeepAFL Aligns with the Concept of Representation Learning.
>
> Here, we mainly aim to address your concern regarding what representation learning means in our work.
>
> First of all, please allow us to commence by reiterating a classic description derived from the seminal work [1] written by **Yoshua Bengio et al.** in 2014:
>
> > “In order to expand the scope and ease of applicability of machine learning, **it would be highly desirable to make learning algorithms less dependent on feature engineering**, so that novel applications could be constructed faster, and more importantly, to make progress towards AI.”
> >
> > "This paper is about representation learning, i.e., **learning representations of the data that make it easier to extract useful information when building classifiers or other predictors**."
> >
> > "A good representation is also one that **is useful as input to a supervised predictor**."
> >
> > "Among the various ways of learning representations, **this paper focuses on deep learning methods: those that are formed by the composition of multiple non-linear transformations, with the goal of yielding more abstract – and ultimately more useful – representations.**"
>
> Meanwhile, to maintain generality and breadth, **we also cite the description of representation learning provided by *Wikipedia*** [2]:
>
> > "In machine learning (ML), **feature learning or representation learning is a set of techniques that allow a system to automatically discover the representations needed for feature detection or classification from raw data. This replaces manual feature engineering and allows a machine to both learn the features and use them to perform a specific task.**"
> >
> > "Labeled data includes input-label pairs where the input is given to the model, and it must produce the ground truth label as the output. **This can be leveraged to generate feature representations with the model which result in high label prediction accuracy.** Examples include supervised neural networks, multilayer perceptrons, and dictionary learning."
> >
> > "Feature learning is intended to **result in faster training or better performance in task-specific settings** than if the data was input directly."
>
> Conceptually, we summarize **two essential core characteristics** of representation learning:
>
> * **Automated Feature Extraction:** It involves learning feature transformations (which are often non-linear) directly from the data, thereby avoiding the complexity and reliance on expert knowledge associated with manual feature engineering.
> * **Utility for Downstream Tasks:** The extracted features are beneficial and useful for adapting to specific downstream tasks, meaning they provide **utility when building predictors (e.g., classifiers)**, ultimately leading to enhanced model performance.
>
> Our DeepAFL directly satisfies these two characteristics:
>
> - The **feature transformations** involve trainable parameters $\mathbf{\Omega}$, which are **automatically learned from the data** via the “sandwiched” least squares. Moreover, the use of the activation function $\sigma(\cdot)$ ensures that the layer-wise transformation is **non-linear**.
> - The feature transformations show **evident and significant benefits for our downstream task (i.e., classification)**. Specifically, when paired with an analytic classifier, the accuracy of our DeepAFL consistently improves on the training & test sets as $T$ increases (as shown in Tables 9-11 of our paper).
>
> In summary, **we maintain that the capability of our DeepAFL indeed aligns with the concept of representation learning**.
>
> Moreover, we want to offer more points regarding your comment below:
>
> > DeepAFL indeed enhances the expressiveness of the analytic mapping in this fixed space, but it does not improve the representational capacity or **transferability** of the underlying model.
>
> We argue that the "transferability" mentioned is not a necessary condition for representation learning but pertains to another field: transfer learning. In fact, the vast majority of current gradient-based supervised methods **also fall far short of transferability**, as they tend to focus narrowly on the current task and suffer from out-of-distribution data.
>
> Moreover, as the development of analytic-learning-based methods is still in its nascent stages, our current core aim is to **achieve direct performance improvement with marginal additional cost**.
>
> While we do not deny that further exploring how to "improve the transferability of the underlying model" is a promising and interesting direction for future work, **it clearly exceeds the scope of our current work**.
>
> We believe that by clarifying the concept of representation learning and its connection with our DeepAFL, we can further enhance the rigor and persuasiveness of our paper. We are actively working on incorporating these clarifications and discussions into our final paper.
>
> Once again, thank you for your valuable question!

---

> ### Author Response · Authors · 2025-11-22
> **Author Response to Reviewer ofER (Part 4 of 12)**
>
> ## W2 & Q4 (2): Re-Weighting/Re-Using Features is a Defining Characteristic of Deep Representation Learning
>
> Next, we wish to discuss the learning process of our DeepAFL (i.e., re-weighting/re-using existing features) and show why it is **not in conflict with our claim of deep representation learning**.
>
> Specifically, this response addresses your following comments:
>
> > This design simply **reprojects** and **reweights** the *fixed backbone features* rather than creating new or more abstract representations.
> >
> > If the backbone is frozen and the residual blocks only **reweight** existing features, are new representations actually being learned?
>
> Here, we want to highlight that our **re-weighting** existing features is a form of **re-using the features learned in the previous layer**. Critically, the re-use of features is precisely recognized as **a canonical characteristic and advantage of deep representation learning** [1], as articulated by **Yoshua Bengio et al**.:
>
> > "**Deep architectures promote the re-use of features.**"
> >
> > "The notion of **re-use**, which explains the power of distributed representations, **is also at the heart of the theoretical advantages behind deep learning**, i.e., constructing multiple levels of representation or learning a hierarchy of features."
> >
> > "The typical computations we allow in each node include: **weighted sum, product, artificial neuron model (such as a monotone nonlinearity on top of an affine transformation)**, computation of a kernel, or logic gates."
>
> From the authoritative citation, we know that one key reason why deep learning possesses representation learning capabilities is precisely its re-use of features. Thus, we argue that **denying or challenging the notion that our DeepAFL learns genuine representations only based on its re-weighting of features is conceptually flawed and misguided.**
>
> To provide you with a more intuitive understanding of this point, please allow us to **draw an analogy between our DeepAFL and the Multi-Layer Perceptron (MLP)**.
>
> Given that the MLP is recognized as the **most fundamental DNN**, if we were to claim that an MLP trained via back-propagation achieves representation learning, you may undoubtedly and readily concur.
>
> Since our DeepAFL incorporates a residual block structure, we also introduce skip connections into the MLP. We believe you would also agree that this modification does not impair its representation learning capability (but rather makes it easier to train).
>
> At this point, what the residual block of the MLP learns is nothing more than subjecting the features from the previous layer to an **affine transformation $\mathbf{W} _ {t+1}$** followed by an **activation function $\sigma(\cdot)$**, where the affine transformation $\mathbf{W} _ {t+1}$ constitutes the learnable parameters of the MLP, as follows:
>
> $g _ {t+1}(\mathbf{\Phi} _ {t}) = \sigma(\mathbf{\Phi} _ {t} \mathbf{W} _ {t+1})$
>
> It is thus straightforward to analyze that when multiple such layers are stacked, the MLP effectively performs nothing more than a **continuous alternation between affine transformations and non-linear activation functions**. In fact, this affine transformation $\mathbf{W} _ {t+1}$ in the MLP is simply a form of **re-weighting/re-using the features from the previous layer**.
>
> More importantly, if we consider the formulation of our DeepAFL, we can clearly see that the approach is **very similar to that of the MLP**, as it also involves the alternating stacking of affine transformations and non-linear activation functions:
>
> $g _ {t+1}(\mathbf{\Phi} _ {t}) = \sigma(\mathbf{\Phi} _ {t}\mathbf{B} _ {t})\mathbf{\Omega} _ {t+1}$
>
> The primary distinction is that the learnable parameters $\mathbf{W} _ {t+1}$ in the MLP act *inside* the activation function, whereas the learnable parameters $\mathbf{\Omega} _ {t+1}$ in our DeepAFL act *outside* the activation function, but this localized difference is effectively diminished when multiple layers are stacked.
>
> More complex DNN also **share this philosophy of feature re-weighting/re-use**. For instance, a CNN can be broadly viewed as an MLP with weight sharing and local connectivity.
>
> Thus, the **re-weighting/re-use of features is by no means a weakness of our DeepAFL**. Instead, it is a crucial underlying factor that enables our DeepAFL to realize the claimed deep representation learning.
> We can fully understand why this misinterpretation may arise, especially since **gradient-based methods currently dominate the field** and **gradient-free deep representation learning appears highly distinct from the main stream**.
>
> We believe that this is precisely **one of the significant contributions of our paper**, as it presents a currently rare and elegant deep representation learning framework in a gradient-free manner, promoting further attention and stimulating critical thinking in this less-explored domain.

---

> ### Author Response · Authors · 2025-11-22
> **Author Response to Reviewer ofER (Part 5 of 12)**
>
> ## W2 & Q4 (3): Further Empirical Evidence Supporting the Representation Learning Capabilities of Our DeepAFL
>
> Then, we would like to further address your question regarding more analyses of intermediate features, as follows.
>
> > Some analyses of intermediate features, such as layer-wise visualization, separability metrics, or similarity, would help substantiate this claim.
>
> We greatly appreciate your constructive suggestion regarding the analyses of intermediate features!
>
> First of all, we would like to respectfully highlight that **we have already provided quantitative intermediate analyses of training and testing accuracies in Tables 9-10 of our paper**.
>
> According to our analyses in W2 & Q4 (1), we argue that the **utility for our downstream task** (i.e., the classification accuracy) is the most direct and fundamental metric to validate the representation learning capabilities of DeepAFL. This perspective also aligns with the classical work by **Yoshua Bengio et al.**, which asserts that better linear separability is indicative of a more meaningful representation [3].
>
> To reinforce this point, we would like to reiterate the impressive performance gains achieved by DeepAFL on both training and testing sets. Let's take the results on the CIFAR-100 dataset as an example. The baseline AFL struggles with severe underfitting, achieving a training accuracy of only **61.55%**. In stark contrast, our DeepAFL (at $T=20$) propels the training accuracy to **85.15%**, representing a massive absolute improvement of **23.6%**. This improved representation translates directly to generalization, yielding a substantial **8.39%** increase in test accuracy.
>
> Crucially, these gains are achieved with high efficiency. The total runtime for DeepAFL is only **91.74 s**, which is a marginal increase of less than 42 s compared to AFL (50.05 s). These results compellingly show that DeepAFL learns significantly more discriminative and useful representations than AFL.
>
> Moreover, in response to your valuable request, we have introduced 3 more metrics to analyze the separability of the intermediate features learned by our DeepAFL, which further substantiates its superior representation learning capabilities.
>
> Specifically, we adopted the following 3 additional metrics for layer-wise feature analyses:
>
> * **Compactness-Separation- Ratio (CSR):** It evaluates the ratio of the average intra-class squared distance (compactness) to the average inter-class squared distance (separation). A **lower CSR value** indicates better separability.
> * **Inverse Fisher Score (IFS):** It is defined as the reciprocal of the Fisher Score, which is a widely adopted metric to distinguish different classes. The IFS is calculated by the ratio of the intra-class variance to inter-class variance. A **lower FS value** implies better separability.
> * **Discriminative Measure (DM):** It is a metric derived from the core principles of the Linear Discriminant Analysis (LDA), quantifying the ratio of within-class scatter to between-class scatter. Specifically, a **lower DM value** signifies better separability.
>
> In summary, all the additional metrics are unified such that a **lower** value indicates **better** representation separability. Thus, we conducted additional experiments on the CIFAR-100 dataset, and the results are presented below:
>
> | Layer | Training CSR | Testing CSR | Training IFS | Testing IFS | Training DM | Testing DM |
> | ----- |-|-|-|-|-|-|
> | 0     | 1.583 | 1.529 | 3.25 | 3.13 | 3.21 | 3.09 |
> | 5     | 1.565 | 1.520 | 3.2  | 3.11 | 3.16 | 3.07 |
> | 10    | 1.555 | 1.516 | 3.17 | 3.10 | 3.14 | 3.06 |
> | 15    | 1.546 | 1.513 | 3.15 | 3.09 | 3.12 | 3.05 |
> | 20    | 1.539 | 1.510 | 3.13 | 3.08 | 3.11 | 3.04 |
>
> From the results, it is evident that all additional metrics show better separability as the layer count of our DeepAFL increases on both the training and test sets, which further substantiates our DeepAFL's representation learning capabilities.
>
> Furthermore, we observe that the metrics on the **test set generally perform better (i.e., yield lower values)** than those on the training set. This is likely because the training set contains a much more samples, leading to greater inherent noise and, consequently, slightly inflated metric values.
>
> Moreover, we also acknowledge that our DeepAFL exhibits a diminishing marginal returns phenomenon as the layer count increases, but this is entirely expected and normal since we did not increase any FL training data volume.
>
> This phenomenon is also common in gradient-based methods (e.g., simply increasing the depth of a DNN can not lead to continuous accuracy improvement up to 100% without encountering a bottleneck). Even the high performance achieved by large models like LLMs is a result of scaling the model *alongside* vast increases in training data volume (i.e., the scaling law).
>
> Thank you again for your valuable comment, and we will incorporate these results into our final paper.

---

> ### Author Response · Authors · 2025-11-22
> **Author Response to Reviewer ofER (Part 6 of 12)**
>
> ## W2 & Q4 (4):  The Monotonic Decrease of Empirical Risk Can Indeed Be Extended to Better Generalization Error Bound in Theory
>
> Last but not least, we want to address your comment regarding the theoretical analyses in our paper:
>
> > Moreover, the theoretical and empirical evidence provided does not demonstrate genuine feature learning. Theorem 2 merely proves that the empirical risk decreases monotonically with layer depth, a sign of better fitting, not of learned representations.
>
> We wish to clarify that the monotonic decrease of empirical risk within our DeepAFL (as proven in Theorems 2 and 3 of our paper) is in fact a very **strong theoretical result**.
>
> Crucially, this conclusion can be straightforwardly **extended to imply a better Generalization Error Bound (GEB)** in statistical learning theory. Intuitively, this is because our DeepAFL does **not increase the complexity of the linear analytic classifier**. At each layer, we utilize the same analytically-optimized linear classifier based on the least-squares method, as defined in Equation (4). Specifically, during the layer-wise optimization process for representation learning in DeepAFL, the data sample size ($N$), the feature dimension ($d _ {\Phi}$), and the regularization parameter ($\lambda$ & $\gamma$) all remain constant.
>
> Below, we provide a brief theoretical explanation.
>
> In statistical learning theory, the Generalization Error Bound (GEB) typically follows the core form:
>
> $\mathcal{R}(f) \leq \hat{\mathcal{R}}(f) + \text{Complexity Term}$
>
> where:
>
> - $\mathcal{R}(f)$ is the **Generalization Risk** (or Generalization Error);
> - $\hat{\mathcal{R}}(f)$ is the **Empirical Risk** (or Empirical Error);
> - The **Complexity Term**, also known as the complexity gap, measures the model's complexity (usually related to the sample size, feature dimension, and the data's geometric structure).
>
> The size of the GEB, $\mathcal{B}(f)$, is what we are typically concerned with:
>
> $\mathcal{B}(f) = \hat{\mathcal{R}}(f) + \text{Complexity Term}$
>
> According to standard *VC-dimension* and *Rademacher* complexity generalization bounds [4], [5], for an analytic linear hypothesis class in $\mathbb{R}^d$, the generalization gap/complexity is bounded by $O\left(\frac{\sqrt{d _ {\Phi} \log (N/\delta)}}{\sqrt{N}}\right)$.
>
> In our analysis:
>
> - $N$ (data size) remains **constant**;
> - $d _ {\Phi}=1024$ (feature dimension) remains **constant**;
> - $\delta \in (0,1)$ (confidence level) remains **constant**.
>
> Therefore, the linear analytic classifiers at each layer of our DeepAFL can be considered to **share the same Complexity Term**.
>
> Notably, the above bound implicitly assumes that the input features $\Phi$ are bounded. This premise is satisfied in our DeepAFL because we incorporate constant **L2-normalization as a constraint** throughout the training process, as shown in Equations (4) and (7) of our paper.
>
> Now, considering the analytic classifiers $f _ t$ and $f _ {t+1}$ trained on the representations of layer $t$ and $(t+1)$ respectively, we have:
>
> $\mathcal{B}({f} _ {t}) = \hat{\mathcal{R}}(f _ {t}) + \text{Complexity Term} _ {t}$
>
> $\mathcal{B}({f} _ {t+1}) = \hat{\mathcal{R}}(f _ {t+1}) + \text{Complexity Term} _ {t+1}$
>
> As we have established that the **Complexity Term is constant** across layers, and our paper proves that the empirical risk is non-increasing (i.e., $\hat{\mathcal{R}}(f _ {t})\geq \hat{\mathcal{R}}(f _ {t+1})$), it directly follows that:
>
> $\mathcal{B}({f} _ {t})\geq \mathcal{B}({f} _ {t+1})$
>
> In this way, we have successfully shown that the theory introduced in our paper can be easily extended to deduce that the **GEB** of the analytic classifier is also **non-increasing**. This result provides **stronger evidence** for the **effectiveness and training-to-testing generalization** of our DeepAFL's representation learning capabilities.
>
> Thank you a lot for your valuable and constructive comment. It has reminded us that these important theoretical implications should be **more clearly elucidated** for readers to fully appreciate our DeepAFL's representation learning capabilities.
>
> We will incorporate these new analyses into our final paper. And we firmly believe this addition will significantly **enhance the quality, clarity, and theoretical soundness** of our manuscript.
>
> ## References
>
> [1] Bengio, Yoshua, Courville, Aaron, and Vincent, Pascal. Representation learning: A review and new perspectives. *IEEE Transactions on Pattern Analysis and Machine Intelligence*, 2013.
>
> [2] Wikipedia contributors. Feature learning. *Wikipedia*.
>
> [3] Alain, Guillaume, and Bengio, Yoshua. Understanding intermediate layers using linear classifier probes. *International Conference on Learning Representations (ICLR)*, 2016.
>
> [4] Vapnik, Vladimir N. An overview of statistical learning theory. *IEEE Transactions on Neural Networks*, 1999.
>
> [5] Mohri, Mehryar, Rostamizadeh, Afshin, and Talwalkar, Ameet. Foundations of machine learning. *MIT Press*, 2018.

---

> ### Author Response · Authors · 2025-11-22
> **Author Response to Reviewer ofER (Part 7 of 12)**
>
> # W3 & Q2: Clarifications on the Advantages and Limitations of DeepAFL
>
> > Deep assumes that all clients share a frozen, pre-trained backbone such as an ImageNet-trained ResNet-18. However, if this backbone has never been trained on data related to the clients’ tasks, its extracted features may already be poorly aligned with the actual data distributions. This feature-domain mismatch is exactly the challenge that federated learning is meant to address—adapting a shared model to diverse client data without violating privacy. The authors themselves note that “a domain shift between the backbone’s pre-training data and the FL training data can further impact AFL’s performance”, yet the proposed method and theory ignore this issue. All experiments assume that the backbone provides perfectly relevant and transferable features, effectively sidestepping the main difficulty that would justify using FL. To be meaningful, DeepAFL should be analyzed and validated in settings where the backbone features are imperfect or mismatched with client data, since that is where its claimed advantages would actually matter.
>
> > The current experiments assume that the backbone provides perfect and well-aligned features. Could the authors evaluate DeepAFL under domain-gap conditions, where the backbone has never seen the clients’ data or task, to test whether the proposed analytic layers can compensate for feature misalignment?
>
> Thank you very much for your constructive and insightful feedback.
>
> We believe there might be a slight misunderstanding regarding the **scope of the problem addressed by our DeepAFL**, as the dependence on the backbone has been repeatedly mentioned across your comments. Thus, we want to clarify our DeepAFL's core advantages and limitations here.
>
> Let's first recall that AFL operates by placing a **linear analytic classifier** after a fixed backbone. This analytic classifier is designed to possess a closed-form solution that exhibits invariance to data heterogeneity, thus offering dual advantages in both performance and efficiency in FL.
>
> Nevertheless, AFL has two fundamental and obvious limitations:
>
> 1. **Dependence on the Backbone:** AFL relies on the backbone as the feature extractor.
>
> 2. **Linear Classification Constraint:** AFL's analytic classifier is restricted to linear classification and often suffers from **underfitting**. When the features provided by the backbone have weak linear separability, AFL struggles to achieve optimal performance, even if the features contain enough information.
>
> In this paper, the core motivation behind our DeepAFL design was primarily to address the **second fundamental limitation** (i.e., the underfitting of analytic classifiers).
>
> We achieve this by employing a **gradient-free deep representation learning** approach to optimize the features outputted by the backbone. This optimization enhances the linear separability of the features, thereby resolving the underfitting issue that plagues the original AFL's linear analytic classifier. Crucially, since our DeepAFL also maintains a **gradient-free manner**, it retains all the excellent properties of AFL, such as **invariance** and **efficiency**.
>
> In fact, the experimental results in our Tables 9–11 provide sufficient evidence that DeepAFL successfully overcomes the underfitting issue. Taking the **CIFAR-100 dataset** as an example, the training set accuracy for AFL is only **61.55%**, which is a clear indication of underfitting, especially compared to its test set accuracy of **58.56%**.
>
> In stark contrast, our DeepAFL (at $T=20$) substantially elevates the training accuracy to **85.15%**, representing a massive absolute improvement of **23.6%**. This improved feature directly yields a substantial **8.39%** increase in test accuracy. Moreover, such a significant performance gain of our DeepAFL requires **negligible computational overhead** and further underscores its superior efficiency.
>
> We also want to emphasize that **the underfitting of analytic classifiers, which our DeepAFL solves, is a general, pervasive, and significant issue.** To a large extent, this issue persists even when the backbone itself is significantly improved. To illustrate this point, we refer to the results in our response to another reviewer (mCa1, W1), where we show DeepAFL's performance with larger ViT backbones.
>
> The results confirm that even with the ViT backbone, which is substantially more powerful than ResNet-18, AFL's analytic classifier still exhibits an underfitting phenomenon. Thus, the performance gain of our DeepAFL over standard AFL remains highly significant (with an improvement of up to **5.06%** when using the ViT-B-16-I-1K backbone). More importantly, the **computational overhead** introduced by our DeepAFL is nearly **negligible** when compared to the resources required to enhance the backbone during pre-training.
>
> Due to space constraints, we will elaborate on our DeepAFL's limitations in the next comment block.

---

> ### Author Response · Authors · 2025-11-22
> **Author Response to Reviewer ofER (Part 8 of 12)**
>
> # W3 & Q2: Clarifications on the Advantages and Limitations of DeepAFL (Continued)
>
> We believe that our preceding discussion clearly establishes the core contributions of our DeepAFL: the resolution of AFL's fundamental and significant limitation of underfitting, while preserving its closed-form solution with superior efficiency and invariance in FL.
>
> Now, let's turn our attention back to the first limitation of AFL mentioned previously (i.e., the dependence on the backbone), which is a persistent concern in your comment. We acknowledge that the current version of DeepAFL, by retaining its gradient-free nature, indeed inherits this limitation of the original AFL.
>
> Specifically, if the backbone suffers from a severe domain gap, the current mechanism of our DeepAFL cannot fundamentally resolve this issue. Let‘s consider an extreme case where the backbone is severely compromised and noisy, such that the mutual information between the features $\mathbf{\Phi} _ 0$ and the labels $\mathbf{Y}$ may become very low (or even uncorrelated). In this situation, it is mechanistically impossible for our DeepAFL to arbitrarily make the originally uncorrelated $\mathbf{\Phi} _ 0$ and $\mathbf{Y}$ linearly separable.
>
> This is because the current version of DeepAFL is essentially designed to optimize the geometrical alignment between the features $\mathbf{\Phi} _ 0$ and the labels $\mathbf{Y}$, thereby increasing their linear separability. **Logically, if the features provided by the backbone are intrinsically uncorrelated with (or have no mutual information with) $\mathbf{Y}$, then it is nearly impossible to optimize the features via gradient-free methods.**
>
> Given your strong emphasis on this point, we would like to further elaborate that the analytic learning community has already begun attempts to address **the dependence on the frozen backbone** [6], and we fully agree that integrating this technique with our DeepAFL to achieve even stronger performance is **a promising and interesting avenue for future work**.
>
> However, this current attempt to unfreeze the backbone fundamentally adopts a mixed gradient-based and gradient-free approach, which inevitably compromises the invariance property of the analytic classifier. **In contrast, the core philosophy and focus of our DeepAFL are distinctly different: we aim to provide a direct and efficient performance improvement for AFL while preserving its crucial theoretical advantages in the FL setting.**
>
> Notably, while we acknowledge that our DeepAFL has not eliminated the dependence on the backbone, we firmly believe that this does not diminish the primary advantage of DeepAFL: **granting the analytic classifier the capability of representation learning while preserving its gradient-free manner.** This key advantage allows our DeepAFL to move beyond merely learning linear mappings on the extracted features, enabling the learning of more complex non-linear relationships from features with high efficiency.
>
> Once again, thank you very much for your valuable feedback, which gave us the opportunity to clarify the contribution and scope of our DeepAFL more clearly. We apologize if the presentation of our contributions and advantages in the original manuscript was not sufficiently clear. We sincerely hope that our detailed explanations and clarifications adequately address your concerns.
>
> Inspired by your review, we will correspondingly revise the descriptions in our paper to incorporate the analyses provided above. We believe this will help readers gain a deeper understanding of the current state of analytic learning and the efforts made in our paper, offering more potential directions for future work.
>
>
> ## References
>
> [6] He, Run, et al. Semantic Shift Estimation via Dual-Projection and Classifier Reconstruction for Exemplar-Free Class-Incremental Learning. *Forty-second International Conference on Machine Learning (ICML)*, 2025.

---

> ### Author Response · Authors · 2025-11-22
> **Author Response to Reviewer ofER (Part 9 of 12)**
>
> # Q1: Use Cases and Motivation of DeepAFL
>
> > Could the authors clarify the real motivation and use case of DeepAFL? If all clients already share a strong frozen backbone, what problem is this method actually addressing? Would it make more sense to frame DeepAFL as a collaborative fine-tuning approach, where clients adapt a shared public backbone to private data with different feature distributions?
>
> Thank you very much for your insightful question.
>
> We understand your concern primarily focuses on the motivation and use case of AFL, as our DeepAFL is a direct improvement based on it and does not alter its primary application methodology.
>
> ## Q1-1: Use Cases of DeepAFL
>
> First and foremost, we want to highlight that our DeepAFL indeed requires a given backbone to serve as an **effective feature extractor** for training, but it **does not mandate** how this backbone is obtained. In fact, we identify at least 3 common ways to construct the backbone:
>
> 1. **Supervised Pre-training:** This is the most prevalent case, and it is the setting we primarily use for experimental comparisons in our paper. In this scenario, our DeepAFL can indeed be viewed as a collaborative fine-tuning approach for FL that you suggested, where clients adapt a shared public backbone to private data with different feature distributions.
> 2. **Self-Supervised Pre-training:** Note that with the rise of large-scale pre-trained models, there is often insufficient labeled data for supervised pre-training. In the case of the self-supervised backbone (such as auto-encoders via reconstruction, and DINO series via contrastive learning), the model inherently does not include a classifier/regressor. That is, these feature extractors only possess generalizable feature extraction ability but lack direct classification/regression ability. Thus, if such a backbone is to be employed in FL, a matching classifier/regressor must be trained for it; otherwise, it cannot be used at all. In this context, our DeepAFL can be seen as a direct and highly effective FL training approach to obtain a classifier/regressor as the backbone has already provided sufficient generalization ability.
> 3. **Domain Adaptation:** If the pre-trained backbone presents a domain shift for the FL system, the server can also perform initial feature domain adaptation by fine-tuning the backbone (using a set of domain-specific or compliant public data). Once this adaptation is complete, the backbone is then frozen, and our DeepAFL can be implemented for subsequent FL training. This case can be viewed as *an engineering strategy* to mitigate the issues arising from a cross-domain backbone. Given that domain adaptation can be performed centrally by the server and DeepAFL incurs only negligible time costs, the aggregate overhead across both stages is still considered to be very small.
>
> Thus, we wish to correct the notion that your original perspective is mainly confined to **Case 1**, where the backbone is supervised pre-trained. In **Case 2**, where the pre-trained model is self-supervised, training a classifier for this model in FL already becomes a distinct and meaningful problem. Furthermore, as you consistently emphasize the potential for domain-gap conditions across multiple comments, we want to reiterate that it can be mitigated by considering domain adaptation in **Case 3** described above, which can be viewed as *an engineering strategy*.
>
> The above clarification summarizes the multiple use cases of our DeepAFL from our perspective. Due to space constraints, we will further clarify the key motivation and rationale of our DeepAFL in the next comment block.

---

> ### Author Response · Authors · 2025-11-22
> **Author Response to Reviewer ofER (Part 10 of 12)**
>
> # Q1: Use Cases and Motivation of DeepAFL (Continued)
>
>
>
> ## Q1-2: Clarifications on Experimental Settings
>
> As we previously mentioned, our DeepAFL has various potential use cases under different backbone conditions. We surmise you might be curious why we primarily focused on (especially experimentally) Case 1 in our paper. Here, we would be very glad to share with you the main reasons and considerations that guided our initial decision:
>
> * **Alignment with AFL:** The original AFL paper primarily uses a supervised pre-trained ResNet-18 as the backbone. As a direct improvement to AFL, our DeepAFL is expected to **maintain experimental alignment with it for fairness**.
>
> * **Problem Simplification:** Regardless of the specific use case mentioned above, the ultimate outcome is the provision of a usable backbone. Therefore, to avoid overly complicating the problem, our DeepAFL aims to only concern itself with how to conduct gradient-free representation learning given a fixed backbone. This allowed us to **focus on our core methodology**.
>
>
>
> ## Q1-3: Pervasiveness of Pre-trained Models in FL
>
> Then, we want to emphasize that the incorporation of **pre-trained models/foundation models** into FL has become a **very common and popular practice** in research across various settings in recent years [7]–[21]. Many recent works acknowledge that incorporating a pre-trained backbone helps introduce prior knowledge, making the FL process more stable [9]–[11]. Specifically, a key commonality between our work and these studies is **the focus on using foundation models for conducting FL, rather than using FL for building foundation models**.
>
> It is worth noting that many existing pre-trained-model-based FL works are even mechanistically constrained to specifically require large, strong foundation models [10], [12], such as ViTs [13], [17], [19], [20], CLIP [14], and Diffusion Models [15], [18]. In fact, they often do not even consider small, weak, or cross-domain backbones. From this perspective, the form of backbone required by our DeepAFL is much more flexible, as backbones of various sizes and capabilities can be directly applied. This distinction makes our DeepAFL more suitable for resource-constrained edge settings compared to many related works.
>
> By the way, in the Continual Learning (CL) community,  which is closely related to FL, the use of pre-trained models has also become a very mainstream and widespread trend [22]-[32], and a substantial number of recent studies choose to freeze the backbone to mitigate catastrophic forgetting [22]-[32].
>
>
>
> ## Q1-4: Necessity of FL with Pre-trained Backbone
>
> As clarified in Q1-3, the main scenario our work addresses is **using foundation models for conducting FL, rather than using FL for building foundation models**. You might then reasonably ask: **Is the FL process really necessary when we already have a pre-trained backbone?**
>
> To further address your concern and validate the necessity of FL with a pre-trained backbone, we provide additional experimental evidence using readily available results from the original AFL paper. Specifically, these additional results include the performance of all clients training locally on the backbone, without any global aggregation, under the settings of $\alpha=0.1$ and $K=100$ on CIFAR-100.
>
> We report the maximum (Local Max) and average (Local Avg) test accuracy of local training among all the clients in the table below:
>
> | Methods  | Local Max | Local Avg | FedAvg | AFL   | DeepAFL (T=20) |
> | -------- | --------- | --------- | ------ | ----- | -------------- |
> | Acc. (%) | 16.36     | 12.04     | 56.62  | 58.56 | 66.98          |
>
> These results clearly demonstrate that, **without global aggregation in FL, the results of local training (12.04% and 16.36%) fall far behind the traditional FL method FedAvg (56.62%)**, let alone AFL and DeepAFL.
>
> In fact, when client data volumes are not massive, **collaborative FL training among clients is highly necessary and beneficial, even in the presence of a pre-trained backbone.** This pattern is also validated in previous studies that use pre-trained backbones [10].
>
> ---
>
> Once again, thank you very much for your valuable feedback, which provided us with the opportunity to further clarify the use cases and motivation of our DeepAFL. We sincerely hope that our detailed explanations and clarifications adequately address your concerns.
>
> Constrained by space limitations, a comprehensive list of references will be presented in the subsequent comment block.

---

> ### Author Response · Authors · 2025-11-22
> **Author Response to Reviewer ofER (Part 11 of 12)**
>
> ## References
>
> [7] Tan, Yue, et al. Federated learning from pre-trained models: A contrastive learning approach. *Advances in Neural Information Processing Systems (NeurIPS)*, 2022.
>
> [8] Chen, Hong-You, et al. On the Importance and Applicability of Pre-Training for Federated Learning. *The Eleventh International Conference on Learning Representations (ICLR)*, 2023.
>
> [9] Legate, Gwen, et al. Guiding the last layer in federated learning with pre-trained models. *Advances in Neural Information Processing Systems (NeurIPS)*, 2023.
>
> [10] Feng, Chun-Mei, et al. Learning federated visual prompt in null space for MRI reconstruction. *Proceedings of the IEEE/CVF Conference on Computer Vision and Pattern Recognition (CVPR)*, 2023.
>
> [11] Zhuang, Weiming, et al. When foundation model meets federated learning: Motivations, challenges, and future directions. *arXiv*, 2023.
>
> [12] Yang, Fu-En, et al. Efficient model personalization in federated learning via client-specific prompt generation. *Proceedings of the IEEE/CVF International Conference on Computer Vision (ICCV)*, 2023.
>
> [13] Chen, Haokun, et al. FedDAT: An approach for foundation model finetuning in multi-modal heterogeneous federated learning. *Proceedings of the AAAI Conference on Artificial Intelligence*. Vol. 38. No. 10, 2024.
>
> [14] Pan, Bikang, et al. Federated learning from vision-language foundation models: Theoretical analysis and method. *Advances in Neural Information Processing Systems (NeurIPS)* , 2024.
>
> [15] Zhang, Tuo, et al. GPT-FL: Generative pre-trained model-assisted federated learning. *Proceedings of the Computer Vision and Pattern Recognition Conference (CVPR)*, 2025.
>
> [16] Wang, Qiang, et al. Traceable federated continual learning. *Proceedings of the IEEE/CVF Conference on Computer Vision and Pattern Recognition (CVPR)*, 2024.
>
> [17] Yu, Hao, et al. Personalized federated continual learning via multi-granularity prompt. *Proceedings of the 30th ACM SIGKDD Conference on Knowledge Discovery and Data Mining (KDD)*, 2024.
>
> [18] Liang, Jinglin, et al. Diffusion-driven data replay: A novel approach to combat forgetting in federated class continual learning. *European Conference on Computer Vision (ECCV)*, 2024.
>
> [19] Masum, Muhammad Anwar, et al. Federated Few-Shot Class-Incremental Learning. *The Thirteenth International Conference on Learning Representations (ICLR)*, 2025.
>
> [20] Yu, Hao, et al. Handling spatial-temporal data heterogeneity for federated continual learning via tail anchor. *Proceedings of the Computer Vision and Pattern Recognition Conference (CVPR)*, 2025.
>
> [21] Zhang, Yifei, et al. pFedMxF: Personalized Federated Class-Incremental Learning with Mixture of Frequency Aggregation. *Proceedings of the Computer Vision and Pattern Recognition Conference (CVPR)*, 2025.
>
> [22] Park, Keon-Hee, et al. Pre-trained vision and language transformers are few-shot incremental learners. *Proceedings of the IEEE/CVF Conference on Computer Vision and Pattern Recognition (CVPR)*, 2024.
>
> [23] Yue, Xianghu, et al. MMAL: Multi-modal analytic learning for exemplar-free audio-visual class incremental tasks. *Proceedings of the 32nd ACM International Conference on Multimedia (MM)*, 2024.
>
> [24] Zhuang, Huiping, et al. GACL: Exemplar-free generalized analytic continual learning. *Advances in Neural Information Processing Systems (NeurIPS)*, 2024.
>
> [25] Zhuang, Huiping, et al. F-OAL: Forward-only online analytic learning with fast training and low memory footprint in class incremental learning. *Advances in Neural Information Processing Systems (NeurIPS)*, 2024.
>
> [26] Peng, Liangzu, et al. TSVD: Bridging theory and practice in continual learning with pre-trained models. *The Thirteenth International Conference on Learning Representations (ICLR)*, 2025.
>
> [27] Tran, Quyen, et al. Boosting Multiple Views for pretrained-based Continual Learning. *The Thirteenth International Conference on Learning Representations (ICLR)*, 2025.
>
> [28] Zhang, Xiang, et al. L3A: Label-Augmented Analytic Adaptation for Multi-Label Class Incremental Learning. *Forty-second International Conference on Machine Learning (ICML)*, 2025.
>
> [29] Tong, Kai, et al. Any-SSR: How Recursive Least Squares Works in Continual Learning of Large Language Model. *Proceedings of the IEEE/CVF International Conference on Computer Vision (ICCV)*, 2025.
>
> [30] Lai, Songning, et al. Learning New Concepts, Remembering the Old: Continual Learning for Multimodal Concept Bottleneck Models. *Proceedings of the 33rd ACM International Conference on Multimedia (MM)*, 2025.
>
> [31] Guo, Zihao, et al. GraphKeeper: Graph Domain-Incremental Learning via Knowledge Disentanglement and Preservation. *The Thirty-ninth Annual Conference on Neural Information Processing Systems (NeurIPS)*, 2025.
>
> [32] Jiang, Kai, et al. Mixture of Noise for Pre-Trained Model-Based Class-Incremental Learning. *The Thirty-ninth Annual Conference on Neural Information Processing Systems (NeurIPS)*, 2025.

---

> ### Author Response · Authors · 2025-11-22
> **Author Response to Reviewer ofER (Part 12 of 12)**
>
> # Q5: Theoretical Guarantees under the Imperfect Backbone
>
> > Theoretical analyses currently assume ideal, fixed backbone features. Do the invariance and convergence results still hold when the features are imperfect, noisy, or mismatched with the clients’ data? If not, discussing or extending the theory to such realistic cases would greatly improve the paper’s relevance.
>
> Thank you for your constructive question.
>
> We understand your concern primarily revolves around whether the theoretical analyses of our DeepAFL still hold when the backbone features are imperfect.
>
> Since the main theoretical results of our DeepAFL comprise two parts: **Invariance to Data Heterogeneity** (Theorem 1) and **Capability of Representation Learning** (Theorems 2-3), we will address them separately.
>
> ## Q5-1: Invariance under the Imperfect Backbone
>
> As for the invariance of our DeepAFL, Theorem 1 essentially states that the training result of our DeepAFL is identical to the centralized analytical solutions obtained on the full dataset $\mathcal{D}= \\{ \mathbf{X}, \mathbf{Y} \\}$. The backbone's primary function is to obtain the local zero-layer feature matrix $\mathbf{\Phi} _ {0}^k \in \mathbb{R}^{N _ {k} \times d _ {\Phi}}$ for each client $k$, given by:
>
> $\mathbf{\Phi} _ 0^{k} = \sigma( \mathrm{Backbone}(\mathbf{X}^{k}, \mathbf{\Theta}) \mathbf{A})$
> When the shared backbone is imperfect (e.g., noisy) and affects feature extraction for every client, **the same effect** also applies to the centralized feature extraction:
>
>
> $\mathbf{\Phi} _ 0 = \sigma( \mathrm{Backbone}(\mathbf{X}, \mathbf{\Theta}) \mathbf{A})$
>
> Therefore, regardless of the quality of the backbone (even if it were an identity mapping), the core conclusion that our DeepAFL's obtained result is identical to the centralized analytical solutions remains unaffected. Thus, our DeepAFL's invariance always holds.
>
> ## Q5-2: Non-increasing Empirical Risk under the Imperfect Backbone
>
> Meanwhile, as for our DeepAFL's capability of representation learning, Theorems 2-3 state that the empirical risk of the analytic classifier is monotonically non-increasing as the number of DeepAFL layers increases. This result still holds regardless of any changes to the backbone, as its validity is entirely independent of the backbone's operation.
>
> Specifically, the guarantee of our DeepAFL's representation learning stems from the convex optimization result provided by least squares for the objectives (4) and (7) in our paper.
>
> Since the derived closed-form solution for our DeepAFL is always the optimal one for objectives (4) and (7), the worst-case scenario is that the empirical risk equals that of the previous layer (i.e., convergence), and it is **impossible** for the empirical risk of a given layer to be greater than that of the previous layer.
>
> In summary, all theoretical results in our DeepAFL (Theorems 1-3) consistently hold no matter how the backbone changes.
>
> ## Q5-3: Impact of the Imperfect Backbone
>
> We understand that you may still intuitively expect the performance of DeepAFL to decline when the backbone deteriorates. This intuition is correct, and the issue lies in the expressiveness of the features extracted by the imperfect backbone.
>
> A poor backbone will reduce the representational capacity of the extracted zero-layer features $\mathbf{\Phi} _ 0$. Specifically, the mutual information between the features $\mathbf{\Phi} _ 0$ and the labels $\mathbf{Y}$ may become very low. While our DeepAFL can still perform further optimization and enhancement, its **performance ceiling** will be limited by the low information content provided by the weaker features $\mathbf{\Phi} _ 0$.
>
> **While the reliance on the quality of the backbone is indeed a potential limitation of our DeepAFL, it does not diminish the primary advantage of DeepAFL: granting the analytic classifier the capability of representation learning**. This allows it to move beyond merely learning linear mappings on the extracted features, enabling the learning of **more complex non-linear relationships**. In fact, our experiments have already validated this direct and effective improvement conferred by our DeepAFL. Moreover, the **exceptional efficiency** of our DeepAFL (e.g., a reduction in computational cost by **more than 99%** compared to gradient-based methods) further highlights its unique benefits.
>
> We will incorporate these analyses into our final paper.
>
> ----
>
> This concludes our responses for your first-period comments.
>
> **It is really a pleasure and an honor to engage in such a meaningful discussion with a professional of your caliber!**
>
> We sincerely hope that our detailed explanations and clarifications can adequately address your concerns. Please feel free to raise any further questions you may have.
>
> **We would be extremely grateful if you would consider increasing your score to further support our work.**
>
> Once again, thank you very much for your valuable review. Looking forward to your feedback!

---

> ### Author Response · Authors · 2025-12-03
> **More Clarifications of DeepAFL's Deep Representation Learning Capabilities (W2 & Q4)**
>
> Dear Reviewer ofER:
>
> Considering the unexpected circumstances regarding the recent ICLR updates, we understand that you may be unable to engage in further discussion with us.
>
> Since your initial review mainly focuses on the DeepAFL's deep representation learning capabilities and its reliance on a backbone, we believe this constitutes your primary concern regarding our work.
>
> We subsequently found that a public reader (Danqi Wang) raised questions (Q3 & Q6) that are very closely aligned with your concerns. We believe you will be highly interested in this topic, and **we wish to provide further clarifications to fully address all your potential concerns in a responsible manner**.
>
> Here, we wish to reiterate that characterizing DeepAFL as possessing **deep representation learning capabilities** is justified and not an overstatement.
>
> As we discussed in Part 4 of our initial responses to you, the effect of DeepAFL can be viewed as highly analogous to an **analytic MLP with residual connections**, operating subsequent to a frozen backbone. Once this perspective is accepted, referring to DeepAFL as performing deep representation learning becomes as natural as attributing the same capability to a standard residual MLP.
>
> In fact, **the core reason** why the current version of DeepAFL requires a backbone is **not due to** its inherent MLP-like nature being unable to extract semantic features from raw data, **but rather because** it has not yet been extended to an **analytic CNN** architecture, making direct application to vision tasks difficult.
>
> To illustrate, if DeepAFL were directly applied to ***tabular data***, the backbone would be unnecessary. In this scenario, DeepAFL would certainly be capable of **directly extracting semantic features from the raw data** due to its **MLP-like effect**. From this viewpoint, we believe that asserting our DeepAFL possesses Deep Representation Learning capabilities is entirely justified.
>
> Thus, the backbone in our paper can be viewed as converting ***raw vision data*** into a feature representation akin to ***tabular data*** with minimal information loss. Then, our DeepAFL is primarily utilized to perform deep representation learning on this derived *"tabular" data*.
>
> Therefore, we conclude that the current limitation of DeepAFL is **not its inability to extract information from raw data** (assuming the raw data is tabular), but rather its current **initial stage (i.e., the MLP-like version)**, which has yet to be extended to a CNN-like version suitable for raw vision data.
>
> You may also be curious why our paper used image classification, a task where DeepAFL is not optimally suited, instead of tabular tasks for experimental validation. Our initial motivation was primarily to **align with the original AFL work**, which also employed image tasks. Additionally, showcasing DeepAFL's ability to perform further representation learning on top of already-extracted backbone features **may better broaden its potential applicability**.
>
> We think DeepAFL would show better performance on tabular data, as it could **directly extract semantic features from the raw data** without potential information loss from the backbone. In fact, if this paper is fortunate enough to be accepted by ICLR, we will consider preparing supplementary experiments on tabular data and **including them in the journal extension of our work**.
>
> Furthermore, we anticipate that you might be interested in whether DeepAFL can be extended from an **"analytic deep residual MLP"** to an **"analytic deep residual CNN"** in the future. As far as we are concerned, we believe this is a **highly promising and feasible future direction**.
>
> Specifically, related work has already investigated deriving **closed-form solutions for CNNs** in the field of analytic learning [33]. In fact, we have already compared our DeepAFL's deepening strategy with this work's strategy in Appendix E.3.7 of our paper, finding their deepening approach to be somewhat weaker. Thus, combining the core ideas of DeepAFL with existing closed-form solutions for CNNs is a very **natural line of future research**.
>
> Starting from our DeepAFL, we believe that many promising future works can be built upon this foundation. We are also committed to them ourselves. If you are interested, we genuinely welcome you to follow our subsequent efforts in the future!
>
> Since the public reader (Danqi Wang) also raised highly relevant questions, we encourage further reference to our responses to him/her.
>
> We sincerely hope that our additional analyses can help you better understand our DeepAFL, particularly its **deep representation learning capabilities**. Thank you again for your support of our work!
>
>
>
> ## References
>
> [33] Zhuang, Huiping, et al. An analytic formulation of convolutional neural network learning for pattern recognition. *Information Sciences*, 2025.

---

### Official Review · Reviewer_mCa1 · 2025-10-31

**Soundness:** 2
**Presentation:** 2
**Contribution:** 2
**Rating:** 4
**Confidence:** 4

**Summary:**

This paper presents DeepAFL, a deep analytic federated learning framework that extends the original Analytic Federated Learning (AFL) approach from a single linear layer to multiple residual analytic layers. Each layer is trained via a closed-form “sandwiched” least-squares solution, completely avoiding gradient-based optimization. The authors provide theoretical proofs showing (1) invariance to data heterogeneity when random projections are shared and (2) a monotonic reduction of empirical risk as the network depth increases. Experiments on CIFAR-10, CIFAR-100, and Tiny-ImageNet demonstrate consistent accuracy improvements and notable reductions in computational and communication costs.

**Strengths:**

1. The paper introduces the first federated learning framework that achieves deep representation learning without gradients while maintaining closed-form analytical updates.
2. The idea of stacking analytic layers in a residual manner is a meaningful and elegant extension of AFL.
3. Reported efficiency gains-around 99% reduction in computation and 50% reduction in communication compared to gradient-based methods-are impressive.

**Weaknesses:**

1. While theoretical invariance is well-supported, the practical scalability to very large models such as ViTs or LLMs has not been evaluated.
2. The related work section omits recent multimodal and representation-based FL methods like FedRep [1] and FedU² [2].
3. No wall-clock runtime, GPU-hour, or FLOP comparisons are provided to substantiate the efficiency claims.

**Questions:**

1. Clarify theoretical assumptions, including shared random seeds, the impact of regularization, and the influence of finite precision.
2. Provide quantitative computational profiling (runtime, FLOPs, GPU-hours) to verify the claimed efficiency improvements.
3. Include comparisons with other analytic or gradient-free deep learning methods beyond AFL.
4. Add a clear visual diagram explaining the “sandwiched” least-squares process for improved readability.

---

> ### Author Response · Authors · 2025-11-21
> **Author Response to Reviewer mCa1 (Part 1 of 5)**
>
> We sincerely thank you for your constructive feedback. We are more than pleased that you recognize the novelty of our DeepAFL as the first deep FL framework to achieve deep representation learning without gradients, the elegance of our residual analytic layer design, and the impressive *potential* of our DeepAFL's efficiency gains.
>
> Here, we would like to respectfully address your raised Weakness (W) and Question (Q) below.
>
> # W1: Practical Scalability
>
> > While theoretical invariance is well-supported, the practical scalability to very large models such as ViTs or LLMs has not been evaluated.
>
> Thank you for your constructive comment!
>
> First, we would like to clarify two main reasons for using ResNet-18 as the backbone in our paper:
>
> * **Alignment with AFL:** As our DeepAFL is built upon AFL, which uses ResNet-18 as its primary backbone, utilizing **the same aligned backbone** facilitates a clearer and more **transparent comparison** between the two methods.
> * **Practicality in FL:** In FL, most scenarios involve resource-constraint **edge devices (in IoT)** as clients. Thus, training and inference overhead associated with the **very large models** might be **impractical** in many real-world settings.
>
> Nevertheless, we understand your concern regarding our DeepAFL's scalability and agree that evaluating it on larger models is beneficial to highlight its superiority.
>
> To fully address your concerns, we have carefully selected 2 different versions of ViTs to conduct new experiments:
>
> * **ViT-B-16-I-1K:** Pre-trained on the ImageNet-1K Dataset (with approx. 1.28 million images and 1,000 classes).
> * **ViT-B-16-I-21K:** Pre-trained on the significantly larger ImageNet-21K Dataset (with 14 million images and 21,843 classes)
>
> Notably, due to the much larger pre-training scale, the latter ViT-B-16-I-21K typically yields better performance than the former ViT-B-16-I-1K. Since the backbone sizes of both ViT models are identical (Base-16), the runtime for our DeepAFL on both models remains essentially the same. Our experiments were conducted with 100 clients on a single 4090 GPU on the CIFAR-100 dataset.
>
> The new experimental results on **ViT-B-16-I-1K** are shown below.
>
> | Model       | Test Accuracy (%) | $\Delta$ Test Accuracy (%) | Cumulative Advantages over AFL (%) | Total Wall-Clock Runtime (s) | $\Delta$ Runtime  (s) | Additional runtime over AFL (s) |
> | --- | --- | --- | --- | :--- | --- | --- |
> | AFL         | 75.45  | 0.00  | 0.00  | 107.19  | 0.00  | 0.00  |
> | Ours (T=5)  | 78.05  | 2.60  | 2.60  | 118.97  | 11.78 | 11.78  |
> | Ours (T=10) | 79.01  | 0.96  | 3.56  | 126.55  | 7.58  | 19.37  |
> | Ours (T=15) | 79.73  | 0.72  | 4.28  | 133.79  | 7.24  | 26.60  |
> | Ours (T=20) | 80.13  | 0.40  | 4.68  | 141.02  | 7.23  | 33.83  |
> | Ours (T=25) | 80.33  | 0.20  | 4.88  | 148.56  | 7.54  | 41.37  |
> | Ours (T=30) | 80.51  | 0.18  | 5.06  | 156.21  | 7.65  | 49.02   |
>
> Concurrently, the new experimental results on **ViT-B-16-I-21K** are shown below.
>
> | Model       | Test Accuracy (%) | $\Delta$ Test Accuracy (%) | Cumulative Advantages over AFL (%) | Total Wall-Clock Runtime (s) | $\Delta$ Runtime  (s) | Additional runtime over AFL (s) |
> | -- | --- | --- | --- | --- | --- | --- |
> | AFL  | 86.35  | 0.00  | 0.00                               | 107.19                       | 0.00                  | 0.00                            |
> | Ours (T=5)  | 87.71             | 1.36                       | 1.36                               | 118.97                       | 11.78                 | 11.78                           |
> | Ours (T=10) | 88.15             | 0.44                       | 1.80                               | 126.55                       | 7.58                  | 19.37                           |
> | Ours (T=15) | 88.49             | 0.34                       | 2.14                               | 133.79                       | 7.24                  | 26.60                          |
> | Ours (T=20) | 88.80             | 0.31                       | 2.45                               | 141.02                       | 7.23                  | 33.83                           |
> | Ours (T=25) | 88.95             | 0.15                       | 2.60                               | 148.56                       | 7.54                  | 41.37                           |
> | Ours (T=30) | 89.08             | 0.13                       | 2.73                               | 156.21                       | 7.65                  | 49.02                           |

---

> ### Author Response · Authors · 2025-11-21
> **Author Response to Reviewer mCa1 (Part 2 of 5)**
>
> # W1: Practical Scalability (Continued)
>
> Through the preceding results, we can analyze that applying our DeepAFL to both versions of the ViT backbone yields **significant performance improvements** with a **very low computational cost**.
>
> Specifically, at the $T=30$ setting, our DeepAFL achieves an accuracy gain of **$5.06\\%$** over AFL with the ViT-B-16-I-1K backbone, and an improvement of **$2.73\\%$** with the ViT-B-16-I-21K backbone.
>
> To achieve this substantial improvement, the required **Total Wall-Clock Runtime** is merely 156.21 s. This represents a total increase in runtime of only 49.02 s compared to the time consumed by AFL itself (107.19 s).
>
> It is important to note that the **Total Wall-Clock Runtime (s)** we measure here is the total time consumed by all 100 clients to complete the training process for all layers (including the clients' forward pass of backbone, the clients' analytic training, and the server's aggregation operations) on a single 4090 GPU.
>
> This implies that, in practice, completing the training of $T=30$ layers of our DeepAFL on a large backbone like ViT-B-16 costs **less than 1.56 s** ($156.21/100 \approx 1.56$) on average per client. In stark contrast, even the simplest and most efficient FL baseline (e.g., FedAvg) requires a Total Wall-Clock Runtime of **over 33,000 s**.
>
> We believe the results above fully demonstrate the **scalability** (stable performance improvement on large ViT backbones) and **efficiency** (extremely low Wall-Clock Runtime) of our DeepAFL.
>
> In addition, we would like to further discuss and clarify some new findings based on the preceding results.
>
> We observe that the performance improvement of our DeepAFL exhibits a **diminishing marginal returns phenomenon** across two aspects:
>
> * **Backbone Strength:** As the backbone capability increases ($\text{ResNet-18} \rightarrow \text{ViT-B-16-I-1K} \rightarrow \text{ViT-B-16-I-21K}$), the absolute advantage of our DeepAFL over AFL weakens under the same $T=30$ setting ($9.15\\% \rightarrow 5.06\\% \rightarrow 2.73\\%$).
> * **Layer Depth:** With the same backbone (e.g., ViT-B-16-I-1K), the marginal performance gain ($\Delta$ Test Accuracy) from every 5 additional layers also diminishes.
>
> We particularly want to clarify that this diminishing marginal returns phenomenon is **entirely expected and normal**:
>
> * As the backbone capability strengthens, the feature representation and linear separability of the features themselves already become very high, making it **increasingly challenging to further extract performance** beyond such a high baseline value.
> * As the number of layers increases, the fitting capacity of our DeepAFL gradually strengthens. However, since the training data volume is not increased, enhancing our DeepAFL will inevitably encounter a **saturation limit** on its fitting capacity.
>
> Notably, the phenomenon of diminishing returns is also **common in gradient-based methods** (e.g., simply increasing the depth of a DNN can not lead to continuous performance improvement up to 100% without encountering a bottleneck). Even the high performance achieved by large models like LLMs is a result of scaling the model *alongside* vast increases in training data volume (i.e., **the scaling law**).
>
> Therefore, the fact that our DeepAFL can achieve stable and significant improvement across various backbone types **without increasing any FL training data volume** is extremely noteworthy. Especially when considering the **overall extremely low runtime cost**, the performance gain delivered by our DeepAFL can be considered to have **virtually no time overhead**.
>
> We are very grateful for this insightful comment! We believe that by clarifying our DeepAFL's practical scalability to large ViTs can further emphasize its strengths and enhance the completeness of our work.
>
> We are committed to and are actively working on incorporating these contents into our final paper.
>
>
> # W2: Supplementing Related Work
>
> > The related work section omits recent multimodal and representation-based FL methods like FedRep [1] and FedU² [2].
>
> Thank you for your kind reminder! We will revise our paper to include these two references and add appropriate discussions.
>
> Specifically, while these FL methods do improve convergence and performance under heterogeneity, they still rely on gradient descent for optimization. In contrast, our DeepAFL completely circumvents this reliance and achieves invariance to data heterogeneity, which represents a significant and unique distinction between our DeepAFL and the referenced works.
>
> We are committed to and are actively working on incorporating these contents into our final paper.

---

> ### Author Response · Authors · 2025-11-21
> **Author Response to Reviewer mCa1 (Part 3 of 5)**
>
> # W3 & Q2: Quantitative Efficiency Metrics
>
> > No wall-clock runtime, GPU-hour, or FLOP comparisons are provided to substantiate the efficiency claims.
>
> > Provide quantitative computational profiling (runtime, FLOPs, GPU-hours) to verify the claimed efficiency improvements.
>
> Thank you for your valuable comment.
>
> We apologize if the presentation of our efficiency metrics in the original manuscript was not sufficiently prominent. **But we indeed presented ample quantitative computational profiling of our DeepAFL** (in comparison with the most efficient gradient-based baseline FedAvg) in Figure 2(a), Figure 3(a), Figure 4(b), Figure 5(b), and Tables 9-11.
> We would like to offer further clarification of these results.
>
> In our paper, the metric for computational cost is precisely defined as the **Total Wall-Clock Runtime**, which primarily comprises three components:
>
> * The total time for all clients to perform **forward propagation** using the backbone for feature extraction.
>
> * The total time for all clients to complete all **analytic computations**.
>
> * The total time for the server to process all **aggregation**.
>
> Our experiments consistently involve $K=100$ clients and were conducted on **a single** NVIDIA GeForce RTX 4090 GPU.
>
> Here, let us use the CIFAR-100 dataset as a case study, as shown in Figure 2(a) and Table 10. When the number of layers $T=20$, the total wall-clock runtime for all clients on a single 4090 GPU is merely 91.74 seconds. This striking result implies:
>
> * The average time cost for a single client to complete the computation across all 20 layers is less than 1 second ($91.74 \div 100 \approx 0.917 \text{ s}$).
>
> * The computation cost for a single client to process a single layer is less than 0.05 seconds ($0.917 \div 20 \approx 0.046 \text{ s}$).
>
> Since the total wall-clock runtime can be divided into GPU and CPU time, and we only utilized a single 4090 GPU, the dedicated GPU time would be even smaller.
>
> In stark contrast, even the simplest and most efficient gradient-based FL baseline (e.g., FedAvg) requires a total wall-clock runtime of **over 33,000 s**. Other FL methods incur even higher costs, with MOON's total wall-clock runtime being approximately 1.5 times that of FedAvg (**about 50,000 s**).
>
> We believe such quantitative results and comparisons on computational costs can strongly support the superior efficiency of our DeepAFL. Moreover, we wish to remind you that in our preceding response to your practical scalability concern (W1), we also provided efficiency analyses on a larger backbone architecture, namely ViT-B-16. Those results present a similar positive trend, further demonstrating the robustness of DeepAFL's efficiency advantages across various backbone scales.
>
> As we understand that you may be surprised by the extreme efficiency of our DeepAFL, we wish to provide further explanation and clarification regarding **how it is achieved**, as follows:
>
> On the one hand, the gradient-based FL methods require additional backward propagation during local training, introducing significantly greater cost compared to our DeepAFL, which only requires forward propagation computation.
>
> On the other hand, the gradient-based FL methods typically require **dual-loop iterations for convergence**, including the **inner-loop multi-epoch local training** and the **outer-loop multi-round aggregation**, which multiplies their computational cost by hundreds or even thousands of times on top of the backward propagation cost.
>
> In contrast, our DeepAFL adopts a **gradient-free manner**, thus requiring **no backward propagation** and **no iterations for convergence**. The main computational overhead only arises from **forward matrix computation (with closed-form solutions)**.
>
> In fact, the efficiency advantage of the analytic learning has been thoroughly discussed in AFL, and our DeepAFL just successfully preserves this key advantage.
>
> Your valuable comment makes us realize we should incorporate more analyses regarding computational cost to further highlight our DeepAFL's superior efficiency.
>
> Thank you again for your valuable comment, and we are actively working on incorporating these contents into our final paper. We sincerely hope this response adequately addresses your concerns.

---

> ### Author Response · Authors · 2025-11-21
> **Author Response to Reviewer mCa1 (Part 4 of 5)**
>
> # Q1: Clarification on Theoretical Assumptions
>
> > Clarify theoretical assumptions, including shared random seeds, the impact of regularization, and the influence of finite precision.
>
> Thank you for your constructive comment.
>
> We fully agree that a more explicit clarification of the theoretical prerequisites can help enhance the reader's understanding of our proposed theorems. Here, we would like to clarify our assumptions point-by-point, as follows:
>
> ## Q1-1: Shared Random Seeds
>
> Since our DeepAFL involves randomly initialized and non-trainable random projections, its theoretical **invariance to data heterogeneity** requires that the random projection matrices $\mathbf{A}$ and $\\{\mathbf{B} _ {t}\\} _ {t=0}^{T-1}$ of all clients must be consistent and identical. This consistency is essential so that the features and representations constructed by all clients reside within the same high-dimensional space.
>
> The simplest and most communication-efficient approach to achieve this is for the server to **broadcast a single set of random seeds** to all participant clients. Each client can then locally generate the necessary random projection matrices based on this set of seeds, avoiding the repeated transmission of the large random projection matrices between the server and clients.
>
> ## Q1-2: Regularization Impact
>
> In our DeepAFL, we introduce Tikhonov Regularization (i.e., L2 Regularization) in the optimization objectives (4) and (7). It primarily serves two key purposes:
>
> * The L2 Regularization helps us mitigate overfitting to **enhance generalization performance**, as is well-established in Ridge Regression.
>
> * The L2 Regularization also ensures the matrix inversion in the solution (e.g., $\mathbf{\Phi} _ t^\top \mathbf{\Phi} _ t + \lambda\mathbf{I}$) is always well-posed (non-singular), thus maintaining the **numerical stability** of our DeepAFL.
>
> To progressively present our theoretical analyses, we provided two different versions of the theorem to show our DeepAFL's representation capabilities:
>
> - Theorem 2 is presented under the basic condition of no regularization ($\gamma=\lambda=0$). Under this assumption, we can only prove that our DeepAFL maintains a non-increasing loss when the loss function has no regularization.
> - We then introduce Theorem 3 in Appendix B.3, providing a more complete theoretical proof to show that our DeepAFL can maintain a non-increasing loss even when the loss function incorporates an arbitrary L2 regularization magnitude.
>
> ## Q1-3: Finite Precision
>
> In our theoretical results, we typically assume infinite precision for clarity. In practical implementations, it is true that finite precision and floating-point arithmetic may introduce minimal numerical errors.
>
> However, these errors are generally well-managed due to the **stability ensured by our introduced regularization terms**. Therefore, the practical impact of finite precision on our DeepAFL is very small. In fact, our experimental evaluations (using Python's default 64-bit precision float type) also provide sufficient **empirical evidence** showing the numerical stability of our DeepAFL.
>
> ---
>
> Thank you again for your valuable comment, and we sincerely hope this response can adequately address your question.

---

> ### Author Response · Authors · 2025-11-21
> **Author Response to Reviewer mCa1 (Part 5 of 5)**
>
> # Q3: Comparisons with Other Analytic Methods
>
> > Include comparisons with other analytic or gradient-free deep learning methods beyond AFL.
>
> Thank you for your valuable feedback.
>
> First and foremost, we would like to respectfully clarify that, to the best of our knowledge, there are currently no other suitable published analytic/gradient-free methods in the field of FL available for a direct comparison. In fact, AFL itself was only recently published at CVPR 2025. Thus, as our DeepAFL is a direct and fundamental improvement upon AFL, our main manuscript mainly focused on comparing our DeepAFL against AFL.
>
> In addition, although there are no other suitable analytic/gradient-free deep learning methods within FL, we would like to kindly remind you that **we did provide experimental results comparing our DeepAFL with other deepening strategies for analytic learning** in **Appendix E.3.7** of our original manuscript.
>
> Specifically, we designed and compared four distinct deepening strategies: (1) cascades with random features, (2) cascades with random features and label encoding, (3) cascades with activated random features, and (4) cascades with activated random features and label encoding.
>
> Notably, because these strategies are not specifically designed for FL, these deepening strategies are not guaranteed to be effectively applicable to the FL setting. Thus, our experiments on these comparisons were conducted in a centralized scenario instead of the FL scenario, primarily to highlight the general superiority of our DeepAFL's deepening strategy.
>
> The detailed experimental results are presented in **Tables 18–23 and Figures 11–13** of our original manuscript. These results visibly demonstrate that our DeepAFL significantly surpasses these baseline deepening strategies, underscoring the superiority of its representation learning capability.
>
> Moreover, we would also like to point out that **we provided detailed ablation studies for our DeepAFL in Appendix E.3.6**, where the four introduced ablation models can also be regarded as analytic/gradient-free deep learning methods for comparison. The ablation results are presented in **Tables 12–17 and Figures 8–10**. These comprehensive ablation studies further confirm the performance superiority of our DeepAFL.
>
> Once again, thank you for your valuable comment, and we sincerely hope this response adequately addresses your concerns.
>
> # Q4: Adding A Visual Diagram for the “Sandwiched” Least-Squares
>
> > Add a clear visual diagram explaining the “sandwiched” least-squares process for improved readability.
>
> Thank you very much for your constructive suggestion! We fully agree that adding a clear visual diagram will significantly help to illustrate our “sandwiched” least-squares process and thus improve the overall readability of our manuscript. We are **actively working** on this addition and **commit** to providing the visual diagram by updating our PDF no later than **November 27th**.
>
> Thank you again for your kind understanding and patience.
>
> ---
>
> This concludes our responses for your first-period comments.
>
> We sincerely hope that our detailed explanations and clarifications can adequately address your concerns. Please feel free to raise any further questions you may have.
>
> We would be extremely grateful if you would consider increasing your score to support our work.
>
> Once again, thank you very much for your valuable review. Looking forward to your feedback!

---

> ### Author Response · Authors · 2025-11-26
> **Additional Visual Diagrams for DeepAFL (W3)**
>
> Dear Reviewer mCa1:
>
> As promised in our initial response regarding your comment Q4, we have **meticulously prepared and completed** the additional visual diagrams for our DeepAFL, **in particular concerning its "sandwiched" least-squares process.**
>
> Specifically, we have submitted a new PDF of our revision, where **Pages 39–40** are dedicated to providing further illustrations and analyses for DeepAFL's detailed process.
>
> - **Figure 16** directly addresses your request by vividly illustrating **why we term this the “sandwiched” least squares problem** and **how it works.**
> - **Inspired by your comment**, we have also included **two additional visual diagrams**, which respectively demonstrate the **layer-by-layer training process** and the **empirical risk reduction process** in our DeepAFL, as shown in **Figures 14 and 15**.
>
> Furthermore, we have provided corresponding textual analysis and explanations for these visual diagrams in **Appendix H**. We believe that this modification will **significantly enhance the quality and readability of our paper.**
>
> With this, all of our results and analyses in response to your initial comments are now complete.
>
> Thank you again for your valuable guidance and patience! We sincerely hope that our response has adequately addressed your concerns and provided sufficient rationale for an improved score.
>
> We are pleased to address any further concerns you may have, and we wish you all the best!

---

### Official Review · Reviewer_f9VW · 2025-11-02

**Soundness:** 3
**Presentation:** 3
**Contribution:** 3
**Rating:** 8
**Confidence:** 3

**Summary:**

The paper aims to address the shortcoming of analytic federated learning: lack of representation learning capability. The idea is to incorporate skip connection into the multi-layer version of the analytic federated learning. The initial feature is extracted from a pre-trained backbone, and then randomly projected and activated to get the zero-th layer feature. The feature is refined at the next layer by adding a residual block consisting of a nonlinear feature transformation and a learnable weight. The nonlinear feature transformation is the same as those for the initial feature, i.e., random projection for stochasticity and activation for nonlinearity. Given the feature at each layer, both the classifier and weight have a closed-form solution to the regularized least-squares. This way, the classifier keeps improving and is shown to converge eventually along the chain of layers. This idea is then adapted to the federated learning setting with only auto-correlation and cross-correlation matrices of relevant client data uploaded to the server, which ensures the invariance of the global model to data heterogeneity as analytic federated learning does. It is mentioned that it is hard to infer client information from the communicated data and the data encryption technique can be integrated to enhance privacy. Extensive experiments are conducted to demonstrate the superiority of the proposed algorithm in terms of both accuracy and efficiency across various settings.

**Strengths:**

1. The idea of adding skip connection to the multi-layer version of analytic federated learning is great.
2. The design of the residual block is nice.
3. The framework addresses well the representation learning issue of the analytic federated learning while keeping all advantages like closed-form solutions and invariance to data heterogeneity.
4. The algorithm is guaranteed to converge
5. Extensive experiments are provided to demonstrate the potential of the algorithm.

**Weaknesses:**

The privacy argument seems weak. Once data encryption techniques are introduced, the computational cost will be increasing as well.

Minor:
In Line 373-374, "and AFL (He et al., 2025). Furthermore, we include the analytic learning-based method AFL (He et al., 2025) as a baseline to further highlight our advantages", where "AFL (He et al., 2025)" is repeated.

**Questions:**

1. what if only partial participation of clients?

---

> ### Author Response · Authors · 2025-11-21
> **Author Response to Reviewer f9VW (Part 1 of 4)**
>
> We sincerely thank you for your thorough and positive feedback. We are highly grateful for your valuable recognition on the **novelty** of our proposed idea, the **design** of our residual block, as well as the extensive efforts regarding our **theoretical and experimental analyses**.
>
> Here, we would like to respectfully address your raised Weakness (W) and Question (Q) below.
>
> # W1: Privacy Argument
>
> > The privacy argument seems weak. Once data encryption techniques are introduced, the computational cost will be increasing as well.
>
> We sincerely appreciate your insightful comment regarding the privacy of our DeepAFL.
>
> Our original intention in mentioning external techniques for enhanced privacy was to highlight our DeepAFL's compatibility, as it can be directly integrated with existing privacy techniques without requiring any extra modifications or adjustments.
>
> We fully understand your concern that a complete work should include a more detailed and specific analysis, particularly addressing the Privacy-Efficiency balance when introducing additional privacy techniques.
>
> Here, we would like to present our clarification across the following 3 points.
>
> ## W1-1: The Privacy of DeepAFL in Plain-text Transmission
>
> In the communication process, the information transmitted consists of Auto-Correlation Matrices ($\mathbf{G} _ {t}^{k}$ and $\mathbf{\Pi} _ {t}^k$) and Cross-Correlation Matrices ($\mathbf{H} _ {t}^{k}$ and $\mathbf{\Upsilon} _ {t}^{k}$).
>
> In effect, these matrices can be viewed as the **sum of the outer products** of each sample's feature/representation vector (with itself or with the label/residual vector). As the data size $N _ k$ on the client increases, the outer-product summation becomes higher, making it more difficult to fully recover the original information (especially given that each client is not required to disclose its local data size $N _ k$).
>
> Moreover, even if we assume that $\mathbf{\Phi} _ {t}^{k}$ or $\mathbf{F} _ {t}^{k}$ could be reverse-engineered, the most information that an attacker could obtain is only the feature embedding instead of the raw data $\mathbf{X}^{k}$. Thus, even when using plain-text transmission, our DeepAFL does not directly leak the client's complete raw data.
>
> ## W1-2: The Additional Privacy Techniques Can Be Lightweight
>
> If high privacy is required in FL, we want to clarify that **the introduced privacy techniques for our DeepAFL do not necessarily involve heavyweight encryption**. In fact, the Secure Aggregation Protocols we cited in our paper are based on lightweight Secret Sharing (instead of Homomorphic Encryption) [1], [2].
>
> The Secure Aggregation Protocols were initially proposed by the **Google team** in 2016 to protect the privacy of traditional FL methods (e.g., FedAvg) and have been widely adopted in the industry.
>
> Below, we list the complexity of it (for a single aggregation round):
>
> | Type | Client | Server |
> | :-: | :-: | :--: |
> |  Computation  | $\mathcal{O}(K^2+K{d _ {\Phi}}^2)$ |  $\mathcal{O}(K^2{d _ {\Phi}}^2)$  |
> | Communication |  $\mathcal{O}(K+{d _ {\Phi}}^2)$   | $\mathcal{O}(K^2+K{d _ {\Phi}}^2)$ |
>
> Notably, we have aligned the symbols in the table with those used in our paper: $K$ is the number of clients, and $d _ {\Phi}$ is the feature dimensions (fixed at $d _ {\Phi}=d _ {F}=1024$ in our paper). As the server is generally assumed to have powerful computational capabilities, we focus on the client-side analysis.
>
> The client's computation cost is $\mathcal{O}(K^2+K{d _ {\Phi}}^2)$. Typically, the number of clients $K$ is much smaller than ${d _ {\Phi}}^2\approx10^{6}$. Thus, the complexity can be simplified to $\mathcal{O}(K{d _ {\Phi}}^2)$.
>
> To clarify what it means, we can compare it with the complexity of our DeepAFL, which is greater than $\mathcal{O}({N} _ {k} d _ {{\Phi}}^2)$, where ${N} _ {k}$ is the data size of the $k$-th client. Typically, $K$ and ${N} _ {k}$ are often not drastically different.
>
> In our experimental setting (e.g., CIFAR-100 dataset with $K=100$ clients), the average data size per client is $\bar{\mathcal{N}} _ {k}=500 > K$. In this case, the computational complexity of the additional protocol is **even lower** than that of our DeepAFL.
>
> As shown in our paper, for the CIFAR-100 dataset involving $K=100$ clients and $T=20$ layers, the total runtime for all clients on a single 4090 GPU is merely 91.74 seconds. This striking result implies:
>
> - The average time cost for a single client to complete the computation for all 20 layers is less than 1 second.
> - The computation cost for a single client to process a single layer is less than 0.05 seconds.
>
> This low per-client and per-layer computational cost strongly supports our DeepAFL's efficiency. Since the computational complexity introduced by privacy techniques is **comparable to or even smaller** than that of our DeepAFL, it will not lead to excessive efficiency concerns in practice.

---

> ### Author Response · Authors · 2025-11-21
> **Author Response to Reviewer f9VW (Part 2 of 4)**
>
> # W1: Privacy Argument (Continued)
>
> ## W1-3: The Gradient-based Methods Have Poorer Privacy-Efficiency Balance
>
> Furthermore, we want to clarify that the gradient-based methods (e.g., FedAvg) have also been widely proven to severely leak private information just by transmitting gradients [1]-[3]. Therefore, they also typically require additional techniques for privacy protection in practice, just like our DeepAFL.
>
> In fact, the Secure Aggregation Protocol we introduced was just initially designed specifically for them. Thus, from a **fair comparison perspective**, if we consider the additional costs incurred by introducing privacy techniques to our DeepAFL, **these gradient-based methods are equally susceptible** to similar costs.
>
> More importantly, as our DeepAFL is gradient-free, it does not require iterative communication for convergence, making the additional cost introduced by privacy techniques more acceptable. Conversely, if these privacy techniques are introduced in gradient-based methods, **they will face much greater cost**, as they require multiple epochs of local training and multiple rounds of global aggregation to converge in FL.
>
> ---
>
> In summary, we think that the introduction of external privacy-enhanced techniques is not a weakness but rather highlights the advantages of our DeepAFL:
>
> * On the one hand, our DeepAFL, as a new gradient-free approach, is directly compatible with off-the-shelf privacy protocols originally designed for gradient-based methods.
> * On the other hand, since all mainstream gradient-based FL methods cannot directly avoid privacy concerns, they also need to adopt privacy protocols. When adopting the same protocol, our DeepAFL offers a greater efficiency advantage over them.
>
> We are very grateful for this valuable comment! We believe that by clarifying the privacy aspect of our DeepAFL, we can further emphasize its strengths and enhance our paper's completeness. We are committed to and are actively working on incorporating these contents into our final paper!
>
> P.S. By the way, we consider the future discussion on offensive \& defensive measures for analytic-based FL methods to be an interesting topic for future work, and we are excited to see more researchers conducting further exploration based on our DeepAFL.
>
>
>
> [1] Bonawitz, Keith, et al. Practical secure aggregation for federated learning on user-held data. arXiv 2016.
>
> [2] Bonawitz, Keith, et al. Practical secure aggregation for privacy-preserving machine learning. CCS, 2017.
>
> [3] So, Jinhyun, et al. Securing secure aggregation: Mitigating multi-round privacy leakage in federated learning. AAAI, 2023.
>
>
>
>
> # W2: Minor Typo
>
> > Minor: In Line 373-374, "and AFL (He et al., 2025). Furthermore, we include the analytic learning-based method AFL (He et al., 2025) as a baseline to further highlight our advantages", where "AFL (He et al., 2025)" is repeated.
>
> Thank you for spotting this repetition as a typo. We will immediately correct this in the final version of the paper. Your meticulous review is of great significance to the rigor and quality of our paper!

---

> ### Author Response · Authors · 2025-11-21
> **Author Response to Reviewer f9VW (Part 3 of 4)**
>
> # Q1: Partial Client Participation
>
> > What if only partial participation of clients?
>
> Thank you for raising this excellent and pertinent question! We believe this question is highly significant, as partial client participation (arising from client stragglers and dropouts) is a common and practical challenge in FL.
>
> To provide a comprehensive and detailed clarification, we will address your question across two primary aspects:
>
> 1. **Mechanism Adjustment:** How the procedural flow of our DeepAFL is adapted to handle partial client participation.
> 2. **Performance Analyses:** How the performance (or robustness) of our DeepAFL changes when confronted with partial participation of clients.
>
> ## Q1-1: Mechanism Adjustment for Partial Client Participation
>
> First, let's briefly review the core concept of the aggregation processes in our DeepAFL.
>
> As detailed in Section 3.2 of our paper, constructing each new layer of our DeepAFL requires two rounds of communication, corresponding to the computation of the classifier weights $\mathbf{W} _ t$ and the feature enhancement matrix $\mathbf{\Omega} _ {t+1}$, respectively.
>
> Specifically, the aggregation process in the first round is defined as:
>
> $\mathbf{G} _ {t}^{1:K} = \sum\nolimits _ {k=1}^{K} \mathbf{G} _ {t}^k, \quad \mathbf{H} _ {t}^{1:K} = \sum\nolimits _ {k=1}^{K} \mathbf{H} _ {t}^k$
>
> And the aggregation process in the second round is:
>
> $\mathbf{\Pi} _ {t}^{1:K} = \sum\nolimits _ {k=1}^{K} \mathbf{\Pi} _ {t}^k, \quad \mathbf{\Upsilon} _ {t}^{1:K} = \sum\nolimits _ {k=1}^{K} \mathbf{\Upsilon} _ {t}^k$
>
> Here, we denote the full set of all clients as $\mathcal{S}$, where the total number of clients is $|\mathcal{S}|=K$.
>
> When only a **partial set of clients** participates in the aggregation for DeepAFL, we assume that at layer $t$, the set of clients participating in the classifier construction is $\mathcal{S} _ t^{\mathbf{W}}$, and the set of clients participating in the feature enhancement is $\mathcal{S} _ t^{\mathbf{\Omega}}$.
>
> Consequently, the aggregated matrices will be constructed using contributions from **only the participating subsets**:
>
> $\mathbf{G} _ {t}^{\mathcal{S} _ t^{\mathbf{W}}} = \sum _ {k \in \mathcal{S} _ t^{\mathbf{W}}} \mathbf{G} _ {t}^k, \quad \mathbf{H} _ {t}^{\mathcal{S} _ t^{\mathbf{W}}} = \sum _ {k \in \mathcal{S} _ t^{\mathbf{W}}} \mathbf{H} _ {t}^k$
>
> $\mathbf{\Pi} _ {t}^{\mathcal{S} _ t^{\mathbf{\Omega}}} = \sum _ {k \in \mathcal{S} _ t^{\mathbf{\Omega}}} \mathbf{\Pi} _ {t}^k, \quad \mathbf{\Upsilon} _ {t}^{\mathcal{S} _ t^{\mathbf{\Omega}}} = \sum _ {k \in \mathcal{S} _ t^{\mathbf{\Omega}}} \mathbf{\Upsilon} _ {t}^k$
>
> Subsequently, the server only needs to utilize these aggregated matrices from the partial participants to construct $\mathbf{W} _ t$ and $\mathbf{\Omega} _ {t+1}$ as before. Since only a subset of clients contributed data, the **effective full dataset** should be redefined as the union of data held by all participating clients (i.e., $\mathcal{D}^{\mathcal{S} _ t^{\mathbf{\Omega}}}$ and $\mathcal{D}^{\mathcal{S} _ t^{\mathbf{W}}}$).
>
> Thus, the **invariance** property of our DeepAFL needs to be redefined as being identical to the centralized analytical solution over the **effective full dataset** comprising the data of the participating clients.
>
> In summary, mechanistically, our DeepAFL can easily and effectively handle partial participation without substantial procedural flow adjustments, while critically maintaining its analytical advantage of being invariant to data heterogeneity.
>
> Due to space constraints, we will provide additional experimental results in the next comment block to validate the robustness of our DeepAFL against partial client participation.

---

> ### Author Response · Authors · 2025-11-21
> **Author Response to Reviewer f9VW (Part 4 of 4)**
>
> # Q1: Partial Client Participation (Continued)
>
> ## Q1-2: Performance Analyses for Partial Client Participation
>
> We understand that merely stating in Q1-1 that the procedural flow of our DeepAFL remains essentially unchanged under partial client participation may not be entirely convincing. For instance, although the closed-form solution is preserved over the effective full dataset, one might still be concerned that partial client participation could introduce **discrepancies or shifts** between the classifiers and features learned across different layers, potentially due to the **inconsistency of the training data** involved.
>
> Thus, to **fully alleviate your concerns** and demonstrate the **superior robustness** of our DeepAFL, we proceeded to conduct extensive experiments to analyze how the performance of our DeepAFL changes across different client participation rates $\eta$.
>
> For simplicity, we assume that in each experiment, the participation rate for every aggregation round is consistent: $\eta=|\mathcal{S} _ t^{\mathbf{W}}|/|\mathcal{S}|=|\mathcal{S} _ t^{\mathbf{\Omega}}|/|\mathcal{S}|$. Furthermore, a subset of clients, corresponding to the ratio $\eta$, is randomly sampled for training before each aggregation step.
>
> Specifically, to **enhance the comprehensiveness** of our experimental design, we considered the following two distinct cases:
>
> 1. **Consistent Participation**: the participating client sets for both aggregations within the same layer are identical, namely $\forall t, \mathcal{S} _ t^{\mathbf{W}}=\mathcal{S} _ t^{\mathbf{\Omega}}$.
> 2. **Inconsistent Participation**: the participating client sets for the two aggregations within the same layer are inconsistent, requiring new sampling before each aggregation, namely $\forall t, \mathcal{S} _ t^{\mathbf{W}} \neq \mathcal{S} _ t^{\mathbf{\Omega}}$.
>
> Intuitively, the latter case (Case 2) is more challenging for our DeepAFL. The inconsistent participation within the same layer may heighten the risk of model instability.
>
> Thus, using $\eta=100\\%$ as the control group, we conduct a broad range of experimental analyses with $\eta$ values including $90\\%$, $80\\%$, $70\\%$, $60\\%$, and $50\\%$, to thoroughly assess the performance variation of our DeepAFL on CIFAR-100.
>
> The results for the first case (consistent participation, $\mathcal{S} _ t^{\mathbf{W}}=\mathcal{S} _ t^{\mathbf{\Omega}}$) are presented below.
>
> | $T$  | $\eta=100\\%$ | $\eta=90\\%$ | $\eta=80\\%$ | $\eta=70\\%$ | $\eta=60\\%$ | $\eta=50\\%$ |
> | -- | - | - | - | ---- | --- | -- |
> | 5    | 64.72 | 64.53 | 64.38 | 63.92 | 62.85 | 61.54 |
> | 10   | 65.96 | 65.46 | 65.26 | 64.49 | 64.76 | 63.17 |
> | 15   | 66.56 | 66.39 | 66.20 | 65.38 | 65.17 | 65.08 |
> | 20   | 66.98 | 66.76 | 66.66 | 66.47 | 66.27 | 65.14 |
>
> The results for the second case (inconsistent participation, $\mathcal{S} _ t^{\mathbf{W}}\neq\mathcal{S} _ t^{\mathbf{\Omega}}$) are presented below.
>
> | $T$  | $\eta=100\\%$ | $\eta=90\\%$ | $\eta=80\\%$ | $\eta=70\\%$ | $\eta=60\\%$ | $\eta=50\\%$ |
> | ---- | --- | --- | -- | -- | --- | -- |
> | 5    | 64.72  | 64.21 | 64.03 | 64.05 | 62.19 | 61.36 |
> | 10   | 65.96  | 65.36 | 65.05 | 64.45 | 64.09 | 62.93 |
> | 15   | 66.56  | 66.22 | 66.17 | 65.64 | 64.60 | 64.98 |
> | 20   | 66.98  | 66.57 | 66.51 | 66.32 | 65.50 | 64.81 |
>
> From the results, we can observe that our DeepAFL exhibits substantial robustness under partial client participation in both cases. Specifically, for a high participation rate of $\eta \geq 70\\%$, the maximum accuracy degradation observed for our DeepAFL (at $T=20$) is less than 0.7% across both cases (e.g., 66.98% - 66.32% = 0.66%).
>
> Furthermore, even under the **extremely challenging scenario** of $\eta=50\\%$ (where up to half of the clients drop out during every aggregation step), our DeepAFL still maintains a high accuracy at $T=20$: 65.14% for Case 1 and 64.81% for Case 2.
>
> Crucially, even at a low layer count ($T=5$) and the most severe dropout rate ($\eta=50\\%$), the performance of our DeepAFL (61.54% and 61.36%) still significantly outperforms the **AFL's accuracy (58.56%) achieved under the condition of 100% client participation**.
>
> We believe that these compelling results can fully demonstrate the superior robustness of our DeepAFL against client partial participation.
>
> We are committed to and are actively working on incorporating these detailed findings and analyses into our final paper. Thank you very much for raising this valuable question! It helps us to further explore and substantiate the robustness of our DeepAFL.
>
> ---
>
> This concludes our responses for your first-period comments.
>
> We sincerely hope that our detailed explanations and clarifications can adequately address your concerns. Please feel free to raise any further questions you may have.
>
> Once again, thank you very much for your valuable review. Looking forward to your feedback!

---

### Public Comment · ~Danqi_Wang1 · 2025-11-13
**Questions on Experimental Baselines and the "Representation Learning" claim**

Thank the author for the work on this interesting paper! I've actually read it with great interest last week, but I have a few questions that I'm hoping the authors can help clarify. My main questions center on the experimental design and the practical limits of the method's "representation learning" claim.

**1. On the Use of Residual Connections:**

I was intrigued by the use of residual connections. ResNet was famously designed to solve the vanishing gradient problem, which is specific to backpropagation. Could the authors elaborate on the intuition for why this architecture, which is so tied to gradient-based learning, is also the right choice for a gradient-free, layer-wise analytic setting? What problem is the skip-connection solving here?

**2. On the Experimental Setup and Baselines:**

My second question is about the experimental setup for the gradient-based baselines (FedAvg, etc.). The paper notes all methods use a ResNet-18 backbone. For the gradient-based methods, was this backbone frozen, or was it fine-tuned during the FL process?

This leads to a critical point about the baselines. If we assume the backbones *were* fine-tuned (as is standard), the results imply that gradient-based fine-tuning simply *fails* under Non-IID. But this raises the question of the IID scenario, which is conspicuously absent.

In an IID setting, the 'heterogeneity invariance' of AFL/DeepAFL provides no advantage. Conversely, a method like FedAvg would be able to fine-tune the backbone efficiently and learn superior representations. I strongly suspect that in an IID setting, a standard FedAvg would significantly outperform the baseline AFL, which is limited by its frozen backbone. **The omission of this key IID benchmark makes it impossible to assess the true trade-off.** We can't see the *cost* that DeepAFL pays in representation power (by keeping the backbone frozen) in order to gain its 'invariance'.

Furthermore, the comparison feels imbalanced in terms of parameters. The paper's gains come from adding analytic layers (T=20) to the classification layers. A fairer "apples-to-apples" comparison would be against a gradient-based method with a similarly deepened classification layers (+20 layers) or, more realistically, against a standard FL method using a deeper backbone from the start (e.g., ResNet-34). It's plausible that a deeper, standard backbone optimized with backpropagation would achieve better performance, potentially even mitigating some heterogeneity issues itself.

**3. On the Fundamental Limits of Representation Learning:**

This brings me to my main concern about the "Representation Learning" claim. The paper's strategy is to deepen the classification layers, but the backbone remains frozen. In practice, isn't it more straightforward to simply use a better, deeper backbone trained offline?

I'm struggling to see how this method overcomes the fundamental limitation of AFL. If the frozen backbone is poorly suited for the downstream task (e.g., due to a large domain shift, as the paper notes), DeepAFL will be just as limited as AFL. No amount of post-hoc analytic layers can recover information that the backbone has already discarded. It seems the 'representation learning' here is not about learning *new* features, but just about finding a more complex non-linear mapping to make the *existing*, *fixed* features more linearly separable.

On that note, looking at the results (Tables 1-2), the improvement from T=5 to T=20 is quite modest (only 2-3% absolute). This suggests a strong case of diminishing returns.

**4. On the Theoretical Part:**

Lastly, the theoretical analysis does not guarantee generalization or the learning of **useful** representations. Can we theoretically demonstrate that adding more classification layers results in a “better” representation, or does it merely become more “discriminative”?

Thank the author for their time and for the work on this interesting problem! I look forward to learn more.

---

> ### Public Comment · ~Danqi_Wang1 · 2025-11-25
> **Follow-up Thoughts**
>
> After reviewing the authors' rebuttal, I have a few additional thoughts I would like to discuss. However, recognizing that these are extended reflections and that the authors have already dedicated significant effort to their response, **please consider these points optional (the authors are under no obligation to reply)**.
>
>
> 1.  Modern LLMs possess extremely large hidden dimensions ($d$). Gradient-based Parameter-Efficient Fine-Tuning (PEFT) methods, such as LoRA, typically scale linearly or better ($O(d \times r)$). In contrast, the method here scales quadratically ($O(d^2)$). Does this asymptotic complexity negate the claimed efficiency advantage when applied to large-scale models? This trade-off might warrant a closer look from other NLP researchers.
>
> 2.  In the response to Reviewer ofER, the authors say that if the backbone features are uncorrelated with the labels (due to domain shift), DeepAFL cannot rectify this alignment. This suggests that DeepAFL may not strictly perform "Deep Representation Learning" (which implies the ability to extract semantic features from raw data), but is arguably better described as **"Deep Non-Linear Classification"** on fixed features. Perhaps an adjustment in terminology would be more precise?
>
> 3.  If a model is trained sequentially on Dataset A and then Dataset B, it inevitably suffers from catastrophic forgetting. While there are numerous optimization techniques to mitigate this in gradient-based approaches, does this analytic framework (relying on direct closed-form solutions) have any inherent or adaptable mechanisms to prevent forgetting in such sequential scenarios?
>
> 4.  Does the invariance property strictly require fixed, shared random projections and synchronized participation at each layer? I am curious if invariance can be theoretically maintained when partial participation or stragglers are not explicitly handled. Given that partial participation and asynchrony are fundamental characteristics of practical FL systems, clarifying the bounds of this invariance is critical.

---

> > ### Author Response · Authors · 2025-11-27
> > **Author Response to the Follow-up Public Comments from Danqi Wang (Part 1 of 4)**
> >
> > Thank you very much for your **valuable follow-up comments**.
> >
> > We genuinely appreciate your **thoughtfulness and consideration** in specifically highlighting that we can treat these points as optional.
> >
> > In a spirit of responsibility, we are glad to further proceed with a discussion of the four follow-up points that you raised.
> >
> > As we have already used **Q1-Q4** to reference and track your initial questions in our previous responses, we will continue this numbering convention and use **Q5-Q8** to refer sequentially to your four new questions here.
> >
> >
> >
> > # Q5: On the Efficiency Advantage
> >
> > > Modern LLMs possess extremely large hidden dimensions ($d$). Gradient-based Parameter-Efficient Fine-Tuning (PEFT) methods, such as LoRA, typically scale linearly or better ($d \times r$). In contrast, the method here scales quadratically ($d^2$). Does this asymptotic complexity negate the claimed efficiency advantage when applied to large-scale models? This trade-off might warrant a closer look from other NLP researchers.
> >
> > Thank you very much for raising this insightful question. This is indeed a critical aspect that warrants careful discussion.
> >
> > We would like to clarify our efficiency claim from two distinct perspectives below.
> >
> >
> >
> > ## Q5-1: The Feature Dimensions of DeepAFL are Flexible and Controllable
> >
> > First, we wish to clarify the quadratic parameter complexity ($d^2$) that you mentioned for our DeepAFL.
> >
> > In our DeepAFL, the relevant dimension $d$ that dictates the complexity is **not** the hidden dimension of the backbone network, $d _ \text{X}$. Instead, the complexity is governed by the intermediate feature dimensions, $d _ {\Phi}$ and $d _ {\text F}$.
> >
> > This is because the zero-layer features $\mathbf{\Phi} _ {0}$ are initially mapped from the backbone output via a random projection matrix $\mathbf{A} \in \mathbb{R}^{d _ \text{X} \times d _ {\Phi}}$, as shown in Equation (2) of our paper. Then, the random features $\mathbf{F} _ t$ learned at each subsequent layer are derived through another random projection matrix $\mathbf{B} _ {t} \in \mathbb{R}^{d _ {\Phi} \times d _ \text{F}}$. In the experiments presented in the paper, we set $d _ {\Phi} = d _ \text{F} = 1024$ as the base values.
> >
> > From this perspective, the complexity of DeepAFL is **not tied to the backbone's hidden dimension** but is instead controlled by **user-defined hyperparameters** ($d _ {\Phi}$ and $d _ {\text F}$). Therefore, the complexity of DeepAFL is more accurately analogous to the $r^2$ scaling, rather than $d^2$.
> >
> > **The parameters $d _ {\Phi}$ and $d _ {\text F}$ can be set independently of $d _ \text{X}$, similar to how the rank $r$ is set in LoRA.** For instance, when the backbone output dimension $d _ \text{X}$ is very large, we can flexibly set relatively smaller values for $d _ {\Phi}$ and $d _ {\text F}$ to further enhance efficiency.
> >
> > Here, we also want to acknowledge that setting $d _ {\Phi}$ and $d _ {\text{F}}$ too small will inevitably compromise the performance of our DeepAFL to some degree, as already confirmed by our sensitivity analyses (i.e., Tables 6–8 in our paper). However, we wish to emphasize that this observation represents a **common efficiency-accuracy trade-off**, and **similar phenomena are also observed in LoRA** (where setting a smaller rank $r$ improves efficiency but concurrently impairs performance to some extent).
> >
> > Our primary assertion is that **our DeepAFL possesses the *ability* to control this trade-off** through its flexible dimensions, rather than being fundamentally constrained by a quadratic dependence on the backbone's fixed hidden dimension.
> >
> > ## Q5-2: The Gradient-Free Manner of DeepAFL Leads to Significantly Lower Practical Cost
> >
> > Beyond the parameter complexity, we want to further clarify that the **gradient-free nature** of our DeepAFL **typically makes it more efficient in practice**, even when compared to gradient-based PEFT methods operating at a similar parameter complexity.
> >
> > This significant advantage stems from the fact that the analytic layers in DeepAFL are trained entirely using **closed-form solutions** and require only a **single recursive forward pass**. In contrast, gradient-based methods fundamentally require the **computationally expensive backward pass**, and this process is often **iterative** with multiple epochs.
> >
> > This practical benefit is partially shown in our experiments on efficiency comparisons. While the parameter complexity of the baselines can be considered comparable to DeepAFL, their total training time is substantially higher. Thus, the **training methodology (gradient-based vs. gradient-free) constitutes an equally, or perhaps even more, critical factor for evaluating efficiency**, beyond the consideration of theoretical parameter complexity alone.
> >
> > Thank you again for your insightful question, and we sincerely hope that this response has adequately addressed it.

---

> > > ### Author Response · Authors · 2025-11-27
> > > **Author Response to the Follow-up Public Comments from Danqi Wang (Part 2 of 4)**
> > >
> > > # Q6: On the Terminology of Deep Non-Linear Classification
> > >
> > > > In the response to Reviewer ofER, the authors say that if the backbone features are uncorrelated with the labels (due to domain shift), DeepAFL cannot rectify this alignment. This suggests that DeepAFL may not strictly perform "Deep Representation Learning" (which implies the ability to extract semantic features from raw data), but is arguably better described as **"Deep Non-Linear Classification"** on fixed features. Perhaps an adjustment in terminology would be more precise?
> > >
> > > Thank you very much for this **excellent insight**!
> > >
> > > Given that the current version of DeepAFL commences its representation learning from features pre-extracted by a backbone, your description of **"Deep Non-Linear Classification"** is indeed appropriate. In this context, the original AFL can essentially be regarded as "Linear Classification."
> > >
> > > However, we still believe that characterizing DeepAFL as having **Deep Representation Learning Capabilities** is also justified and not an overstatement. As we discussed in our response to your Q3-3, the effect of DeepAFL can be viewed as highly analogous to an **analytic MLP with residual connections**, operating subsequent to a frozen backbone. Once this perspective is accepted, referring to DeepAFL as performing **Deep Representation Learning** becomes as natural as attributing the same capability to an MLP.
> > >
> > > In fact, **the core reason** why the current version of DeepAFL requires a backbone is **not due to** its inherent MLP-like nature being unable to extract semantic features from raw data, **but rather because** it has not yet been extended to an **analytic CNN** architecture, making direct application to vision tasks difficult.
> > >
> > > To illustrate, if DeepAFL were directly applied to ***tabular data***, the backbone would be unnecessary. In this scenario, DeepAFL would certainly be capable of **directly extracting semantic features from the raw data** due to its **MLP-like effect**. From this viewpoint, we believe that asserting our DeepAFL possesses Deep Representation Learning capabilities is entirely justified.
> > >
> > > Thus, the backbone in our paper can be viewed as converting ***raw vision data*** into a feature representation akin to ***tabular data*** with minimal information loss. Then, our DeepAFL is primarily utilized to perform deep representation learning on this derived *"tabular" data*.
> > >
> > > Therefore, we conclude that the current limitation of DeepAFL is **not its inability to extract information from raw data** (assuming the raw data is tabular), but rather its current **initial stage (i.e., the MLP-like version)**, which has yet to be extended to a CNN-like version suitable for raw vision data.
> > >
> > > You may be curious why our paper used image classification, a task where DeepAFL is not optimally suited, instead of tabular tasks for experimental validation. Our initial motivation was primarily to **align with the original AFL work**, which also employed image tasks. Additionally, showcasing DeepAFL's ability to perform further representation learning on top of already-extracted backbone features **may better broaden its potential applicability**.
> > >
> > > We think DeepAFL would show better performance on tabular data, as it could **directly extract semantic features from the raw data** without potential information loss from the backbone. In fact, if this paper is fortunate enough to be accepted by ICLR, we will consider preparing supplementary experiments on tabular data and **including them in the journal extension of our work**.
> > >
> > > Furthermore, we anticipate that you might be interested in whether DeepAFL can be extended from an **"analytic deep residual MLP"** to an **"analytic deep residual CNN"** in the future. As far as we are concerned, we believe this is a **highly promising and feasible future direction**.
> > >
> > > Specifically, related work has already investigated deriving **closed-form solutions for CNNs** in the field of analytic learning [10]. In fact, we have already compared our DeepAFL's deepening strategy with this work's strategy in Appendix E.3.7 of our paper, finding their deepening approach to be somewhat weaker. Thus, combining the core ideas of DeepAFL with existing closed-form solutions for CNNs is a very **natural line of future research**.
> > >
> > > Starting from our DeepAFL, we believe that many promising future works can be built upon this foundation. We are also committed to them ourselves. If you are interested, we genuinely welcome you to follow our subsequent efforts in the future!
> > >
> > > Thank you again for your insightful question, and we sincerely hope that this response has adequately addressed it.
> > >
> > > ## References
> > >
> > > [10] Zhuang, Huiping, et al. An analytic formulation of convolutional neural network learning for pattern recognition. *Information Sciences*, 2025.

---

> > > > ### Author Response · Authors · 2025-11-27
> > > > **Author Response to the Follow-up Public Comments from Danqi Wang (Part 3 of 4)**
> > > >
> > > > # Q7: On the Closed-Form Solutions against Catastrophic Forgetting
> > > >
> > > > > If a model is trained sequentially on Dataset A and then Dataset B, it inevitably suffers from catastrophic forgetting. While there are numerous optimization techniques to mitigate this in gradient-based approaches, does this analytic framework (relying on direct closed-form solutions) have any inherent or adaptable mechanisms to prevent forgetting in such sequential scenarios?
> > > >
> > > > Thank you for your insightful question.
> > > >
> > > > Your intuition is indeed **spot-on**. Catastrophic forgetting is a crucial issue in the field of Continual Learning (CL), also known as incremental or life-long learning.
> > > >
> > > > In fact, in our DeepAFL, we establish the **invariance** of the closed-form solutions within the FL setting, characterized by two main features:
> > > >
> > > > * The **final result remains invariant** regardless of the spatial Non-IID variations.
> > > >
> > > > * The result obtained via the distributed computation is **identical** to the centralized analytic solution computed on the full dataset across all clients.
> > > >
> > > > This invariance can indeed be **effectively extended** to the CL setting, as we are essentially shifting the **spatial Non-IID variations** into the **temporal dimension**. In this sequential scenario, the invariance still holds, and the result is **consistent** with the **joint analytic solution** computed on the full dataset across **all learning phases**.
> > > >
> > > > In this context, we can consider that catastrophic forgetting is addressed, since the outcomes are identical to those of joint analytic learning. Thus, in CL, this invariance within the analytic framework is often termed **absolute memorization** [11], [15], [19]. In the CL community, numerous works have adopted the analytic framework (i.e., pre-trained backbone + analytic classifier), and many of them have already achieved SOTA performance due to the desirable properties [11]-[22].
> > > >
> > > > However, to the best of our knowledge, there is currently no analytic method in the CL field that possesses the gradient-free representation learning capability demonstrated by our DeepAFL. As discussed in **Section 5** of our paper, we also identified that introducing the core idea of DeepAFL into CL represents a **valuable and non-trivial direction for future work**.  In fact, we are currently actively working towards this goal.
> > > >
> > > > Thank you again for your insightful question. We sincerely hope that this response has adequately addressed your query.
> > > >
> > > > ## References
> > > >
> > > > [11] Zhuang, Huiping, et al. ACIL: Analytic class-incremental learning with absolute memorization and privacy protection. *Advances in Neural Information Processing Systems (NeurIPS)*, 2022.
> > > >
> > > > [12] McDonnell, Mark D., et al. RanPAC: Random projections and pre-trained models for continual learning. *Advances in Neural Information Processing Systems (NeurIPS)*, 2023.
> > > >
> > > > [14] Yue, Xianghu, et al. MMAL: Multi-modal analytic learning for exemplar-free audio-visual class incremental tasks. *Proceedings of the 32nd ACM International Conference on Multimedia (MM)*, 2024.
> > > >
> > > > [15] Zhuang, Huiping, et al. GACL: Exemplar-free generalized analytic continual learning. *Advances in Neural Information Processing Systems (NeurIPS)*, 2024.
> > > >
> > > > [16] Zhuang, Huiping, et al. F-OAL: Forward-only online analytic learning with fast training and low memory footprint in class incremental learning. *Advances in Neural Information Processing Systems (NeurIPS)*, 2024.
> > > >
> > > > [17] Peng, Liangzu, et al. TSVD: Bridging theory and practice in continual learning with pre-trained models. *The Thirteenth International Conference on Learning Representations (ICLR)*, 2025.
> > > >
> > > > [18] Tran, Quyen, et al. Boosting Multiple Views for pretrained-based Continual Learning. *The Thirteenth International Conference on Learning Representations (ICLR)*, 2025.
> > > >
> > > > [19] Zhang, Xiang, et al. L3A: Label-Augmented Analytic Adaptation for Multi-Label Class Incremental Learning. *Forty-second International Conference on Machine Learning (ICML)*, 2025.
> > > >
> > > > [20] Tong, Kai, et al. Any-SSR: How Recursive Least Squares Works in Continual Learning of Large Language Model. *Proceedings of the IEEE/CVF International Conference on Computer Vision (ICCV)*, 2025.
> > > >
> > > > [21] Guo, Zihao, et al. GraphKeeper: Graph Domain-Incremental Learning via Knowledge Disentanglement and Preservation. *The Thirty-ninth Annual Conference on Neural Information Processing Systems (NeurIPS)*, 2025.
> > > >
> > > > [22] Jiang, Kai, et al. Mixture of Noise for Pre-Trained Model-Based Class-Incremental Learning. *The Thirty-ninth Annual Conference on Neural Information Processing Systems (NeurIPS)*, 2025.

---

> > > > > ### Author Response · Authors · 2025-11-27
> > > > > **Author Response to the Follow-up Public Comments from Danqi Wang (Part 4 of 4)**
> > > > >
> > > > > # Q8: On the Invariance under Partial Participation and Asynchrony
> > > > >
> > > > > > Does the invariance property strictly require fixed, shared random projections and synchronized participation at each layer? I am curious if invariance can be theoretically maintained when partial participation or stragglers are not explicitly handled. Given that partial participation and asynchrony are fundamental characteristics of practical FL systems, clarifying the bounds of this invariance is critical.
> > > > >
> > > > > Thank you for your insightful question.
> > > > >
> > > > > We believe this follow-up question was inspired by our preceding responses to Reviewer f9VW's Q1 and Reviewer mCa1's Q1. Here, we aim to provide a more detailed analysis.
> > > > >
> > > > > The core meaning of invariance in our original paper can be stated as: *the model weights derived from our DeepAFL are identical to the centralized analytical solutions over the **full dataset** $\mathcal{D}$, regardless of how the full dataset is distributed among the clients.*
> > > > >
> > > > > Let’s now consider the **partial client participation** in our DeepAFL. We denote the full set of all clients as $\mathcal{S}$, where the total number of clients is $|\mathcal{S}|=K$. We assume that at layer $t$, the set of clients participating in the classifier construction is $\mathcal{S} _ t^{\mathbf{W}}$, and the set of clients participating in the feature enhancement is $\mathcal{S} _ t^{\mathbf{\Omega}}$.
> > > > >
> > > > > Under partial client participation, the **original invariance property cannot be strictly satisfied**. This is highly intuitive, as the total data observed by DeepAFL during each training step is not equivalent to the full dataset of all clients due to dropouts, i.e., $\bigcup _ {k \in \mathcal{S} _ t^{\mathbf{W}}} \mathcal{D} _ k \neq \mathcal{D}$ and $\bigcup _ {k \in \mathcal{S} _ t^{\mathbf{\Omega}}} \mathcal{D} _ k \neq \mathcal{D}$.
> > > > >
> > > > > In this case, we can **extend the original invariance property** by defining an ***effective full dataset***. Since only a subset of clients contributed data, the effective full dataset (i.e., $\mathcal{D}^{\mathcal{S} _ t^{\mathbf{W}}}$ and $\mathcal{D}^{\mathcal{S} _ t^{\mathbf{\Omega}}}$) should be defined as the union of data held by all participating clients, as follows:
> > > > >
> > > > > $\mathcal{D}^{\mathcal{S} _ t^{\mathbf{\Omega}}} = \bigcup _ {k \in \mathcal{S} _ t^{\mathbf{\Omega}}} \mathcal{D} _ k$
> > > > >
> > > > > $\mathcal{D}^{\mathcal{S} _ t^{\mathbf{W}}} =\bigcup _ {k \in \mathcal{S} _ t^{\mathbf{W}}} \mathcal{D} _ k $
> > > > >
> > > > > Thus, accounting for partial participation and asynchrony, the invariance property of our DeepAFL can be **redefined** with the help of the ***effective full dataset***. Specifically, for each analytic layer $t$, the model weights $\mathbf{W} _ {t}$ and $\mathbf{\Omega} _ {t}$ are identical to the **centralized analytical solution** over the ***effective full dataset***, $\mathcal{D}^{\mathcal{S} _ t^{\mathbf{W}}}$ and $\mathcal{D}^{\mathcal{S} _ t^{\mathbf{\Omega}}}$, respectively.
> > > > >
> > > > > Moreover, we kindly recommend referencing our response to **Reviewer f9VW's Q1**, which also shows our DeepAFL's robustness against partial participation in practice.
> > > > >
> > > > > Furthermore, as you also questioned the impact of **shared random projections** on our DeepAFL, we would like to clarify that the shared random projections, $\mathbf{A}$ and $\\{\mathbf{B} _ {t}\\} _ {t=0}^{T-1}$, are indeed an **essential prerequisite** for the invariance property, as explicitly stated in our Theorem 1.
> > > > >
> > > > > If different clients use different random projections at the same layer, features would be mapped to entirely different high-dimensional spaces, thus disrupting the aggregation process.
> > > > >
> > > > > Yet, maintaining the same random projections across all clients is not difficult. It simply requires the server to synchronize and confirm the random projections (or their corresponding random seeds) for each layer with every client upon their initial registration.
> > > > >
> > > > > Since Reviewer mCa1 previously raised a very similar question, we encourage further reference to our Response to Reviewer mCa1 in Q1, which provides more exhaustive analyses.
> > > > >
> > > > > Thank you again for your insightful question. We sincerely hope that this response has adequately addressed your query.
> > > > >
> > > > > ---
> > > > >
> > > > > This concludes our responses to your thoughtful follow-up questions.
> > > > >
> > > > > Although you were not an official reviewer of our paper, we **sincerely appreciate** the high quality of your questions and the beneficial discussion we have conducted.
> > > > >
> > > > > **If you find our responses to have addressed your concerns, we would be deeply grateful if you could kindly offer further positive remarks in your final comment.** Your support would significantly aid the Area Chair and other reviewers in better evaluating the merits of our work.
> > > > >
> > > > > Once again, thank you a lot for your strong interest in our work!

---

> ### Author Response · Authors · 2025-11-25
> **Author Response to the Initial Public Comments from Danqi Wang (Part 1 of 9)**
>
> We are genuinely pleased that our work has garnered attention from you **as an external reader** during the ICLR review period. **Thank you sincerely for your valuable interest in our work and your generous recognition of its contribution!**
>
> Specifically, we find all your questions to be highly professional and insightful, and we have been carefully considering them over the last few days. **Yet, we must express our sincere apologies that our response to you is slightly delayed.** This is because, in the past few days, our primary focus has been dedicated to completing the additional experiments and generating the required illustrations requested by the other reviewers. **We sincerely appreciate your patience and understanding.**
>
> Here, we would like to respectfully address your initially raised questions point by point, as follows.
>
> # Q1: On the Use of Residual Connections
>
> > I was intrigued by the use of residual connections. ResNet was famously designed to solve the vanishing gradient problem, which is specific to back-propagation. Could the authors elaborate on the intuition for why this architecture, which is so tied to gradient-based learning, is also the right choice for a gradient-free, layer-wise analytic setting? What problem is the skip-connection solving here?
>
> We think this is an excellent and insightful question! We fully agree that analyzing the similarities in the role of residual connections between gradient-based and gradient-free learning is highly beneficial for the advancement of both topics.
>
> Regarding the point you raised, we wish to respectfully remind you that **we have provided related discussions in Appendix F of our original paper**. Recognizing that you may be looking for more in-depth analyses, we would like to further elaborate on our work's rationale **from three distinct perspectives**:
>
> 1. **Similar Problem:** Performance Saturation and Degradation
> 2. **Similar Optimization:** Feature Reuse and Correction
> 3. **Similar Philosophy:** Ensemble Learning and Boosting
>
> The detailed analyses are provided in the following Subsections *Q1-1* to *Q1-3* below.
>
> ## Q1-1: Similar Problem: Performance Saturation and Degradation
>
> First, let‘s revisit the historical context surrounding the proposal of ResNet in early deep learning. When network depth surpassed a certain threshold, researchers observed an anomaly: performance not only ceased to improve but instead exhibited a trend of ***saturation or even degradation***. This phenomenon is often termed the *"Degradation Problem"*.
>
> In our proposed **DeepAFL**, attempting to deepen the analytic layers intrinsically faces a very similar issue. Indeed, our original paper provides extensive **ablation studies** in Tables 12–17 and Figures 8–10. Specifically, Ablation (1) just presents the results of our DeepAFL architecture *without* the residual skip connections.
>
> The results clearly show that our DeepAFL consistently outperforms Ablation (1), confirming the effectiveness of residual connections within our DeepAFL. More critically, when observing the performance of Ablation (1) as the layers become deeper (as shown in the blue lines of Figures 8–10), the trend exhibits a strong tendency for performance to **saturate readily**.
>
> Specifically, focusing on Figure 8 (i.e., the results on CIFAR-10), Ablation (1) shows a distinct **degradation in test accuracy** as the layers become deeper. These empirical results precisely indicate that, without residual connections, both gradient-based and gradient-free learning appear to confront a very similar problem of ***performance saturation and degradation*** with increasing depth. This observation served as a key inspiration and motivation for incorporating residual connections into our DeepAFL.

---

> ### Author Response · Authors · 2025-11-25
> **Author Response to the Initial Public Comments from Danqi Wang (Part 2 of 9)**
>
> ## Q1-2: Similar Optimization: Feature Reuse and Correction
>
> Then, we want to highlight that, whether in gradient-based or gradient-free approaches, the skip connections perform a **similar optimization** on the feature learning process. Specifically, the skip connections enhance **feature reuse** and help avoid feature loss in deep networks [1]. In this way, the network does not need to relearn effective features already acquired by shallow layers in the deeper stages.
>
> Thus, in ResNet, the network's task is no longer to **relearn the entire feature** (with global and drastic non-linear transformations) but to learn a **residual correction** (with local and minimal deformation) to the existing features. This encourages the network to learn only the smallest modification necessary to make the manifold flatter and closer to linearly separable.
>
> In our gradient-free DeepAFL, the aim of the residual connection is similar. Specifically, each analytic layer in our architecture contains a random projection $\mathbf{B} _ {t}$ with an activation function $\sigma(\cdot)$, which may easily damage the effective features learned by the preceding layer during the forward pass. This is empirically validated by our earlier analysis of Figure 8 for the ablation results on CIFAR-10. Specifically, as the layers become deeper, the training accuracy of Ablation (1) shows virtually no improvement, and its test accuracy even exhibits a distinct degradation.
>
> More importantly, **the key theoretical property of our DeepAFL for representation learning**, namely the non-increasing nature of its empirical risk with added layers, is also **directly ensured by the use of skip connections**. Specifically, when network features converge, each residual block can be minimally set to zero to achieve an unchanging empirical risk, guaranteeing that the overall empirical risk of our DeepAFL will not increase. **If we remove the skip connections, our Theorems 2-3 will no longer hold, and the representation learning capability of our DeepAFL will be severely compromised.**
>
>
>
> ## Q1-3: **Similar Philosophy:** Ensemble Learning and Boosting
>
> Finally, we consider the residual connection from the perspective of ensemble learning, a concept that offers a unifying philosophical framework for both gradient-based and gradient-free deep architectures.
>
> A key insight into the success of ResNet is that it behaves like an **ensemble of numerous relatively shallow networks** [2]. By unrolling the recursive definition, we can define ResNet as the ensemble of weak and shallow neural networks of varying sizes as follows:
>
> $\mathbf{\Phi} _ {T} =\mathbf{\Phi} _ {T-1}+g _ {t}(\mathbf{\Phi} _ {T-1})=\cdots=\mathbf{\Phi} _ {0}+\sum _ {t=1}^{T}g _ {t}(\mathbf{\Phi} _ {t-1})$
>
> This ensemble perspective aligns remarkably well with the philosophy of boosting, a sequential ensemble technique. In Boosting, models (or "weak learners") are trained sequentially, with each new model attempting to **correct the errors** made by the previous ones. Therefore, each layer in the ResNet can be viewed as a weak learner, which already inspired much subsequent development in gradient boosting [2]–[5].
>
> In our gradient-free DeepAFL, each layer's task is similarly focused on the **residual error** left unexplained by the previous layers, also aligning with the philosophy of gradient boosting. **A deeper explanation is that, under the chosen MSE lossof our DeepAFL, the negative gradient of the loss function is precisely the residual.** Therefore, by continuously fitting the residuals from previous layers, our DeepAFL can be seen as learning pseudo-gradients.
>
> In summary, the deep structure in both paradigms can be conceptually viewed as an iterative, sequential process where each layer contributes a minimal and additive correction to the cumulative representation, thereby boosting the overall feature quality and predictive power. This shared philosophy may help unify the understanding of the residual connections in both gradient-based and gradient-free learning approaches.
>
> ---
>
> We sincerely hope that our preceding clarifications can adequately address your question.
>
> Thank you again for your insightful comments!
>
> ## References
>
> [1] Huang, Gao, et al. Densely connected convolutional networks. *Proceedings of the IEEE Conference on Computer Vision and Pattern Recognition (CVPR)*. 2017.
>
> [2] Veit, Andreas, et al. Residual networks behave like ensembles of relatively shallow networks." *Advances in Neural Information Processing Systems (NeurIPS)*, 2016.
>
> [3] Huang, Furong, et al. Learning deep ResNet blocks sequentially using boosting theory." *International Conference on Machine Learning (ICML)*, 2018.
>
> [4] Nitanda, Atsushi, et al. Functional gradient boosting based on residual network perception. *International conference on machine learning (ICML)*. 2018.
>
> [5] Suggala, Arun, et al. Generalized boosting. *Advances in Neural Information Processing Systems (NeurIPS)*, 2020.

---

> ### Author Response · Authors · 2025-11-25
> **Author Response to the Initial Public Comments from Danqi Wang (Part 3 of 9)**
>
> # Q2: On the Experimental Setup and Baselines
>
> > My second question is about the experimental setup for the gradient-based baselines (FedAvg, etc.). The paper notes all methods use a ResNet-18 backbone. For the gradient-based methods, was this backbone frozen, or was it fine-tuned during the FL process?
> >
> > This leads to a critical point about the baselines. If we assume the backbones *were* fine-tuned (as is standard), the results imply that gradient-based fine-tuning simply *fails* under Non-IID. But this raises the question of the IID scenario, which is conspicuously absent.
> >
> > In an IID setting, the 'heterogeneity invariance' of AFL/DeepAFL provides no advantage. Conversely, a method like FedAvg would be able to fine-tune the backbone efficiently and learn superior representations. I strongly suspect that in an IID setting, a standard FedAvg would significantly outperform the baseline AFL, which is limited by its frozen backbone. The omission of this key IID benchmark makes it impossible to assess the true trade-off. We can't see the *cost* that DeepAFL pays in representation power (by keeping the backbone frozen) in order to gain its 'invariance'.
> >
> > Furthermore, the comparison feels imbalanced in terms of parameters. The paper's gains come from adding analytic layers (T=20) to the classification layers. A fairer "apples-to-apples" comparison would be against a gradient-based method with a similarly deepened classification layers (+20 layers) or, more realistically, against a standard FL method using a deeper backbone from the start (e.g., ResNet-34). It's plausible that a deeper, standard backbone optimized with back-propagation would achieve better performance, potentially even mitigating some heterogeneity issues itself.
>
> Thank you for your valuable questions.
>
> First of all, we would like to clarify our experimental setup. To maintain fairness in comparisons, the results for all baselines in our paper were directly employed from the benchmark provided in the **original AFL paper**. Specifically, within the original AFL paper, the gradient-based approaches follow a similar setup to AFL, where the backbone is frozen during the FL process.
>
> Then, we would like to address your two related concerns sequentially, including the **trade-off between gradient-based and gradient-free approaches** and the **fairer "apples-to-apples" comparisons**.
>
>
>
> ## Q2-1: Trade-Off Between Gradient-based and Gradient-free Approaches
>
> We fully agree that the ideal IID scenario that you mentioned is indeed very meaningful, particularly for analyzing the *cost* that DeepAFL pays in representation power (by keeping the backbone frozen) to gain its *invariance*, **as we all know there is no *free lunch***.
>
> Thus, we conducted additional experiments for standard FedAvg under the ideal scenario (IID condition and unfrozen backbone). Taking CIFAR-100 as an example, the test set best accuracy for standard FedAvg under these conditions reaches 78.33%, which can be considered an upper-bound performance. This result is indeed higher than AFL (58.56%) and our DeepAFL (66.98%), as we expected.
>
> However, we wish to emphasize that such ideal upper-bound performance is often unachievable in FL, as Non-IID data is widely acknowledged as a pervasive issue. Thus, our gradient-free DeepAFL retains a unique advantage in practical Non-IID scenarios due to its superior invariance. **Logically, its advantage is expected to become more pronounced as the degree of Non-IID increases.**
>
> Meanwhile, **excellent efficiency** is also a major advantage of our gradient-free DeepAFL, as it is fundamentally a one-shot method and does not require back-propagation. In fact, as the gradient-based baselines typically require multi-epoch computation and multi-round communications, even if they freeze the backbone and only optimize the classifier, their computational and communication overheads are far higher than those of our DeepAFL. This overhead would only increase if we considered unfreezing the backbone or introducing a multi-layer neural network after the backbone.
>
> In summary, the primary advantages of our DeepAFL lie in its **invariance** and **efficiency** in FL. Although its current characteristic of freezing the backbone might result in a certain gap from the upper-bound performance, we believe this does not undermine its core advantages, **especially in highly Non-IID and resource-constrained FL scenarios.**
>
> We hope this additional analysis can help you better understand the trade-off between the gradient-based and gradient-free approaches.
>
> P.S. We also believe that it is **a promising future direction** to consider combining gradient-based and gradient-free approaches to construct a *hybrid one* based on our DeepAFL, as we anticipate this has the potential to **integrate the advantages of both methodologies simultaneously**.

---

> ### Author Response · Authors · 2025-11-25
> **Author Response to the Initial Public Comments from Danqi Wang (Part 4 of 9)**
>
> ## Q2-2: "Apples-to-Apples" Comparisons
>
> Regarding your suggestion for a fairer "apples-to-apples" comparison, we would like to point out that we have conducted relevant experiments in our **Response to Reviewer ZKVc in Q1**. Here, we would like to further clarify and re-present these results for you.
>
> Specifically, we introduced **multiple trainable layers** ($T=2, 5, 10, 20$) for the gradient-based baselines. We took two baselines, FedAvg and FedDyn, as examples and conducted the aforementioned experiments on CIFAR-100 under the Dirichlet distribution with $\alpha=0.1$. We kept the gradient-based methods' pre-trained backbone fixed and only trained their additional layers. The feature dimension of the additional layers for the gradient-based methods was kept consistent with that of our DeepAFL (i.e., 1024).
>
> The results are presented in the table below:
>
> |        | FedAvg | FedDyn |
> | ------ | ------ | ------ |
> | $T=1$  | 56.62% | 57.55% |
> | $T=2$  | 55.83% | 57.22% |
> | $T=5$  | <5%    | 56.26% |
> | $T=10$ | <3%    | <3%    |
> | $T=20$ | <1%    | <1%    |
>
> Through these results, we can observe that the performance of the gradient-based methods gradually and constantly degrades as $T$ increases.
>
> When $T$ increases beyond a certain threshold (e.g., $T\geq10$), the excessive number of parameters, combined with the Non-IID data in the FL scenario, may cause the training process to collapse.
>
> These results fully show that directly increasing the number of network layers after the backbone for the baselines, similar to our DeepAFL setup, would have a **detrimental effect on their performance**. Thus, the "apples-to-apples" comparison seems to make the baseline's performance **even worse**.
>
> This phenomenon is likely because, as the number of layers deepens, the extent of the baseline's back-propagation increases, making convergence more difficult, especially when considering the Non-IID data.
>
> ---
>
> We sincerely hope that our preceding explanations and clarifications can adequately address your concerns.
>
> Thank you again for your insightful comments!

---

> ### Author Response · Authors · 2025-11-25
> **Author Response to the Initial Public Comments from Danqi Wang (Part 5 of 9)**
>
> # Q3: On the Representation Learning Claim
>
> > This brings me to my main concern about the "Representation Learning" claim. The paper's strategy is to deepen the classification layers, but the backbone remains frozen. In practice, isn't it more straightforward to simply use a better, deeper backbone trained offline?
> >
> > I'm struggling to see how this method overcomes the fundamental limitation of AFL. If the frozen backbone is poorly suited for the downstream task (e.g., due to a large domain shift, as the paper notes), DeepAFL will be just as limited as AFL. No amount of post-hoc analytic layers can recover information that the backbone has already discarded. It seems the 'representation learning' here is not about learning new features, but just about finding a more complex non-linear mapping to make the existing, fixed features more linearly separable.
> >
> > On that note, looking at the results (Tables 1-2), the improvement from T=5 to T=20 is quite modest (only 2-3% absolute). This suggests a strong case of diminishing returns.
>
> We sincerely thank you for your constructive feedback. Here, we would like to address your concerns by clarifying four key points below:
>
> * **DeepAFL Aligns with the Concept of Representation Learning**
> * **Enhanced Backbones Cannot Fully Resolve AFL's Underfitting**
> * **More Separable Features Intrinsically Mean Better Representations**
> * **Diminishing Returns are Normal Phenomena in Representation Learning**
>
> The detailed analyses are provided in the following Subsections *Q3-1* to *Q3-4* below.
>
>
>
> ## Q3-1: DeepAFL Aligns with the Concept of Representation Learning
>
> First of all, considering your primary concern regarding our *"Representation Learning"* claim, we want to clarify why our DeepAFL aligns with the concept of representation learning.
>
> According to the descriptions of representation learning by **Yoshua Bengio et al.** [6] and ***Wikipedia*** [7], we, conceptually summarize **two essential core characteristics** of representation learning:
>
> * **Automated Feature Extraction:** It involves learning feature transformations (which are often non-linear) directly from the data, thereby avoiding the complexity and reliance on expert knowledge associated with manual feature engineering.
>
> * **Utility for Downstream Tasks:** The extracted features are beneficial and useful for adapting to specific downstream tasks, meaning they provide **utility when building predictors (e.g., classifiers)**, ultimately leading to enhanced model performance.
>
> Our DeepAFL just directly satisfies these two characteristics:
>
> * The **feature transformations** involve trainable parameters $\mathbf{\Omega}$, which are **automatically learned from the data** via the “sandwiched” least squares. Moreover, the use of the activation function $\sigma(\cdot)$ ensures that the layer-wise transformation is **non-linear**.
>
> * The feature transformations show **evident and significant benefits for our downstream task (i.e., classification)**. Specifically, when paired with an analytic classifier, the accuracy of our DeepAFL consistently improves on the training & test sets as $T$ increases (as shown in Tables 9-11 of our paper).
>
> Thus, **we maintain that the capability of our DeepAFL indeed aligns with the concept of representation learning**.
>
> Since Reviewer ofER previously raised a very similar question, we encourage further reference to our **Response to Reviewer ofER in W2 & Q4**, which provides more exhaustive analyses.

---

> ### Author Response · Authors · 2025-11-25
> **Author Response to the Initial Public Comments from Danqi Wang (Part 6 of 9)**
>
> ## Q3-2: Enhanced Backbones Cannot Fully Resolve AFL's Underfitting
>
> Then, we want to clarify the core scope of our DeepAFL.
>
> Recall that standard AFL operates by placing a **linear analytic classifier** after a fixed backbone. Thus, AFL faces two fundamental and obvious limitations:
>
> 1. **Dependence on the Backbone:** AFL relies on the backbone as the feature extractor.
> 2. **Linear Classification Constraint:** AFL's analytic classifier is restricted to linear classification and often suffers from **underfitting**. When the features provided by the backbone have weak linear separability, AFL struggles to achieve optimal performance, even if the features contain enough information.
>
> In this paper, the core motivation behind our DeepAFL design was primarily to address the **second fundamental limitation** (i.e., the underfitting of linear analytic classifiers).
>
> We now directly address your concern:
>
> > The paper's strategy is to deepen the classification layers, but the backbone remains frozen. In practice, isn't it more straightforward to simply use a better, deeper backbone trained offline?
>
> We want to emphasize that the underfitting of analytic classifiers, which our DeepAFL solves, is a **general, pervasive, and significant issue**. To a large extent, **simply using a better, deeper backbone trained offline cannot fully resolve this underfitting issue.**
>
> To illustrate this critical point, we refer to the results presented in our Response to Reviewer mCa1 in W1, where we specifically investigate DeepAFL's performance when integrated with much **larger ViT backbones**.
>
> The results confirm that even with the large ViT backbone, which is substantially more powerful than ResNet-18, AFL's analytic classifier still exhibits an underfitting issue. Specifically, the performance gain of our DeepAFL over standard AFL remains highly significant (with an improvement of up to 5.06% when using the ViT-B-16-I-1K backbone). **Thus, our DeepAFL consistently provides additional performance gains even when a *"better"* backbone is already in use.**
>
> More importantly, the **computational overhead** introduced by our DeepAFL is nearly **negligible** when compared to the resources required to enhance the backbone during pre-training. Thus, the low overhead behind the performance gain is also a key advantage of DeepAFL, especially when compared to the substantial resources (e.g., massive pre-training data and computing power) required for improving the backbone.
>
> In essence, we can view the enhancement of the pre-trained backbone's performance (offline) and the improvement of the classifier's fitting capacity as two largely orthogonal problems. In practice, addressing only one of these problems may often be insufficient to achieve the optimal performance gains.
>
> While we acknowledge that our DeepAFL has not eliminated the dependence on the backbone, we firmly believe that this does not diminish the primary advantage of DeepAFL: **granting the analytic classifier the capability of representation learning while preserving its gradient-free manner.** This key advantage allows our DeepAFL to move beyond merely learning linear mappings on the extracted features, enabling the learning of more complex non-linear relationships from features with high efficiency.
>
> Meanwhile, we would like to further clarify that the analytic learning community has already begun attempts to address the dependence on the frozen backbone [8], and we fully agree that integrating this technique with our DeepAFL to achieve potentially stronger performance is a promising and interesting avenue for future work.
>
> Since Reviewer ofER previously raised a very similar question, we encourage further reference to our **Response to the Reviewer ofER in W3 & Q2**, which provides more exhaustive analyses.

---

> ### Author Response · Authors · 2025-11-25
> **Author Response to the Initial Public Comments from Danqi Wang (Part 7 of 9)**
>
> ## Q3-3: More Separable Features Intrinsically Mean Better Representations
>
> We appreciate your comment:
>
> > It seems the 'representation learning' here is not about learning new features, but just about finding a more complex non-linear mapping to make the existing, fixed features more linearly separable.
>
> We want to clarify that **more separable features intrinsically mean better representations**
>
> As we previously analyzed, due to the frozen backbone, our DeepAFL is indeed not learning "entirely new" features from the raw input. Instead, it operates on the features extracted by the backbone by finding a more complex non-linear mapping to optimize their geometric configuration and enhance their linear separability.
>
> However, **this distinction does not conflict with our claim of representation learning**.
>
> The goal of representation learning is to transform the features to make it easier to extract useful information when building a predictor. By increasing the linear separability of the existing features through a non-linear mapping, our DeepAFL **indeed enhances the utility for downstream tasks** (i.e., classification) and directly **solves the underfitting issue of the analytic classifier**.
>
> In fact, the notion that ***improved linear separability of features directly reflects the extraction of increasingly abstract and useful representations*** in deep networks has already been well-established.
>
> For instance, **Guillaume Alain and Yoshua Bengio** proposed in their work to use **linear classifier probes** to understand intermediate layers, asserting that better linear separability is indicative of a more meaningful representation [9].
>
> To provide you with a more intuitive understanding of this point, please allow us to **draw an analogy between our DeepAFL and the Multi-Layer Perceptron (MLP)** with residual connections.
>
> Given that the MLP is recognized as the **most fundamental DNN**, if we were to claim that an MLP trained via back-propagation achieves representation learning, you may undoubtedly concur.
>
> At this point, what the residual block of the MLP learns is nothing more than subjecting the features from the previous layer to an **affine transformation $\mathbf{W} _ {t+1}$** followed by an **activation function $\sigma(\cdot)$**, where the affine transformation $\mathbf{W} _ {t+1}$ constitutes the learnable parameters of the MLP, as follows:
>
> $g _ {t+1}(\mathbf{\Phi} _ {t}) = \sigma(\mathbf{\Phi} _ {t} \mathbf{W} _ {t+1})$
>
> This process can also be precisely interpreted as deriving a more complex non-linear mapping to enhance the linear separability of the existing features from the previous layer.
>
> If we consider the formulation of our DeepAFL, we can clearly see that the approach is **very similar to that of the MLP**, as it also involves the alternating stacking of affine transformations and non-linear activation functions:
>
> $g _ {t+1}(\mathbf{\Phi} _ {t}) = \sigma(\mathbf{\Phi} _ {t}\mathbf{B} _ {t})\mathbf{\Omega} _ {t+1}$
>
> The primary distinction is that the learnable parameters $\mathbf{W} _ {t+1}$ in the MLP act *inside* the activation function, whereas the learnable parameters $\mathbf{\Omega} _ {t+1}$ in our DeepAFL act *outside* the activation function, but this localized difference is effectively diminished when multiple layers are stacked.
>
> Thus, if AFL is approximated as attaching a linear classification layer after the backbone, then **our DeepAFL can be analogously viewed as attaching an MLP with complex non-linear mapping after the backbone**. Crucially, this approximated *"MLP"* in our DeepAFL is trained in a gradient-free manner, thus enabling its **invariance and high efficiency**.
>
> We sincerely hope that our above analogy can help you better understand **how our DeepAFL operates to improve the features**.

---

> ### Author Response · Authors · 2025-11-25
> **Author Response to the Initial Public Comments from Danqi Wang (Part 8 of 9)**
>
> ## Q3-4: Diminishing Returns are Normal Phenomena in Representation Learning
>
> Finally, we also want to discuss the diminishing returns that you mentioned:
>
> > On that note, looking at the results (Tables 1-2), the improvement from T=5 to T=20 is quite modest (only 2-3% absolute). This suggests a strong case of diminishing returns.
>
> Specifically, we want to clarify that the observed phenomena of diminishing marginal returns are **entirely expected and normal** in representation learning.
>
> Notably, the phenomenon of diminishing returns is also **common in gradient-based representation learning methods** (e.g., simply increasing the depth of a DNN can not lead to continuous performance improvement up to 100% without encountering a bottleneck). Even the high performance achieved by large models like LLMs is a result of scaling the model *alongside* vast increases in training data volume (i.e., the scaling law).
>
> Therefore, the fact that our DeepAFL can achieve stable and significant improvement across various backbone types **without increasing any training data volume** is already highly noteworthy. Especially when considering the **overall extremely low runtime cost**, the performance gain delivered by our DeepAFL can be considered to have **virtually no time overhead**.
>
>
>
> ----
>
> We sincerely hope that our preceding explanations and clarifications can adequately address your concerns on our DeepAFL's representation learning.
>
> Thank you again for your insightful comments!
>
> ### References
>
> [6] Bengio, Yoshua, Courville, Aaron, and Vincent, Pascal. Representation learning: A review and new perspectives. *IEEE Transactions on Pattern Analysis and Machine Intelligence*, 2013.
>
> [7] Wikipedia contributors. Feature learning. *Wikipedia*.
>
> [8] Run He, et al. Semantic Shift Estimation via Dual-Projection and Classifier Reconstruction for Exemplar-Free Class-Incremental Learning. *Forty-second International Conference on Machine Learning (ICML)*, 2025.
>
> [9] Alain, Guillaume, and Bengio, Yoshua. Understanding intermediate layers using linear classifier probes. *International Conference on Learning Representations (ICLR)*, 2016.

---

> ### Author Response · Authors · 2025-11-25
> **Author Response to the Initial Public Comments from Danqi Wang (Part 9 of 9)**
>
> # Q4: On the Theoretical Analyses
>
> > Lastly, the theoretical analysis does not guarantee generalization or the learning of useful representations. Can we theoretically demonstrate that adding more classification layers results in a “better” representation, or does it merely become more “discriminative”?
>
> Thank you for your valuable questions.
>
> As we have previously discussed the notion that **"more discriminative" can be considered a category of "better" representation** in our response to your Q3-3, we believe your Q4 mainly focuses on our theoretical analyses regarding generalization guarantee.
>
> Here, we would like to categorize the *"generalization"* ability you mentioned into two main meanings for a comprehensive discussion:
>
> * **Generalization from Training Data to Test Data:** Since our theory only provides the non-increase of empirical error, we surmise you are curious whether this theory can be extended to an analysis of the generalization error. In this case, we assume that the training and test data remain IID, which is the common practice and prerequisite in almost all supervised statistical learning theory.
>
> * **Generalization (or Transferability) from IID to Out-Of-Distribution (OOD):** Whether the features learned by our DeepAFL can be applied to other cross-domain applications with transferability.
>
> For the first point (i.e., Training-to-Testing generalization), we want to state that our proposed monotonic decrease of empirical risk within our DeepAFL (as proven in Theorems 2 and 3 of our paper) is a very strong theoretical result. Crucially, this conclusion can be straightforwardly **extended to imply a better Generalization Error Bound (GEB)** in statistical learning theory.
>
> Intuitively, this is because our DeepAFL does **not increase the complexity of the linear analytic classifier**. At each layer, we utilize the same analytically-optimized linear classifier based on the least-squares method, as defined in Equation (4). Specifically, during the layer-wise optimization process for representation learning in DeepAFL, the data sample size ($N$), the feature dimension ($d _ {\Phi}$), and the regularization parameter ($\lambda$ & $\gamma$) all remain constant.
>
> Since more detailed theoretical explanations about our *Training-to-Testing generalization* have been provided in our responses to the other reviewers, we will not reiterate these details here for the sake of brevity. We encourage further reference to our Responses to **Reviewer ZKVc in W2** and **Reviewer ofER in W2 & Q4**, for more detailed analyses.
>
> Then, for the second point (i.e., IID-to-OOD generalization/transferability), we want to clarify that "transferability" is not a necessary condition for successful representation learning but pertains to the distinct field of transfer learning. Since we did not claim the transferability ability of our DeepAFL, its lack of immediate cross-domain transfer capability does not constitute a severe weakness for our current work.
>
> In fact, **the vast majority of current gradient-based supervised methods also fall far short of robust transferability**, as they tend to focus narrowly on the current task and inherently suffer when faced with out-of-distribution data. Furthermore, as the development of analytic-learning-based methods is still in its nascent stages, our current core aim is to **achieve direct performance improvement with marginal additional cost**.
>
> While it clearly **exceeds the scope of our current work**, we agree that further exploring how to improve the IID-to-OOD generalization/transferability is an **interesting and promising direction for future work**.
>
> We sincerely hope that our clarifications can adequately address your questions.
>
> ---
>
> This concludes the contents of our response to your **initially raised four questions**.
>
> Once again, thank you very much for your **valuable recognition and professional insights**.
>
> The process of responding to your questions **prompted significant reflection and yielded substantial new insights for us**, helping us to gain a deeper understanding of our own DeepAFL and seriously consider its potential future work directions.
>
> We have also noticed that **you have raised several new questions**. Please be assured that we will provide the **corresponding responses** **within the next 3 days**.
>
> It is really a pleasure and an honor to engage in such a meaningful discussion with a professional of your caliber!

---

### Comment · Area_Chair_8WkT · 2025-11-27

Dear Reviewers,

Thank you for the time and effort you have dedicated to reviewing this paper and providing thoughtful feedback. The authors have now submitted their responses to your comments. I kindly ask that you engage in the discussion with them and assess whether your concerns and questions have been fully addressed before the December 2 deadline.

Please also keep in mind that the author–reviewer relationship is reciprocal; the engagement you offer here reflects the same level of consideration you would expect when you are on the author side.

Thank you for your continued support and cooperation.

Best regards,
AC

---

### Author Response · Authors · 2025-11-30
**The Summary of Our Rebuttal**

Dear AC, SAC, and PC,

We are aware of the unexpected circumstances regarding the recent ICLR updates and regret that we cannot continue discussions with the reviewers under the original arrangement.

**We fully recognize the significant responsibility and workload the new AC faces under this situation.** To assist you in efficiently evaluating the merits of our work, **we would like to provide a concise summary of our rebuttal**.

# Current Review Status

To date, our work has received insightful comments from four official reviewers and one external public reader (Danqi Wang). Specifically, the initial scores from the four reviewers are **8, 6, 6, and 4 (Average: 6)**. According to statistics from Paper Copilot, this average places our submission in the top tier of all ICLR submissions (approximately **Top 3.85%–9.24%**), **reflecting strong initial recognition from the reviewers**.

On the same day the reviews were released, we received public comments from an **external reader**, Danqi Wang. We were pleasantly surprised by his/her strong endorsement of our work. This public reader raised highly professional questions (4 initial ones followed by 4 subsequent ones), leading to a very fruitful discussion. **Given that attracting external engagement during the ICLR rebuttal phase is rare, we believe this evidence demonstrates the broad potential impact of our work.**

After we addressed the public reader's questions, we noticed he/she posted a positive final comment (**explicitly praising our DeepAFL as “great work”**). However, this final comment was subsequently deleted, possibly due to the system's deletion to prevent abuse of public comments in the current situation. Fortunately, we captured a **snapshot** of this final comment before it disappeared. **We believe that the positive endorsement from this public reader would have further assisted the reviewers in fully evaluating the merits of our work, had this unexpected deletion not occurred.**

Although the previous AC kindly reminded reviewers to address our responses, they had not yet managed to engage in further discussion before the unexpected situation occurred. **Given the reviewers' initially positive and constructive attitude, we believe they were likely occupied during the first two weeks of the rebuttal and would have engaged in meaningful discussion during the final week had the disruption not occurred.** We also firmly believe our detailed responses and clarifications have fully addressed their concerns and provided strong grounds for raising their scores.

# Summary of Responses and Revisions

Throughout the rebuttal phase, we have comprehensively addressed the concerns raised by all parties.

Specifically, we have provided clarifications on the following key aspects:

* Insights of Residual Connections (Danqi-Q1)
* Experimental Setup (Danqi-Q2, ofER-W1, ofER-Q3, ZKVc-Q1)
* Claim of Deep Representation Learning (Danqi-Q3, Danqi-Q6, ofER-W2, ofER-Q4, ZKVc-W2)
* Theoretical Analyses' Extensions, Assumptions, and Bounds  (Danqi-Q4, Danqi-Q8, mCa1-Q1, ofER-Q5)
* Efficiency and Quantitative Overhead (Danqi-Q5, mCa1-W3, mCa1-Q2)
* Analytical Solutions in Continual Learning (Danqi-Q7)
* Partial Client Participation (f9VW-Q1)
* Practical Scalability to Large ViTs (mCa1-W1)
* Privacy Argument (f9VW-W1)
* Content Modifications (f9VW-W2, mCa1-W2)
* Sandwiched Least Squares (mCa1-Q4, ZKVc-Q2)
* Comparisons with More Baselines (mCa1-Q3, ZKVc-W3)
* Advantages and Limitations of DeepAFL (Danqi-Q3, ofER-W3, ofER-Q2)
* Motivation and Use Case (ofER-Q1)
* Robustness to Noisy Data (ZKVc-W1)

Meanwhile, we have given the following additional new results:

* New Experimental Results:

  * "Apples-to-Apples" Comparisons (Danqi-Q2, ZKVc-Q1)
  * Partial Client Participation (f9VW-Q1)
  * Practical Scalability to Large ViTs (mCa1-W1)
  * IID Results (Danqi-Q2, ofER-W1, ofER-Q3)
  * Separability of Intermediate Features (ofER-Q2, ofER-W4, ZKVc-W2)
  * Local Training w/o FL (ofER-Q1)
  * Robustness to Noisy Data (ZKVc-W1)
  * Two New Baselines (ZKVc-W3)
  * Reducing Redundant Representations (ZKVc-Q2)

* New Theoretical Results:

  * Training-to-Testing Generalization (ofER-Q2, ofER-W4, ZKVc-W2)
  * Complexity of Privacy Techniques (f9VW-W1)

* New Illustrations:

  * More Visual Diagrams (mCa1-Q4)

Moreover, we are also making our best efforts to **document these detailed analyses and results in our revised PDF (marked in blue)**. Any remaining minor content will be incorporated into our camera-ready version.

In summary, we believe our work has already received great recognition from the public reader & reviewers, and we are highly optimistic that our detailed responses merit further score increases. **We would be deeply grateful if you would carefully evaluate the merits of our work and consider recommending it for a *Spotlight* presentation.**

Thank you again for your valuable time and consideration.

Sincerely,

Authors of ICLR 5209

---

### Meta-Review · Area_Chair_FwY2 · 2026-01-06

**Summary:**

major concerns from the reviewers:

1. Validity of "Representation Learning" Claims
2. Methodological Fairness & Benchmarking
3. Computational Efficiency & Scalability
4. Theoretical & Architectural Justification

**Reviewer Concerns:**

I believe most of the concerns have been sufficiently addressed by the rebuttal.  The issue of the dependence on the backbone architecture, and of cross-domain transfer capability are still outstanding, which are recognized by the authors as limitations of the current work. Despite these outstanding issues, I think the contribution of the current work is still significant and novel.

**Reviewer Scores:**

I think the reviewers would probably remain or raise their original scores after the rebuttal. Either would lead to an accept decision.

---

### Decision · Program_Chairs · 2026-01-26

Accept (Poster)